# SCENIC+: single-cell multiomic inference of enhancers and gene regulatory networks

Carmen Bravo González-Blas[1,2,4], Seppe De Winter [1,2,4], Gert Hulselmans[1,2], Nikolai Hecker [1,2], Irina Matetovici [1,3], Valerie Christiaens[1,2], Suresh Poovathingal[1], Jasper Wouters [1,2], Sara Aibar[1,2] & Stein Aerts [1,2] ✉

Joint profiling of chromatin accessibility and gene expression in individual cells provides an opportunity to decipher enhancer-driven gene regulatory networks (GRNs). Here we present a method for the inference of enhancer-driven GRNs, called SCENIC+. SCENIC+ predicts genomic enhancers along with candidate upstream transcription factors (TFs) and links these enhancers to candidate target genes. To improve both recall and precision of TF identification, we curated and clustered a motif collection with more than 30,000 motifs. We benchmarked SCENIC+ on diverse datasets from different species, including human peripheral blood mononuclear cells, ENCODE cell lines, melanoma cell states and *Drosophila* retinal development. Next, we exploit SCENIC+ predictions to study conserved TFs, enhancers and GRNs between human and mouse cell types in the cerebral cortex. Finally, we use SCENIC+ to study the dynamics of gene regulation along differentiation trajectories and the effect of TF perturbations on cell state. SCENIC+ is available at scenicplus.readthedocs.io.

Cell identity is encoded by gene regulatory networks (GRNs), in which transcription factors (TFs) interact with sets of *cis*-regulatory elements (CREs) to control transcription of target genes. CREs are often cell-type-specific and consist of specific TF-binding site (TFBS) combinations. In-depth knowledge of GRNs is important for mechanistic understanding of biological aspects underlying development[1,2], evolution[3,4] and disease[5]; however, knowledge of TF–target relationships at the *cis*-regulatory level is still limited.

Experimental techniques, including chromatin immunoprecipitation and sequencing (ChIP-seq), have yielded a wealth of TF-binding datasets. Nevertheless, for tissues with high cell-type diversity it remains challenging to map TFBSs because of the need for large amounts of homogenous cells. In addition, for most TFs, high-quality antibodies are lacking. Alternative approaches have recently been described that have increased cellular resolution (for example, single-cell CUT&Tag[6], nano-CT[7] and NTT-seq[8]) or that rely on genetic tagging (for example, DamID[9] and nanoDam[10]), yet such methods are still difficult to scale to all TFs.

Computational modeling is an alternative for identifying TFBSs. For example, SCENIC combines single-cell RNA-sequencing (scRNA-seq) coexpression networks with TF motif discovery[11,12], but it cannot identify the exact CRE targeted by the TF and it only uses a small proportion of a gene's putative regulatory space[13,14]. With single-cell chromatin-accessibility data, the accuracy of TFBS predictions can be improved substantially[15]. In fact, genomic regions that are specifically accessible in a cell type often represent enhancers and are enriched for TFBS combinations[2,14,16–18].

Here, we developed SCENIC+, a computational framework that combines single-cell chromatin accessibility and gene expression data with motif discovery to infer enhancer-driven GRNs (eGRNs).

## SCENIC+ uses more than 30,000 TF motifs to predict eGRNs

SCENIC+ is a three-step workflow that involves identifying candidate enhancers, identifying enriched TF-binding motifs and linking TFs to candidate enhancers and target genes (Fig. 1a and Supplementary

[1]VIB Center for Brain & Disease Research, Leuven, Belgium. [2]Department of Human Genetics, KU Leuven, Leuven, Belgium. [3]VIB Tech Watch, VIB Headquarters, Ghent, Belgium. [4]These authors contributed equally: Carmen Bravo González-Blas, Seppe De Winter. ✉e-mail: stein.aerts@kuleuven.be

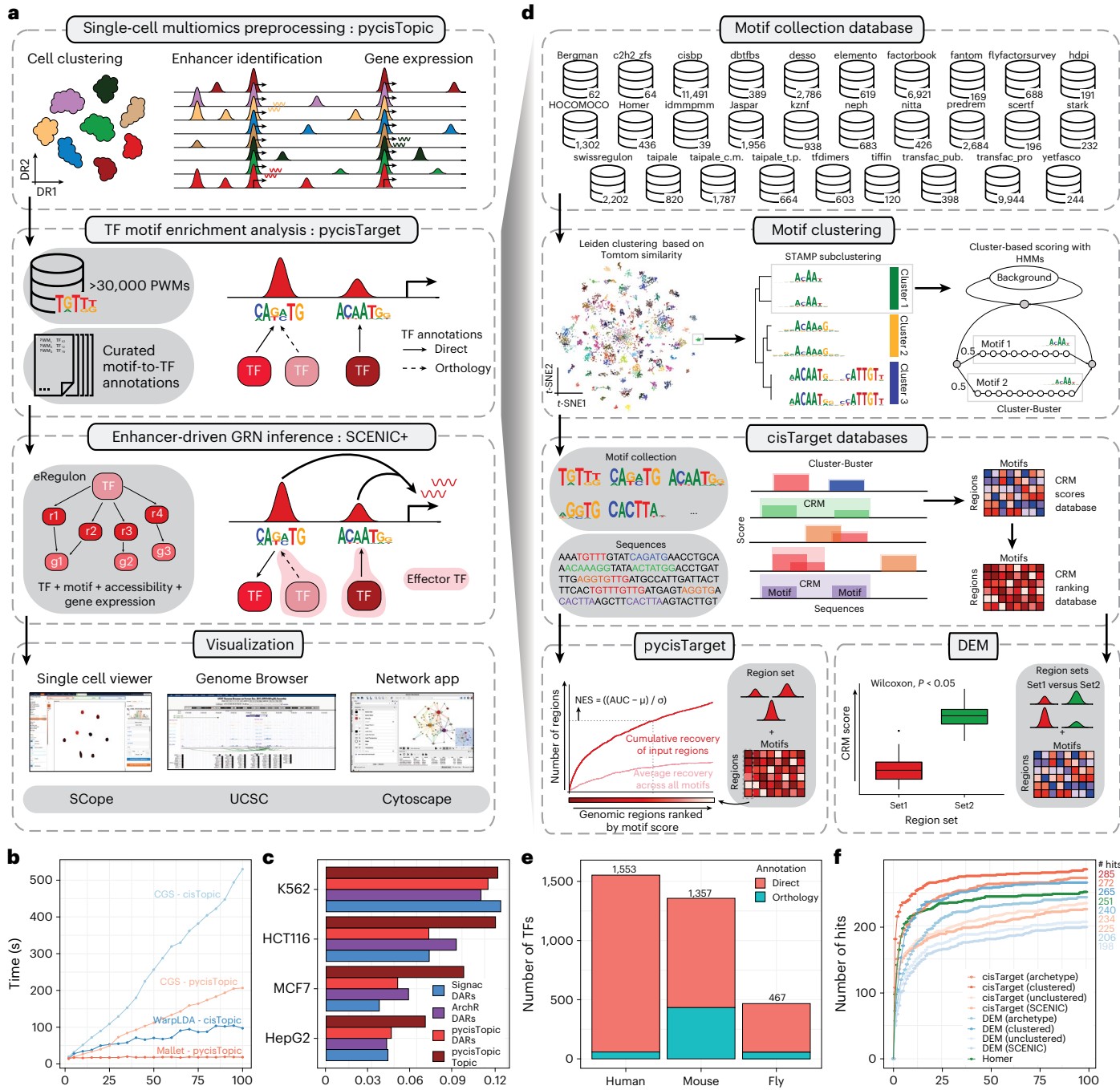

**Fig. 1 | The SCENIC+ workflow and motif collection. a**, SCENIC+ workflow. Topics and DARs inferred with pycisTopic are transformed into cistromes of directly bound regions by identifying modules that present significant enrichment of the regulator's binding motif using pycisTarget. SCENIC+ integrates region accessibility, TF and target gene expression and cistromes to infer eGRNs, in which TFs are linked to their target regions and these to their target genes. PWM, position weight matrix; UCSC, University of California, Santa Cruz. **b**, Running-time comparison per topic model using cisTopic with Collapsed Gibbs Sampling or WarpLDA (blue) and pycisTopic with Collapsed Gibbs Sampling or MALLET (red) for parameter optimization. **c**, Bar-plots showing the area under the recovery curve (AUC; enhancer recovery) on the top 10% of the ranking based on STARR-seq signal, for the top 5,000 DARs identified by Signac, pycisTopic and ArchR and top 5,000 regions from the cell-line-specific topics identified by pycisTopic. The AUC value is scaled by dividing by the maximum possible AUC at 10% of the ranking. Promoter regions were excluded from the

analysis. **d**, Workflow to create motif databases for SCENIC+. The SCENIC+ motif collection includes 34,524 unique motifs gathered from 29 motif collections, which were clustered with a two-step strategy. Input regions are scored for each cluster of motifs using hidden Markov models (HMMs), where each motif of the cluster is used as a hidden state. The score-based motif database is used in the DEM algorithm, whereas the ranking-based database is used for cisTarget. NES, normalized enrichment score. **e**, Number of TFs in the SCENIC+ motif collection annotated by direct evidence or orthology. **f**, Recovery of TFs from 309 ENCODE ChIP-seq datasets using different databases and motif enrichment methods, namely Homer, pycisTarget and DEM. The unclustered databases include all annotated motifs before clustering (singlets), the archetype databases use the consensus motifs of the clusters based on STAMP and the clustered databases use the motif clusters, scoring regions using all motifs in the cluster. The *x* axis shows the positions in which the TFs targeted in the ChIP-seq experiment can be found and the *y* axis shows the cumulative number of TFs that are found at that position.

Note 1). The output is a set of enhancer-driven regulons (eRegulons) that form an eGRN.

To find candidate enhancers, single-cell analysis of accessible chromatin (scATAC-seq) data is preprocessed using pycisTopic, a faster Python reimplementation of cisTopic[16] (Fig. 1b and Extended Data Fig. 1a–f). SCENIC+ uses both differentially accessible regions (DARs) and topics, sets of co-accessible regions, across cell types or states as enhancer candidates. Topics are more enriched for functional enhancer regions compared to DARs (Fig. 1c and Extended Data Fig. 1g).

To discover potential TFBSs in candidate enhancers we make use of motif enrichment analysis. For this we created the largest motif collection to date (Supplementary Note 2) and built a Python package called pycisTarget. pycisTarget implements two algorithms for motif enrichment analysis: the cisTarget ranking-and-recovery-based algorithm[11,19–21] and a Wilcoxon rank-sum test called differential enrichment of motifs (DEM) (Supplementary Note 3).

The motif collection is a secondary database containing 32,765 unique motifs collected from 29 collections (Fig. 1d and Extended Data Fig. 2a,b) along with TF annotations. The collection spans a total of 1,553 TFs, 1,357 TFs and 467 TFs, respectively in human, mouse and fly (Fig. 1e and Extended Data Fig. 2c). We clustered all motifs based on motif-to-motif similarity and found that scoring candidate regions using all motifs within a cluster yields a significantly higher precision and recall compared to using a single 'archetype' motif per cluster (Fig. 1f and Extended Data Fig. 2d–h). Both the cisTarget and DEM algorithm outperform Homer[22] (Fig. 1f and Extended Data Fig. 2d), with the DEM algorithm enabling detection of differential motifs in sets of regions with a similar motif content (Extended Data Fig. 2i–j).

SCENIC+ next uses GRNBoost2 (ref. 23) to quantify the importance of both TFs and enhancer candidates for target genes and it infers the direction of regulation (activating/repressing) using linear correlation. Motif enrichment analysis results are combined with GRNBoost2 inferences using a second enrichment analysis to recover the best TF for each set of motifs. This forms the eRegulon, a TF with its set of target regions and genes.

The overall running time and memory of the workflow ranges from 1 h and 21 Gb to 44 h and 461 Gb for the smallest and largest tested dataset, respectively (Extended Data Fig. 3).

## Illustration of SCENIC+ on PBMC multiome data

We first analyzed a publicly available single-cell multiomics dataset containing 9,409 human peripheral blood mononuclear cells (PBMCs) to showcase and validate SCENIC+. Dimensionality reduction based on eRegulon enrichment scores separates the main biological cell states (Fig. 2a). SCENIC+ identified 53 activator eRegulons, targeting a total of 23,470 regions and 6,142 genes. A total of 89% of genes have between 1–10 predicted enhancers and 49% of enhancers are predicted to most likely regulate the most proximal gene (Fig. 2b).

SCENIC+ recovers well-known master regulators of B cells (EBF1, PAX5 and POU2F2/POU2AF1), T cells (TCF7, GATA3 and BCL11B), natural killer (NK) cells (EOMES, RUNX3 and TBX21), dendritic cells (SPIB and IRF8) and monocytes (SPI1 and CEBPA) (Fig. 2c)[24–28]. The majority of the top five cell-type-specific TFs show co-binding to shared enhancers. Such cooperativity is not observed for TFs that are not specific for the same cell type (Fig. 2d). In particular, for B cells SCENIC+ suggests cooperativity between EBF1(+), PAX5(+) and POU2F2/AF1(+) (Fig. 2e), with a strong overlap of most of their predicted target enhancers with EBF1, PAX5 and POU2F2 ChIP-seq data (Fig. 2f–g).

In conclusion, SCENIC+ infers key regulators of different PBMC types and genomic target regions of these regulators in a high-throughput manner. This can be exploited to infer cooperativity.

## Validation of SCENIC+ predictions using ENCODE data

We next used simulated single-cell multiome data from eight ENCODE deeply profiled cell lines (Fig. 3a)[29,30] (GM12878, IMR90, MCF7, HepG2,

PC3, K562, Panc1 and HCT116) to validate the quality of TFs, target regions, region-to-gene relationships and target genes predicted by SCENIC+. We benchmarked these predictions to other tools that predict (e)GRNs using multiomics data, namely CellOracle[31], Pando[32], FigR[33] and GRaNIE[34], and included SCENIC[11,12] as a baseline (Supplementary Note 4). SCENIC+ identified 178 TFs. GRaNIE, FigR, SCENIC and Pando identified fewer TFs (39, 71, 108 and 157 TFs, respectively), while CellOracle identified 235 TFs (Fig. 3b). On average SCENIC+ predicts 471 and 1,152 target genes and regions per eRegulon (Fig. 3b).

To assess whether the predicted GRNs contain enough information to recapitulate all biological cell states we performed principal-component analysis (PCA) based on regulon enrichment scores. SCENIC+ was able to separate all cell lines, whereas other methods mixed two or more cell lines (Fig. 3c).

Next, we evaluated to what extent the identified TFs are biologically relevant by quantifying the recovery of highly differentially expressed TFs and TFs with many direct ChIP-seq peaks[35,36]. SCENIC+ achieved the best recovery for both metrics, followed by SCENIC (Fig. 3d, Extended Data Fig. 4a–c). Notably, TFs identified by SCENIC+ include most of the known lineage TFs, such as GATA1, TAL1, MYB and LMO2 for K562 (refs. 37–40) or HNF1A, HNF4A, FOXA2 and CEBPB for HepG2 (ref. 40) or ESR1 and GRHL2 for MCF7 (ref. 41). CellOracle had little recovery of differentially expressed TFs, whereas it recovers a large fraction of non-cell-line-specific TFs (for example GABPA, YY1 and SP1; Extended Data Fig. 4d).

As a third criterion, we evaluated the precision and recall of the predicted target regions of each TF based on TF ChIP-seq data in the ENCODE cell lines. For this, we used both unprocessed ChIP-seq peaks as well as direct ChIP-seq peaks from UniBind[35,36] and predicted TF binding by Enformer[42]. Overall, the predicted target regions of SCENIC+ and GRaNIE have the highest precision and recall, followed by Pando and CellOracle (Fig. 3e and Extended Data Fig. 4e–g). Furthermore, the predicted target regions by SCENIC+ have the highest enhancer activity as measured by STARR-seq (Extended Data Fig. 4h).

As a fourth criterion, we assessed the quality of predicted region-to-gene associations making use of deeply sequenced Hi-C data on five of the cell lines. SCENIC+ predicts a total of 402,838 links and has an average correlation coefficient of 0.25 with the Hi-C data (Fig. 3f and Extended Data Fig. 4i,j). The other methods identify fewer links, ranging from 13,123 to 311,168, and have a lower correlation with Hi-C data (Fig. 3f and Extended Data Fig. 4i,j).

Next, we evaluated target gene predictions using three methods. First, we reasoned that correct target gene predictions would allow for accurate estimation of target gene expression given the expression of the upstream TFs. To this end, we trained a regression model using each method's predicted eGRN as a scaffold. Predicted gene expression values using links from SCENIC+ had an average correlation coefficient of 0.61 with real expression values; this correlation was lower for Pando, GRaNIE, FigR and CellOracle (Extended Data Fig. 5a). Second, we quantified recovery of genes that change after knockdown of TFs. Across 157 TF perturbation datasets on the ENCODE cell lines, predicted target genes of SCENIC+ had the highest enrichment score per TF (Extended Data Fig. 5b) and the highest precision and recall (Fig. 3g and Extended Data Fig. 5c). Finally, to better account for indirect effects of TF knockdown experiments either due to indirect interactions or cooperativity (Extended Data Fig. 5d,e), we performed in silico TF perturbations based on the GRNs inferred by each method. While only a fraction of the variation in gene expression can be explained by any of the GRNs, eGRNs inferred by SCENIC+ agree the best with the experimental data (Extended Data Fig. 5f and Supplementary Table 1).

We performed these benchmark analyses using either motif (or ChIP-seq based) databases derived from all consensus peaks or derived from the SCREEN[43] regions, resulting in similar performance (data not shown). We also assessed the effect of sample size and coverage

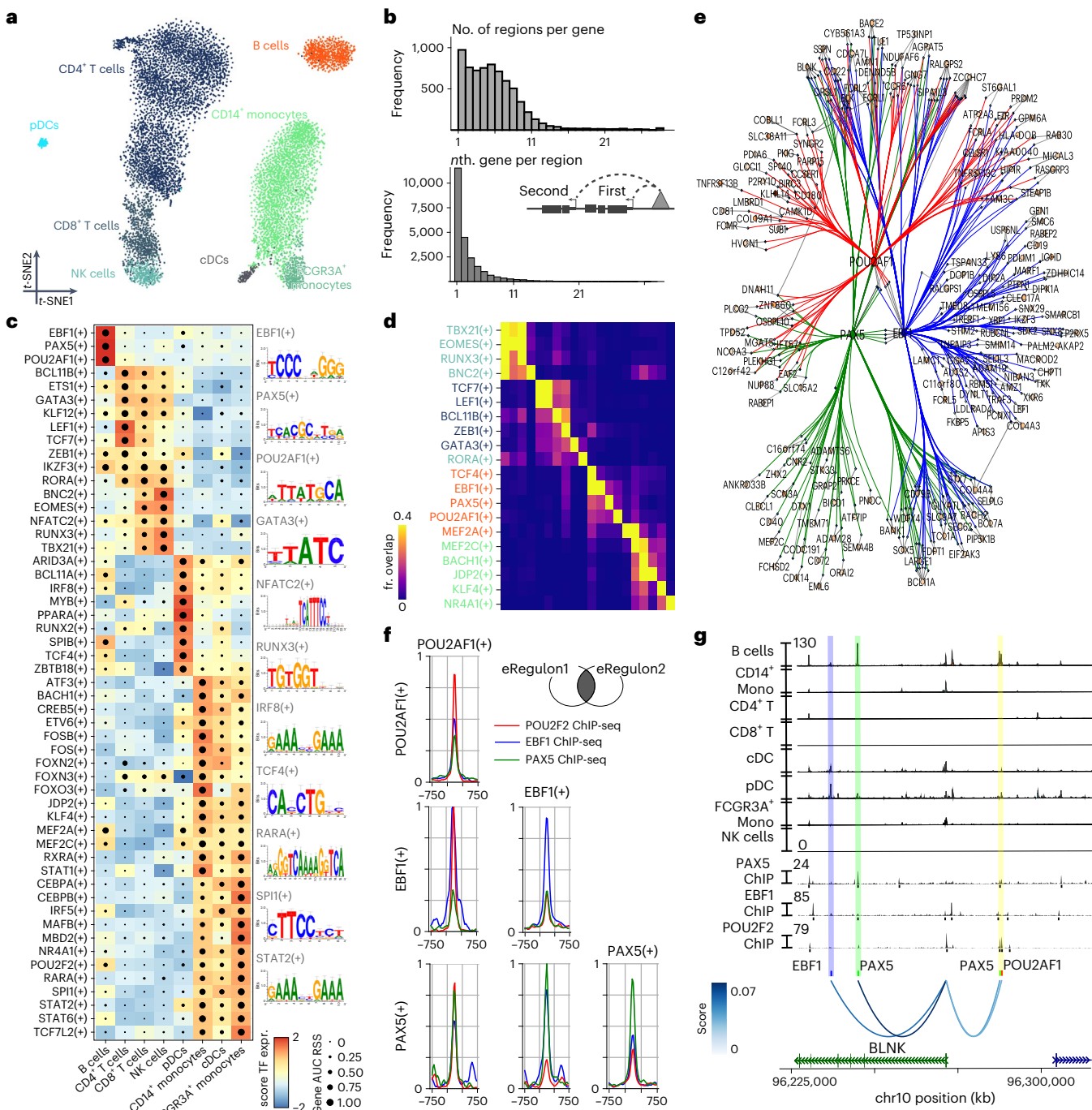

**Fig. 2 | SCENIC+ analysis on peripheral blood mononuclear cells.**
**a**, t-SNE dimensionality reduction of 9,409 cells based on target gene and target region enrichment scores of eRegulons. pDC, plasmacytoid dendritic cell; cDC, conventional dendritic cell. **b**, Top: distribution of the number of regions linked to each gene. Bottom: distribution showing whether the nth closest region to the target gene has the highest region-to-gene importance score. **c**, Heat map/dot-plot showing TF expression of the eRegulon on a color scale and cell-type specificity (RSS) of the eRegulon on a size scale. Cell types are ordered on the basis of their gene expression similarity. **d**, Overlap of target regions of eRegulons. The overlap is divided by the number of target regions of the eRegulon in each row. fr., fraction. **e**, Visualization of the eGRN formed by EBF1, PAX5, POU2AF1 and POU2F2. TF target nodes are restricted to highly variable genes and regions. **f**, Aggregated ChIP-seq signal of EBF1, PAX5 and POU2F2 in GM12878 on target regions of either EBF1, PAX5 or POU2AF1 and combinations of two of these factors. **g**, Chromatin-accessibility profiles across cell types and ChIP-seq signal together with peak calls of EBF1, PAX5 and POU2F2 in GM12878 on chr10:96226082–96316945. Region–gene links are shown as arcs. Region–gene gradient-boosting machine feature importance scores are encoded as colors (from light to dark blue). Predicted target sites of eRegulons are shown using colored ticks and semi-transparent boxes.

on the predictions of SCENIC+. SCENIC+ does not perform well with very few cells with low coverage (80 cells, 3,000 ATAC-seq fragments and 5,000 RNA-seq reads), but works accurately at standard coverage (Extended Data Fig. 6).

Finally, to test whether the same conclusions can be drawn from a real single-cell multiome dataset we repeated the benchmark analyses on the PBMC data (Extended Data Fig. 7). GRaNIE is not present in this benchmark as it was developed for bulk datasets and did not scale

to this larger (10,000 cells) dataset. GRNs from all methods were able to recapitulate all biological cell states, except the GRN inferred by FigR (Extended Data Fig. 7d). SCENIC+ and Pando performed best in terms of identifying biologically relevant TFs, with Pando finding additional TFs compared to SCENIC+ (Extended Data Fig. 7e,j). SCENIC+ had the highest precision and recall in terms of target region predictions (Extended Data Fig. 7f). Note that even when only scATAC-seq is used to identify TFs per cell type (for example, ArchR[44] and Signac[45]), the majority of known cell-type-specific TFs could still be recovered with accurate target region predictions (Extended Data Fig. 7g,h), showing that motif discovery is a powerful means for cell-type-specific TF prediction; however, using scATAC-seq alone resulted in large amounts of false-positive TF predictions, representing TFs that are not expressed in the cell type but have a similar motif (Extended Data Fig. 7i).

## SCENIC+ prioritizes functional enhancers

SCENIC+ uses automatic thresholding procedures, on the TF–gene, region–gene and region–motif scores, to obtain discrete sets of eRegulons; however, in some circumstances it may be beneficial to obtain a ranking of TFs, regions and genes based on their importance. For this reason, we implemented a ranking that quantitatively ranks TF–region–gene triplets. This ranking is the aggregated ranking[46] of the TF–region scores, TF–gene scores and region–gene scores (Fig. 3h).

We tested whether the triplet ranking can be used to prioritize potential enhancers. Indeed, regions in the top 10% of triplets have a higher ChIP-seq signal, as measured experimentally and in silico by Enformer and higher enhancer activity as measured by STARR-seq (Fig. 3i).

To further illustrate this, we focused on three master regulators of HepG2 cells: HNF4A, FOXA2 and CEBPB. Predicted target regions of these TFs have a high ChIP-seq coverage for these TFs, as measured experimentally and in silico by Enformer (Fig. 3j–l). Regions with high ChIP-seq coverage also have a higher TF-to-region ranking, compared to those with low coverage (Fig. 3j–l). In comparison, the predicted target regions of the same TFs by GRaNIE, Pando or CellOracle are very sparse. Only for HNF4A the predicted target regions by GRaNIE correspond well to those of SCENIC+; however, GRaNIE identified a subset of target regions that have a low Enformer score, even though they overlap with a ChIP-seq peak (Fig. 3j).

Next, we zoomed in on the target region of HNF4A that was predicted to be most important according to the triplet ranking. This region is also predicted to be targeted by FOXA2 and CEBPB and is predicted to regulate SPP1 (Fig. 3m), a marker gene of HepG2 cells (average log fold change (FC) of 9.24). The region is specifically accessible in HepG2 cells and has a high ChIP-seq signal for HNF4A, FOXA2 and CEBPB (Fig. 3n). Altogether, this region is a strong enhancer candidate for SPP1 in HepG2.

## SCENIC+ simulates phenotype switching of cancer cell states

Gene regulatory network analysis of cancer cells holds promise to identify stable (attractor) cell states and their regulators. As a case study we performed scATAC-seq on nine melanoma cell lines that represent different melanoma states[47,48] and combined these data with previously published scRNA-seq data for the same lines[48].

Cells clustered in three states based on eRegulon enrichment scores (Fig. 4a). Furthermore, a Boolean model[49] based on the top 25% TF-to-TF edges from the SCENIC+ network was sufficient to recapitulate all the main cell states (Extended Data Fig. 8a,b).

SCENIC+ recovered the known regulators for the melanocytic (MEL) state (MITF, SOX10, TFAP2A and RUNX3), the mesenchymal (MES) state (JUN, NFIB and ZEB1) and the intermediate sub-state of MEL governed by the MEL TFs supplemented with SOX6, EGR3 and ETV4 (Fig. 4b and Extended Data Fig. 8c–g)[48,50,51]. It was previously suggested that RUNX motifs are part of the MEL enhancer code[17,48] but which member of the RUNX family was unclear. Using SCENIC+, we predict that it is most likely RUNX3 (Fig. 4b).

It is known that melanoma cells can dynamically shift state from MEL to MES and vice versa, driving metastasis and therapeutic resistance, a process called phenotype switching[52]. Knockouts of specific TFs can drive this process[48].

To simulate phenotype switching and to prioritize TFs that underlie this process, we took inspiration from CellOracle[31] and GRaNPA[34], by using SCENIC+ as a feature selection method and training a random forest (RF) regression model for each gene to predict its expression based on the expression of their upstream TFs. After fitting the model, we use it to predict the effect of a TF perturbation by setting the expression of the TF to zero. To account for indirect effects (TFs targeting other TFs), perturbed gene expression values are propagated over several iterations. The effect of the simulated perturbation can be visualized by co-embedding the simulated gene expression matrix with the original one (Fig. 4c).

As proof of principle, we simulated the effect of SOX10 KD on the MEL state. Notably, the simulated cells, after SOX10 KD, suggest that they switch to a more MES-like state, whereby MES genes are upregulated and MEL genes are downregulated and this effect stabilizes after four iterations of simulation (Fig. 4d). This predicted effect of SOX10 knockdown was strongest in the intermediate cell lines and is fully recapitulated by experimental SOX10 KD, followed by RNA-seq[48] (Fig. 4e and Extended Data Fig. 8d).

Encouraged by this result, we simulated perturbations of all the identified TFs. Simulated knockdowns of RUNX3, SOX10 and MITF show the strongest potential to switch cells from MEL to MES; whereas knockdowns of ZEB1, SOX4 or SMAD3 are predicted to cause the reverse switch from MES to MEL (Fig. 4f–g), consistent with the role of these TFs in epithelial-to-mesenchymal transition (EMT)[48,53–57]. We also identified

---

**Fig. 3 | Benchmark of SCENIC+ and other single-cell multiomics GRN inference methods using ENCODE deeply profiled cell lines. a**, Diagram of benchmarking strategy. **b**, Number of TFs identified per method and distributions of the number of target genes and regions per regulon and method. **c**, PCA based on target gene and region enrichments and ARI quantification (4,000 cells). **d**, Cumulative recovery, per method, of TFs ranked in descending order by maximum logFC based on differential gene expression between all cell lines. **e**, F1 score distributions from the comparison of regulon target regions, per method and UniBind. **f**, Correlation between Hi-C links for top 100 marker genes and region–gene scores per method. Two-sided Wilcoxon rank-sum test comparing mean correlation of links versus shuffled links. The Holm method was used to correct for multiple testing. **g**, F1 score distributions from the comparison of regulon target genes, per method and TF perturbation data. **h**, Diagram of triplet ranking. **i**, Distributions of experimental and predicted TF ChIP-seq coverage and STARR-seq logFC target regions and other consensus peaks (not in eRegulon). **j–l**, Heat maps showing experimental and predicted ChIP-seq coverage on the union of predicted target regions per method with binary heat map indicating regions found per method and scatter-plot showing TF-to-region (TF2R) ranking of SCENIC+ target regions, for the TFs HNF4A (**j**), FOXA2 (**k**) and CEBPB (**l**). **m**, Network for top ten edges, targeted by any of FOXA2, HNF4A or CEBPB. Open and closed circles represent regions and genes and their color is proportional to the accessibility/gene expression logFC, respectively. Region-to-gene edges width and color represent importance scores. Arrow indicates the highlighted SPP1 enhancer (chr4:88107462–88107963). **n**, Chromatin-accessibility profiles across cell lines and HNF4A, FOXA2 and CEBPB ChIP-seq coverage on the SPP1 locus, with region-to-gene links and the SPP1 enhancer highlighted. For box-plots in **b**, **e**–**g** and **i**, the top/lower hinge represents the upper/lower quartile and whiskers extend from the hinge to the largest/smallest value no further than 1.5 × interquartile range from the hinge, respectively. The median is used as the center. NA, data are not available for the method. GRaNIE* was run with simulated single-cell data instead of bulk.

MXI1 and ZNF487 as potential EMT regulators, warranting further research. This strategy can thus be used to prioritize TFs regulating cell state and state transitions.

## Conservation and divergence of eGRNs in the mammalian brain

The mammalian cortex consists of a highly diverse but evolutionary conserved set of excitatory (pyramidal) and inhibitory neurons[58–60].

Although several marker TFs have been described for some of these cell types, little is known about how precise TF combinations, their binding sites and their target genes underlie neuronal identity. We reasoned that two independent SCENIC+ analyses on human and mouse cortex could reveal conserved, and thereby, high-confidence eGRNs underlying cortical cell types. This evaluation, using comparative genomics, also serves as a benchmark for robustly detecting eGRNs despite potential species and dataset-specific biases.

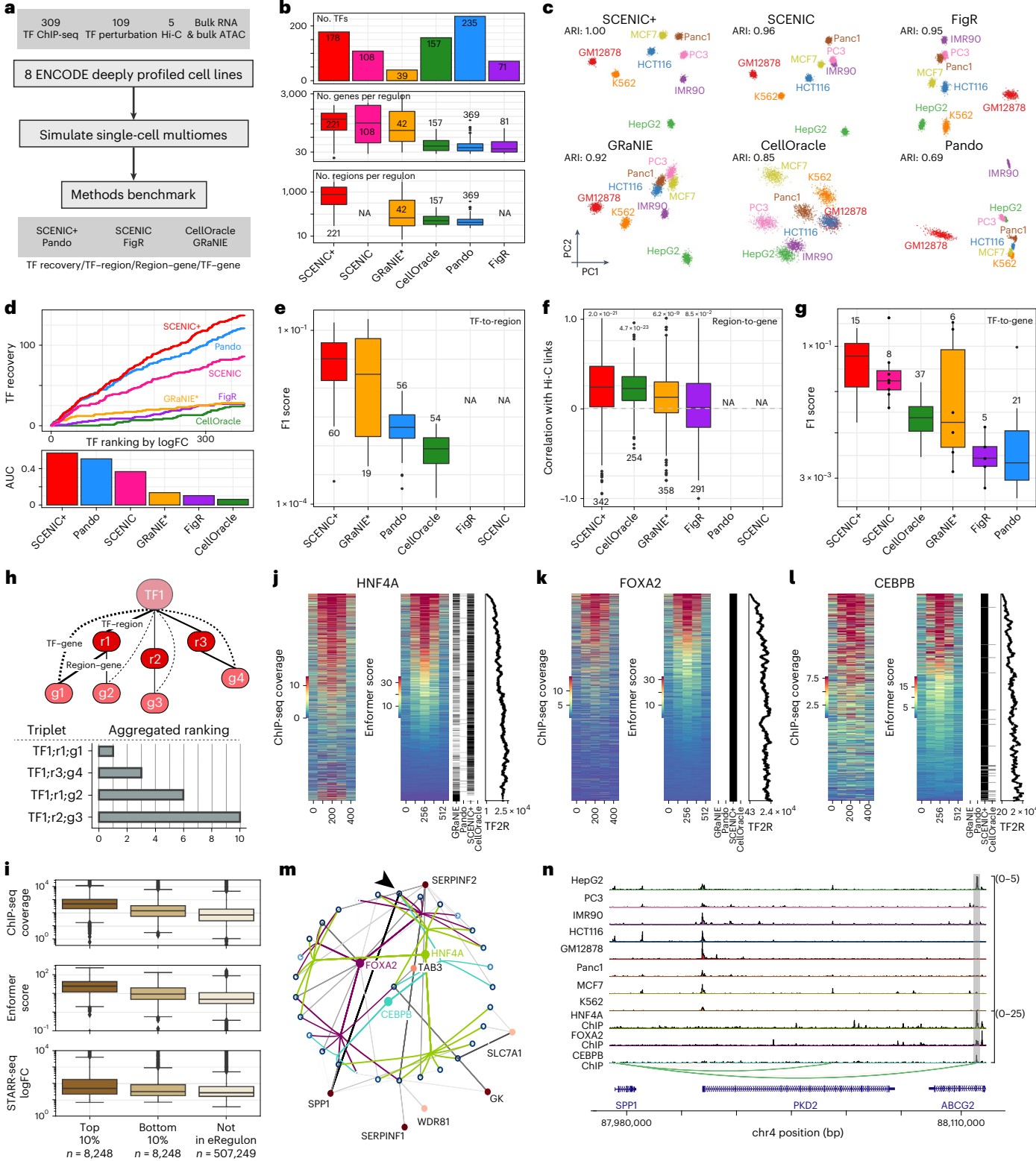

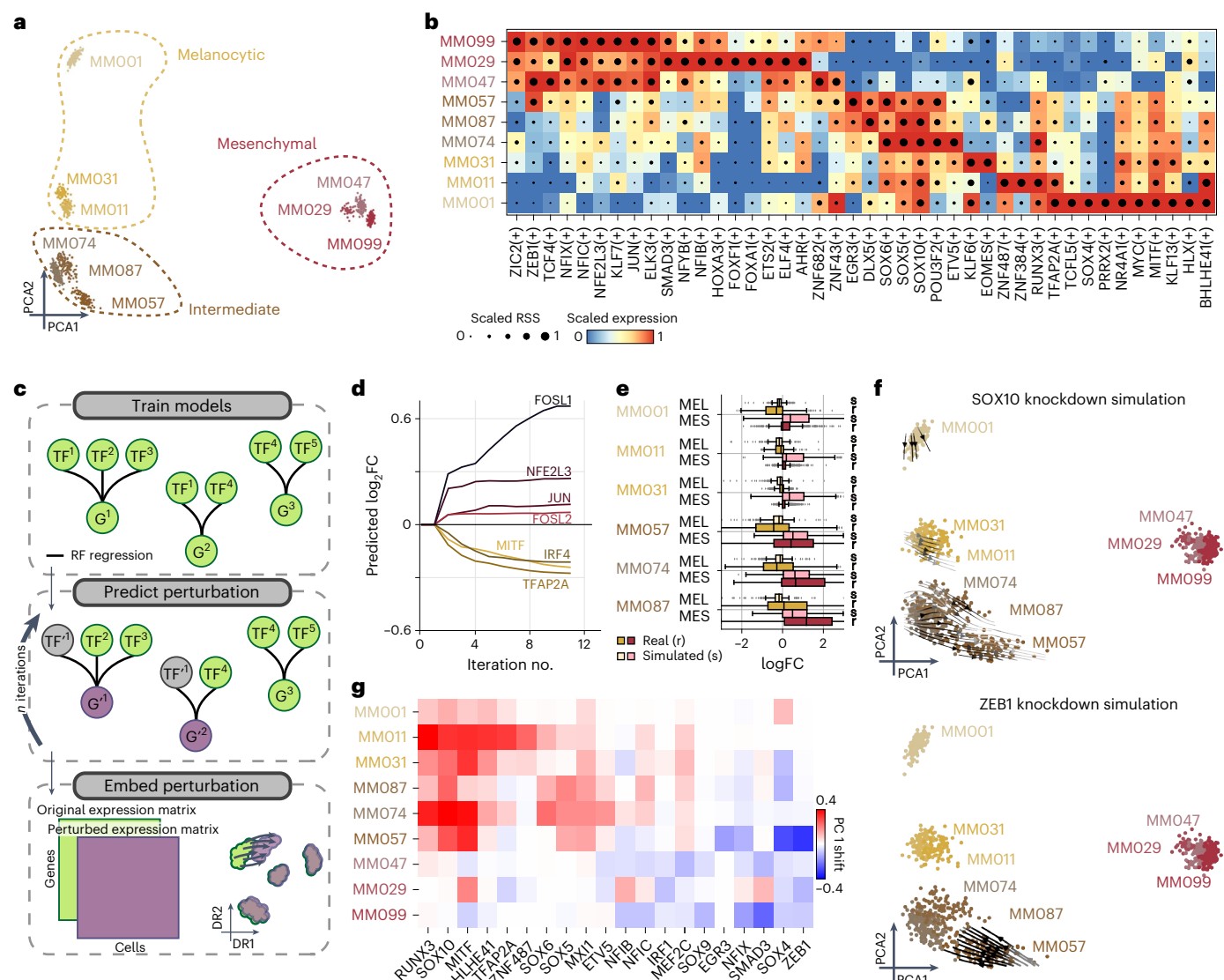

**Fig. 4 | SCENIC+ analysis using separate scATAC-seq and scRNA-seq data on a mix of human melanoma lines. a**, PCA of 936 pseudo-multiome cells based on target gene and target region enrichment scores. **b**, Heat map/dot-plot showing TF expression of the eRegulon on a color scale and cell-type specificity (RSS) of the eRegulon on a size scale. **c**, Illustration of how predictions from SCENIC+ can be used to simulate TF perturbations. Top: SCENIC+ is used as a feature selection method and RF regression models are fitted for each gene using TF expressions as predictors for gene expression. Middle: the expression of TF(s) is altered in silico and the effect on gene expression is predicted using the regression models, which is repeated for several iterations to simulate indirect effects. Bottom: the original and simulated gene expression matrices are co-embedded in the same dimensionality reduction to visualize the predicted effect of the perturbation on cell states. **d**, Predicted logFC of mesenchymal (red shades) and melanocytic (yellow shades) marker genes over several iterations of SOX10 knockdown simulation. **e**, Simulated (s) and actual (r) distribution of logFCs of melanocytic ($n = 523$) and mesenchymal ($n = 722$) marker genes after SOX10 knockdown across several melanoma lines. Upper/lower hinge represents upper/lower quartile, whiskers extend from the hinge to the largest/smallest value no further than $1.5 \times$ interquartile range from the hinge respectively. The median is used as the center. **f**, Simulated shift after SOX10 and ZEB1 knockdown represented using arrows. Arrows are shaded based on the distance traveled by each cell after knockdown simulation. **g**, Heat map representing the shift along the first principal component of each melanoma line after simulated knockdown of several TFs.

For the mouse cortex, we performed 10x single-cell multiome and for the human cortex we re-used a previously published multiome dataset[60]. We were able to identify matching cell types in both species, including layer-specific excitatory neurons, interneurons derived from the medial and caudal ganglionic eminences (MGEs and CGEs, respectively) and non-neuronal populations (microglia, astrocytes, endothelial cells, oligodendrocytes and oligodendrocyte progenitor cells (OPCs); Fig. 5a,b).

SCENIC+ identified 125 and 142 high-quality eRegulons for mouse and human, respectively, out of which 60 are found in both species (Fig. 5c,d). Notably, we observed a high correlation of the specificity scores of eRegulons for these orthologous TFs in matching cell types (Extended Data Fig. 9a), implying that cell-type identity can be decomposed into these 60 eRegulons. Eight out of 60 conserved TFs have not been described before in the context of the cortex. These include Smad3/SMAD3 in the excitatory neurons of the upper cortical layers, Pparg/PPARG and Bhlhe40/BHLHE40 in L4 excitatory neurons, Etv5/ETV5 and Nfat5/NFAT5 in L5/6 excitatory neurons, Thrb/THRB and Pbx1/PBX1 in L6 excitatory neurons and Meis1/MEIS1 in oligodendrocytes (Fig. 5c,d and Extended Data Fig. 9b–d). Projection of SCENIC+ regulons onto spatial transcriptomics data further validated layer-specific GRNs in the mammalian cortex (Extended Data Fig. 9e–j and Supplementary Note 5).

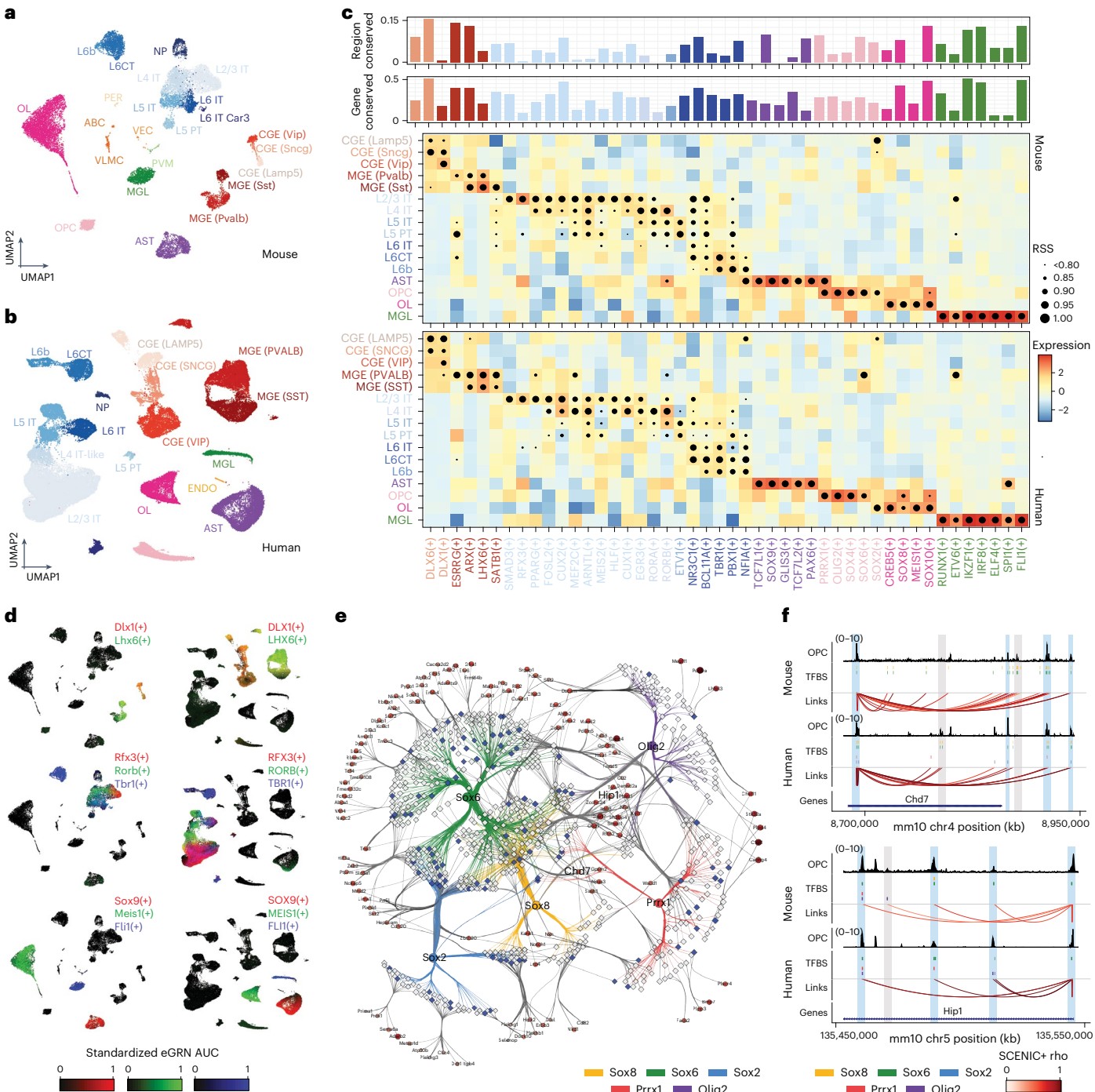

**Fig. 5 | SCENIC+ reveals regulatory lexicon conservation across mammalian brains. a**, Uniform Manifold Approximation and Projection (UMAP) dimensionality reduction of 19,485 mouse cortex cells based on target gene and region enrichment scores. **b**, UMAP dimensionality reduction of 84,159 human motor cortex cells based on target gene and region enrichment scores. **c**, Heat map/dot-plot showing TF expression of the eRegulon on a color scale and cell-type specificity (RSS) of the eRegulon on a size scale. The bar-plot above indicates the percentage of the regulon that is conserved in the other species, for predicted target regions (top) and target genes (bottom). **d**, Mouse and human UMAPs colored by enrichment scores for selected regulons using RGB encoding. **e**, Mouse-based OPC eGRNs with conserved TFs. Regions are shown as a diamond shape and their size represents the logFC of the region accessibility in OPCs compared to the rest of the cells. Regions conserved in the human brain

are shown in blue and regions only found in the mouse analysis are shown in gray. Genes are shown as a circular shape and their color and size represent the logFC of the gene expression in OPCs compared to the rest of the cells. TF−region links are colored by TF and region−gene links are colored by region−gene correlation coefficients. **f**, OPC coverage, TFBSs and region−gene links in two loci, Chd7 and Hip1. Data are shown in the mouse genome (mm10) and human data have been lifted over (mm10). Peaks found in both human and mouse are highlighted in blue, whereas peaks only accessible in one of the species are highlighted in gray. ABC/VLMC, vascular leptomeningeal cell; AST, astrocyte; CT, cortico-thalamic; ENDO, endothelial cell; IT, intratelencephalic; MGL, microglia; NP, near-projecting; PER, pericyte; PVM, perivascular macrophage; PT, pyramidal-tract; OL, oligodendrocyte; VEC, vascular endothelial cell.

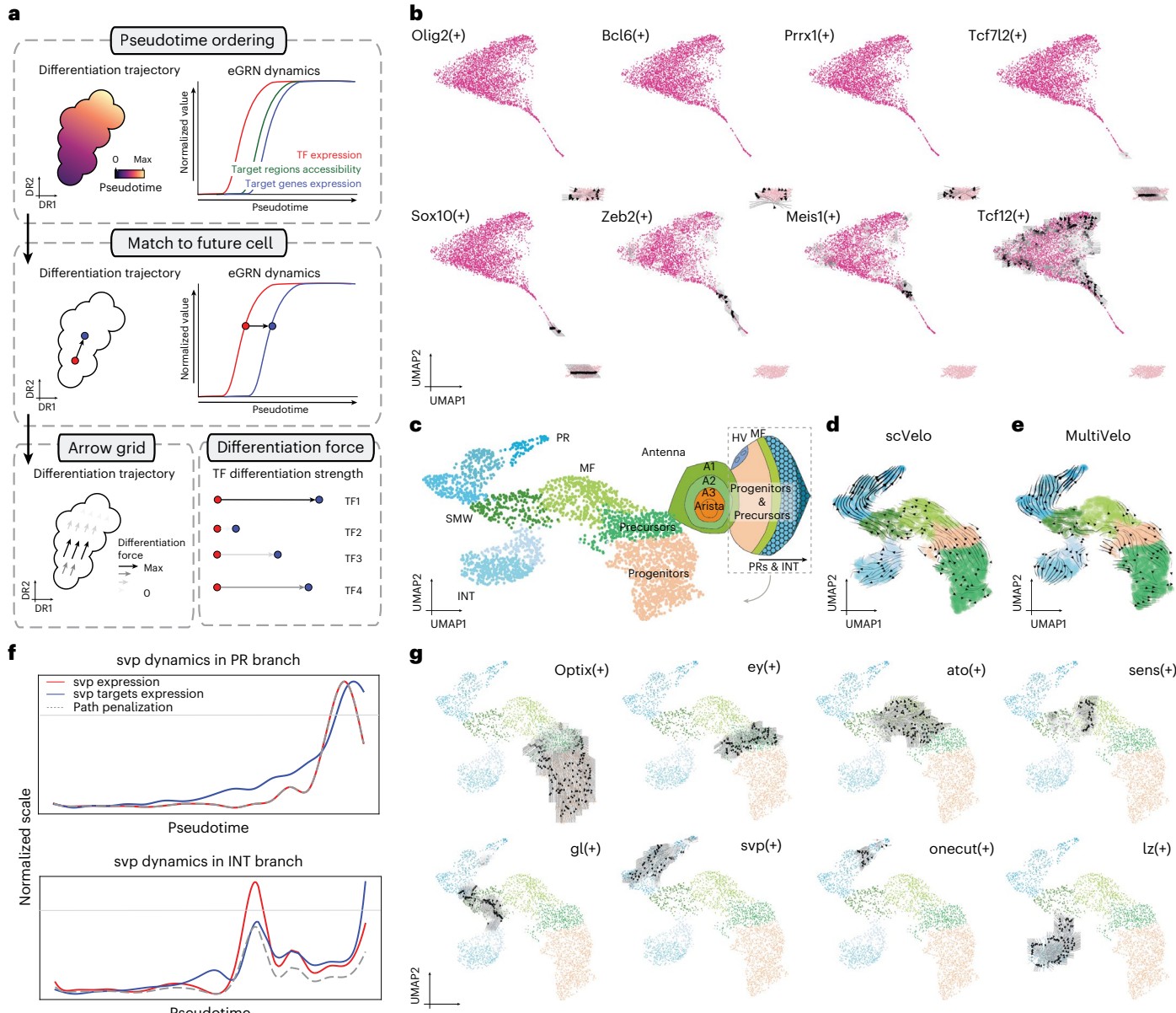

**Fig. 6 | Identification of differentiation drivers from SCENIC+ eGRNs.**
**a**, Computational approach to infer differentiation drivers from a SCENIC+ analysis. First, differentiating cells are ordered by pseudotime. Second, for each eRegulon, a standardized GAM is fitted along the pseudotime axis for its expression and its target genes (or regions) enrichment scores and each cell in a certain quantile of the GAM TF expression curve is mapped to its future cells in the same quantile in the GAM regulon enrichment curve. Finally, the differentiation force of a cell and regulon is defined as the distance from the TF expression curve to its future cell in the regulon enrichment curve. **b**, Arrow grid representation along the differentiation of OPCs to mature oligodendrocytes in the mouse cortex (4,435 cells). **c**–**e**, UMAP dimensionality reduction of 3,104 pseudocells from the fly eye based on target gene and region enrichment scores, with a schematic representation of the fly eye-antennal disc (**c**), scVelo velocity arrows (**d**) and

MultiVelo velocity arrows (**e**). **f**, Representation of svp dynamics along the two paths in eye disc differentiation. The gray horizontal line represents the TF expression threshold for arrows to be drawn. For cells below this threshold, the GRN velocity values are set to 0. The gray dashed line represents the penalization curve, which is the GAM fitted curve drawn using the standardized data across all possible paths for the cells in that path. Those points where the penalization and the TF expression curve disagree are considered artifacts (the TF gene seems to be expressed even if there is low expression, due to the standardization of the TF curve in that specific path). The red curve represents the GAM fitted curve using the standardized TF expression data (along the path) and the blue curve represents the GAM fitted curve using the standardized gene enrichment scores (along the path). **g**, Arrow grid representation along the eye disc differentiation.

eRegulons identified in only one of the two species can be either species-specific TFs or false negatives in one of the two analyses. To distinguish one from the other, we assessed the correlation coefficient of cell-type-specificity scores of each mouse eRegulon to its human orthologous matching eRegulon by converting the mouse predicted target genes to human orthologous genes. We found an additional 51 eRegulons with a correlation coefficient >0.6.

This indicates that these regulators are likely conserved, but were missed in the human analysis. For example, while Pou3f1/POU3F1 and Fezf2/FEZF2, previously described regulators of L5 PT and L5/6 neurons, respectively[59], were only found in the mouse analysis, the human-based mouse eRegulons are enriched in the corresponding cell types in the human dataset, matching the expression of these TFs (Extended Data Fig. 9b).

Next, we assessed the conservation of predicted target genes and regions across human and mouse. Out of the 102,746 regions found within human eGRNs, 84,861 could be lifted over (82%), whereas only 69% of all accessible regions (697,721) could be lifted over. Out of these 84,861 conserved mouse regions, 61,973 were accessible in the mouse cortex. In addition, 312,591 (out of 379,749) region–gene links from the human cortex could be lifted over, of which 283,900 corresponded to the same region–gene pair in the mouse cortex. On average, 28% and 6% of eRegulon target genes and regions, respectively, for each orthologous TF were conserved between the two species (Fig. 6c). We observed a strong correlation (0.68) of the fraction of conserved regions to the fraction of conserved genes per regulon. Thus, despite high conservation of TFs per cell type, the target genes (and even more so the target regions) are less conserved. This has also been observed in previous studies. For example, Bakken et al. reported 25% and 5% conservation of differentially expressed genes and DARs, respectively, across cell types in the human and marmoset cortex[60]. Stergachis et al. performed DNase I footprinting across 25 mouse tissues, finding that only around 20% of TF footprints are conserved in human, whereas 95% of the TF code is shared[61]. Genomic relocation and turnover of TFBSs and enhancers may partly explain these observations; however, the sparsity of the single-cell datasets may also contribute to these findings, as we are only capturing a fraction of the transcriptome and epigenome in each cell. Overall, we identify 4,798 and 8,318 conserved TF–region and TF–gene relationships, respectively (Supplementary Table 2). Given the sparsity of direct TF–enhancer and TF–target gene relationships in the literature, this is the largest set of conserved TF–target interactions in the mammalian cortex.

We further studied eGRN conservation in OPCs. While mature oligodendrocytes are driven by SOX10 (see further below), OPCs show higher activity of Sox2/SOX2, Sox6/SOX6 and Sox8/SOX8, alongside Olig2/OLIG2 and Prrx1/PRRX1 (Fig. 5e). These TFs have indeed been described previously in the literature as key drivers of OPC proliferation, migration, quiescence and differentiation[62,63]. Out of 636 regions predicted to be targeted by at least one of these five TFs in mouse and linked to at least one conserved target gene in both human and mouse, 102 TFBS are conserved across the two species (16%).

To further examine the relationship between target region conservation and TFBS presence, we zoomed in on two example loci, Chd7 and Hip1. We observed three distinct scenarios related to enhancer turnover: (1) a chromatin-accessibility peak and TFBSs are present in one of the species, whereas in the other species there is no accessibility and no TFBSs (two cases in the Chd7 loci); (2) a chromatin-accessibility peak and the same TFBSs are found in both species (two cases in the Hip1 loci); and (3) a chromatin-accessibility peak and at least one TFBS are shared across the two species, but additional non-shared TFBSs can be found. For the latter case, we also observed cross-species variations in the peak shape and size for peaks where different/additional TFBSs are found (for example, more accessibility in the species where additional TFBSs are found or a different peak shape when different TFBSs are found), whereas peaks with the same TFBSs have a similar shape (Fig. 5e).

Altogether, comparative analysis with SCENIC+ reveals TF lexicon conservation across mammalian brains, but divergence of their target genes and target regions.

## Predicting TFs driving differentiation using GRN velocity

Single-cell omics data are often used to sample cells during a dynamic biological process such as differentiation. Models, such as RNA velocity[64,65] and MultiVelo[66], which try to reconstruct the most likely trajectory from such data are available; however, these approaches do not include gene regulatory information to model dynamics.

We reasoned that regulatory relationships derived by SCENIC+ could provide additional intrinsic cues to predict cell-state dynamics.

For example, the expression of a TF may precede accessibility of its binding sites and chromatin accessibility in turn may precede target gene expression[67] (Fig. 6a). Therefore, we have developed a procedure to quantify the putative differentiation force of a TF. In this approach, cells are ordered along a pseudotime axis and each cell is matched to its future cell based on its current TF expression value and the cell with the best-matching future target gene expression. The differentiation force of a TF in each cell is then defined as the distance to its future cell, along the pseudotime axis. These forces can be plotted as arrows on a grid in any cell embedding (Fig. 6a).

We first applied this approach to a linear differentiation trajectory from OPCs to mature oligodendrocytes in the mouse brain. This revealed a set of TFs (Olig2, Bcl6 and Prrx1) that maintain OPC identity. On the other hand, Tcf7l2 and Sox10 had a delay between TF and target gene expression. This can be seen as arrows pointing toward newly forming oligodendrocytes (NFOLs). A final set of TFs (Zeb2, Meis1 and Tcf12) were identified as potential drivers of the maturation from NFOLs to oligodendrocytes (Fig. 6b). Notably, Meis1 has been previously described to be involved in early neurogenesis and hematopoiesis[68] but not in oligodendrocyte maturation. In line with this, SCENIC+ also identified Meis1 as a conserved TF in human and mouse oligodendrocytes (Fig. 5c).

Next, we applied GRN velocity to a branched differentiation trajectory from progenitor cells to photoreceptors or interommatidial cells in the developing fly retina. For this, we performed single-cell (sc) ATAC-seq on the eye field and integrated these data with scRNA-seq and scATAC-seq data on the developing eye (Fig. 6c)[14]. SCENIC+ identified 105 eRegulons that are active in the eye part. Of note, SCENIC+ found a repressor eRegulon for Cut (Ct) that is expressed in the antennae. It has been already shown that this acts as a repressor of the eye field[69] and here we predict that it directly represses 13 other TFs, including Spineless (*ss*), Eyeless (*ey*), Twin of eyeless (*toy*) and Optix.

As inferred by scVelo and MultiVelo, cells follow a differentiation trajectory from progenitors to the morphogenetic furrow (MF) and to the second mitotic wave (SMW), which forms a branch point to either photoreceptor cells (PRs) or interommatidial cells (INTs) (Fig. 6d,e). GRN velocity revealed strong differentiation arrows for Optix, Toy and Ey in progenitors, followed by Hairy (*hry*), Anterior open (*aop*), Rotund (*rn*) and Atonal (*ato*) in the MF. BarH1 (*B-H1*), BarH2 (*B-H2*), Sine oculis (*so*) and Glass (*gl*) were found to trigger the differentiation from MF toward both PRs and INTs. Lozenge (*lz*) was found to be the key driver of INT identity and Tramtrack (*ttk*) as the key driver for their maturation. In the photoreceptor branch, Senseless (*sens*) and Rough (*ro*) were identified as key regulators of differentiation. These are followed first by Asense (*ase*), Lola, Seven up (*svp*) and Scratch (*scrt*) and later by Shaven (*sv*) and Onecut (*onecut*) in mature photoreceptors (Fig. 6f,g). Notably, these findings are consistent with a previously described differentiation cascade in the eye disc[14].

## Discussion

CREs are key to control differential gene expression across cell types, during development, in evolution and in disease[1–5,70]. Yet, only few GRNs have been characterized to the level of detail where they include CREs as nodes[2,14]. We lack such GRNs mainly due to challenges associated with high-throughput experimental identification and validation of TFBSs. For this reason, we need computational methods that can identify TFBSs on a genome-wide scale and at the cell-type-specific level. Single-cell chromatin accessibility and gene expression profiling combined with sequence analysis is ideally suited for this and led to the concept of eGRNs[2,14,31–34]. In this work we present SCENIC+, a computational method to efficiently infer eGRNs.

By applying SCENIC+ to single-cell multiome data across a range of biological systems and across species we showed that SCENIC+ can accurately identify key TF combinations for each cell type. More notably, it can confidently link these TFs to CREs and target genes.

By comparing SCENIC+ to other methods, we could identify several elements that improve the quality of eGRN inference. First, the use of topic modeling improves unsupervised prioritization of informative regions. Second, the use of multiple motifs per TF and the use of a large motif collection improve the recall to identify important TFs. Finally, the use of motif enrichment analysis instead of motif scanning that is used in alternative methods reduces the false-positive rate of TFBS predictions.

One biological application where eGRN inference plays a pivotal role is in evolutionary genomics. For example, within the mammalian cortex, the majority of cell types were found to be conserved[60,71–73]; however, hundreds of genes are differentially expressed between orthologous cell types[60]. Comparison of eGRNs inferred across species can provide insights into these discrepancies. By mapping human and mouse eGRNs in the cortex, we found that cell-type-specific TF combinations are strongly conserved; however, TFBSs and enhancers show high turnover in line with earlier experimental findings[61,74]. This alludes to the fact that the unique combination of TFs and their interactions (the core regulatory complex[75]) define a cell type.

Another biological application is to study the regulatory underpinnings of dynamic cell-state changes. For this, we developed two downstream methods that exploit the inferred eGRN. One method predicts the effect of a TF perturbation on the transcriptome, which can be used to screen for the most important TFs needed to maintain a certain cell state. Another method, called GRN velocity, models the effect of each TF in a differentiation trajectory. This technique is complementary to other methods that infer directionality in differentiation trajectories (such as scVelo[65] and MultiVelo[66]).

There are limitations with this study and eGRN inference methods in general that may be overcome with future technological advances. Benchmarking these methods is challenging due to the lack of standardized ground-truth data. For example, to evaluate the predicted target genes we relied on transcriptome changes after the perturbation of a TF, which also causes indirect downstream effects and requires one experiment per targeted TF. Another challenge is the validation of enhancer–gene relationships, for which we used Hi-C data. Hi-C has a limited resolution and the relationship between physical enhancer–promoter distance and gene expression is still unclear[76–78] and warrants further research[79]. Furthermore, even though we show that eGRNs can be used to model transcriptome changes upon perturbation, their power is still limited. Further improvements may require more sophisticated models, for example using deep neural networks[42], to yield both quantitative and biologically explainable predictions[80]. Finally, eGRN inference is still biased toward activation and is less accurate in identifying repressive interactions (Extended Data Fig. 10 and Supplementary Note 6).

In conclusion, in this study we present SCENIC+, a tool to infer eGRNs from single-cell multiomics data. SCENIC+ and the code for downstream analyses is available at https://github.com/aertslab/scenicplus.

## Online content

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

## Methods

### SCENIC+ workflow

The SCENIC+ workflow consists of three main analysis steps: (1) unsupervised identification of enhancers with shared accessibility patterns from scATAC-seq data; (2) prediction of TFBSs via motif enrichment analysis; and (3) prediction of eGRNs combining TF expression, TFBSs, region accessibility and gene expression. These steps are performed using three Python modules: pycisTopic, pycisTarget and SCENIC+. Detailed explanations are described in Supplementary Note 1. Links to the tools, SCENIC+ code and tutorials are available at scenicplus.readthedocs.io.

**pycisTopic.** *Consensus peak calling.* Pseudobulk fragment bed files per cell type were generated using the fragments file and cell-type annotations provided by the user. Peaks were called using MACS2 (ref. [81]) with parameters –format BEDPE –keep-dup all –shift 73 –ext_size 146. An iterative approach described by Corces et al.[82] was used to obtain a consensus peak set. Briefly, each peak's summit was extended with a 'peak_half_width' (default 250 bp) in each direction and overlapping and less-significant peaks were filtered out. The original peak was kept if there was only a single peak. The original peak with the highest score was kept if there were two or more overlapping peaks. This process was repeated until there were no more overlapping peaks. The process of consensus peak generation was repeated twice: first for each cell type separately and, second, after peak score normalization within the cell type, using the union of peaks across cell types.

*Quality control.* The sample-level statistics that we used to assess the overall quality of the sample were:

- Barcode rank plot
- Insertion size
- Sample transcription start site (TSS) enrichment
- Fraction of reads in peaks (FRiP) distribution
- Duplication rate

The barcode-level statistics that we used to differentiate good quality cells versus the rest were:

- Total number of unique fragments per cell barcode
- TSS enrichment per cell barcode
- FRiP per cell barcode

Fragment count matrices were generated from the fragments files by counting the number of fragments that overlap with consensus peaks per high-quality cell barcodes.

Topic modeling was performed either using the serial Latent Dirichlet allocation (LDA) algorithm with a collapsed Gibbs sampler[83] or using MALLET[84] using the same default parameters as in cisTopic[16]. The model with the optimal number of topics was selected as the model based on the topic selection metrics, namely coherence, log-likelihood and the metrics described in refs. [85] and [86] (Supplementary Note 1).

Region–topic probabilities were binarized either using the Otsu method or by taking the top-*n* regions per topic.

Dropouts in scATAC-seq data were imputed by matrix multiplication of the region–topic and cell–topic matrices.

DARs were calculated using a Wilcoxon rank-sum test on the imputed probability matrix and selecting regions with a logFC > 0.5 and Benjamini–Hochberg adjusted *P* values < 0.05.

**pycisTarget.** *Generation of cisTarget database.* For the generation of the cisTarget database, a matrix with regions as rows (clusters of) motifs as columns and either raw scores (DEM) or ranking of these scores (cisTarget) was generated by scoring the DNA sequence of consensus peaks using Cluster-Buster[87]. Briefly, Cluster-Buster uses HMMs to score clusters of motifs given a set of DNA sequences. Each motif within a cluster is used as a separate hidden state in the model. Cluster-Buster was run separately for each (cluster of) motif(s) on the DNA sequence of all consensus peaks and the maximum cis-regulatory module score per region was used as the score for each region.

*cisTarget algorithm.* For the cisTarget algorithm[19–21], for each (cluster of) motif(s) a recovery curve approach was used using a set of regions, for which to calculate motif enrichment and the ranking database containing ranked (cluster of) motif(s) scores in descending order for the (cluster of) motif(s) of interest. The recovery curve was defined as the cumulative number of regions within the region set found at each position of the ranking. Enrichment was calculated as a normalized AUC at the top 0.5% ranking (NES).

$$NES = \frac{AUC - mean(AUC)}{s.d.(AUC)}$$

where

mean (AUC) is the average AUC value across all motifs

s.d. (AUC) is the standard deviation of AUC values across all motifs

By default, motifs that obtain an NES >3.0 are kept. To obtain the target regions for each motif (motif-based cistrome) the regions at the top of the ranking (leading edge) are retained. The top of the ranking is defined by an automated thresholding method that retains regions with a ranking below the rank at max, which is defined by the following formula:

$$RankAtMax = max(rcc_{motif} - [\mu(rcc_{allmotifs}) + 2 \times s.d.(rcc_{allmotifs})])$$

where

$rcc_{motif}$ is the recovery curve of the motif of interest.

$\mu(rcc_{allmotifs})$ is the average recovery curve over all motifs.

$s.d.(rcc_{allmotifs})$ is the standard deviation of the recovery curve over all motifs.

*DEM algorithm.* For each (cluster of) motif(s) a Wilcoxon rank-sum test was performed between a foreground and a background set of regions using the score distributions for the (cluster of) motif(s). Motifs with an adjusted *P* value < 0.05 (Bonferroni) and logFC > 0.5 were kept. Regions containing the motif (motif-based cistrome) were obtained by taking regions with a cis-regulatory module score >3 for each enriched motif.

**SCENIC+.** *Generation of pseudo-multiome data.* In cases of non-multiome data, pseudo-multiome data were generated by sampling a predefined number of cells from each data modality within the same cell-type annotation label and averaging the raw gene expression and imputed chromatin-accessibility data across these cells to create a multiome meta-cell containing data of both modalities.

*Calculating TF-to-gene and region-to-gene scores.* The Arboreto Python package (v.0.1.6) was used to calculate importance scores. TF-to-gene importance scores were calculated using gradient-boosting machine regression by predicting raw TF expression from raw gene expression counts and using the importance score of each feature (gene) as the TF-to-gene importance score. Pearson correlation was used to separate positive (>0.03) from negative (<−0.03) interactions. The importance score of a TF for itself was set to the maximum importance score across all genes added with an arbitrary small value of $1 \times 10^{-5}$. Region-to-gene importance scores were calculated using

gradient-boosting machine regression by predicting TF expression from imputed region accessibility, using all regions within a gene's search space and using the importance score of each feature (region) as the region-to-gene importance score. Spearman rank correlation was used to separate positive (>0.03) from negative (<−0.03) interactions. A gene's search space was defined as a minimum of 1 kb and a maximum of 150 kb upstream/downstream of the start/end of the gene or the promoter of the nearest upstream/downstream gene. The promoter of a gene was defined as the transcription starting site of that gene ±10 bp.

*Binarizing region-to-gene importance scores.* Region-to-gene importance scores were binarized by taking the 85th, 90th and 95th quantile of the region-to-gene importance scores, the top 5, 10 and 15 regions per gene based on the region-to-gene importance scores and a custom implementation of the BASC[88] method on the region-to-gene importance scores.

*eRegulon creation.* For each TF, TF–region–gene triplets were generated by taking all regions that are enriched for a motif annotated to the TF and all genes linked to these regions, based on the binarized region-to-gene links. Gene set enrichment analysis (GSEA) was performed by ranking all genes based on their TF-to-gene importance score and calculating enrichment of the set of genes within the TF–region–gene triplet using the gsea_compute function from GSEApy (v.0.10.8). Genes in the top of the ranking (leading edge) were retained and were the target genes of the eRegulon. This analysis was run separately for TF–gene and region–gene relationships with positive and negative correlation coefficients. eRegulons with fewer than ten predicted target genes or obtained from region–gene relationships with a negative correlation coefficient were discarded.

*eRegulon enrichment.* All consensus peaks and all genes were ranked respectively by their imputed chromatin accessibility and raw gene expression counts per cell. Enrichment for eRegulon target regions and target genes is defined as the AUC at 5% of the ranking and calculated using the AUCell function from the ctxcore Python package (v.0.1.2.dev2+g1ffcf0f).

*eRegulon dimensionality reduction.* The eRegulon enrichment scores for regions and genes were normalized for each cell and used as input into the UMAP, t-distributed stochastic neighbor embedding (*t*-SNE) or PCA from the Python package UMAP (v.0.5.2), fitsne (v.1.2.1) or Scikit-Learn (v.0.24.2), respectively.

*eRegulon specificity scores.* eRegulon specificity scores were calculated, per cell type and eRegulon, using the RSS algorithm as described elsewhere[12,89] using target region or target gene eRegulon enrichment scores as input. Briefly, the Jensen–Shannon divergence was calculated by comparing the distribution of enrichment scores per cell type to the distribution that was set to all zeros, except for the cell type of interest, where it was set to one.

*Triplet ranking.* For all TF–region–gene triplets from eRegulons, rankings of TF-to-gene importance scores, region-to-gene importance scores and the best-ranked position of the region across all motifs annotated to the TF were aggregated as described by Aerts et al.[46]

## SCENIC+ motif collection

The SCENIC+ motif collection includes more than 49,504 motifs from 29 motif collections (Supplementary Note 2 and Supplementary Table 3). Identical motifs across collections (after rescaling) were merged, resulting in 34,524 motifs. Motif-to-motif similarities using TomTom[90] (MEME v.5.4.1). Motifs with equal length and similarity $q$ value < $10^{-40}$ were merged, resulting in 32,766 motifs (unclustered motif collection). For motif clustering, motifs, with an information content >5 that were similar to at least on other motif with $q$ value < $10^{-5}$ and not one of 1,265 dimer motifs nor part of the Factorbook and Desso collection, were used (11,526 motifs), and the remaining were kept as singlets (9,685 motifs). Motif similarity $q$ values were transformed as follows:

$$-\log_{10}\left(\text{TomTom } q_{\text{value}}\right) + 10^{-45}$$

Seurat[91] (v.4.0.3) was used to normalize, scale and perform PCA. Leiden clustering was performed on the top 100 principal components with a resolution of 25, resulting in 199 clusters. Sub-clustering was performed using STAMP[92] (v.1.3; using the -cc -sd −chp options) resulting in 1,986 subclusters. TF annotations per subcluster were merged based on direct and orthology evidence. These subclusters together with singlets and dimer motifs form the clustered motif collection.

## Benchmarking pycisTarget

Four different cisTarget databases were generated: (1) a database was generated using the unclustered motif collection; (2) a database was generated using the STAMP-consensus motif per cluster; (3) a database was generated using the clustered motif collection; and (4) a database was generated using the clustered motif collection but Transfac Pro motifs were removed. Motif enrichment analyses using these databases and the cisTarget and DEM algorithm and Homer[22] were performed on 309 ChIP-seq datasets from ENCODE[29,30] that were also included in UniBind[35,36] (Supplementary Note 3). The enrichment of motifs annotated to the TFs for which ChIP-seq was performed was assessed.

## DEM on SOXE cistromes

cisTarget and DEM were run on regions enriched for motifs annotated to SOX10 in melanoma cell lines (see Melanoma cell line analysis; $n = 18,506$), SOX10 in oligodendrocytes (see Comparative analysis in the mammalian brain using SCENIC+; $n = 2,553$) and SOX9 in astrocytes (see Comparative analysis in the mammalian brain using SCENIC+; $n = 6,817$). For DEM, one-versus-all comparisons were made.

## Comparison of cisTopic and pycisTopic

A simulated single-cell epigenomics dataset from five melanoma cell lines (three melanocytic and two mesenchymal) with 100 cells[16] was downloaded from https://github.com/aertslab/cisTopic. cisTopic (v.2.1.0) using Collapsed Gibbs Sampling and WarpLDA and pycisTopic (v.1.0.1.dev21+g8aa75d8) using Collapsed Gibbs Sampling and MALLET, using 150 iterations and 21 cores for 21 models (starting from 2 topics and from 5–100, increasing by 5), were run. For all models α was set to 50 divided by the number of topics and β was set to 0.1, as previously described[16,93].

## Cell-type discovery benchmark with ArchR, Signac and pycisTopic

scATAC-seq datasets from ENCODE deeply profiled cell lines were simulated (see Benchmark of GRN inference methods), with different coverages (20,000, 10,000 and 3,000 fragments per cell) and numbers of cells (25,000, 10,000, 1,000 and 80 cells). In all cases, the bulk consensus peaks were used to generate the fragment count matrix (see Benchmark of GRN inference methods). pycisTopic was run as described in the corresponding sections. ArchR and Signac were run using default parameters. Briefly, Signac (v.1.9.0) was run using latent semantic indexing (LSI), using the top 30 PCs (excluding the first PC as recommend) for dimensionality reduction and clustering. ArchR (v.1.0.2) was run with default parameters, using iterative LSI, using the top 30 PCs for dimensionality reduction and clustering. Dimensionality reduction was performed using UMAP, using the PC matrix (ArchR and Signac) or the topic contribution matrix (pycisTopic). To calculate the adjusted Rand index (ARI) in the power analysis based on simulated data from ENCODE, hierarchical clustering was performed on these matrices, making eight partitions based on the hierarchical tree using the cutree() function from the stats R package. In the mouse cortex, batch correction (per sample) was performed using the recommended approaches from each method. For pycisTopic, data were corrected using harmonypy (v.0.0.5) on the scaled cell–topic matrix (see Comparative analysis in the mammalian brain using SCENIC+). For Signac, the integrated LSI approach was used, as described in the scATAC-seq data integration vignette from the package. Briefly, LSI was performed

in each sample separately, integration anchors were identified with FindIntegrationAnchors() (using dims of 2:30) and LSI embeddings were integrated using IntegrateEmbeddings() (using dims.to.integrate of 1:30). For ArchR, the addHarmony() function was used to correct the iterative LSI embedding. Dimensionality reduction was performed using UMAP, using the corrected PC matrix (ArchR and Signac) or the corrected topic contribution matrix (pycisTopic).

## Enhancer discovery benchmark with ArchR, Signac and pycisTopic

DARs and regulatory topics were inferred using a simulated single-cell scATAC-seq dataset from ENCODE deeply profiled cell lines (see Benchmark of GRN inference methods). Briefly, pycisTopic was run with default parameters, using MALLET with 500 iterations for topic modeling and generating 21 topics (2 topics and from 5–100, increasing by 5), selecting the model with 40 topics based on the model selection metrics (see Benchmark of GRN inference methods). Signac[45] (v.1.9.0) was run using LSI, using the top 30 PCs (excluding the first PC as recommended) for dimensionality reduction and clustering. DARs were determined using the FindMarkers() function, keeping regions with $P$ value < 0.005. ArchR[44] (v.1.0.2) was run with default parameters, using iterative LSI, using the top 30 PCs for dimensionality reduction and clustering. DARs were determined using the getMarkerFeatures() function, using the bulk consensus peaks (as used to generate the fragment count matrix used for pycisTopic and ArchR) as peak set, accounting for potential biases based on TSS enrichment and cell coverage and using false discovery rate (FDR) ≤ 0.1 and logFC ≥ 0.5 as threshold.

These regions were compared to whole-genome STARR-seq data on K562, HepG2, HCT116 and MCF7. Data were downloaded from ENCODE[29,30] (ENCFF045TVA (K562), ENCFF047LDJ (HepG2), ENCFF428KHI (HCT116) and ENCFF826BPU (MCF7)) and intersected with the consensus peaks. Promoter regions, defined as the TSS of each gene ±500 bp, were excluded from this analysis. For each cell line, regions with STARR-seq data available were ranked based on the logFC value. Enrichment of the top 500 DARs and region–topic contributions (for cell-line-specific topics) in the top of this ranking was assessed for each of the four cell lines separately by calculating the AUC at 10% of the ranking.

## SCENIC+ time and memory complexity analysis

Several scATAC-seq datasets from the deeply profiled cell lines with different coverages (20,000, 10,000 and 3,000 fragments per cell) and numbers of cells (25,000, 10,000, 1,000 and 80 cells) were simulated. In all cases, the bulk consensus peaks were used to generate the fragment count matrix (see Benchmark of GRN inference methods). For pycisTopic, only the mandatory steps for the SCENIC+ workflow (namely object creation, topic modeling, dimensionality reduction, dropout imputation and DAR inference) were run using default parameters. pycisTarget was run using default parameters with cisTarget and DEM, using the bulk consensus peaks motif databases (see Benchmark of GRN inference methods). SCENIC+ was run with default parameters (Supplementary Table 5). The analyses were run on an Intel(R) Xeon(R) Platinum 8360Y (IceLake), with 300 GB memory and 20 cores.

## PBMC analysis

**Data.** Filtered feature barcode matrices and fragments files were downloaded from the 10x Genomics website (https://cf.10xgenomics.com/samples/cell-arc/1.0.0/pbmc_granulocyte_sorted_10k/pbmc_granulocyte_sorted_10k_filtered_feature_bc_matrix.h5 and https://cf.10xgenomics.com/samples/cell-arc/1.0.0/pbmc_granulocyte_sorted_10k/pbmc_granulocyte_sorted_10k_atac_fragments.tsv.gz).

**Quality control of scRNA-seq and cell-type annotation.** The scRNA-seq part of the multiome dataset was preprocessed using Scanpy[94] (v.1.8.2). Briefly, genes expressed in fewer than three cells were removed. Cells that expressed fewer than 200 genes or more

than 6,000 genes or had more than 30% counts in mitochondrial genes were removed. Doublets were detected and removed using Scrublet[95] (v.0.2.3) with a doublet score threshold of 0.17. This resulted in 11,101 high-quality cells. Cells were annotated using ingest label transfer, using the sc.tl.ingest function included in Scanpy[94] (v.1.8.2) and by matching transferred labels to Leiden clusters (resolution 0.8) based on maximum overlap using the annotated PBMC dataset included in the Scanpy package as a reference (sc.datasets.pbmc3k_processed()).

**Quality control of scATAC-seq and topic modeling.** The scATAC-seq part of the multiome dataset was preprocessed using pycisTopic (v.1.0.1.dev21+g8aa75d8). Briefly, consensus peaks (342,044) were called as described above using the downloaded fragments file and cell-type labels from the scRNA-seq side. Cells with fewer than $1 \times 10^{3.3}$ total number of unique fragments, FRiP <0.45 and TSS enrichment <5 were removed. Doublets were detected and removed using Scrublet[95] (v.0.2.3) with a doublet score threshold of 0.33. This resulted in 10,955 high-quality cells. Topic modeling was performed as described above using LDA with the collapsed Gibbs sampler. A model of 16 topics was selected based on the stabilization of the metrics described in refs. [85,86,96] and log-likelihood.

**Motif enrichment analysis.** Motif enrichment analysis was performed using pycisTarget (v.1.0.1.dev17+gd2571bf) as described above. For this, a custom score and ranking database was generated using create_cisTarget_databases Python package using the DNA sequences of consensus peaks and all annotated motifs as input. Motif enrichment was performed using both the cisTarget and DEM algorithm on cell-type-based DARs (logFC > 1.5), top 3,000 regions per topic and topics binarized using the Otsu method. The motif enrichment analysis was run both including promoters and excluding them. Promoters were defined as regions within 500 bp up- or downstream of the TSS of each gene. TSSs for each gene were downloaded from BioMart (http://sep2019.archive.ensembl.org) using the pybiomaRt package (v.0.2.0).

**SCENIC+ analysis.** The raw gene expression count matrix, imputed accessibility and motif enrichment results were used as input into the SCENIC+ workflow, keeping 9,409 cells with both high-quality ATAC-seq and RNA-seq profiles. The SCENIC+ workflow was run using default parameters and as described above. Briefly, a search space of a maximum between either the boundary of the closest gene or 150 kb and a minimum of 1 kb upstream of the TSS or downstream of the end of the gene was considered for calculating region–gene relationships using gradient-boosting machine regression. TF–gene relationships were calculated using gradient-boosting machine regression between all TFs and all genes. Genes were considered as TFs if they were included in the TF list available on http://humantfs.ccbr.utoronto.ca/ (ref. [97]). Final eRegulons were constructed using the GSEA approach in which region–gene relationships were binarized based on gradient-boosting machine regression importance scores using the 85th, 90th and 95th quantile; the top 5, 10 and 15 regions per gene and the BASC method[88] for binarization. Only eRegulons with a minimum of ten target genes were retained. For each eRegulon cell enrichment scores (AUC) of target genes and regions were calculated using the AUCell algorithm[11]. eRegulons for which the correlation coefficient between semi-pseudobulked per cell type (100 meta-cells per cell type and 5 cells per meta-cell) TF expression and region AUC scores was >0.7 or <−0.8 were considered as high quality and used for downstream analysis. This resulted in 63 regulons, with a median of 296 genes and 528 regions per regulon. Analyses can be explored in SCope at https://scope.aertslab.org/#/scenic-v2 and the UCSC Genome Browser at https://genome-euro.ucsc.edu/s/Seppe%20De%20Winter/scenicplus_pbmc.

**ChIP-seq enrichment in eRegulon target regions.** PAX5, EBF1 and POU2F2 ChIP-seq bigWig and summit bed files were downloaded from ENCODE (https://www.encodeproject.org/)[29,30] using the following

accession numbers: ENCFF702MTT and ENCSR000BHD for PAX5; ENCFF107LDM and ENCSR000BGU for EBF1; and ENCFF803HIP and ENCFF934JFA for POU2F2 for bigWig and summit bed files, respectively. The target regions of PAX5(+), EBF1(+) and POU2AF1(+) and regions targeted by any combination of two eRegulons were intersected with the ChIP-seq summits; the original region was kept if a region did not intersect with the summit. ChIP-seq coverage was calculated on these regions using the pyBigWig package (v.0.3.18) using 50 bins. Coverage was min–max normalized by using the minimum and maximum across all regions per ChIP-seq dataset. ChIP-seq data, along with pseudobulk accessibility, were visualized using the Signac[45] R package (v.1.3.0).

**Comparison of region-to-gene links with Hi-C data.** Hi-C data on GM12878 were downloaded from ENCODE (https://www.encodeproject.org/)[29,30] (ENCFF053VBX). SCALE-normalized scores across bins of 5 kb using Juicer Tools[98] (v.2.13.05) were extracted, keeping only links with scores >10 and involving a bin that overlaps at least one of the consensus peaks and a TSS (±1,000 bp), resulting in 4,842,692 region–gene links. Spearman rank correlations between the Hi-C scores and region-to-gene importance scores (gradient boost importance score and Spearman correlation coefficient) were calculated for all B-cell marker genes (logFC > 1.5 and adjusted $P$ value < 0.05).

**CellOracle analysis.** CellOracle was run as described at https://morris-lab.github.io/CellOracle.documentation/. Briefly, scRNA-seq data were analyzed using Scanpy[94] (v.1.9.1), using flavor = 'cell_ranger' and $n$_top_genes = 2,000 to identify highly variable genes and 14 PCs for dimensionality reduction. scATAC-seq data were processed with Cicero[99] (v.1.3.4.11) with a window of 500,000, which inferred 21,592 region–gene connections with coaccess ≥ 0.8 and TSS annotation obtained from BioMart (http://sep2019.archive.ensembl.org). Next, gimmemotifs[100] (v.0.17.1) was used to identify TFBSs using the gimmemotifs motif collection with default parameters. After inferring the links, we filtered links with the filter_links function using $P$ = 0.001, weight = 'coef_abs' and threshold_number = 2,000. This resulted in 100 regulons, with a median of 40 genes and 43 regions per regulon.

**Pando analysis.** Pando was run as described at https://quadbiolab.github.io/Pando/. Briefly, the data were processed using Signac[45] (v.1.8.0). MotifMatchR (v.1.16.0) was used with motifs from Jaspar and CisBP. To infer the GRN the infer_grn function was used with default parameters (peak_to_gene_method = 'GREAT', upstream = $1 \times 10^5$, downstream = 0 and extend = $1 \times 10^6$). The GRN edges were filtered using the find_modules function with default parameters ($P$_thresh = 0.1, nvar_tresh = 2, min_genes_per_module = 1 and rsq_thresh = 0.05). This resulted in 525 regulons, with a median of 24 genes and 28 regions per regulon.

**FigR analysis.** FigR was run as described at https://buenrostrolab.github.io/FigR/. Briefly, we used the pycisTopic cell–topic matrix (with 16 topics, as derived in the SCENIC+ workflow) to infer the $k$NN matrix for smoothing the data (with $k$ = 30). We kept peak–gene correlations with $P$ value < 0.05 and kept TF–domains of regulatory chromatin (DORC) associations with at least five significant peaks per gene for each DORC and abs(score) > 1. This resulted in 455 regulons, with a median of four genes per regulon.

**SCENIC analysis.** SCENIC was run as described at https://pyscenic.readthedocs.io/en/latest/ using default parameters and using the v.9 motif collection (SCENIC motif collection). This resulted in 253 regulons, with a median of 49 per regulon.

**Signac analysis.** Signac[45] (v.1.9.0) was run using LSI, using the top 30 PCs (excluding the first PC as recommend) for dimensionality reduction and clustering. DARs were determined using the FindMarkers() function, keeping regions with $P$ value < 0.005. Motif enrichment on DARs was performed using the JASPAR2022 motif collection (as recommended) using the FindMotifs() function from Signac (v.1.9.0), keeping motifs with adjusted $P$ value < 0.01. Regions with enriched motifs were used to create cistromes, based on the motif annotations. ChromVAR cistromes were derived using the RunChromVAR() function from the Signac package.

**ArchR analysis.** ArchR[44] (v.1.0.2) was run with default parameters, using iterative LSI, using the top 30 PCs for dimensionality reduction and clustering. Marker peaks were identified using the getMarkerFeatures(), using FDR ≤ 0.1 and logFC ≥ 0.5 as thresholds and the consensus peaks from pycisTopic as peak set, accounting for potential biases based on TSS enrichment and cell coverage. Motif enrichment on DARs was performed using the peakAnnoEnrichment, using the cisbp motif collection (default) and FDR ≤ 0.1 and logFC ≥ 0.5 as thresholds. We kept motifs with $\log_{10}$(adjusted $P$ value) > 2. Regions with enriched motifs were used to create cistromes based on the motif annotations. We assessed the enrichment of the tracks in the ENCODE TFBS annotation from ArchR using the same approach. Finally, we also ran ChromVAR using the wrapper functions included in ArchR, addBgdPeaks() and addDeviationsMatrix(), using the cisbp motif collection.

**Dimensionality reduction based on eRegulon enrichment scores obtained from different methods.** AUC values were calculated on target genes and target regions (if present) of all regulons obtained from SCENIC+, CellOracle, Pando, FigR and SCENIC using AUCell. This was performed separately for target genes/regions that have a positive/negative contribution (separate regulons were generated for repressors and activators). AUC scores were scaled using the StandardScaler function from Scikit-Learn (v.0.24.2) and $t$-SNE dimensionality reductions were generated using the TSNE function from Scikit-Learn (v.0.24.2) separately for target gene and target region (if present) and for the combined enrichment scores. Scaled AUC matrices were clustered using Leiden clustering with a resolution of 2.0. ARI were calculated by comparing this clustering with the cell-type labels, using the adjusted_rand_score function from Scikit-Learn (v.0.24.2).

**Comparison of predicted target regions with ChIP-seq data.** Accuracy of target regions of TFs inferred by SCENIC+, CellOracle, Pando, ArchR and Signac was assessed using ChIP-seq data. All TF ChIP-seq data on the cell lines BLaER1 (B-cell precursor), GM08714 (B lymphocyte), GM12878 (B lymphocyte), GM12891 (B lymphocyte), GM12892 (B lymphocyte), HL-60 (promyeloblast), K562 (hematopoietic cancer) and NB4 (blood cancer) were downloaded from ENCODE (https://www.encodeproject.org/)[29,30]. This resulted in a total of 527 ChIP-seq tracks and 333 unique TFs, 142 of which overlapped with the union of TFs identified by SCENIC+, CellOracle, Pando, FigR and SCENIC, and 259 of which overlapped with the union of TFs identified by SCENIC+, ArchR and Signac. All consensus peaks were scored for these ChIP-seq tracks using pyBigWig (v.0.3.18) by taking the maximum coverage across the consensus peak. Precision and recall were assessed by binarizing the ChIP-seq scores of all TFs on the consensus peaks. For SCENIC+, CellOracle and Pando this threshold was set dynamically by choosing the threshold that optimizes the F1 score for each TF and method. For Signac and ArchR this threshold was set dynamically by taking the 99th percentile on the ChIP-seq scores.

**Gene Ontology enrichment analysis on TFs identified by several methods.** Gene ontology enrichment analysis was run using Gprofiler assessing enrichment of human protein atlas and human phenotype gene ontology terms for TFs found exclusively by Signac, ArchR, Pando and CellOracle or the union of TFs found by Signac, ArchR and SCENIC+ and the union of TFs found by SCENIC+, Pando and CellOracle.

## Benchmark of GRN inference methods

**Simulated single-cell multiomics data.** Data from ENCODE deeply profiled cell lines were downloaded from https://www.encodeproject.org/ (refs. 29,30). Single-cell multiome data was simulated using bulk RNA-seq and ATAC-seq data from eight ENCODE deeply profiled cell lines, namely MCF7 (breast cancer, ENCFF136ANW and ENCFF772EFK, for RNA-seq and ATAC-seq, respectively), HepG2 (hepatocellular carcinoma, ENCFF660EXG and ENCFF239RGZ), PC3 (prostate cancer, ENCFF874CFD and ENCFF516GDK), GM12878 (B cell, ENCFF626GVO and ENCFF415FEC), K562 (leukemia, ENCFF833WFD and ENCFF512VEZ), Panc1 (pancreatic cancer, ENCFF602HCV and ENCFF836WDC), IMR90 (lung fibroblast, ENCFF027FUC and ENCFF848XMR) and HCT116 (colon cancer, ENCFF766TYC and ENCFF724QHH). Briefly, 500 single-cell multiomics profiles were simulated by randomly sampling 50,000 reads and 20,000 fragments from each bulk RNA-seq and ATAC-seq profiles, respectively, resulting in a dataset with 4,000 simulated single cells. The scRNA-seq count matrix was generated using featureCounts (Subread v.1.6.3) and the GRCh38.86 genome annotation. After calling peaks with MACS2 (ref. 81) (v.2.1.2.1) on the bulk ATAC-seq samples, we generated a set of 642,982 consensus peaks that was used to generate the scATAC-seq matrix, as previously described. Analyses can be explored in SCope at https://scope.aertslab.org/#/scenic-v2 and the UCSC at https://genome.ucsc.edu/s/cbravo/SCENIC%2B_DPCL.

**Methods.** The simulated dataset was analyzed with different state-of-the-art methods. For all methods, we used a search space of 150 kb for inferring region–gene relationships and kept regulons with at least 25 target genes:

- **SCENIC+:** pycisTopic was run with default parameters, using MALLET with 500 iterations for topic modeling and generating 21 topics (2 topics and from 5–100, increasing by 5), selecting the model with 40 topics based on the model selection metrics. SCENIC+ was run using both consensus peaks (642,982 peaks) and SCREEN regions[43]. SCENIC+ was run with default parameters, using http://oct2016.archive.ensembl.org/ as the BioMart host. High-quality regulons were selected based on the correlation between gene-based regulon AUC and region-based regulon AUC (>0.7). This resulted in 178 regulons, with a median of 437 genes and 774 regions per regulon.
- **CellOracle[31]:** CellOracle was run as described at https://morris-lab.github.io/CellOracle.documentation/. Briefly, scRNA-seq data were analyzed using Scanpy[94] (v.1.8.2), using flavor = 'cell_ranger' and $n$_top_genes = 3,000 to identify highly variable genes and seven PCs for dimensionality reduction. scATAC-seq data were processed with Cicero[99] (v.1.6.2) with window = 150,000, which inferred 13,123 region–gene connections with coaccess ≥ 0.8. Next, gimmemotifs[100] (v.0.17.1) was used to identify TFBSs using the gimmemotifs motif collection with default parameters. After inferring the links, we filtered links with the filter_links function using $P$ = 0.001, weight = 'coef_abs' and threshold_number = 2,000. This resulted in 157 regulons, with a median of 49 genes and 50 regions per regulon.
- **Pando[32]:** Pando was run as described at https://quadbiolab.github.io/Pando/. Briefly, the data were processed using Signac[45] (v.1.3.0) and SCREEN regions[43] were used in the initiate_grn function. Motif-MatchR (v.1.10.0) was used with motifs from Jaspar and CisBP. To infer the GRN (infer_grn), we used peak_to_gene_method = 'Signac', method = 'glm', upstream = 150,000, downstream = 150,000. This resulted in 235 regulons, with a median of 43 genes and 42 regions per regulon.
- **FigR[33]:** FigR was run as described at https://buenrostrolab.github.io/FigR/. Briefly, we used the pycisTopic cell–topic matrix (with 40 topics, as derived in the SCENIC+ workflow) to infer the $k$NN matrix

for smoothing the data. We kept peak–gene correlations with $P$ value < 0.05 and kept TF–DORC associations with abs(score) > 1, resulting in 10,757 TF–gene pairs. This resulted in 71 regulons, with a median of 39 genes per regulon.

- **GRaNIE[34]:** GRaNIE was run as described at https://grp-zaugg.embl-community.io/GRaNIE/, initially using the bulk data; however, its performance was very poor (likely to the reduced size of the dataset), only finding 26 TFs and 11,106 TF–region–gene links. Notably, when we applied it to our simulated single-cell dataset (with adaptations), its recovery increased (finding 39 TFs and 44,666 TF–region–gene links); hence, we report the latter results. Briefly, we used the binary matrix (with a pseudocount of 1) as input, normalizing the scATAC-seq data using 'Deseq_sizeFactor' and the scRNA-seq data with 'quantile'. Motif scanning was performed using the default motif collection (Hocomoco). In add-Connections_peak_gene we used promoterRange = 150,000. To filter the GRN, we used 0.2 as the FDR threshold. This resulted in 39 regulons, with a median of 176 genes and 68 regions per regulon.
- **SCENIC[11,12]:** SCENIC was run as described at https://pyscenic.readthedocs.io/en/latest/. To assess the effect of clustering in the motif collection, we also benchmarked SCENIC using the non-clustered and the clustered databases at the same time, obtaining similar results. For each collection, we generated two gene–motif rankings (10 kb around the TSS or 500 bp upstream the TSS). Using both collections at the same time, we found 108 regulons with a median of 322 genes.

**Cell-state recovery.** To assess whether the predicted GRNs for each of the tools correspond with the number of stable states in the biological data, AUC values were calculated based on predicted target genes for each TF using the AUCell method[11,12]. PCA was performed to assess whether the two first PCs recapitulated the different cell lines. To calculate the ARI, hierarchical clustering was performed on the two first PCs, making eight partitions based on the hierarchical tree using the cutree() function from the stats R package.

**TF recovery.** To assess whether the methods recover relevant TFs for the cell lines, TFs included in UniBind for these cell lines (https://unibind.uio.no/) were ranked based on the number of target regions in the database. Next, a cumulative recovery curve was generated for each method, where the value is the cumulative number of TFs found at each ranked position. AUCs were calculated using the first 40 positions of the ranking. Tau values for each TF were calculated using tispec (v.0.99.0) and plots using UpsetR (v.1.4.0). As a complementary approach, differential gene expression analysis was performed between the cell lines, keeping genes with adjusted $P$ value < 0.01 and logFC > 1.5. Next, TFs were ranked based on their maximum logFC across all cell types (in descending order) and we built a cumulative recovery curve for each method, where the value is the cumulative number of TFs found at each ranked position. AUCs were calculated using the first 40 positions of the ranking. Precision–recall curves were calculated using the precrec R package (v.0.12.9).

**TF–region.** To assess the quality of the TF–region relationships inferred by the methods, three different standards were used: (1) UniBind peaks[35,36] (optimal ChIP-seq peaks with TF motif); (2) ENCODE ChIP-seq coverage[29,30]; and (3) Enformer prediction scores[42]. For all, precision, recall and the F1 metric were calculated. For UniBind, predicted TF target regions from 309 ChIP-seq datasets performed on the deeply profiled cell lines used in this study were used (Supplementary Table 4). For the ChIP-seq and Enformer standard, all consensus peaks were scored using either the ChIP-seq data (using pyBigWig (v.0.3.18)) or the Enformer model. For both only TF ChIP-seq data or predicted data that were generated in one of the eight deeply profiled cell lines used in this study was used. For the ChIP-seq data this was a total of 539 ChIP-seq

tracks for 217 unique TFs. For the Enformer model this was a total of 334 predicted ChIP-seq tracks for 126 unique TFs. For both, the maximum coverage (either predicted or real) for each consensus peak was used as the score of that consensus peak. Precision and recall were assessed by binarizing the scores of all TFs on the consensus peaks. This threshold was set dynamically by choosing the threshold that optimizes the F1 score for each TF and method.

Enhancer activity was assessed on the predicted target regions using whole-genome STARR-seq data on K562, HepG2, HCT116 and MCF7. Data were downloaded from ENCODE[29,30] (ENCFF045TVA, ENCF-F047LDJ, ENCFF428KHI and ENCFF826BPU) and intersected with the consensus peaks and taking the maximum score, for each consensus peak, across the four cell lines. We assessed the distribution of scores for all predicted target regions.

FigR and SCENIC were excluded from the comparison as they do not report TF–region relationships.

**Region–gene.** To assess the quality of the region–gene relationships inferred by the methods, Hi-C data on five of the cell lines (namely IMR90 (ENCFF685BLG), GM12878 (ENCFF053VBX), HCT116 (ENCF-F750AOC), HepG2 (ENCFF020DPP) and K562 (ENCFF080DPJ)) from ENCODE ([https://www.encodeproject.org/](https://www.encodeproject.org/))[29,30] were used. Briefly, for each dataset, SCALE-normalized scores across bins of 5 kb were extracted using Juicer Tools[98] (v.2.13.05), keeping only links with score >10 and involving a bin that overlaps at least one of the consensus peaks and a TSS (±1,000 bp), resulting in 4,076,222 region–gene links on average. Finally, for each cell line, the correlations between the scores given by the methods and the Hi-C scores for the top 100 marker genes of that cell line were calculated. In this comparison, Pando and SCENIC were excluded. Pando reports a score per TF–region–gene triplet, and because generally several TFs can bind to the same region, this results in several scores for the same region–gene pair. SCENIC does not calculate region–gene relationships.

To assess the quality of region–gene relationships inferred by several regression models region–gene links were calculated using the following linear regression models implemented in Scikit-Learn (v.0.24.2): ridge regression, lasso regression, elastic net regression, least-angle regression, stochastic gradient descent and support vector machine using a linear kernel and the methods available in SCENIC+: Spearman rank correlation, gradient-boosting machine regression and RF regression for the top 100 differentially expressed genes for all cell lines. For all methods, the correlation between the feature importance scores/coefficients and the Hi-C scores was assessed.

**TF–gene.** To assess the quality of the predicted target genes for each TF, two different approaches were used. First, we tested the predictability capacity of the methods, in other words, how well the regulons predicted by the methods could predict the transcriptome. Briefly, for each method and for each gene a gradient-boosting machine regression model (Scikit-Learn v.0.24.2) was trained using as features the expression of the TFs that are predicted to regulate the gene, using 80% of the data. Next, these models were used to predict gene expression for the remaining 20% of the cells. As a quality metric, the correlation between the observed and predicted values was used. Second, we assessed whether gene expression changes upon TF knockdown coincide with predicted target genes of the different methods, using 157 TF perturbation datasets from ENCODE ([https://www.encodeproject.org/](https://www.encodeproject.org/))[29,30] (Supplementary Table 1). The logFC between control and perturbed samples were calculated with DESeq2 (ref. [101]) (v.1.28.1) and the effect of these perturbation on all regulons of the different methods was assessed by performing a GSEA, where genes are ranked based on the logFC compared to control data and predicted target genes for the TFs are used as gene sets. F1 score, precision and recall, across all TFs, were calculated using the threshold where the F1 score is maximized in each case. In addition, we also performed in silico TF perturbations

based on the eGRNs returned by the method (see Melanoma cell line analysis). Briefly, regression models per gene that were previously trained were used, but now the TF expression was set to zero to simulate a knockdown. The predicted expression matrix was used again to predict downstream changes over five iterations. Predicted logFC values were compared to DESeq2 logFC using Spearman rank correlation.

**Benchmark of motif and TF ChIP-seq-based databases.** TF ChIP-seq databases for cisTarget and DEM were generated using 1,660 TF ChIP-seq tracks from ENCODE[29,30]. Briefly, bigWig files were downloaded from [https://www.encodeproject.org/](https://www.encodeproject.org/) and the average signal on the target regions (in this case, consensus peaks) was calculated using bigWigAverageOverBed (Kent), resulting in a score-based database with regions as rows, TF ChIP-seq tracks as columns and average signal scores as values. Regions were ranked in decreasing order per track based on their score, resulting in TF ChIP-seq rank databases. The code used to generate these databases is available at [https://github.com/aertslab/create_cisTarget_databases](https://github.com/aertslab/create_cisTarget_databases). Each track was annotated to the TF that was targeted in the experiment. cisTarget and DEM were run as previously explained. SCENIC+ was run using cistromes derived from using only the motif-based databases, only the track-based databases or both. The derived eGRNs were compared using the benchmarks described above.

**Sensitivity analysis benchmark of SCENIC+.** To assess the sensitivity of SCENIC+, single-cell multiomics datasets were simulated using the ENCODE cell lines, with different numbers of cells and coverage. Specifically, we simulated a low-coverage dataset (with 3,000 scATAC-seq fragments per cell and 5,000 scRNA-seq reads per cell), a medium-coverage dataset (with 10,000 scATAC-seq fragments per cell and 20,000 scRNA-seq reads per cell) and a high-coverage dataset (with 20,000 scATAC-seq fragments per cell and 50,000 scRNA-seq reads per cell). From each of these datasets, four datasets with different numbers of cells, 80, 1,000, 10,000 and 25,000 cells, were derived. SCENIC+ was run in each dataset as previously described and the derived eGRNs from each analysis were compared using the benchmarks described above.

**ChIP-seq and Enformer-predicted ChIP-seq coverage for HNF4A, FOXA2 and CEBPB predicted target regions**
Experimental and Enformer-predicted ChIP-seq coverage of HNF4A (ENCFF080FZD), FOXA2 (ENCFF626IVY) and CEBPB (ENCFF003HJB) in HepG2 cells on the union of regions predicted to be targeted by corresponding TFs by the tools GRaNIE, Pando, CellOracle and SCENIC+ were visualized using heat maps with the color scaled to the fifth percentile region and 95th percentile region, based on Enformer scores on all consensus peaks, for both the ChIP-seq and the Enformer heat map. The regions were sorted based on the maximum Enformer score across each region.

**Melanoma cell line analysis**
**scATAC-seq (10x Genomics).** Two rounds of scATAC-seq (10x Genomics) were performed on a mix of MM050, MM099, MM116, MM001, MM011, MM057 and MM087 (run 1) and a mix of MM031, MM074, MM047 and MM029 (run 2).

*Cell culture.* Patient-derived melanoma cell lines used in this study were obtained from the laboratory of G.-E. Ghanem (Institut Jules Bordet). The identity of each line was determined using RNA-seq and ATAC-seq. Cell cultures used for experiments providing data to this study were tested for *Mycoplasma* contamination and were found to be negative. Cells were cultured in Ham's F10 nutrient mix (Thermo Fisher Scientific) supplemented with 10% fetal bovine serum (Thermo Fisher Scientific) and 100 µg ml⁻¹ penicillin–streptomycin. Cell cultures were kept at 37 °C and 5% $CO_2$. Before nuclei isolation, cells were washed with 1× PBS (Thermo Fisher Scientific), detached using trypsin

(Thermo Fisher Scientific) and centrifuged at 250*g* for 5 min to remove the medium.

*Nuclei isolation*. To isolate nuclei from the different melanoma cell lines, protocol CG000169 (10x Genomics) was followed. Briefly, cells were washed with PBS + 0.04% BSA and cell concentration was determined with the LUNA-FL Dual Fluorescence Cell Counter. For each cell line, 500,000 cells were resuspended in 100 µl nuclei lysis buffer (10 mM Tris-HCl, pH 7.4; 10 mM NaCl; 3 mM MgCl$_2$; 0.1% Tween-20; 0.1% NP40; 0.01% digitonin and 1% BSA in nuclease-free water). After 5 min incubation on ice, 1 ml chilled wash buffer was added to the lysed cells (10 mM Tris-HCl, pH 7.4; 10 mM NaCl; 3 mM MgCl$_2$; 0.1% Tween-20 and 1% BSA in nuclease-free water). The lysed cell suspension was centrifuged at 500*g* for 5 min at 4 °C and the pellet was resuspended in 1× nuclei buffer.

*Library preparation*. Single-cell libraries were generated using the GemCode Single-Cell Instrument and Single-Cell ATAC Library & Gel Bead Kit v.1-v.1.1 and ChIP kit (10x Genomics). Briefly, single nuclei suspended in 1× nuclei buffer were incubated for 60 min at 37 °C with a transposase that fragments the DNA in open regions of the chromatin and adds adaptor sequences to the ends of the DNA fragments. After generation of nanoliter-scale gel-bead-in-emulsions (GEMs), GEMs were incubated in a C1000 Touch Thermal Cycler (Bio-Rad) under the following program: 72 °C for 5 min; 98 °C for 30 s; 12 cycles of 98 °C for 10 s, 59 °C for 30 s, 72 °C for 1 min; and hold at 15 °C. After incubation, single-cell droplets were broken and the single-strand DNA was isolated and cleaned using Cleanup Mix containing Silane Dynabeads. Illumina P7 sequence and a sample index were added to the single-strand DNA during library construction via PCR: 98 °C for 45 s; 9–13 cycles of 98 °C for 20 s, 67 °C for 30 s, 72 °C for 20 s; 72 °C for 1 min; and hold at 15 °C. The sequencing-ready library was cleaned up with SPRIselect beads.

*Sequencing*. Before sequencing, the fragment size of every library was analyzed using the Bioanalyzer high-sensitivity chip. All 10x scATAC libraries of run 1 were sequenced on NextSeq2000 instrument (Illumina) with the following sequencing parameters: 51 bp for read 1, 8 bp for index 1, 24 bp for index 2 and 51 bp for read 2 and those of run 2 on a NovaSeq6000 instrument (Illumina) had the following sequencing parameters: 50bp for read 1, 8 bp for index 1, 16 bp for index 2 and 49 bp for read 1. For both scATAC-seq runs, reads were mapped to GRCh38 and fragments files generated using CellRanger-ATAC count command (v.2.0.0) using default parameters.

**Cell line annotation of scATAC-seq and scRNA-seq cells.** To determine the identity of the scRNA-seq cells, demuxlet was used as described in Wouters et al.[48]. To determine the identity of scATAC-seq cells, demuxlet was used using sample-specific mutations obtained from bulk ATAC-seq data on individual cell lines. Genotypes of the individual cell lines were called using the bcftools mpileup command (v.1.11; options, −max-depth 8,000 and −skip-indels) and bcftools call command (v.1.11; options, −multiallelic-caller, -variants-only, −skip-variants indels and −output-type b). Only variants that were single-nucleotide polymorphisms and not homozygous across samples were kept. To run demuxlet for the scATAC-seq, bam files were filtered to only contain reads covering single-nucleotide polymorphisms and having a cell barcode (based on CB tag); reads were piled up using the popscle dsc-pileup command using default parameters and demuxlet was run using the popscle demuxlet command (using the option −field GT).

**Quality control of scRNA-seq.** The scRNA-seq data of baseline MM-lines, as available on the Gene Expression Omnibus (GEO) under accession no. GSE134432, were preprocessed using Scanpy (v.1.8.2)[94]. Briefly, genes expressed in fewer than three cells were removed and cells that were expressed in fewer than 200 genes or more than 4,000 genes or had more than 15% counts in mitochondrial genes were removed. Doublets were detected and removed using Scrublet (v.0.2.3)[95] with default parameters. This resulted in 3,557 high-quality cells.

**Quality control of scATAC-seq and topic modeling.** The scATAC-seq data were preprocessed using pycisTopic (v.1.0.1.dev21+g8aa75d8). Briefly, cell line annotation was used to call consensus peaks (total of 360,302 peaks), as described above. Cells with fewer than 3.25 and 3.8 log number of unique fragments per cell, FRiP < 0.5 and TSS enrichment <4 and <5, respectively, for scATAC-seq run 1 and run 2 and doublets based on Scrublet (v.0.2.3)[95] calls with a threshold of 0.25 were removed. This resulted in 5,509 high-quality cells. Topic modeling was performed as described above using LDA with the collapsed Gibbs sampler. A model of 30 topics was selected based on the stabilization of the metrics described in refs. 85,86,96 and log-likelihood.

**Motif enrichment analysis.** Motif enrichment analysis was performed using pycisTarget (v.1.0.1.dev17+gd2571bf) as described above. For this, a custom score and ranking database was generated using create_cisTarget_databases Python package using the DNA sequences of consensus peaks and all annotated motifs as input. Motif enrichment was performed using both the cisTarget and DEM algorithm on cell line- and cell state (melanocytic, mesenchymal or intermediate)-based DARs (logFC > 1.5) and the top 3,000 regions per topic. The motif enrichment analysis was run both including promoters and excluding them. Promoters were defined as regions within 500 bp up- or downstream of the TSS of each gene. TSSs for each gene were downloaded from BioMart (http://www.ensembl.org), using the pyBioMart package (v.0.2.0).

**SCENIC+ analysis.** Pseudo-multiome data were generated for cell lines present in both scRNA-seq and scATAC-seq data, as described above using five cells per meta-cell for a total of 936 meta-cells. The SCENIC+ workflow was run using default parameters and as described above. Briefly, a search space of a maximum between either the boundary of the closest gene or 150 kb and a minimum of 1 kb upstream of the TSS or downstream of the end of the gene was considered for calculating region–gene relationships using gradient-boosting machine regression. TF–gene relationships were calculated using gradient-boosting machine regression between all TFs and all genes. Genes were considered as TFs if they were included in the TF list available on http://humantfs.ccbr.utoronto.ca/. (ref. 97). Final eRegulons were constructed using the GSEA approach in which region–gene relationships were binarized based on gradient-boosting machine regression importance scores using the 85th, 90th and 95th quantile; the top 5, 10 and 15 regions per gene and the BASC method[88] for binarization. Only eRegulons with a minimum of ten target genes were retained. For each eRegulon cellular enrichment scores (AUC) of target genes and regions were calculated using the AUCell algorithm[11]. eRegulons for which the correlation coefficient between semi-pseudobulked per cell type (100 meta-cells per cell type and 5 cells per meta-cell) TF expression and region AUC scores was >0.65 or <−0.75 were considered as high quality and used for downstream analysis. This resulted in 51 regulons, with a median of 176 genes and 183 regions per regulon. Analyses can be explored in SCope at https://scope.aertslab.org/#/scenic-v2 and UCSC Genome Browser at http://genome-euro.ucsc.edu/s/Seppe%20De%20Winter/scenicplus_mix_melanoma.

**ChIP-seq enrichment in eRegulon target regions.** MITF, SOX10 and TFAP2A ChIP-seq fastq files were downloaded from GEO using the respective accession nos. GSE61965 (MITF and SOX10) and GSE67555 (TFAP2A). Reads were mapped to Grch38 using Bowtie2 (v.2.4.4) using default parameters. Genomic coverage was calculated and stored as bigWig files using the bamCoverage function from deepTools (v.3.5.0; options -binSize 10, −effectiveGenomeSize 2913022398 and −normalizeUsing RPGC). ChIP-seq coverage was calculated on the target regions of MITF(+), SOX10(+) and TFAP2A(+) and the regions targeted by any combination of two eRegulons using the pyBigWig package (v.0.3.18) using 50 bins. Coverage was min−max normalized by using the minimum and maximum across all regions per ChIP-seq dataset.

**Perturbation simulation.** To simulate TF perturbations, first an RF regression model was trained to predict gene expression from TF expression, using the GradientBoostingRegressor fit function from the Scikit-Learn Python package (v.0.24.2) for each gene using the TFs predicted to regulate that gene as predictors for the gene. TFs that were predicted to regulate their own expression were excluded from the list of predictors before training the model. To simulate the effect of a TF knockout, a simulated gene expression matrix was generated by predicting the expression of each gene using the expression of the predictor TFs, while setting the expression of the TF of interest to zero. This simulation was repeated over several iterations, always using the newly simulated TF expression values as new predictor values for each gene. To visualize the effect of the perturbation in an embedding, first the shift of the cells in the original embedding (delta embedding) is estimated as described in Kamimoto et al.[31] based on eRegulon gene-based AUC values calculated using the simulated gene expression matrix. The delta embedding is used to draw arrows on the original embedding, using the streamplot or quiverplot function from the matplotlib (v.3.5.2) Python package. To prioritize TFs for their effect of driving melanocytic–mesenchymal transitions, or vice versa, average shifts along the first principal component, based on the delta embedding after five iterations of simulation, were calculated.

**SOX10 KD RNA-seq analysis.** Fastq files of bulk RNA-seq of MM001, MM011, MM031, MM057, MM074 and MM087 after SOX10 knockdown and non-targeting controls[48] were downloaded from GEO accession no. GSE134432. After trimming, sequencing adaptors from reads using fastq-mcf (v.1.05; default parameters) reads were mapped to GRCh38 using STAR (v.2.7.9a; options –alignIntronMax 1, –alignIntronMin 2), only reads with a mapping quality of minimum 4 were kept and a count matrix was generated using htseq-count (v.0.9.1; options -a 0, -m union and -t exon). The logFC values of SOX10 knockdown over non-targeting control were calculated using the rlogTransformation command from DESeq2 (ref. 101) (v.1.34.0).

**Comparison of target regions with enhancer activity assays.** Consensus regions were intersected with STARR-seq data, based on H3K27ac and ATAC-seq peaks, from Mauduit et al.[18]. Enhancer activity was compared in consensus regions in eRegulons versus consensus peaks not in eRegulons and the signal was correlated with the triplet ranking.

**CellOracle analysis.** CellOracle was run as described at https://morris-lab.github.io/CellOracle.documentation/. Briefly, scRNA-seq data were analyzed using Scanpy[94] (v.1.9.1), using flavor = 'cell_ranger' and $n$_top_genes = 2,000 to identify highly variable genes and nine PCs for dimensionality reduction. scATAC-seq data were processed with Cicero[99] (v.1.3.4.11) with window = 500,000, which inferred 21,001 region–gene connections with coaccess ≥ 0.8 and TSS annotation obtained from BioMart (http://may2015.archive.ensembl.org). Next, gimmemotifs[100] (v.0.17.1) was used to identify TFBSs using the gimmemotifs motif collection with default parameters. After inferring the links, we filtered links with the filter_links function using $P$ = 0.001, weight = 'coef_abs' and threshold_number = 2,000. This resulted in 97 regulons, with a median of 36 genes and 39 regions per regulon.

**Pando analysis.** Pando was run as described at https://quadbiolab.github.io/Pando/. Briefly, computationally paired data (see SCENIC+ analysis) were processed using Signac[45] (v.1.8.0). MotifMatchR (v.1.16.0) was used with motifs from Jaspar and CisBP. To infer the GRN the infer_grn function was used with default parameters (peak_to_gene_method = 'GREAT', upstream = $1 \times 10^5$, downstream = 0 and extend = $1 \times 10^6$). The GRN edges were filtered using the find_modules function with default parameters ($P$_thresh = 0.1, nvar_tresh = 2, min_genes_per_module = 1 and rsq_thresh = 0.05). This resulted in 887 regulons, with a median of 237 genes and 724 regions per regulon.

**FigR analysis.** FigR was run as described at https://buenrostrolab.github.io/FigR/. Briefly, computationally paired data (see SCENIC+ analysis) were preprocessed using Seurat[91] (v.4.3.0) and we used the cell-principal components matrix (with ten PCs) based on the scRNA-seq to infer the $k$NN matrix for smoothing the data (with $k$ = 30). We kept peak–gene correlations with $P$ value < 0.05 and kept TF–DORC associations with at least five significant peaks per gene for each DORC and abs(score) > 1. This resulted in 514 regulons, with a median of three genes per regulon.

**SCENIC analysis.** SCENIC was run as described at https://pyscenic.readthedocs.io/en/latest/ using default parameters and using the v.9 motif collection (SCENIC motif collection).

**GRaNIE analysis.** GRaNIE was run as described at https://grp-zaugg.embl-community.io/GRaNIE/. Briefly, computationally paired data (see SCENIC+ analysis) were used as input with a pseudocount of 1 added to the scATAC-seq matrix. The scATAC-seq data were normalized using the 'Deseq_sizeFactor' and the scRNA-seq data with 'quantile'. Motif scanning was performed using the default motif collection (HOCOMOCO) and default settings were used to infer the GRN. To filter the GRN, we used default settings. This resulted in 45 regulons, with a median of five genes and one region per regulon.

**Dimensionality reduction based on eRegulon enrichment scores obtained from different methods.** AUC values were calculated on target genes and target regions (if present) of all regulons obtained from SCENIC+, CellOracle, Pando, FigR, GRaNIE and SCENIC using AUCell. This was conducted separately for target genes/regions that have a positive/negative contribution (separate regulons were generated for repressors and activators). AUC scores were scaled using the StandardScaler function from Scikit-Learn (v.0.24.2) and $t$-SNE dimensionality reductions were generated using the TSNE function from Scikit-Learn (v.0.24.2) separately for target gene and target region (if present) enrichment scores and for the combined enrichment scores. Scaled AUC matrices were clustered using Leiden clustering with a resolution of 2.0. ARIs were calculated by comparing this clustering with the cell-type labels, using the adjusted_rand_score function from Scikit-Learn (v.0.24.2).

**BoolODE analysis.** Boolean networks were generated and formatted as described in the documentation of BoolODE (https://murali-group.github.io/Beeline/BoolODE.html) for all TF–TF edges inferred by GRaNIE, SCENIC and CellOracle and for the top 10%, 25% and 50% of edges, based on the triplet score, of SCENIC+. The number of TFs targeting a single gene was randomly down sampled to 15 if the gene was targeted by more than 15 TFs. BoolODE was run separately for all Boolean networks simulating 500 cells with the model type hill, a simulation time of 20, with kinetic parameters set to one sampled value using a standard deviation of 0.5. Simulated gene expression matrices from the final BoolODE iteration were scaled using the StandardScaler function and PCA was run using the PCA function from Scikit-Learn (v.0.24.2) using 50 principal components. The PC matrices of the simulated and real data were integrated using harmonypy, using default settings. The distance between each simulated cell and its five nearest neighbors was quantified in the harmony-corrected and integrated data using the NearestNeighbors function from Scikit-Learn (v.0.24.2). The first two PCs of the integrated data were used for visualization.

**Comparative analysis in the mammalian brain using SCENIC+ Human cortex data.** Human motor cortex data were downloaded from https://data.nemoarchive.org/publication_release/Lein_2020_M1_study_analysis/Multimodal/sncell/SNARE/human/processed/counts/counts/M1/ (scRNA-seq count matrix and cell metadata) and

https://data.nemoarchive.org/biccn/grant/u01_zhangk/zhang/multimodal/sncell/SNARE_ATACseq/human/processed/align/BICCN-H_20190523_SNARE2-AC_snapTools_190808/ (scATAC-seq fragment files).

**Mouse cortex data.** *Mouse cortex dissection*. All animal experiments were conducted according to the KU Leuven ethical guidelines and approved by the KU Leuven Ethical Committee for Animal Experimentation (approved protocol no. ECD P007/2021). Mice were maintained in a specific-pathogen-free facility under standard housing conditions (temperature 20–24 °C and humidity 45–65%) with continuous access to food and water. Mice used in the study were 57 d old and were maintained on 14 h light, 10 h dark–light cycle from 7:00 to 21:00. In this study, cortical brain tissue from male P57 BL/6Jax was used. Animals were anesthetized with isoflurane and decapitated. Cortices were collected and immediately snap-frozen in liquid nitrogen.

*Sample preparation*. Five multiome experiments were performed, with small variations in sample preparation. For the sample labeled as '10x_complex' we used a modified protocol from the Nuclei Isolation from Complex Tissues for Single-Cell Multiome ATAC + Gene Expression Sequencing Protocol (CG000375) from 10x Genomics. Briefly, a ~1-cm$^3$ frozen piece of mouse cortex tissue was transferred to 0.5 ml ice-cold homogenization buffer (salt-Tris solution of 10 mM NaCl, 10 mM Tris, pH 7.4, 3 mM MgCl$_2$, 0.1% IGEPAL CA-63, 1 mM dithiothreitol and 1 U μl$^{-1}$ Protector RNase inhibitor (Sigma)) in a Dounce homogenizer mortar and thawed for 2 min. The tissue was homogenized with ten strokes of pestle A and ten strokes of pestle B until a homogeneous nuclei suspension was achieved. The resulting homogenate was filtered through a 70-μm cell strainer (Corning). The homogenizer and the filter were rinsed with an additional 0.5 μl homogenization buffer. The tissue material was pelleted at 500g and the supernatant was discarded. The tissue pellet was resuspended in wash buffer (1% BSA in PBS + 1 U μl$^{-1}$ Protector RNase inhibitor (Sigma)). The obtained nuclei were stained with 7AAD (Thermo Fisher Scientific) and viability sorted on a BD FACS Fusion into 5-ml low-bind Eppendorf tubes containing BSA with RNase inhibitor. The sorted nuclei were pelleted at 500g and the supernatant was discarded. Next, the nuclei were permeabilized by resuspending the pellet in 0.1× lysis buffer (salt-Tris solution of 10 mM NaCl, 10 mM Tris, pH 7.4, 3 mM MgCl$_2$, 0.1% IGEPAL CA-63, 0.01% digitonin, 1% BSA, 1 mM dithiothreitol and 1 U μl$^{-1}$ Protector RNase inhibitor (Sigma)) and incubated on ice for 2 min. Then, 1 ml wash buffer (salt-Tris solution of 10 mM NaCl, 10 mM Tris, pH 7.4, 3 mM MgCl$_2$, 0.1% Tween-20, 1% BSA, 1 mM dithiothreitol and 1 U μl$^{-1}$ Protector RNase inhibitor (Sigma)) was added and the nuclei were pelleted at 500g and the supernatant was discarded. The nuclei pellet was resuspended in diluted nuclei buffer (1× Nuclei buffer Multiome kit (10x Genomics)), 1 mM dithiothreitol and 1 U μl$^{-1}$ Protector RNase inhibitor (Sigma)).

For the sample labeled as '10x_no_perm' we used the above-described modified protocol from the Nuclei Isolation from Complex Tissues for Single-Cell Multiome ATAC + Gene Expression Sequencing Protocol (CG000375) from 10x Genomics, omitting the nuclei permeabilization step. After sorting the nuclei were pelleted at 500g and the supernatant was discarded. The nuclei pellet was resuspended in diluted nuclei buffer (1× Nuclei buffer Multiome kit (10x Genomics)), 1 mM dithiothreitol and 1 U μl$^{-1}$ Protector RNase inhibitor (Sigma)).

For the sample labeled as 'TST' nuclei isolation we used a modified protocol from De Rop et al.[102]. Briefly, a ~1-cm$^3$ frozen piece of mouse cortex tissue was transferred to 0.5 ml ice-cold homogenization buffer (salt-Tris solution of 146 mM NaCl, 10 mM, Tris pH 7.5, 1 mM CaCl$_2$, 21 mM MgCl$_2$, 250 mM sucrose, 0.03% Tween-20, 1% BSA, 25 mM KCl, 1× cOmplete, Mini, EDTA-free Protease Inhibitor Cocktail (Roche), 1 mM dithiothreitol and 1 U μl$^{-1}$ Protector RNase inhibitor (Sigma)) in a Dounce homogenizer mortar and thawed for 2 min. The tissue was

homogenized with ten strokes of pestle A and ten strokes of pestle B until a homogeneous nuclei suspension was achieved. The resulting homogenate was filtered through a 70-μm cell strainer (Corning). The homogenizer and the filter were rinsed with an additional 0.5 ml homogenization buffer. The tissue material was pelleted at 500g and the supernatant was discarded. The tissue pellet was resuspended in a homogenization buffer without Tween-20. An additional 1.65 ml homogenization buffer was topped up and mixed with 2.65 ml gradient medium (75 mM sucrose, 1 mM CaCl$_2$, 50% Optiprep (Stemcell), 5 mM MgCl$_2$, 10 mM Tris, pH 7.5, 0.5× cOmplete, Mini, EDTA-free Protease Inhibitor Cocktail (Roche), 1 mM dithiothreitol and 1 U μl$^{-1}$ of Protector RNase inhibitor (Sigma)). Then, 4 ml 29% iodoxanol cushion was prepared with a diluent medium (250 mM sucrose, 150 mM KCl, 30 mM MgCl$_2$ and 60 mM, Tris pH 8) and was loaded into a 13.2-ml ultracentrifuge tube. The 5.3 ml sample in homogenization buffer + gradient medium was gently layered on top of the 29% iodoxanol cushion. The sample was centrifuged at 10,139.3g at 4 °C for 30 min, and the supernatant was gently removed without disturbing the nuclei pellet. The nuclei pellet was resuspended in diluted nuclei buffer (1× Nuclei buffer Multiome kit (10x Genomics)), 1 mM dithiothreitol, 1 U μl$^{-1}$ Protector RNase inhibitor (Sigma)).

For the sample labeled as '10x_complex_UC' we designed a protocol where we combined the '10x_complex' with 'TST'. Briefly, a ~1-cm$^3$ frozen piece of mouse cortex tissue was transferred to 0.35 ml ice-cold homogenization buffer (salt-Tris solution of 10 mM NaCl, 10 mM Tris, pH 7.5, 3 mM MgCl$_2$, 0.1% IGEPAL CA-63, 1 mM dithiothreitol and 1 U μl$^{-1}$ of Protector RNase inhibitor (Sigma)) in a Dounce homogenizer mortar and thawed for 2 min. The tissue was homogenized with ten strokes of pestle A and ten strokes of pestle B until a homogeneous nuclei suspension was achieved. The resulting homogenate was filtered through a 70-μm cell strainer (Corning). The homogenizer and the filter were rinsed with an additional 0.65 ml homogenization buffer. The homogenate was incubated on ice for 5 min, pelleted at 500g and the supernatant was discarded. Then, 1 ml permeabilization buffer (PBS 1×, BSA 1%, 0.1% IGEPAL CA-63, 0.01% digitonin and 1 U μl$^{-1}$ Protector RNase inhibitor (Sigma)) was added and incubated on ice for 2 min. Next, the pellet was resuspended, incubated on ice for extra 5 min and pelleted at 500g. The pelleted nuclei were resuspended in 1 ml wash buffer (PBS 1×, BSA 1% and 0.5 U μl$^{-1}$ Protector RNase inhibitor (Sigma)). An additional 1.65 ml wash buffer was topped up and mixed with 2.65 ml gradient medium (75 mM sucrose, 1 mM CaCl$_2$, 50% Optiprep (Stemcell), 5 mM MgCl$_2$, 10 mM Tris, pH 7.5 and 1 mM dithiothreitol). Then, 4 ml 29% iodoxanol cushion was prepared with a diluent medium (250 mM sucrose, 150 mM KCl, 30 mM MgCl$_2$ and 60 mM, Tris pH 8) and was loaded into a 13.2-ml ultracentrifuge tube. Then, 5.3 ml sample in wash buffer + gradient medium was gently layered on top of the 29% iodoxanol cushion. The sample was centrifuged at 10,139.3g, 4 °C for 30 min and the supernatant was gently removed without disturbing the nuclei pellet. The nuclei pellet was resuspended in diluted nuclei buffer (1× Nuclei buffer Multiome kit (10x Genomics)), 1 mM dithiothreitol and 1 U μl$^{-1}$ Protector RNase inhibitor (Sigma)).

For the sample labeled as 'TST_NP40_004' we designed a protocol starting from 'TST'. Briefly, a ~1-cm$^3$ frozen piece of mouse cortex tissue was transferred to 0.5 ml ice-cold homogenization buffer (salt-Tris solution of 146 mM NaCl, 10 mM Tris, pH 7.5, 1 mM CaCl$_2$, 21 mM MgCl$_2$, 250 mM sucrose, 0.03% Tween-20, 0.01% BSA, 25 mM KCl, 1× cOmplete, Mini, EDTA-free Protease Inhibitor Cocktail (Roche), 1 mM dithiothreitol and 1 U μl$^{-1}$ Protector RNase inhibitor (Sigma)) in a Dounce homogenizer mortar and thawed for 2 min. The tissue was homogenized with ten strokes of pestle A and ten strokes of pestle B until a homogeneous nuclei suspension was achieved. The resulting homogenate was filtered through a 70-μm cell strainer (Corning). The homogenizer and the filter were rinsed with an additional 500 ml homogenization buffer. The homogenate was pelleted at 500g and the supernatant was discarded. Then, 0.3 ml permeabilization buffer (salt-Tris solution of 10 mM NaCl,

10 mM Tris, pH 7.5, 3 mM MgCl$_2$, 1%, BSA, 25 mM KCl, 250 mM sucrose, 0.04 % IGEPAL CA-63, 0.01% digitonin, 1 mM dithiothreitol, 1× cOmplete, Mini, EDTA-free Protease Inhibitor Cocktail (Roche) and 0.5 U μl$^{-1}$ Protector RNase inhibitor (Sigma)) was added and incubated on ice for 5 min. The homogenate was pelleted at 500$g$ and the supernatant was discarded. The tissue pellet was resuspended in 1 ml wash buffer (salt-Tris solution of 10 mM NaCl, 10 mM Tris, pH 7.5, 3 mM MgCl$_2$, 1%, BSA, 25 mM KCl, 250 mM sucrose, 1 mM dithiothreitol, 1× cOmplete, Mini, EDTA-free Protease Inhibitor Cocktail (Roche) and 0.5 U μl$^{-1}$ Protector RNase inhibitor (Sigma)). An additional 1.65 ml wash buffer was topped up and mixed with 2.65 ml gradient medium (75 mM sucrose, 1 mM CaCl$_2$, 50% Optiprep (Stemcell), 5 mM MgCl$_2$, 10 mM Tris, pH 7.5 and 1 mM dithiothreitol). Then, 4 ml 29% iodoxanol cushion was prepared with a diluent medium (250 mM sucrose, 150 mM KCl, 30 mM MgCl$_2$ and 60 mM Tris, pH 8) and was loaded into a 13.2-ml ultracentrifuge tube. Then, 5.3 ml sample in wash buffer + gradient medium was gently layered on top of the 29% iodoxanol cushion. The sample was centrifuged at 10,139.3$g$, 4 °C for 30 min, and the supernatant was gently removed without disturbing the nuclei pellet. The nuclei pellet was resuspended in diluted nuclei buffer (1× Nuclei buffer Multiome kit (10x Genomics)), 1 mM dithiothreitol and 1 U μl$^{-1}$ Protector RNase inhibitor (Sigma)).

*Nuclei suspension quality control*. Nuclei yield, morphology and presence of clumps/debris were evaluated by mixing 9 μl sample with 1 μl arginine orange/propidium iodide stain, loaded onto a LUNA-FL slide and visualized with the LUNA-FL Automated cell counter (Logos Biosystems).

*Library preparation*. Single-nuclei libraries were generated using the 10x Chromium Single-Cell Instrument and NextGEM Single-Cell Multiome ATAC + Gene Expression kit (10x Genomics) according to the manufacturer's protocol. Briefly, the single mouse brain nuclei were incubated for 60 min at 37 °C with a transposase that fragments the DNA in open regions of the chromatin and adds adaptor sequences to the ends of the DNA fragments. After generation of nanoliter-scale GEMs, GEMs were incubated in a C1000 Touch Thermal Cycler (Bio-Rad) under the following program: 37 °C for 45 min, 25 °C for 30 min and hold at 4 °C. Incubation of the GEMs produced 10x barcoded DNA from the transposed DNA (for ATAC) and 10x barcoded, full-length cDNA from poly-adenylated mRNA (for GEX). Next quenching reagent (Multiome 10x kit) was used to stop the reaction. After quenching, single-cell droplets were dissolved and the transposed DNA and full-length complementary DNA were isolated using Cleanup Mix containing Silane Dynabeads. To fill gaps and generate sufficient mass for library construction, the transposed DNA and cDNA were amplified via PCR at 72 °C for 5 min; 98 °C for 3 min; seven cycles of 98 °C for 20 s, 63 °C for 30 s, 72 °C for 1 min; 72 °C for 1 min; and hold at 4 °C. The pre-amplified product was used as input for both ATAC library construction and cDNA amplification for gene expression library construction. Illumina P7 sequence and a sample index were added to the single-strand DNA during ATAC library construction via PCR at 98 °C for 45 s; 7–9 cycles of 98 °C for 20 s, 67 °C for 30 s, 72 °C for 20 s; 72 °C for 1 min; and hold at 4 °C. The sequencing-ready ATAC library was cleaned up with SPRIselect beads (Beckman Coulter). Barcoded, full-length pre-amplified cDNA was further amplified via PCR at 98 °C for 3 min; 6–9 cycles of 98 °C for 15 s, 63 °C for 20 s, 72 °C for 1 min; 72 °C for 1 min; and hold at 4 °C. Subsequently, the amplified cDNA was fragmented, end-repaired, A-tailed and index adaptor-ligated, with SPRIselect cleanup in between steps. The final gene expression library was amplified by PCR at 98 °C for 45 s; 5–16 cycles of 98 °C for 20 s, 54 °C for 30 s, 72 °C for 20 s. 72 °C for 1 min; and hold at 4 °C. The sequencing-ready GEX library was cleaned up with SPRIselect beads.

*Sequencing*. Before sequencing, the fragment size of every library was analyzed using the Bioanalyzer high-sensitivity chip. All 10x Multiome ATAC libraries were sequenced on NovaSeq6000 instruments (Illumina) with the following sequencing parameters: 50 bp for read

1, 8 bp for index 1 (i7), 16 bp for index 2 (i5) and 49 bp for read 2. All 10x Multiome GEX libraries were sequenced on NovaSeq6000 instruments with the following sequencing parameters: 28 bp for read 1, 10 bp for index 1 (i7), 10 bp for index 2 (i5) and 75 bp for read 2. The generated fastq files were processed with CellRanger-ARC (v.2.0.0) count function, with include introns = true option. Reads were aligned to *Mus musculus* reference genome (ATAC-CellRanger-ARC-mm10-2020-A-2.0.0).

**Human cortex data analysis.** High-quality cells (84,159) selected by Bakken et al.[60] were used for the analysis. scRNA-seq data were analyzed using Seurat[91] (v.4.0.3), using 47 PCs for dimensionality reduction and Leiden clustering (with resolution of 0.6). This resulted in 30 clusters (corresponding to 19 cell types) that were manually annotated based on marker gene expression. These cell-type labels were used to create pseudobulks from which peaks were called with MACS2 (ref. [81]) (v.2.1.2.1) and consensus peaks were derived using the iterative-filtering approach (as previously described), resulting in 697,721 regions. Topic modeling was performed using MALLET, using 500 iterations and models with 10 topics and from 25 to 500 by an increase of 25, selecting the model with 50 topics based on the model selection metrics. pycisTarget (v.1.0.1.dev17+gd2571bf) was run using a custom database with the consensus regions, on DARs and binarized topics (with Otsu thresholding), with and without promoters and using pycisTarget and DEM. SCENIC+ was run with default parameters, using http://jul2018.archive.ensembl.org/ as a BioMart host. High-quality regulons were selected based on the correlation between gene-based regulon AUCs and region-based regulon AUCs (>0.4) and on the number of target genes (>30). This resulted in 142 regulons, with a mean of 315 genes and 768 regions per regulon.

**Mouse cortex data analysis.** scRNA-seq data were first analyzed using VSN (v.0.27.0). Briefly, cells with at least 100 genes expressed and less than 1% of mitochondrial reads were kept. Doublets were removed using Scrublet[95] (v.0.2.3), with default parameters. Fifty PCs were used as input for harmony, which was used to correct batch effects due to the sample preparation protocol and the corrected PCs were used for dimensionality reduction and Leiden clustering (resolution 1). This resulted in 41 clusters that were annotated based on marker gene expression. Cell types not belonging to the cortex (such as medium spiny neurons from the striatum) were removed and the dataset was reanalyzed using Seurat[91] (v.4.0.3), using 52 PCs as input for harmony, as previously described, which were used for dimensionality reduction. This resulted in a dataset with 21,969 high-quality cells (based on scRNA-seq). These cell-type labels were used to create pseudobulks from which peaks were called with MACS2 (ref. [81]) (v.2.1.2.1) and consensus peaks were derived using the iterative-filtering approach (as previously described), resulting in 568,403 regions. We further filtered the dataset based on the scATAC-seq quality, keeping cells with at least 1,000 fragments, FRiP > 0.4 and TSS > 4, resulting in 19,485 cells. Topic modeling was performed using MALLET, using 500 iterations and models with 2 topics and from 5–100 by an increase of 5, selecting the model with 60 topics based on the model selection metrics. pycisTarget (v.1.0.1.dev17+gd2571bf) was run using a custom database with the consensus regions on DARs and binarized topics (with Otsu thresholding), with and without promoters and using pycisTarget and DEM. SCENIC+ was run with default parameters, using http://nov2020.archive.ensembl.org/ as a BioMart host. High-quality regulons were selected based on the correlation between gene-based regulon AUCs and region-based regulon AUCs (>0.4) and on the number of target genes (>30). This resulted in 125 regulons, with a mean of 295 genes and 694 regions per regulon.

**Cross-species comparison.** Mouse gene names were converted to their orthologous human gene names based on the orthology table at http://www.informatics.jax.org/downloads/reports/HOM_MouseHumanSequence.rpt. Gene-based human and mouse-to-human regulons were

intersected, calculating the percentage of agreement as the number of overlapping genes divided by the size of the smallest regulon. Human region-based regulons were lifted over to mm10 using UCSC Liftover (https://genome.ucsc.edu/cgi-bin/hgLiftOver) with default parameters and regions overlapping mouse cortex consensus peaks and linked to the same (orthologous) gene in the two species were kept. Region-based human-to-mouse and mouse regulons were intersected, calculating the percentage of agreement as the number of overlapping regions divided by the size of the smallest regulon. Human bigWig files were lifted over to the mm10 genome using CrossMap (v.0.6.0). Analyses can be explored in SCope at https://scope.aertslab.org/#/scenic-v2 and UCSC Genome Browser at https://genome-euro.ucsc.edu/s/cbravo/SCENIC%2B_Cortex.

## GRN velocity

**GRN velocity calculation.** First, cells were ordered by pseudotime. For each TF ($TF$), a standardized generalized additive model (GAM) was fitted along the pseudotime axis for its expression and its target genes' (or regions, $TG$ and $TR$, respectively) AUC values, using the LinearGAM() function from pyGAM (v.0.8.0) with a spline term and automatic parameter grid search.

$$g(TF) = \beta_0 + s(\text{pseudotime})$$

$$g(TG) = \beta_0 + s(\text{pseudotime})$$

$$g(TR) = \beta_0 + s(\text{pseudotime})$$

Next, each cell ($C$) in a certain quantile of the GAM TF expression model was mapped to a future cell in the same quantile of the GAM regulon AUC curve (posterior in the pseudotime axis). By default, the curve fitted on the target genes AUC values ($g(TG)$) was used. If there was no posterior cell in that quantile the cell was mapped to itself. Only positive interactions were considered in this analysis.

$$C^{TF}_{g(TF)} \leftrightarrow C^{TG}_{g(TG)} \text{ where } g(TF) = g(TG)$$

$$\text{if pseudotime}\left(C^{TF}_{g(TF)}\right) < \text{pseudotime}\left(C^{TG}_{g(TG)}\right) \text{else } C^{TF}_{g(TF)} \leftrightarrow C^{TF}_{g(TF)}$$

We define the differentiation force of a cell as the distance from the TF expression curve to its matching cell in the regulon AUC curve.

$$GRN_{\text{force}} = \text{pseudotime}\left(C^{TG}_{g(TG)}\right) - \text{pseudotime}\left(C^{TF}_{g(TF)}\right)$$

Differentiation forces can be plotted as an arrow grid in any dimensionality reduction of the data and prioritized per group of cells to identify key drivers in differentiation transitions. To visualize a regulon differentiation force, the distance in the embedding between matching cells is calculated (delta embedding).

$$\vec{\Delta} = C^{TG}_{g(TG)_{(x,y)}} - C^{TF}_{g(TF)_{(x,y)}} \text{ where } g(TF) = g(TG)$$

$$\text{if pseudotime}\left(C^{TF}_{g(TF)}\right) < \text{pseudotime}\left(C^{TG}_{g(TG)}\right) \text{ else } \vec{\Delta} = (0, 0)$$

The delta embedding is used to draw arrows between the cells, using the streamplot or quiverplot function from the matplotlib (v.3.5.2) Python package.

When having multiple differentiation paths, the same strategy is applied separately in each path ($p$) and then an average across all the paths is taken ($P$). In addition, as the data are standardized before fitting the GAM model, a penalization curve standardized on the whole dataset is used ($TF_{\text{global}}$). This will prevent false arrows from being drawn if a TF

(or its target) is not present in the branch. By default, the penalization threshold ($t_p$) is 0.03.

$$\vec{\Delta} = \sum_p^P C^{TG|p}_{g(TG|p)_{(x,y)}} - C^{TF}_{g(TF)_{(x,y)}}$$

$$\text{if pseudotime}\left(C^{TF}_{g(TF)}|p\right) < \text{pseudotime}\left(C^{TG|p}_{g(TG|p)}\right)$$

$$\text{if } |g(TF_p)_{C^{TF}_{g(TF)}} - g(TF_{\text{global}})_{C^{TF}_{g(TF)}}| < t_p$$

**GRN velocity along oligodendrocyte differentiation in the mouse cortex.** Oligodendrocyte cells (OPCs and oligodendrocytes) were subsetted from our in-house mouse cortex dataset, resulting in 4,435 cells. The eRegulon AUC matrix was processed using Scanpy[94] (v.1.8.2) and the embedding-based pseudotime was derived using the diffmap and dpt functions. Differentiation arrows were inferred for cells above the 70% quantile of TF expression and default parameters. Prioritization of differentiation forces was conducted using the regulon specificity score (RSS) metric using arrow lengths in each cell for each regulon as input values.

**GRN velocity along the fly retina differentiation.** *Data*. scRNA-seq and scATAC-seq from the third instar larvae eye-antennal disc from Bravo et al.[14] were used. In addition, we performed an additional scATAC-seq run using only eye discs (by cutting the antenna out of the eye-antennal disc), using the same protocol as described by Bravo et al.[14]. The analysis can be explored in SCope at https://scope.aertslab.org/#/scenic-v2 and UCSC Genome Browser at http://genome.ucsc.edu/s/cbravo/SCENIC%2B_EAD.

*SCENIC+ analysis*. scATAC-seq annotated cells by Bravo et al.[14] were used to create pseudobulks from which peaks were called with MACS2 (ref. 81) (v.2.1.2.1) and consensus peaks were derived using the iterative-filtering approach (as previously described), resulting in 39,732 regions. Additional *Drosophila* cisTarget regions that did not overlap with these peaks were used, resulting in a dataset with 127,711 regions. cisTarget regions are defined by partitioning the entire noncoding *Drosophila* genome based on cross-species conservation, resulting in more than 136,000 bins with an average size of 790 bp, and we found that using this region set increases the resolution for rare cell types, in which peak calling is difficult due to low amounts of cells.

Combining high-quality cells from all the runs (based on Cell-Ranger), a dataset with 23,317 cells was obtained. Topic modeling was performed using MALLET, using 500 iterations and models with 2 topics and from 5–100 by an increase of 5, selecting the model with 80 topics based on the model selection metrics. pycisTarget (v.1.0.1.dev17+gd2571bf) was run using a custom database with the consensus regions on DARs and binarized topics (with Otsu thresholding), with and without promoters and using pycisTarget and DEM. Next, scATAC-seq and scRNA-seq data were mapped into a virtual template of the eye-antennal disc using ScoMAP (v.0.1.0), as described by Bravo et al.[14]. This resulted in a dataset with 5,058 multiome pseudocells, for which both scRNA-seq and scATAC-seq measurements are available. These data were used as input for SCENIC+. SCENIC+ was run with default parameters, using http://dec2017.archive.ensembl.org as BioMart host and a 50 kb window for the inference of region–gene links (instead of 150 kb). High-quality regulons were selected based on the correlation between gene-based regulon AUCs and region-based regulon AUCs (>0.4) and on the number of target genes (>10). This resulted in 153 regulons, with a mean of 216 genes and 323 regions per regulon.

*GRN velocity*. Differentiating cells in the eye disc were subsetted, resulting in a dataset with 3,104 cells. The eRegulon AUC matrix was processed using Scanpy[94] (v.1.8.2). Cell annotations were refined based on Leiden clustering on the eRegulon AUC matrix, resulting in nine clusters along the fly retina differentiation. To identify branching

points, PAGA (included in Scanpy) was used, using 0.24 as threshold for paga_compare. The UMAP representation was recalculated using init_pos = 'paga'. The embedding-based pseudotime was derived using the diffmap and dpt functions. Differentiation arrows were inferred for cells above the 70% quantile of TF expression; a value of 0.2 was used as branch penalization and default parameters along the two differentiation paths (from progenitors to photoreceptors and from progenitors to interommatidial cells). Prioritization of differentiation forces was conducted using the RSS metric using arrow lengths in each cell for each regulon as input values. Prioritization of differentiation forces was conducted using the RSS metric using arrow lengths in each cell for each regulon as input values.

*Comparison with scVelo and MultiVelo.* Spliced and unspliced scRNA-seq count matrices were generated using featureCounts, counting only reads in exons or in the whole gene body, respectively. Velocity was calculated using scVelo (v.0.2.5) in the virtual template cells, using velocity with mode = 'stochastic'. scVelo pseudotime was calculated using velocity_pseudotime with default parameters. MultiVelo was run with default parameters, using the imputed scATAC-seq data, the spliced and unspliced scRNA-seq counts of the cells paired in the virtual template and the region–gene links derived by SCENIC+, using all genes.

### Spatial GRN mapping

**Human cerebellum 10x Visium.** *Data.* 10x Visium data from the human cerebellum were downloaded from https://www.10xgenomics.com/resources/datasets/human-cerebellum-whole-transcriptome-analysis-1-standard-1-2-0. The 10x single-cell multiome data from the human cerebellum were downloaded from https://www.10xgenomics.com/resources/datasets/frozen-human-healthy-brain-tissue-3-k-1-standard-1-0-0. The analysis can be explored in SCope at https://scope.aertslab.org/#/scenic-v2 and UCSC Genome Browser at https://genome-euro.ucsc.edu/s/cbravo/SCENIC%2B_cerebellum.

*SCENIC+ analysis and regulon mapping.* Cells with at least 500 scRNA-seq reads and less than 5% of mitochondrial reads were kept and doublets were removed using Scrublet[95] (v.0.2.3) with default parameters. scRNA-seq data were analyzed using Seurat[91] (v.4.0.3), using 37 PCs for dimensionality reduction and Leiden clustering (with resolution 0.6). This resulted in 15 clusters (corresponding to 13 cell types) that were manually annotated based on marker gene expression. These cell-type labels were used to create pseudobulks from which peaks were called with MACS2 (ref. 81) (v.2.1.2.1) and consensus peaks were derived using the iterative-filtering approach (as previously described), resulting in 435,834 regions. The dataset was further filtered based on the scATAC-seq quality, keeping cells with at least log(unique fragments) > 3.5, FRiP > 0.2 and TSS > 4, resulting in 1,736 cells. Topic modeling was performed using MALLET, using 500 iterations and models with 2 topics and from 5–50 by an increase of 5, selecting the model with 40 topics based on the model selection metrics. pycisTarget (v.1.0.1.dev17+gd2571bf) was run, using a custom database with the consensus regions, on DARs and binarized topics (with Otsu thresholding), with and without promoters and using pycisTarget and DEM. SCENIC+ was run with default parameters, using http://www.ensembl.org as a BioMart host. High-quality regulons were selected based on the correlation between gene-based regulon AUCs and region-based regulon AUCs (>0.6) and on the number of target genes (>10). This resulted in 111 regulons, with a mean of 100 genes and 171 regions per regulon. The 10x Visium data were processed using Seurat (v.4.0.3). SCENIC+ regulons were scored in the spots using AUCell (with the spot–gene matrix as input), with default parameters.

**Molecular Cartography in the mouse cortex.** *Gene panel selection.* One hundred genes were selected based on their gene expression patterns (marker genes for a cell type or group of cell types) on our in-house mouse cortex dataset and literature (Supplementary Table 6). In addition, dimensionality reduction using only these 100 genes was

performed to ensure that all cell types could be distinguished with this gene panel.

*Probe design.* The probes for the 100 selected genes were designed using Resolve's proprietary design algorithm. Briefly, the probe design was performed at the gene level. For every targeted gene, all full-length protein-coding transcript sequences from the ENSEMBL database were used as design targets if the isoform had the GENCODE annotation tag 'basic'[103]. To speed up the process, the calculation of computationally expensive parts, especially the off-target searches, the selection of probe sequences was not performed randomly, but limited to sequences with high success rates. To filter highly repetitive regions, the abundance of *k*-mers was obtained from the background transcriptome using Jellyfish[104]. Every target sequence was scanned once for all *k*-mers and those regions with rare *k*-mers were preferred as seeds for full probe design. A probe candidate was generated by extending a seed sequence until a certain target stability was reached. A set of simple rules was applied to discard sequences that were found experimentally to cause problems. After these fast screens, every kept probe candidate was mapped to the background transcriptome using ThermonucleotideBLAST[105] and probes with stable off-target hits were discarded. Specific probes were then scored based on the number of on-target matches (isoforms), which were weighted by their associated APPRIS level[106], favoring principal isoforms over others. A bonus was added if the binding site was inside the protein-coding region. From the pool of accepted probes, the final set was composed by greedily picking the highest scoring probes. Gene names and catalog numbers for the specific probes designed by Resolve Biosciences are included in Supplementary Table 6.

*Tissue sections.* Mouse brain samples were fixed with PAXgene Tissue FIX solution (Resolve Biosciences) for 24 h at room temperature followed by 2 h in PAXgene Tissue Stabilizer (Resolve Biosciences) at room temperature. Samples were cryoprotected in a 30% sucrose solution (w/v) overnight at 4 °C and frozen in 2-methylbutane (Sigma-Aldrich, 106056) on dry ice. Frozen samples were sectioned with a cryostat (Leica CM3050) and 10-µm thick sections were placed within the capture areas of cold Resolve Biosciences slides. Samples were then sent to Resolve Biosciences on dry ice for analysis. Upon arrival, tissue sections were thawed and rehydrated with isopropanol, followed by 1-min washes in 95% ethanol and 70% ethanol at room temperature. The samples were used for Molecular Cartography (100-plex combinatorial single molecule fluorescence in situ hybridization) according to the manufacturer's instructions (protocol v.1.3; available for registered users), starting with the aspiration of ethanol and the addition of buffer DST1 followed by tissue priming and hybridization. Briefly, tissues were primed for 30 min at 37 °C followed by overnight hybridization of all probes specific for the target genes (see below for probe design details and target list). Samples were washed the next day to remove excess probes and fluorescently tagged in a two-step color development process. Regions of interest were imaged as described below and fluorescent signals were removed during decolorization. Color development, imaging and decolorization were repeated for multiple cycles to build a unique combinatorial code for every target gene that was derived from raw images as described below.

*Imaging.* Samples were imaged on a Zeiss Celldiscoverer 7, using the ×50 Plan Apochromat water immersion objective with an NA of 1.2 and the ×0.5 magnification changer, resulting in a ×25 final magnification. Standard CD7 LED excitation light source, filters and dichroic mirrors were used together with customized emission filters optimized for detecting specific signals. Excitation time per image was 1,000 ms for each channel (4,6-diamidino-2-phenylindole (DAPI) was 20 ms). A *z*-stack was taken at each region with a distance per *z*-slice according to the Nyquist–Shannon sampling theorem. The custom CD7 CMOS camera (Zeiss Axiocam Mono 712, 3.45-µm pixel size) was used. For each region, a *z*-stack per fluorescent color (two colors) was imaged per imaging round. A total of eight imaging rounds were conducted for each position, resulting in 16 *z*-stacks per region. The completely

automated imaging process per round (including water immersion generation and precise relocation of regions to image in all three dimensions) was realized by a custom Python script using the scripting API of the Zeiss ZEN software (open application development).

*Spot segmentation.* The algorithms for spot segmentation were written in Java and are based on the ImageJ library functionalities. Only the iterative closest point algorithm is written in C++ based on the libpointmatcher library (https://github.com/ethz-asl/libpointmatcher).

*Preprocessing.* As a first step, all images were corrected for background fluorescence. A target value for the allowed number of maxima was determined based upon the area of the slice in μm² multiplied by the factor 0.5. This factor was empirically optimized. The brightest maxima per plane were determined, based upon an empirically optimized threshold. The number and location of the respective maxima was stored. This procedure was conducted for every image slice independently. Maxima that did not have a neighboring maximum in an adjacent slice (called a $z$ group) were excluded. The resulting maxima list was further filtered in an iterative loop by adjusting the allowed thresholds for (Babs-Bback) and (Bperi-Bback) to reach a feature target value (Babs, absolute brightness; Bback, local background; and Bperi, background of periphery within one pixel). These feature target values were based upon the volume of the three-dimensional (3D) image. Only maxima still in a $z$ group of at least two after filtering were passing the filter step. Each $z$ group was counted as one hit. The members of the $z$ groups with the highest absolute brightness were used as features and written to a file. They resemble a 3D point cloud.

*Final signal segmentation and decoding.* To align the raw data images from different imaging rounds, images had to be corrected. To do so the extracted feature point clouds were used to find the transformation matrices. For this purpose, an iterative closest point cloud algorithm was used to minimize the error between two point clouds. The point clouds of each round were aligned to the point cloud of round one (reference point cloud). The corresponding point clouds were stored for downstream processes. Based upon the transformation matrices the corresponding images were processed by a rigid transformation using trilinear interpolation. The aligned images were used to create a profile for each pixel consisting of 16 values (16 images from two color channels in eight imaging rounds). The pixel profiles were filtered for variance from zero normalized by total brightness of all pixels in the profile. Matched pixel profiles with the highest score were assigned as an ID to the pixel. Pixels with neighbors having the same ID were grouped. The pixel groups were filtered by group size, number of direct adjacent pixels in group and number of dimensions with size of two pixels. The local 3D maxima of the groups were determined as potential final transcript locations. Maxima were filtered by number of maxima in the raw data images where a maximum was expected. Remaining maxima were further evaluated by the fit to the corresponding code. The remaining maxima were written to the results file and considered to resemble transcripts of the corresponding gene. The ratio of signals matching to codes used in the experiment and signals matching to codes not used in the experiment were used as estimation for specificity (false positives).

*Downstream analysis.* Final image analysis was performed in ImageJ (v.2.3.0/1.53f) using the Polylux tool plugin (v.1.6.1) from Resolve Biosciences to examine specific Molecular Cartography signals. Nuclei segmentation was performed using CellProfiler (v.4.2.1) based on the DAPI signal, using 30 and 100 as minimum and maximum diameter of objects, an adaptive threshold strategy and Otsu as thresholding method. Nuclei were expanded by 50 pixels. Cell-type labels and whole-transcriptome gene expression data from our in-house mouse cortex atlas were mapped using Tangram (v.1.0.2), after library size correction, log normalization and correction of the gene expression values by sample using Combat (Scanpy[94] v.1.8.2). SCENIC+ regulons were scored on the nuclei using AUCell (with the nuclei–gene matrix as input), using default parameters.

## Reporting summary

Further information on research design is available in the Nature Portfolio Reporting Summary linked to this article.

## Data availability

Data generated in this manuscript, namely scATAC-seq in melanoma cell lines, 10x multiome in the mouse cortex and scATAC-seq in the *Drosophila* eye disc, are available in GEO under accession code GSE210749. GRCh38.86 genome annotation used in this study is available at https://ftp.ensembl.org/pub/release-86/gtf/homo_sapiens/Homo_sapiens.GRCh38.86.chr.gtf.gz. The GRCh38 genome index used in this study is available at https://cf.10xgenomics.com/supp/cell-arc/refdata-cellranger-arc-GRCh38-2020-A-2.0.0.tar.gz. The mm10 genome index used in this study is available at https://cf.10xgenomics.com/supp/cell-arc/refdata-cellranger-arc-mm10-2020-A-2.0.0.tar.gz. Data from ENCODE deeply profiled cell lines were downloaded from https://www.encodeproject.org/, including bulk RNA-seq and ATAC-seq for eight cell lines, namely MCF7 (ENCFF136ANW and ENCFF772EFK, for RNA-seq and ATAC-seq, respectively), HepG2 (ENCFF660EXG and ENCFF239RGZ), PC3 (ENCFF874CFD and ENCFF516GDK), GM12878 (ENCFF626GVO and ENCFF415FEC), K562 (ENCFF833WFD and ENCFF512VEZ), Panc1 (ENCFF602HCV and ENCFF836WDC), IMR90 (ENCFF027FUC and ENCFF848XMR) and HCT116 (ENCFF766TYC and ENCFF724QHH); and Hi-C data on five of the cell lines (IMR90 (ENCFF685BLG), GM12878 (ENCFF053VBX), HCT116 (ENCFF750AOC), HepG2 (ENCFF020DPP) and K562 (ENCFF080DPJ)). STARR-seq data were downloaded from ENCODE (ENCFF045TVA (K562), ENCFF047LDJ (HepG2), ENCFF428KHI (HCT116), ENCFF826BPU (MCF7)). ChIP-seq bigWig and summit bed files were downloaded from ENCODE using the following accession numbers: ENCFF702MTT and ENCSR000BHD for PAX5; ENCFF107LDM and ENCSR000BGU for EBF1; ENCFF803HIP and ENCFF934JFA for POU2F2 for bigWig and summit bed files respectively. The bulk RNA-seq experiments upon perturbation in these cell lines and ChIP-seq datasets are described in Supplementary Tables 1 and 4, respectively. The 10x multiome data on PBMCs were downloaded from the 10x website. scRNA-seq data of baseline MM-lines and bulk RNA-seq data after SOX10 knockdown were downloaded from GEO (GSE134432). MITF, SOX10 and TFAP2A ChIP-seq data were downloaded from GEO (GSE61965 (MITF and SOX10) and GSE67555 (TFAP2A)). SNARE-seq2 data on the human cortex were downloaded from Bakken et al.[60] scATAC-seq and scRNA-seq data from the *Drosophila* eye-antennal disc were downloaded from GEO (GSE115476). The 10x Visium data and 10x single-cell multiome data from the human cerebellum were downloaded from the 10x website. All analyses can be explored in SCope (https://scope.aertslab.org/#/scenic-v2) and UCSC in the following sessions: PBMCs (https://genome-euro.ucsc.edu/s/Seppe%20De%20Winter/scenicplus_pbmc), ENCODE cell lines (https://genome.ucsc.edu/s/cbravo/SCENIC%2B_DPCL), melanoma (http://genome-euro.ucsc.edu/s/Seppe%20De%20Winter/scenicplus_mix_melanoma), mouse and human cortex (https://genome-euro.ucsc.edu/s/cbravo/SCENIC%2B_Cortex), eye-antennal disc (http://genome.ucsc.edu/s/cbravo/SCENIC%2B_EAD) and human cerebellum (https://genome-euro.ucsc.edu/s/cbravo/SCENIC%2B_cerebellum). The SCENIC+ motif collection is available at https://resources.aertslab.org/cistarget/motif_collections.

## Code availability

pycisTopic is available at https://github.com/aertslab/pycisTopic and deposited in Zenodo at https://doi.org/10.5281/zenodo.7857024. pycisTarget is available at https://github.com/aertslab/pycistarget and deposited in Zenodo at https://doi.org/10.5281/zenodo.7857022. SCENIC+ is available at https://github.com/aertslab/scenicplus and deposited in Zenodo at https://doi.org/10.5281/zenodo.7857017. Detailed tutorials and documentation on the SCENIC+ workflow are available at scenicplus.readthedocs.io and tutorials on pycisTopic and pycisTarget (within the SCENIC+ workflow and as standalone packages) are available at pycisTopic.readthedocs.io and pycistarget.readthedocs.io,

respectively. Code to generate custom cisTarget databases is available at https://github.com/aertslab/create_cisTarget_databases. Our implementation of Cluster-Buster is available at https://github.com/ghuls/cluster-buster/tree/change_f4_output. Notebooks to reproduce the analyses presented in this manuscript are available at https://github.com/aertslab/scenicplus_analyses.

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

## Acknowledgements

Computing was performed at the Vlaams Supercomputer Center. This work was funded by the following grants to S. Aerts: ERC Consolidator Grant (724226_cis-CONTROL), ERC Proof of Concept (963884), Special Research Fund (BOF) KU Leuven (grants C14/18/092 and C14/22/125), Foundation Against Cancer (2020-062), EOS (G0I2722N/40007513) and FWO (grants G0B5619N and G094121N); PhD fellowships from the FWO to C.B.G.-B. (11F1519N) and S.D.W. (1191323N) and postdoctoral fellowships from FWO to N.H. (1273822N) and Stichting tegen Kanker (Foundation Against Cancer) to J.W. (2019-100). We thank members of various groups that make curated position weight matrices publicly available, including T. Hughes (cisbp), M. Bulyk (Uniprobe), A. Mathelier (Jaspar), V. Makeev (Hocomoco) and many others, listed in Supplementary Table 3. We thank Resolve Biosciences, especially J. Aerts, for performing the Molecular Cartography experiments in the mouse cortex; and Janssen Pharmaceutica, VIB Tech Watch and the VIB single-cell accelerator for help and funding for generating the mouse cortex data. We thank D. Daaboul for proofreading the manuscript.

## Author contributions

C.B.G.-B., S.D.W. and S. Aerts conceived the study. C.B.G.-B. developed pycisTopic, C.B.G.-B. and S.D.W. co-developed pycisTarget and the SCENIC+ modules and workflow and G.H. developed the code to generate custom cisTarget databases. C.B.G.-B. and G.H. made the SCENIC+ motif collection. C.B.G.-B. and S.D.W. performed the computational analyses, with the assistance of G.H., N.H. and S. Aibar. I.M. and S.P. generated the mouse cortex multiome data and J.W. performed the single-cell ATAC-seq experiments on the melanoma cell lines, with the assistance of V.C. C.B.G.-B., S.D.W. and S. Aerts wrote the manuscript.

## Competing interests

The authors declare no competing interests.

## Additional information

**Extended data** is available for this paper at https://doi.org/10.1038/s41592-023-01938-4.

**Correspondence and requests for materials** should be addressed to Stein Aerts.

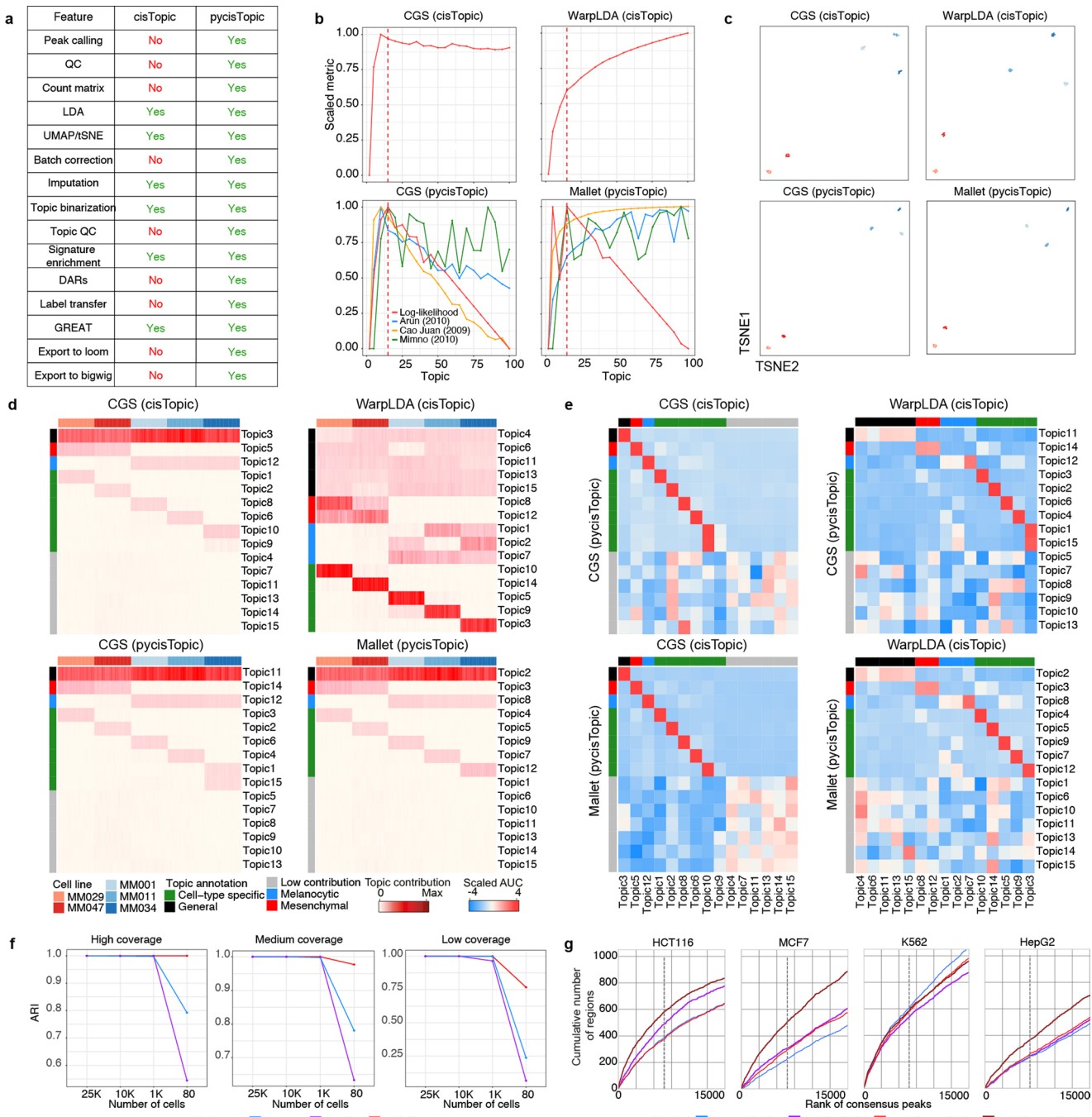

**Extended Data Fig. 1 | Cell type and enhancer discovery benchmark with pycisTopic, cisTopic, Signac and ArchR. a.** Feature comparison between cisTopic and pycisTopic. **b.** Model selection for models (for 100 cells simulated from melanoma cells lines) with different parameter optimization methods, namely Collapsed Gibbs Sampler (CGS) and WarpLDA with cisTopic and CGS and Mallet with pycisTopic. cisTopic relies on the log-likelihood per model; while pycisTopic incorporates additional measurements including coherence (Minmo (2010)), a density-based metric (Cao Juan (2009) and a divergence-based metrics (Arun (2010)). **c.** Cell-topic dimensionality reduction for each of the models (100 cells). Red clusters denote the 2 mesenchymal cell lines, blue clusters depict the 3 melanocytic cell lines. **d.** Cell-topic enrichment heat map for each of the models. General topics are shown in black; mesenchymal, in red; melanocytic, in blue;

cell line specific in green; and low contributing in gray. **e.** AUCell enrichment of topics between different models. **f.** Adjusted Rand Index (ARI) for pycisTopic, Signac and ArchR in simulated datasets with different coverage per cell (3 K, 10 K, or 20 K fragments per cell) and number of cells, using as ground truth the bulk label from which cells were simulated. Data was simulated from bulk ATAC-seq and bulk RNA-seq data from ENCODE's Deeply Profiled Cell Lines. **g.** Recovery curves for top 5 K Differentially Accessible Regions (DARs) identified by Signac, pycisTopic and ArchR and top 5 K regions in the cell line specific topics identified by pycisTopic. Genome-wide STARR-seq in HCT116, MCF7, K562 and HepG2 is ranked in descending order (x axis) when a region of the ranking is found in a region set an increasing step along the y axis is taken. Dashed line represents the top 10% of the ranking.

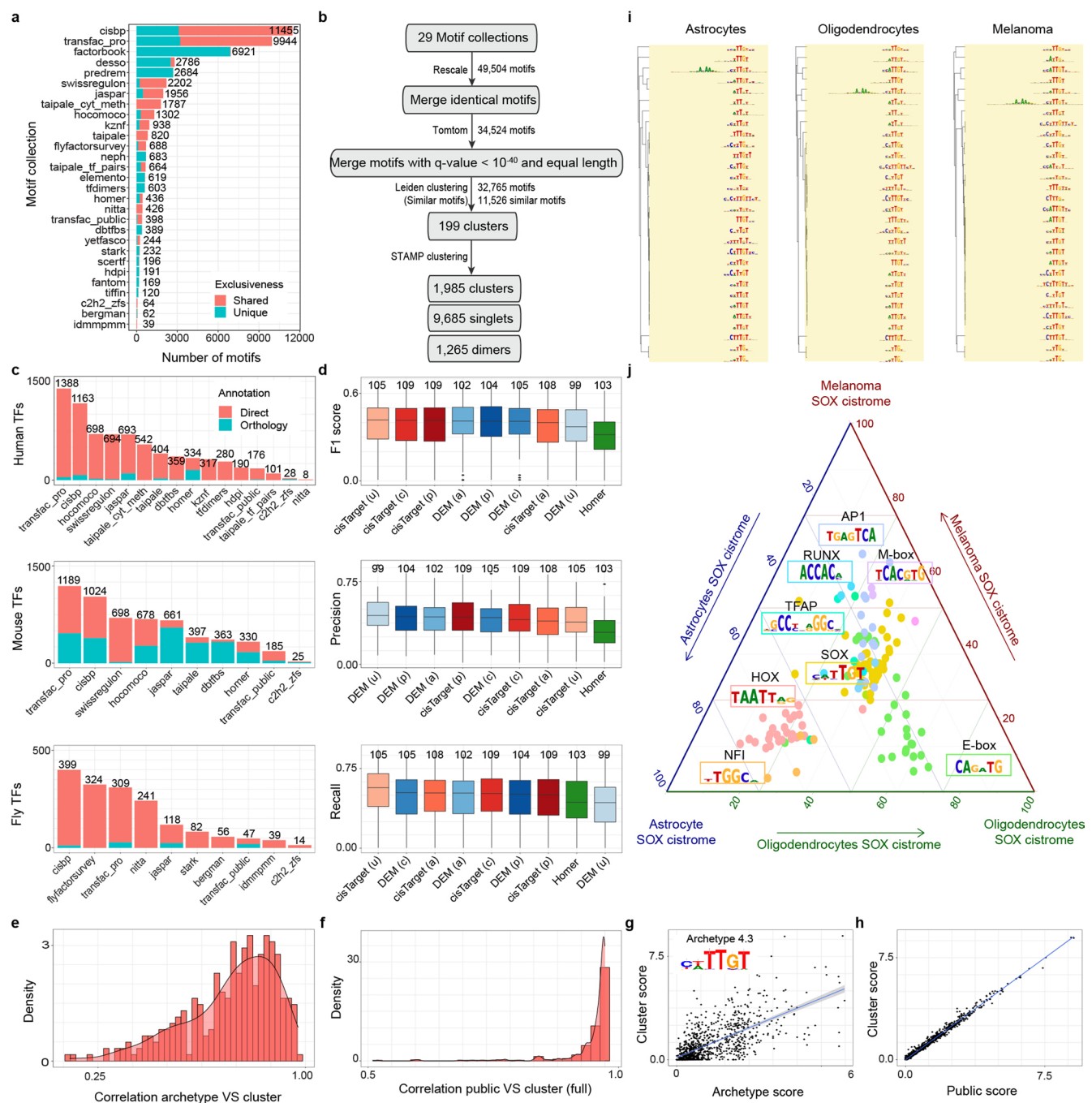

**Extended Data Fig. 2 | The SCENIC+ motif collection. a**. Number of motifs per motif collection that are shared or unique for one collection. **b**. Workflow depicting the motif collection cluster strategy. **c**. Number of motifs annotated directly or by orthology per motif collection. **d**. F1 score (top), precision (middle) and recall (bottom) distributions of TF cistromes from motif enrichment on 309 TF ChIP-seq data sets from ENCODE, using different databases and motif enrichment methods, namely Homer, pycisTarget and DEM. The unclustered databases (u) include all annotated motifs before clustering (singlets), the archetype databases (a) use the consensus motifs of the clusters based on STAMP and the clustered databases uses the motif clusters (c), scoring regions using all motifs in the cluster and the public databases (p) is the clustered database without licensed Transfac Pro motifs. Upper/lower hinge represent upper/lower quartile, whiskers extend from the hinge to the largest/smallest value no further than 1.5 times the interquartile range from the hinge respectively. Median is used as center. **e**. Distribution of the correlation between scores (on chr19) using archetypes or all motifs in a cluster. **f**. Distribution of the correlation between scores (on chr19) using all motifs in a cluster or all motifs except for Transfac Pro motifs. **g**. Correlation between scores (on chr19) for cluster 4.3 using the archetype or all motifs. **h**. Correlation between scores (on chr19) for cluster 4.3 using all motifs or all motifs except for Transfac Pro motifs. **i**. Top 30 motifs identified by cisTarget using regions in the SOX cistromes from melanoma, oligodendrocytes and astrocytes clustered using motifStack. Colors indicate the TF family of the motifs (in this case, SOX). **j**. Ternary plot showing enrichment scores of motifs found in melanoma, oligodendrocyte and astrocyte SOX regions. Each corner represents a cell-type-specific SOX topic, dots represent enriched motifs and axes represent average enrichment scores for each topic. The colors of the dots are used to indicate the TF family to which the motifs belong.

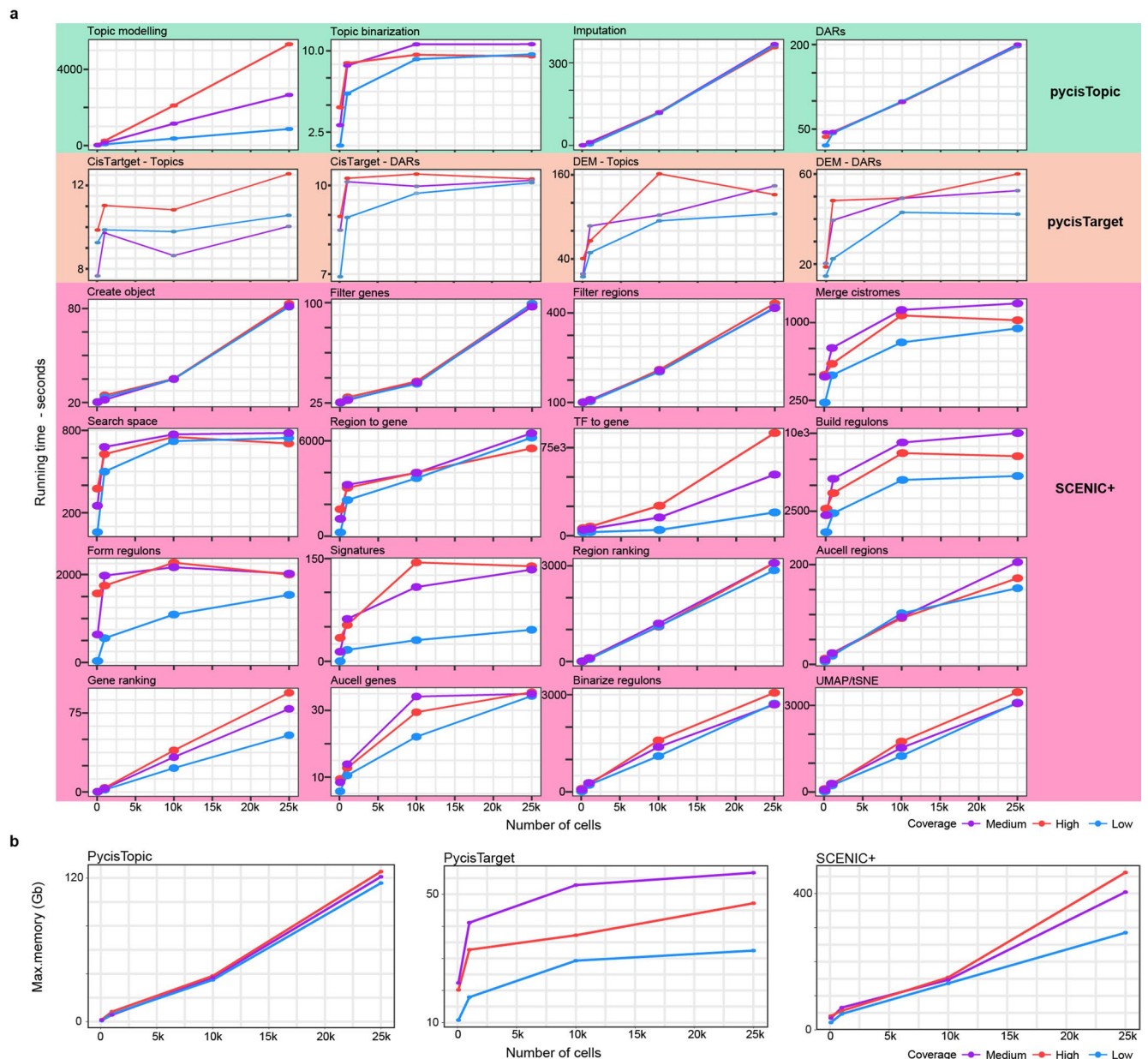

**Extended Data Fig. 3 | Time and memory complexity analysis of the SCENIC+ workflow using simulated datasets with different coverage per cell (3 K, 10 K, or 20 K fragments per cell) and number of cells. a**. Running times for the minimal preprocessing steps with pycisTopic, pycisTarget and SCENIC+.

The times specified for topic modeling correspond to the average running time for one model. The running times specified for pycisTarget correspond to the average running time for one region set. **b**. Maximum memory used for the minimal preprocessing steps with pycisTopic, pycisTarget and SCENIC+.

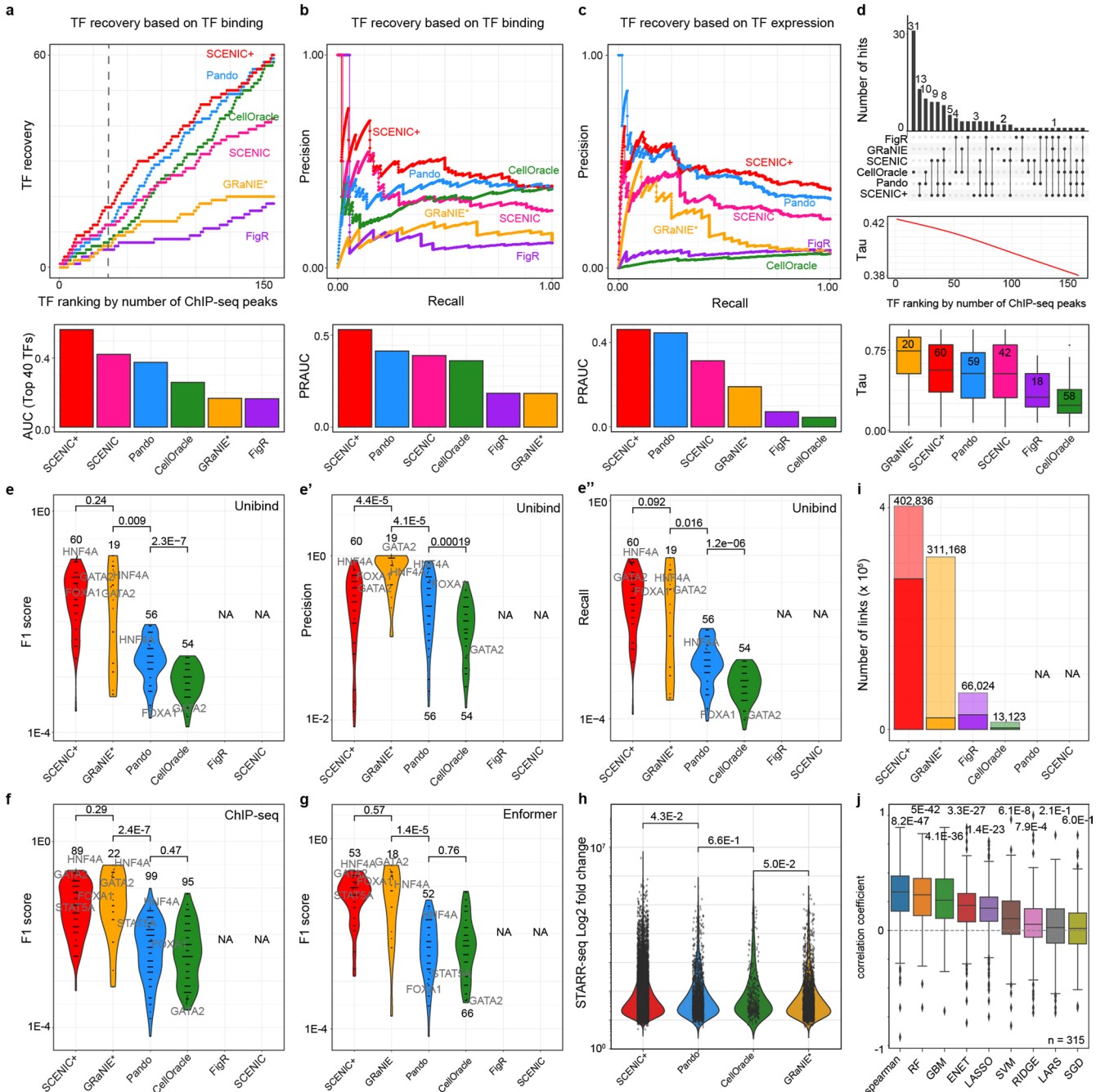

**Extended Data Fig. 4 | TF, target region and region-to-gene relationships recovery performance by single-cell multiomics methods. a.** Cumulative TF recovery, TFs are ranked based on the number of Unibind peaks in descending order (top) and Area Under the Curve (AUC) per method on top 40 TFs (bottom). **b-c.** Precision-recall curves of TFs found per method using different thresholds on the TF ChIP-seq based ranking (**b**, top) and LogFC of TF expression (**c**, top) and AUC values (**b, c**, bottom). **d.** Overlap between identified TFs per method (top), GAM fitted Tau values for the TFs (middle) and distribution of Tau values per method. **e-g.** Violin plots of F1 score (**f, g**), precision and recall (**e', e'**) distributions from the comparison of regulon target regions, per method and Unibind (**e**), ChIP-seq peaks (**f**) and Enformer predicted ChIP-seq (**g**). The numbers indicate the number of regulons. **h.** Violin plot showing distribution of maximum enhancer activity as measured using STARR-seq data from ENCODE on K562, HepG2, HCT116 and MCF7 regions. **i.** Barplots showing the number of region-gene links found per method. Non-transparent bars show the number of

links in the eGRN, transparent bars show the number region-gene links before eGRN construction. Pando and SCENIC are excluded from the comparison since they do not report (unique) region-gene relationships. **j.** Correlation between Hi-C links for the top 100 markers genes for each of the cell lines where Hi-C is available (IMR90, GM12878, HCT116, HepG2 and K562) and region-gene scores from different region-gene inference models (Spearman correlation, Random Forest (RF), GBM (Gradient Boosting Machine), ENET (Elastic Net), Lasso, Support Vector Machine (SVM) with linear kernel, Ridge, Least-Angle Regression (LARS) and Stochastic Gradient Descent (SGD)). For boxplots in panels d and j: Upper/lower hinge represent upper/lower quartile, whiskers extend from the hinge to the largest/smallest value no further than 1.5 times the interquartile range from the hinge respectively. Median is used as center. Difference in mean between methods (e-h) and shuffled links (i) assessed using two-sided Wilcoxon rank-sum test, correction for multiple testing using Benjamini–Hochberg procedure. GRaNIE* was run with simulated single-cell data instead of bulk.

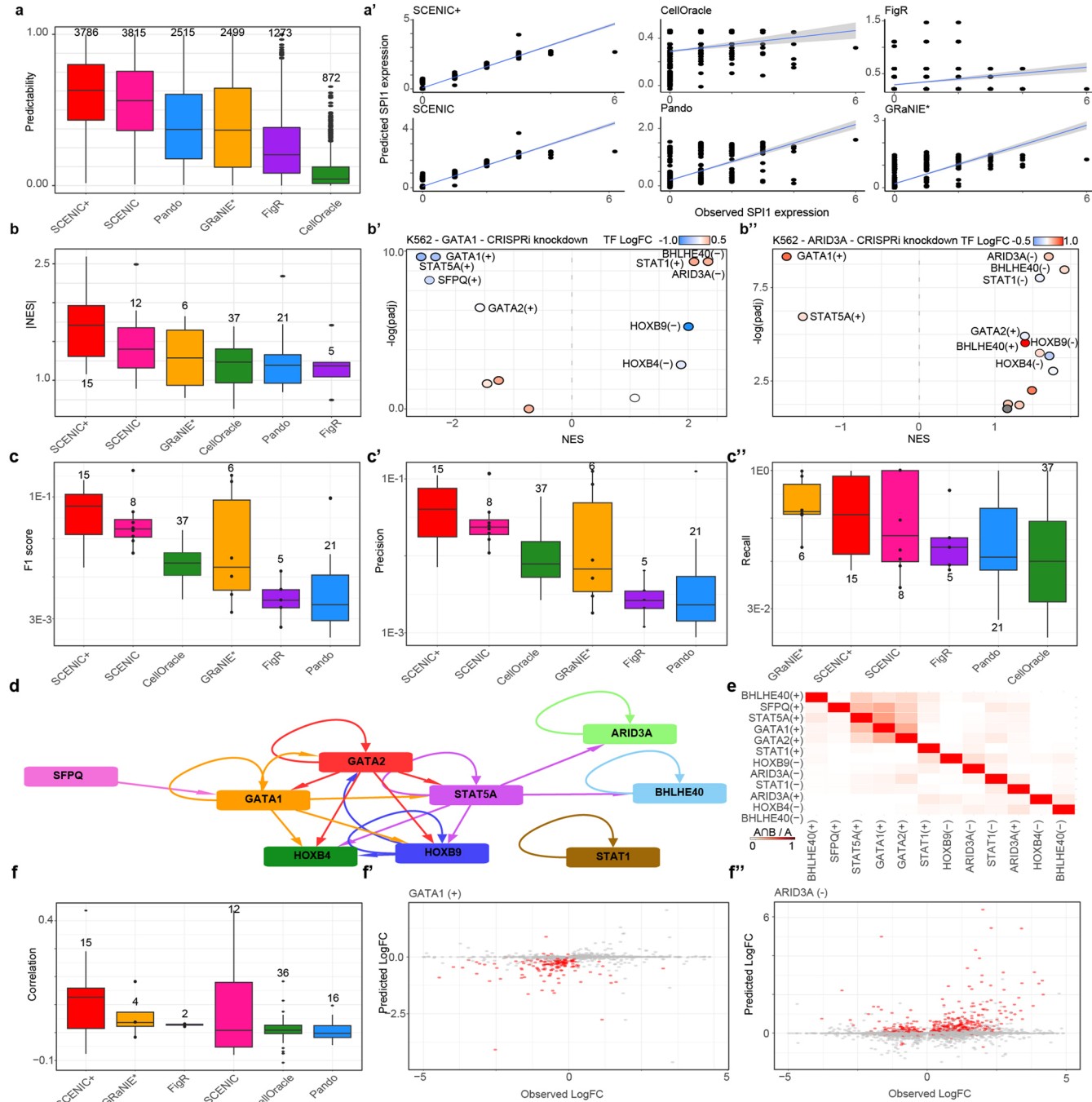

**Extended Data Fig. 5 | Target gene recovery performance by single-cell multiomics methods. a**. Boxplot depicting the correlation between observed and predicted gene expression values using the eGRNs inferred from each method, together with scatter plots showing the correlation between the predictions by each method and the observed expression values for SPI1 (**a'**). **b**. NES distribution based on GSEA analysis using TF knockdown data as ranking and target genes derived by each method as gene set, with examples on K562 upon STAT5A (**b'**) and HOXB9 (**b'**) knockdowns showing GSEA -log10 adjusted p value and NES for different eGRNs found by SCENIC+. **c**. Boxplots represent the F1 score (**c**), precision (**c'**) and recall (**c'**) distributions of the predicted target genes per TF compared to TF perturbation data. **d**. Network showing TF-target

gene interactions for selected genes. **e**. Heat map showing the overlap between the regions of the regulons indicated by the rows and columns, divided by the size of the regulons in the columns. **f**. Spearman correlation between predicted LogFC with in silico TF perturbation for each method versus the observed LogFC changes upon TF perturbation, together with the comparison between predicted and observed LogFC changes upon GATA1 KD (**f'**) and ARID3A KD (**f'**). Dots in red indicate genes in the GATA1 or ARID3A regulons, respectively. In boxplots, upper/lower hinge represent upper/lower quartile, whiskers extend from the hinge to the largest/smallest value no further than 1.5 times the interquartile range from the hinge respectively. Median is used as center.

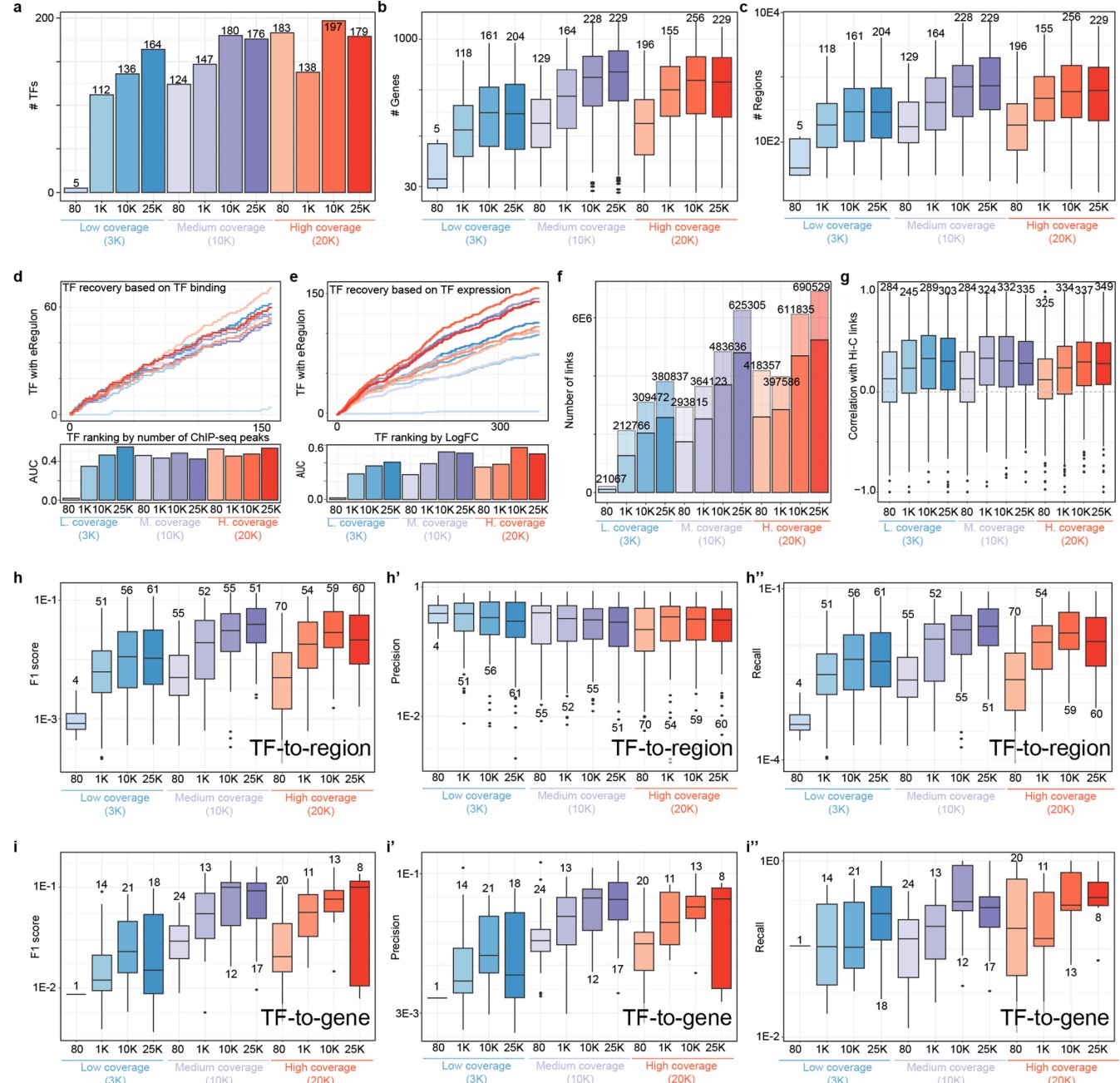

**Extended Data Fig. 6 | Performance of SCENIC+ upon variations in coverage and sample size. a.** Number of TFs identified per analysis. **b.** Number of genes per regulon per analysis. **c.** Number of regions per regulon per analysis. **d.** Cumulative TF recovery for each method using as x axis TFs ranked based on the number of ChIP-seq peaks and AUC values per method using the top 40 TFs. **e.** Cumulative TF recovery for each method using as x axis TFs ranked based on the maximum LogFC across the cell lines and AUC values per method. **f.** Number of region-gene links inferred (non-transparent links indicate that the links are included in the final eGRN). **g.** Boxplot showing the correlation with the Hi-C links for the top 100 marker genes for each of the cell line where Hi-C is available (IMR90, GM12878, HCT116, HepG2 and K562). **h.** F1 score (**h**), precision (**h'**) and recall (**h'**) distributions of the predicted regions per TF using Unibind regions as standard. **i.** Boxplots representing the F1 score (**i**), precision (**i'**) and recall (**i'**) distributions of the predicted target genes per TF compared to TF perturbation data. In boxplots, upper/lower hinge represent upper/lower quartile, whiskers extend from the hinge to the largest/smallest value no further than 1.5 times the interquartile range from the hinge respectively. Median is used as center.

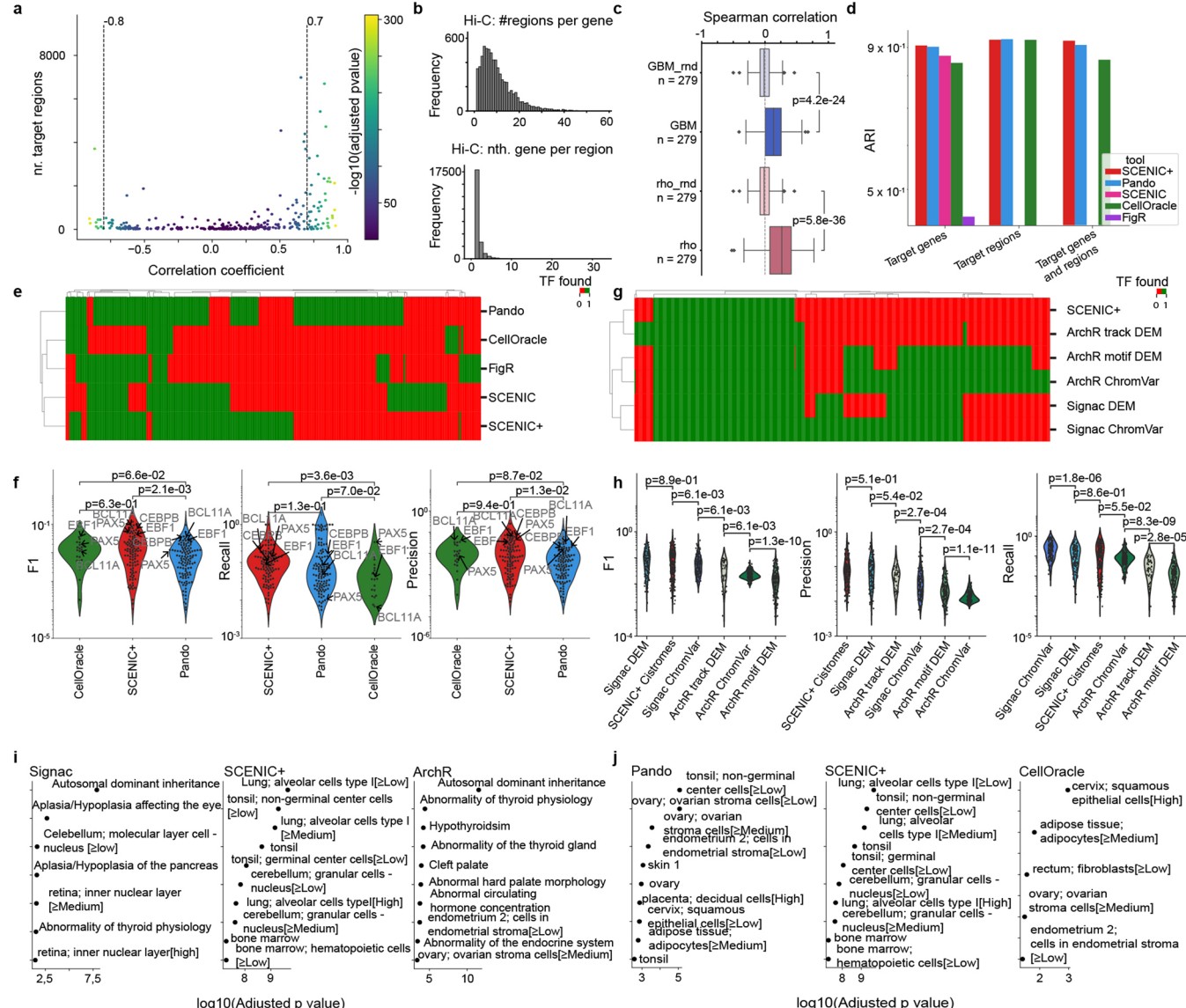

**Extended Data Fig. 7 | Benchmark of SCENIC+ and other methods on PBMC single-cell multiomics data. a**. Scatter plot showing number of target regions versus TF expression-to-region AUC Pearson correlation coefficients for each eRegulon inferred in the PBMC data set. eRegulons are selected based on a threshold on the correlation coefficient, indicated by dotted line. **b**. Distribution of the number of regions linked to each gene based on Hi-C in GM12878 (using a minimum score of 1) and the rank, based on absolute distance, for each region and the gene with the highest Hi-C score in GM12878. **c**. Boxplots showing the distribution of Spearman correlation coefficients between Hi-C scores in GM12878 and region-to-gene importance score and region-to-gene correlation coefficients (rho) as calculated by SCENIC+ for B-cell marker genes. Upper/lower hinge represent upper/lower quartile, whiskers extend from the hinge to the largest/smallest value no further than 1.5 times the interquartile range from the hinge respectively. Median is used as center. Random controls are obtained by shuffling the gradient boost importance scores (GBM_rnd) and correlation

coefficients (rho_rnd). Difference in the mean to the random control is assessed using the Mann-Whitney U test. **d**. Adjusted Rand Index (ARI) quantifying how well cell types are separated based on the AUC scores for the PBMC data set. **e**. Heat maps showing whether a TF is found across different methods comparing SCENIC+ to Signac and ArchR. Signac and ArchR were run using different options. (1) DEM: Differentially Enriched Motifs or ChIP-seq tracks in differentially accessible regions and (2) ChromVAR deviations. **f**. Scatter plot showing enrichment of top 10 Human Protein Atlas and Human Phenotype GO terms for TFs found exclusively by Signac, Archr or all methods including SCENIC+. **g**. Heat maps showing whether a TF is found across different methods. GRaNIE is not included because the analysis ran out of memory (tested on a machine with 72 cores Intel(R) Xeon(R) Platinum 8360Y CPU @ 2.40 GHz and 2 TB of memory). **h**. Scatter plot showing enrichment of top 10 Human Protein Atlas and Human Phenotype GO terms for TFs found exclusively by Pando, CellOracle or all methods including SCENIC+.

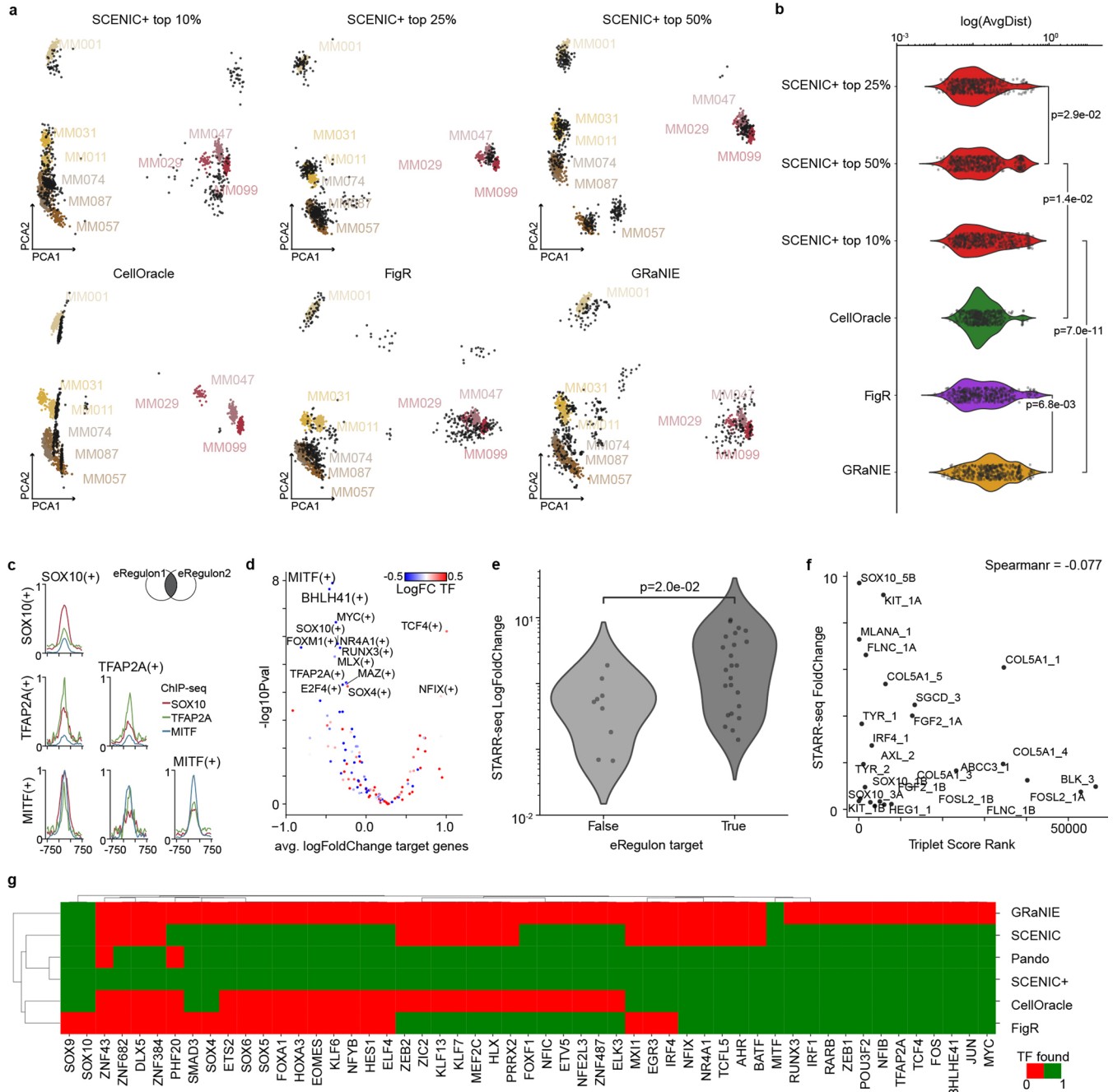

**Extended Data Fig. 8 | Benchmark of SCENIC+ and other methods on the melanoma cell lines data set. a.** Boolean networks were generated from gene regulatory networks inferred from SCENIC+, CellOracle, FigR and GRaNIE. For SCENIC+ the top 10%, 25% and 50% of edges based on the triplet score were used. 500 cells were simulated using the boolODE method using a simulation time of 20 and the hill activation function. Simulated cells were co-embedded in PCA space with real cells after Harmony batch effect correction. **b.** Violin and jitter plot of the average distance of each simulated cell to its three nearest neighbors in the first 2 principal components of PCA space. Difference in mean is assessed using two-tailed Mann-Whitney U test and p values are adjusted using the Benjamini–Hochberg procedure. y axis is sorted by the median average distance. **c.** ChIP-seq enrichment of SOX10, MITF and TFAP2A in target regions of SOX10, MITF and TFAP2A and all combinations of two. Signal is scaled across all comparisons between 0 and 1. **d.** -log10 p value (t-test) and average log2 fold change of target genes of eRegulons after SOX10 knockdown in MM001. Color scale encodes log2 fold change of the expression of the TF corresponding to each eRegulon after SOX10 knockdown. **e.** Scatter and jitter plot showing enhancer activity as measured by the STARR-seq method[18] in regions targeted by any of the regulons and regions targeted by None. **f.** Scatter plot comparing enhancer activity as measure by the STARR-seq method (y axis)[18] to the minimum of the triplet score over all TFs targeting and genes targeted by the region (x axis). Labels of the regions are according to the labels in Mauduit *et al.*[18]. Difference in mean is assessed using two-tailed Mann-Whitney U test and p values are adjusted using the Benjamini–Hochberg procedure. x axis is sorted by the median average distance. **g.** Heat map showing whether a TF is found across different e(GRN) inference methods (present: green; absent: red). Only TFs found by SCENIC+ are shown.

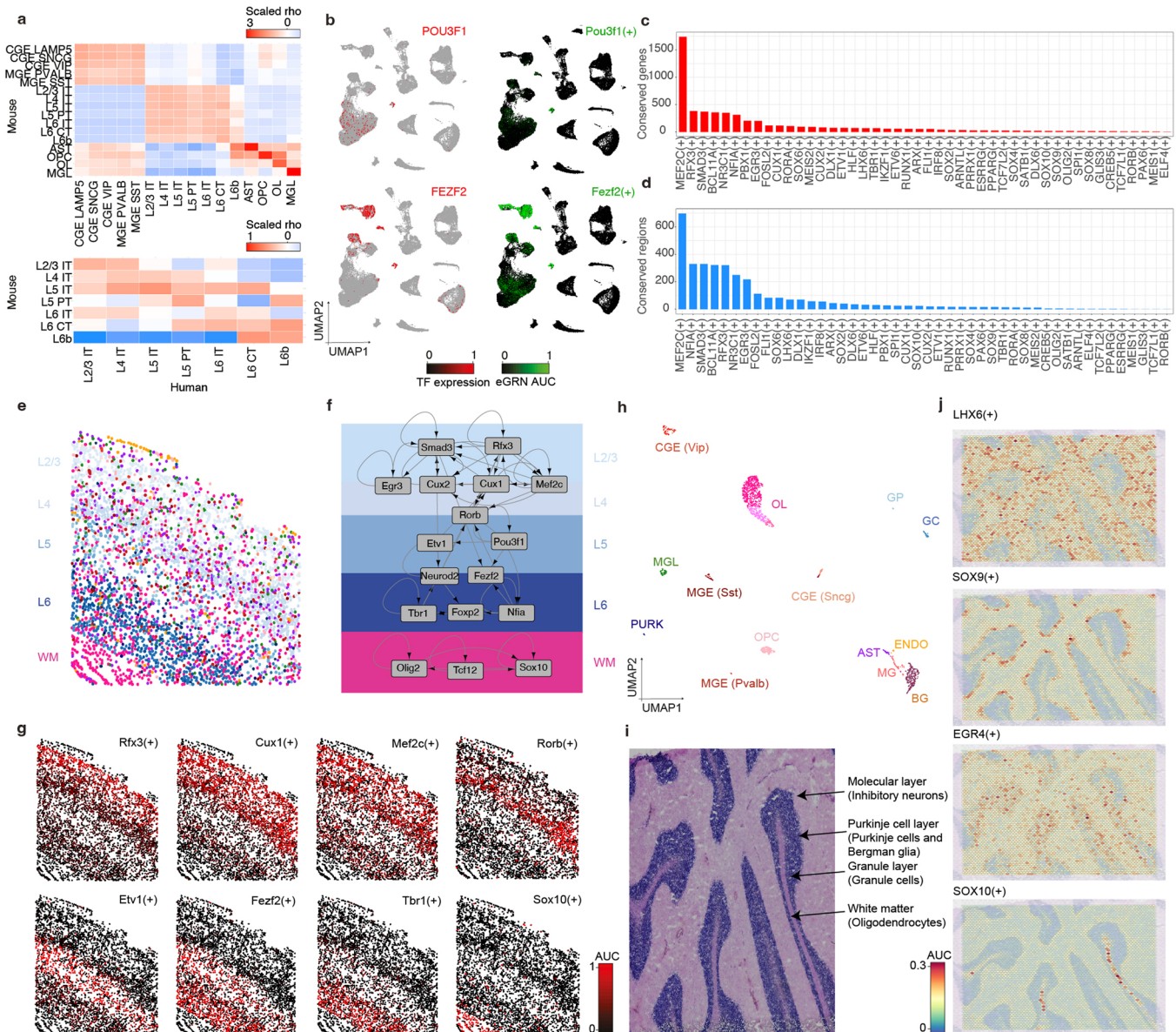

**Extended Data Fig. 9 | Conservation and spatial visualization of enhancer-GRNs in the mammalian brain. a.** Heat map showing the scaled correlation between the RSS values for each regulon in each cell type. **b.** Human cortex UMAP (84,159) showing TF expression (red) and AUC enrichment of the mouse regulon (converted to human genes). **c.** Barplot showing the number of conserved genes between the matching human and mouse regulons. **d.** Barplot showing the number of conserved regulons between the matching mouse and human regulons. **e.** Mapping of cell types in the mouse cortex into our smFISH map using Tangram. **f.** Visualization of regulons AUC enrichment in our smFISH map of the mouse cortex. **g.** Representative layer-specific gene regulatory network. The

network depicted from L2/3 to L6 corresponds to excitatory neurons, while in the white matter corresponds to oligodendrocytes. **h.** SCENIC+ UMAP containing 1,736 cells from the human cerebellum. **i.** Human cerebellum 10x Visium slide annotated with anatomical regions in the cerebellum. **j.** Visualization of regulons AUC enrichment on the 10x Visium data. AST: Astrocytes, BG: Bergman Glia, CGE: Caudal Ganglionic Eminence, ENDO: Endothelial cells, GC: Granule Cell, GP: Granule cell Progenitor, MGE: Medial Ganglionic Eminence, MGL: Microglia, MG: Muller Glia, OL: Oligodendrocyte, OPC: Oligodendrocyte Precursor Cells, PURK: Purkinje cells, WM: White Matter.

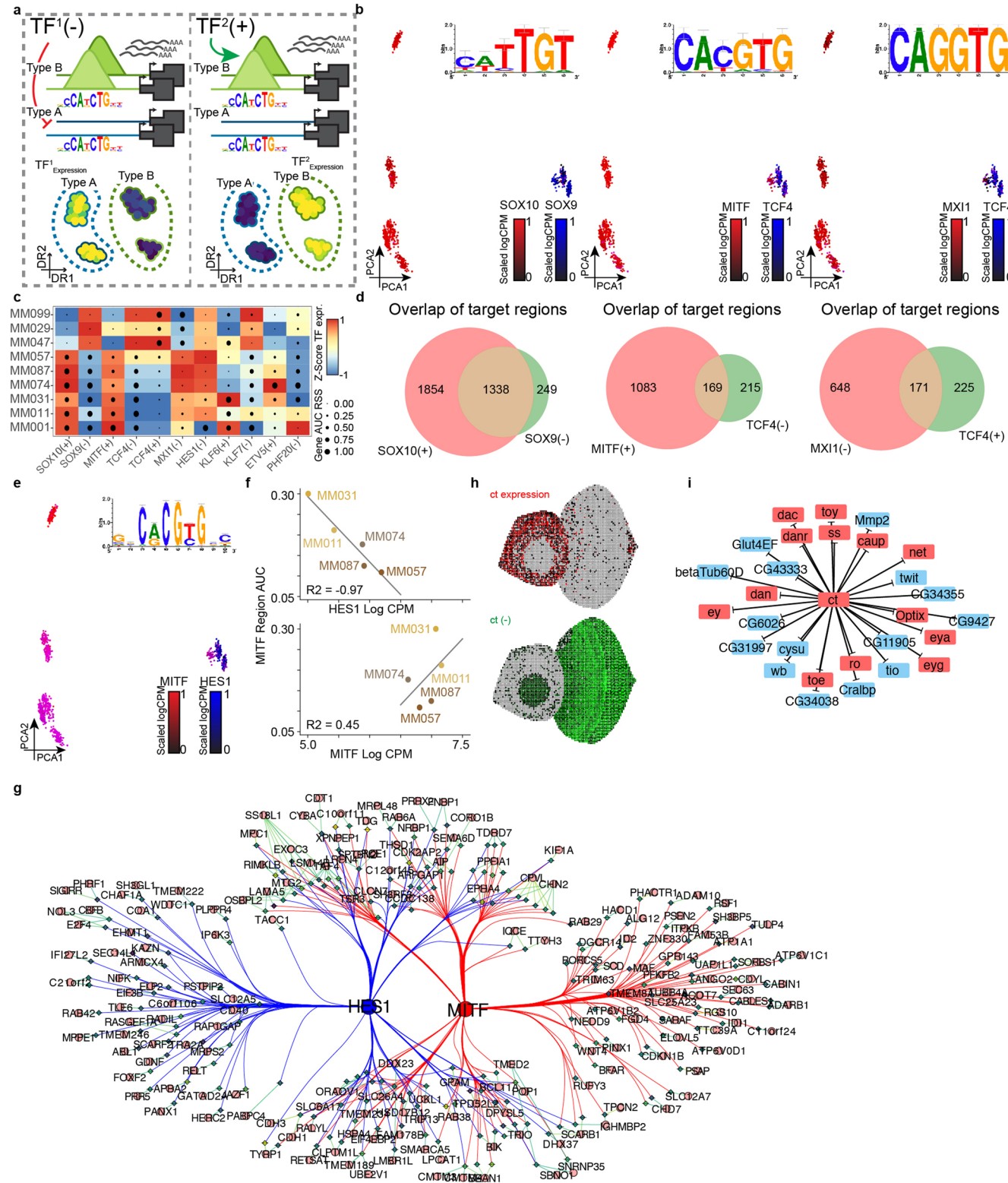

**Extended Data Fig. 10 | See next page for caption.**

**Extended Data Fig. 10 | Repressor predicitons of SCENIC+ in melanoma and eye-antennal disc. a**. TFs of the same family for which the expression is anti-correlated in a system can cause spurious repressor predictions. Scenario 1 (left): TF[1] is a potential repressor which is expressed in cell type A and actively closes chromatin in that cell type. Scenario 2 (right): TF[2] is a potential activator of the same TF family as TF[1] which is expressed in cell type B and opens the chromatin in that cell type. Both scenarios lead to the same gene expression and chromatin-accessibility measurements and can thus not be disentangled if both TF[1] and TF[2] are present in the same system. **b**. Principal-Component Analysis (PCA) projection of 936 pseudo mutli-ome cells based on cellular enrichment (AUC scores) of predicted target genes and regions from SCENIC+ eRegulons colored by gene expression. Shared motif used by the pair of TFs in each plot is shown on the top right. **c**. Heat map-dotplot showing TF expression of the eRegulon on a color scale and cell type specificity (RSS) of the eRegulon on a size scale.

**d**. Venn diagram showing overlap of predicted target regions of SOX10 and SOX9, MITF and TCF4; and MXI1 and TCF4. **e**. Principal-Component Analysis (PCA) projection of 936 pseudomutli-ome cells based on cellular enrichment (AUC scores) of predicted target genes and regions from SCENIC+ eRegulons colored by the expression of MITF and HES1. Shared motif used by the pair of TFs in each plot is shown on the top right. **f**. Log(CPM) expression of HES1 (top, x axis) and MITF (bottom, x axis) versus MITF target region AUC value (y axis). Line fit using linear regression, least squares method. **g**. Network showing subset of MITF and HES1 target regions. Diamonds represent regions circles represent genes and are color-coded by the average accessibility LogFC of corresponding regions in the melanocytic state. **h**. Virtual eye-antennal disc with 5,058 pseudocells colored by Ct expression and AUC values of the repressive Ct regulon. **i**. Targets of the Ct repressive regulon, showing in red targets that are transcription factors.

# Reporting Summary

## Statistics

For all statistical analyses, confirm that the following items are present in the figure legend, table legend, main text, or Methods section.

| n/a | Confirmed | |
|---|---|---|
| ☐ | ☒ | The exact sample size (*n*) for each experimental group/condition, given as a discrete number and unit of measurement |
| ☐ | ☒ | A statement on whether measurements were taken from distinct samples or whether the same sample was measured repeatedly |
| ☐ | ☒ | The statistical test(s) used AND whether they are one- or two-sided *Only common tests should be described solely by name; describe more complex techniques in the Methods section.* |
| ☐ | ☒ | A description of all covariates tested |
| ☐ | ☒ | A description of any assumptions or corrections, such as tests of normality and adjustment for multiple comparisons |
| ☐ | ☒ | A full description of the statistical parameters including central tendency (e.g. means) or other basic estimates (e.g. regression coefficient) AND variation (e.g. standard deviation) or associated estimates of uncertainty (e.g. confidence intervals) |
| ☐ | ☒ | For null hypothesis testing, the test statistic (e.g. *F*, *t*, *r*) with confidence intervals, effect sizes, degrees of freedom and *P* value noted *Give P values as exact values whenever suitable.* |
| ☐ | ☒ | For Bayesian analysis, information on the choice of priors and Markov chain Monte Carlo settings |
| ☒ | ☐ | For hierarchical and complex designs, identification of the appropriate level for tests and full reporting of outcomes |
| ☐ | ☒ | Estimates of effect sizes (e.g. Cohen's *d*, Pearson's *r*), indicating how they were calculated |

*Our web collection on statistics for biologists contains articles on many of the points above.*

## Software and code

Policy information about availability of computer code

| | |
|---|---|
| Data collection | Motifs from 29 collections were downloaded to build the SCENIC+ motif collection. A description of the collections and references are described in Supplementary Table 3. The SCENIC+ motif collection is available at https://resources.aertslab.org/cistarget/. |
| Data analysis | PycisTopic (v1.0.1.dev21+g8aa75d8) is available at https://github.com/aertslab/pycisTopic . Pycistarget (v1.0.1.dev17+gd2571bf) is available at https://github.com/aertslab/pycistarget. SCENIC+ (v0.1.dev373+geb5f14e) is available at https://github.com/aertslab/scenicplus. Detailed tutorials and documentation on the SCENIC+ workflow are available at scenicplus.readthedocs.io, while tutorials on pycisTopic and pycistarget (within the SCENIC+ workflow and as standalone packages) are available at pycisTopic.readthedocs.io and pycistarget.readthedocs.io, respectively. Code to generate custom cisTarget databases is available at https://github.com/aertslab/create_cisTarget_databases. Our implementation of Cluster-Buster is available at https://resources.aertslab.org/cistarget/programs/cbust. Notebooks to reproduce the analyses presented in this manuscript are available at https://github.com/aertslab/scenicplus_analyses. Analyses were performed using R 4.0.3 and Python 3.8. Other relevant software and versions: Precrec (v0.12.9), Scrublet (v0.2.3), harmonypy (v0.0.5), Arboreto (v0.1.6), GSEApy (v0.10.8), ctxcore (v0.1.2), umap (v0.5.2), fitsne (v1.2.1), Scikit-Learn (v0.24.2), loomxpy (v0.4.1), Kent (v2.7), networkx (v2.7.1), pyvis (v0.1.3.1), Cytoscape (v3.9.0), Seurat (v4.0.3), STAMP (v1.3), cisTopic (v2.1.0), Scanpy (v1.8.2), pybiomart (v0.2.0), pybigwig (v0.3.18), Signac (v1.3.0), MACS2 (v2.1.2.1), Cicero (v1.6.2), gimmemotifs (v0.17.1), MotifMatchR (v1.10.0), tispec (v0.99.0), UpsetR (v1.4.0), Juicer Tools (v2.13.05), DESeq2 (v1.28.1), cellrager-atac (v2.0.0), cellranger-arc (v2.0.0), bcftools (v1.11), Bowtie2 (v2.4.4), STAR (v2.7.9a) , fastq-mcf (v1.05), deepTools (v3.5.0), matplotlib (v3.5.2), HTseq (v0.9.1), VSN (v0.27.0), CrossMap (v0.6.0), pyGAM (v0.8.0), scVelo (v0.2.5), ScoMAP (v0.1.0), CellProfiler (v4.2.1) Tangram (v1.0.2), ImageJ (v2.3.0/1.53f) and Polylux tool plugin (v1.6.1). |

For manuscripts utilizing custom algorithms or software that are central to the research but not yet described in published literature, software must be made available to editors and reviewers. We strongly encourage code deposition in a community repository (e.g. GitHub). See the Nature Portfolio guidelines for submitting code & software for further information.

## Data

Policy information about availability of data

All manuscripts must include a data availability statement. This statement should provide the following information, where applicable:
- Accession codes, unique identifiers, or web links for publicly available datasets
- A description of any restrictions on data availability
- For clinical datasets or third party data, please ensure that the statement adheres to our policy

Data generated in this manuscript, namely scATAC-seq in melanoma cell lines, 10x multiome in the mouse cortex and scATAC-seq in the Drosophila eye disc are available in GEO (GSE210749). GRCh38.86 genome annotation used in this study is available on: https://ftp.ensembl.org/pub/release-86/gtf/homo_sapiens/Homo_sapiens.GRCh38.86.chr.gtf.gz, GRCh38 genome index used in this study is available on: https://cf.10xgenomics.com/supp/cell-arc/refdata-cellranger-arc-GRCh38-2020-A-2.0.0.tar.gz,  mm10 genome index used in this study is available on: https://cf.10xgenomics.com/supp/cell-arc/refdata-cellranger-arc-mm10-2020-A-2.0.0.tar.gz. Data from ENCODE Deeply Profiled Cell Lines was downloaded from https://www.encodeproject.org/,  including bulk RNA-seq and ATAC-seq for 8 cell lines, namely MCF7 (ENCFF136ANW and ENCFF772EFK, for RNA-seq and ATAC-seq, respectively), HepG2 (ENCFF660EXG and ENCFF239RGZ), PC3 (ENCFF874CFD and ENCFF516GDK), GM12878 (ENCFF626GVO and ENCFF415FEC), K562 (ENCFF833WFD and ENCFF512VEZ), Panc1 (ENCFF602HCV and ENCFF836WDC), IMR90 (ENCFF027FUC and ENCFF848XMR) and HCT116 (ENCFF766TYC and ENCFF724QHH); and Hi-C data on 5 of the cell lines (IMR90 (ENCFF685BLG), GM12878 (ENCFF053VBX), HCT116 (ENCFF750AOC), HepG2 (ENCFF020DPP) and K562(ENCFF080DPJ)). STARR-seq data was downloaded from ENCODE (ENCFF045TVA (K562), ENCFF047LDJ (HepG2), ENCFF428KHI (HCT116), ENCFF826BPU (MCF7)). bulk RNA-seq experiments upon perturbation in these cell lines and ChIP-seq data sets are described in Supplementary Table 1 and Supplementary Table 4, respectively. 10x multiome data on PBMC was downloaded from the 10x website. scRNA-seq data of baseline MM-lines and bulk RNA-seq data after SOX10 knockdown were downloaded from GEO (GSE134432). MITF, SOX10 and TFAP2A ChIP-seq data were downloaded from GEO (GSE61965 (MITF and SOX10) and GSE67555 (TFAP2A)).  SNARE-seq2 data on the human cortex was downloaded from Bakken et al., 202160. scATAC-seq and scRNA-seq data from the Drosophila eye-antennal disc were downloaded from GEO (GSE115476). 10x Visium data and 10x single cell multiome data from the human cerebellum was downloaded from the 10x website. All analyses can be explored in SCope (https://scope.aertslab.org/#/scenic-v2) and UCSC in the following sessions: PBMC (https://genome-euro.ucsc.edu/s/Seppe%20De%20Winter/scenicplus_pbmc), ENCODE cell lines (https://genome.ucsc.edu/s/cbravo/SCENIC%2B_DPCL), melanoma (http://genome-euro.ucsc.edu/s/Seppe%20De%20Winter/scenicplus_mix_melanoma), mouse and human cortex (https://genome-euro.ucsc.edu/s/cbravo/SCENIC%2B_Cortex),  eye-antennal disc (http://genome-euro.ucsc.edu/s/cbravo/SCENIC%2B_EAD) and human cerebellum (https://genome-euro.ucsc.edu/s/cbravo/SCENIC%2B_cerebellum). The SCENIC+ motif collection is available at: https://resources.aertslab.org/cistarget/motif_collections.

## Human research participants

Policy information about studies involving human research participants and Sex and Gender in Research.

| | |
|---|---|
| Reporting on sex and gender | NA |
| Population characteristics | NA |
| Recruitment | NA |
| Ethics oversight | NA |

Note that full information on the approval of the study protocol must also be provided in the manuscript.

# Field-specific reporting

Please select the one below that is the best fit for your research. If you are not sure, read the appropriate sections before making your selection.

☒ Life sciences    ☐ Behavioural & social sciences    ☐ Ecological, evolutionary & environmental sciences

For a reference copy of the document with all sections, see nature.com/documents/nr-reporting-summary-flat.pdf

# Life sciences study design

All studies must disclose on these points even when the disclosure is negative.

| | |
|---|---|
| Sample size | No statistical method was used to predetermine sample size. Sample sizes were chosen based on the maximum amount of samples that were available for each analysis (for example, all available ChIP-seq datasets in ENCODE on the Deeply Profiled Cell Lines, used in this study, were used). For each analysis the sample size was sufficient to derive statistically meaningful results passing multiple testing procedures. |
| Data exclusions | Single cells with low quality on the scRNA-seq (e.g. low number of reads, high percentage of mitochondrial reads) and/or the scATAC-seq data (e.g. low TSS enrichment, FRiP, number of fragments) were excluded, as described in the methods section. |
| Replication | We provide a Github repository with code to replicate the analyses at: https://github.com/aertslab/scenicplus_analyses |
| Randomization | Not applicable. Each analysis was independent. |

| Blinding | Blinding was not relevant to our study as we report an analysis software as main finding. |

# Reporting for specific materials, systems and methods

We require information from authors about some types of materials, experimental systems and methods used in many studies. Here, indicate whether each material, system or method listed is relevant to your study. If you are not sure if a list item applies to your research, read the appropriate section before selecting a response.

## Materials & experimental systems

| n/a | Involved in the study |
|-----|----------------------|
| ☒ | ☐ Antibodies |
| ☐ | ☒ Eukaryotic cell lines |
| ☒ | ☐ Palaeontology and archaeology |
| ☐ | ☒ Animals and other organisms |
| ☒ | ☐ Clinical data |
| ☒ | ☐ Dual use research of concern |

## Methods

| n/a | Involved in the study |
|-----|----------------------|
| ☒ | ☐ ChIP-seq |
| ☒ | ☐ Flow cytometry |
| ☒ | ☐ MRI-based neuroimaging |

## Eukaryotic cell lines

Policy information about cell lines and Sex and Gender in Research

| Cell line source(s) | Patient-derived melanoma cell lines used in this study were obtained from the laboratory of Pr. Ghanem-Elias Ghanem (Istitut Jules Bordet, ULB, Belgium). |
| Authentication | The identity of each line has been determined using RNA-seq and ATAC-seq. |
| Mycoplasma contamination | Cell cultures used for experiments providing data to this study were tested for myoplasm contamination and were found to be negative. |
| Commonly misidentified lines (See ICLAC register) | No commonly misidentified cell lines have been used. |

## Animals and other research organisms

Policy information about studies involving animals; ARRIVE guidelines recommended for reporting animal research, and Sex and Gender in Research

| Laboratory animals | Mice were maintained in a specific pathogen-free facility under standard housing conditions (temperature 20-24C and humidity 45-65%) with continuous access to food and water. Mice used in the study were 57 days old and were maintained on 14 hr light, 10 hr dark light cycle from 7 to 21 hr. In this study, cortical brain tissue from male P57 BL/6Jax was used. Animals were anesthetized with isoflurane, and decapitated. Cortices were collected and immediately snap frozen in liquid nitrogen. |
| Wild animals | No wild animals were used in this study. |
| Reporting on sex | Sex is not relevant for this study as we report an analysis software as main finding, therefore sex was not considered in the study design. The findings in this study apply to only one sex (male mice). Sex of mice was determined visually based on anogenital distance and pigmentation. |
| Field-collected samples | No field collected samples were used in this study. |
| Ethics oversight | All animal experiments were conducted according to the KU Leuven ethical guidelines and approved by the KU Leuven Ethical Committee for Animal Experimentation (approved protocol numbers ECD P007/2021). |

Note that full information on the approval of the study protocol must also be provided in the manuscript.

