## [Peer Review File · Nature Methods]

Peer Review Information

Manuscript Title: SCENIC+: single-cell multiomic inference of enhancers and gene regulatory networks

Corresponding author name(s): Stein Aerts

Editorial Notes:

Reviewer Comments & Decisions:

Decision Letter, initial version:
--

31st Oct 2022

Dear Professor Aerts,

Your Article, "SCENIC+: single-cell multiomic inference of enhancers and gene regulatory networks", has now been seen by 3 reviewers. As you will see from their comments below, although the reviewers find your work of considerable potential interest, they have raised a number of concerns. We are interested in the possibility of publishing your paper in Nature Methods, but would like to consider your response to these concerns before we reach a final decision on publication.

We therefore invite you to revise your manuscript to address these concerns, with additional analysis, benchmarking, validation and other changes.

* include a point-by-point response to the reviewers and to any editorial suggestions

* please underline/highlight any additions to the text or areas with other significant changes to facilitate review of the revised manuscript

- * address the points listed described below to conform to our open science requirements
- * ensure it complies with our general format requirements as set out in our guide to authors at www.nature.com/naturemethods
- * resubmit all the necessary files electronically by using the link below to access your home page

[REDACTED]

We hope to receive your revised paper within eight weeks. We are very aware of the difficulties caused by the COVID-19 pandemic to the community. If you cannot send it within this time, please let us know. In this event, we will still be happy to reconsider your paper at a later date so long as nothing similar has been accepted for publication at Nature Methods or published elsewhere.

OPEN SCIENCE REQUIREMENTS

REPORTING SUMMARY AND EDITORIAL POLICY CHECKLISTS

DATA AVAILABILITY

All novel DNA and RNA sequencing data, protein sequences, genetic polymorphisms, linked genotype and phenotype data, gene expression data, macromolecular structures, and proteomics data must be deposited in a publicly accessible database, and accession codes and associated hyperlinks must be provided in the "Data Availability" section.

CODE AVAILABILITY

Please include a "Code Availability" subsection in the Online Methods which details how your custom code is made available. Only in rare cases (where code is not central to the main conclusions of the paper) is the statement "available upon request" allowed (and reasons should be specified).

For more information on our code sharing policy and requirements, please see: <https://www.nature.com/nature-research/editorial-policies/reporting-standards#availability-of-computer-code>

MATERIALS AVAILABILITY

Authors reporting new chemical compounds must provide chemical structure, synthesis and

characterization details. Authors reporting mutant strains and cell lines are strongly encouraged to use established public repositories.

ORCID

Sincerely,

Lin Tang, PhD
Senior Editor
Nature Methods

Reviewers' Comments:

Reviewer #1:

Remarks to the Author:

The manuscript describes SCENIC+, a Python package for gene regulatory network analysis using single cell profiles of gene expression and chromatin accessibility. This multimodal data has the potential to deepen our understanding of gene regulation. The development of software analysis tools for this type of data will be of interest to the research community and the software makes use of broadly adopted data structures which will allow for integration with related Python packages.

1. Overall, the paper shows many methods and results that are very similar to those presented in previous work by the same group. Much of the new work appears to be analysis which are suggestive of interesting biological findings, but without the necessary follow-up experiments to provide solid scientific evidence. The software development for this project seems to make some useful algorithms more readily available to the community. Nevertheless, by covering such a broad range of topics, given today's high level of activity in single cell algorithm development, the manuscript does not appear to be particularly innovative and convincing in any particular aspect. The core of the paper is on gene regulatory analysis using multimodal single cell data. In revisions of this manuscript the focus might be kept on this aspect, taking care to explicitly present the modeling assumptions along with

unbiased evaluations of these assumptions, including sound benchmarking standards. It is unfortunate that many of the examples presented in the paper are not based on true multimodal data, but are generated using single cell RNA-seq and ATAC-seq data from separate experiments. This suggests there is no additional value in multiomic experiments instead of experiments on separate modalities. The manuscript might include a comment about the value (or not) of true multiomic data.

2. SCENIC+ provides a Python module, `pycisTopic` for inference of transcription factor (TF) binding sites from TF motif data. The module facilitates motif-based analysis using a compendium of motifs collected from a large number of individual motif resources along with algorithms for inferring TF binding using this collection. In particular, the suggested approach is based on Cluster Buster (Frith et al, Nucleic Acids Research, 2003). The Aerts lab has published multiple papers based on a similar strategy of collecting and integrating TF binding motifs, for example Janky et al. PLoS Comput. Biol., 2014, and the paper describing SCENIC (Aibar, Nature Methods, 2017). The area under recovery curve appears to be the same as presented in the first SCENIC paper. It seems the contribution in the current manuscript is mainly in the increased scale of the motif collection and perhaps the reimplementing of old software into a new Python package. A clearer description of what is novel in this work would be helpful.

3. Although the motif compendium might be a useful resource, the resulting motif compendium itself does not appear to have been made available online. Instead results of mappings to various selected regions of the genome are provided, which reduces the value of this work. In addition, the reported reimplementing of Cluster-Buster does not appear to be available as source code, but as some sort of undefined binary file. Perhaps there are other links to the motif collection and source code which were not clear on the website?

4. The third step of SCENIC+ is TF to target gene prediction, making use of chromatin accessibility data. This part seems important and would benefit from a better description. It appears that this part is also reliant on previous approaches developed in the Aerts lab but the description does not clearly specify the new developments. There is also work by other labs that do a similar type of analysis (for example Duren, Z. et al Proc. Natl. Acad. Sci. U. S. A, 2018). What are the advantages and disadvantages of the SCENIC+ approach relative to other methods that try to infer GRNs using gene expression and chromatin accessibility data?

5. The illustration of SCENIC+ on PBMC multiome data lacks a description of how the method adds insights into the system that would not be observed through other approaches. For example, the master regulators of B cells, T cells, dendritic cells and monocytes are identified by SCENIC+, Would these regulators not be identifiable by other tools such as chromVAR, Signac, or the original SCENIC software?

6. In the validation of SCENIC+ on ENCODE data the first task presented is the prediction of regulatory TFs recovered by several methods. In the main text it would be good to define what recovering TFs means. It could mean any of the following: TFs that are expressed in a certain cell type, TFs that are actively regulating genes in a cell type, TFs that define chromatin accessibility, TFs that are necessary to define the cell states, or TFs that were chosen by ENCODE for unknown reasons. The last is not meaningful as the TF selection might be biased towards the research interests of the investigators and the availability of ChIP antibodies.

7. In Fig 3 numbers of TF, regions etc are shown. It would be more informative to show performance

metrics such as precision and recall than to show these numbers. For example, one could trivially show a method that always predicts all the TFs and all regions, but this would be meaningless.

8. In the assessment of the quality of the predicted target regions it appears there is a component of DNA sequence analysis used in the definition of the "silver" standard. This benchmarking approach is highly problematic for several reasons. First, there is a degree of circularity between benchmark and analysis. An analysis method that aligns well with the use of motifs in the benchmark generation will tend to do well. Second, this approach suggests that the TF binding sites that can be recognized to bind directly to DNA are of greater functional importance than the rest. The manuscript does not provide any evidence to support this claim. Third, the inability to make accurate predictions of TF binding based on sequence might be limited because current characterizations of DNA binding are not adequate. In this regard it might be helpful to contrast with approaches such as that proposed by Avsec, Žiga, et al. *Nature Genetics*, 2021. The most appropriate benchmarking comparison for this part is the ENCODE-DREAM in vivo Transcription Factor Binding Site Prediction Challenge <https://www.synapse.org/#!Synapse:syn6131484/wiki/402026>. This benchmark would be more objective and enable a direct comparison with the results obtained by the best performing DREAM challenge methods.

9. In the evaluation of the TF target gene by knockdown experiment it would be helpful to provide more information, including the TF, the true TF-regulated gene set, the prediction and the summary result. In addition, if the TF binding has been assayed by ENCODE it would be of interest to compare the performance of SCENIC+ using motifs with TF ChIP-seq data. Some insights into this evaluation would also be of value, for example which types of TFs are well predicted and what aspects of SCENIC+ were most important in making these predictions.

10. Although the paper does some GRN benchmarking, many of the benchmarks are not entirely convincing. It would be of interest to adapt some of the benchmarks proposed by Pratapa et al *Nature Methods*, 2020 to multomic data.

11. One of the analyses shown by Pratepa et al relates to the number of stable states consistent with the gene regulatory networks. Can it be shown that the numbers of stable states derived from the networks presented in the manuscript are consistent with the observed biological systems?

12. The melanoma phenotype switching is derived from an earlier scRNA-seq study by the same group and new scATAC-seq data for the same 10 melanoma cell lines were generated for the current study. Although the manuscript describes many TFs discovered by SCENIC+, many of these were already discovered and reported in the earlier study and the manuscript does not clearly demonstrate what was novel in the SCENIC+ finding and what was the important difference between the SCENIC+ findings and SCENIC. In addition, it would be helpful to include a comparison with methods such as chromVAR and Signac for this section.

13. The analysis with CellOracle was not particularly informative as it did not make the case for SCENIC+, which seems to be topic of the manuscript.

14. The section on conservation between human and mouse cortex also seems tangential to the main subject of the paper. The results presented in this section could reflect aspects of biology, or certain biases in the analysis that are conserved between human and mouse. Separating the two might require some deeper analysis, which would go beyond the scope of the current paper.

15. GRN velocity is a concept which might be worth expanding on. Related ideas and methods are reported by Ma et al, Cell, 2020 and ref 69 (Li et al). and it would be of interest to compare the results produced by the respective methods. The paper does not present algorithms that have been developed specifically for multiomic data latent space analysis (for example Gong et al, Genome Biology, 2021, Minoura et al. Cell Reports Methods, 2022; Lynch et al, Nature Methods, 2022; Lin et al. scJoint: Nature Biotech, 2022). As the pseudotime analysis depends on the latent space the trajectory inference, to what degree do the results of the GRN velocity change depend on these upstream processing steps?

16. In the fly retina analysis of GRN velocity single cell ATAC-seq and RNA-seq data are not generated from the same cells using multimodal analysis. This seems problematic as there might be differences between the gene expression and the chromatin states of the cells which would be hidden by the matching procedure. Again, no comparison is made with other methods. It is not clear that the GRN velocity approach was important in this analysis, or whether similar conclusions could be reached using alternative methods.

17. In the spatial GRN analysis, true multiomic data was once again not used. In this section GRNs are inferred then mapped onto spatial data. This section does seem to add much to the manuscript.

Reviewer #2:

Remarks to the Author:

González-Blas and colleagues propose a novel computational method (SCENIC+) for inference of enhancer-based gene regulatory networks (eGRNs) from multimodal (scRNA- and scATAC-seq) single cell sequencing. For this, they leverage expertise and previous methods from the own group such as the use of cisTopic for denoising sparse single cell ATAC-seq data, a meta-database of TF motifs (Motif Collection database/cisTargets), methods to characterize cis-regulatory regions (pyCisTarget & DEM) and association of TF to enhancers to genes (based on Boosting trees/Arboreto). These are all combined together as SCENIC+. Some of the individual components (DEM, Motif Collection database, Velocity) are novel. Some components were improved, i.e. cisTopic has a faster implementation/estimation algorithms. This is altogether a very important framework for inference of regulatory networks. A few recent competing methods (FigR, GRaNIIE, Inferelator, scMega) provide equivalent core features (estimation eGRNs), while SCENIC+ includes additional aspects as methods for insilico perturbation and differentiation analysis. The method is evaluated in distinct data sets including ENCODE chromatin data (used for benchmarking), PMBC multimode data, melanoma cell lines, human/mouse cortex cells, fly eye-antennal disc and mammalian brain with mostly convincing results.

The method is overall complex including many sub-methods and options. Some important details were missing, and descriptions of some methods were hard to follow. As a judge this as a method manuscript, focus could be given in method description/benchmarking, and less in the interesting but mostly anecdotal data analyses. For example, estimation of differentiation velocities and spatial mapping are interesting, but there is lack of methods details and rigorous evaluation. This makes it hard to conclude their value. Overall, this manuscript would benefit from focus on main contributions.

1. The reported improvements on cisTopic are welcomed (efficient algorithm and additional model

selection approaches), but some important method descriptions are missing. For example, how is a final topic decided from the four available model selection criteria (lines 824-834 & Fig. S2b)? Authors should also give more details on how to choose the topic binarization (lines 843-851). For example, in which type of analysis "balanced region sets" are more desirable? Finally, it is unclear if topics are altogether used by SCENIC+, as analysis seen to focus on cell types/DARs. Further explanation and a table enumerating how these (and other main parameters /methodological choices) were set per data set would be helpful.

2. pyCisTarget is rather complex part of SCENIC+. It consists of 3 main components: a novel TF motif db, methods for motif enrichment (DEM, homer and cisTarget) and a cisTarget database.

2.1 The nature of cisTarget database is unclear and hard to follow. In my understanding, this is a matrix encoding region vs. motif enrichment scores. How are the regions defined? I Aren't this data set specific? Can regions derived in a data set (PBMCs) be used in other data (cortex cells)? Or do authors recommend using pre-defined SCREEN or whole genome regions? This should be clarified (and again choices for each analyzed data explained).

2.2. Authors mention the detection of CRMs (or cluster of motifs) using Cluster-Buster (line 131), whoever scores seem to be based on individual motifs (see point 2.1 and figure 1d). This needs clarification.

2.3 In their evaluation (Fig. 1F) CisTarget outperforms DEM. So, it is unclear why DEM is need in SCENIC+.

2.4 The benchmarking of distinct motif data bases and representation is an interesting aspect (Fig. 1f and Sup. S3). Fig. 1F has strong results but underlying experiments are unclear. It is hard to follow some of the analysis in Sup. S3, as for example several of the terms [(u),(c),(p) in panel h) are not clearly defined. Since, this database is a SCENIC+ novel aspect, more details on the benchmarking are needed.

3. Another major contribution of SCENIC+ is the creation of the eGRNs. Authors should formalize or schematized the description of eRegulon (line 1016-1022), which is unclear. Also, some important aspects (need for a specificity score - line 1037) is not justified in the manuscript.

4. The extensive benchmarking (Fig. 3) is welcome, but some aspects are difficult to. What does the recovery of TFs (F3c & S5a) mean? Are these true positives? What is the set of positives here? What are "marker gene TFs" (line 209)? Altogether, CellOracle (or Pando according to fig. 3) does perform better in absolute TF numbers. The justification is based on lack of cell specificity of CellOracle, whoever shouldn't the benchmarking approach account for cell specificity? True positives are only TFs predicted in the correct cellular context, i.e. cell type used for the ChIP-seq experiment? Authors should improve their benchmarking approach to account of cell specificity of TFs.

5. Benchmarking results on TF-region, Region-Gene and TF-gene are mostly convincing. Authors should discuss however the best precision of Granie of the TF-region problem. It is also unclear why SCENIC is evaluate with Rho and Importance in figure 3f (and Rho is also not defined anywhere). This should be justified. Also, the interpretation of the NES vs. FC plots in panel 3g is unclear. Why has STAT5 a negative NES and HOXB9 a positive after their KO? Repressor TFs? This should be explained in the text.

6. The in-silico perturbation analysis is an interesting approach. Here, authors could reinforce the novelty of the proposed method in relation to CellOracle (linear vs. non-linear?). The need for several interactions needs to be better justified in the main text (predicted TF expression is used in interactions?). Also, can the model predict which TFs have best perturbation effects (or why only SOX10 and ZEB1 were chosen)? Could this approach be benchmarked more systematically by using the ENCODE data (explored in Fig. 3)?

7. The mouse/human eGRN is neat but descriptive. It is unclear if any new finding comes from this analysis.

8. The GRN velocity is an interesting (and novel) contribution. However, more details are needed. For example, I miss how the trajectory analysis was performed in the oligodendrocytes data and how the sub-trajectories ("OPC to NFOL", "NFOL Maturation" and NFOL to OL") are defined. Authors need to describe this. Velocity fields of Olig2 and Bcl6 point towards the center of the OPC cluster. What does this mean? The drosophila eye analysis is more complete/convincing, possibly due to a better capture of cells at distinct differentiation stages.

9. How is the differentiation force defined? Method (line 1716) describes the use of a "penalization curve", but no further details are given. In the scheme (Fig. 6a), red curves anticipate the blue curves but, in the examples/real data (Fig. 6g), we also observe the opposite. How does the model deal with this? Are only positive values considered? What would the negative vectors mean? Authors need to both formalize the method and include a benchmarking more like analysis of this novel methodology.

10. The spatial analysis of eGRNs is interesting, but its value is unclear (besides visualization). For example, if one plots the cellular deconvolution results of the cell types with LHX7, SOX9 specific TF activity, would one not obtain similar results as shown in panel Fig. 7C? What novel information can be gained here?

11. We were able to successfully install/run the method in linux. In a mac, we encountered issues with package dependencies (annoy package).

Minor points:

Line 66 - The statement that scATAC-seq is similar ChIP-seq accuracy is based on pre-print (23). Authors should tune this down.

Line 73 - Authors should also cite previous works introducing some of these analyses, i.e. no reference to Cicero, which first explored peak-gene links in scATAC-seq.

Line 103 - Mixture of HMMs have been previously explored for a similar problem. Please cite 10.1093/bioinformatics/bti1001) (and discuss any relevant distinction).

Line 104-108 - This phrase is complex and content rich (i.e. what is leading edge analysis?). Authors should improve this.

Line 112 - From the method description I got the impression that GBMs are used to find relations (and Pearson is only used for finding signals), but the main text indicates correlation is also important. Please, clarify.

Line 197 – Are these “TF” ChIP-seq?

Line 259 - How are TF ranked? By increasing or decreasing number of bindings?

Line 998 –The phrase regarding TF-to-TF regulation is unclear. Please clarify.

Line 1104 - The definition of archetypes is unclear here.

Fig. 4e – Colors of the boxplot are difficult to discern.

Supplement – Several links in the supplement pdf are not working. Please revise.

Supplementary Figure 5 – What is R and L (after SCENIC + (C)).

Reviewer #3:

Remarks to the Author:

Manuscript by Gonz. Iez-Blas et al described a valuable computation tool for inferring enhancers and gene regulatory networks based on single cell multiomic datasets. The tool overall can be very useful for mining large scale single cell datasets and providing great insights on how cell types are regulated via binding of transcription factor and their targets. The database with curation of 30K+ motifs and motif-TF annotation is an incredible resource for the community. Overall, the authors did a great job of presenting this remarkable tool and showcase various interesting applications. On other hands, some improvements are necessarily to enhance usability of this tool and presentation of this manuscript.

Here is list of major key points regarding the Manuscript

- PycisTopic is proposed as a new tool for both enhancer discovery and co-accessibility of enhancers, which is also possible via other popular methods including ArchR. Since these early processing steps play a pivotal role in the final eGRN it would be useful to benchmark pycisTopic against current methods. Essentially an analysis like Figure 1f for enhancer discovery would be informative to the reader/user.
- The illustration of SCENIC+ on PBMC data captures 40% of genes with a proximal enhancer that regulates expression. Recent studies have used HiC to look at the 3D chromatin organization for PBMC populations including Monocytes. Using the HiC data as an external validation for region-gene links should be done.
- Limited correlation of HiC interactions in the 5 cell lines as shown in Figure 3f as well as an overall higher number of region-gene links could point towards more false-positive interactions being identified. How does SCENIC+ address the presence of noisy interactions in the eGRN building process? For example, do multiple models (including Gradient Boosting Machine regression) agree on region-to-gene importance or binary links? Comparison of different regression models to calculate TF-to-gene and region-to-gene would be worthwhile.
- It is unclear how scalable the method is to number of cells and number of clusters. It will be nice to present a complexity analysis and running time analysis.
- It is unclear how sensitive this tool is to analyze fine resolution cell types. How many cluster-specific peaks are needed to identify cluster specific eRegulons? Some power analysis can be very helpful here.
- The paper focused a lot on the confirmative analysis, showing previously categorized regulons, but very shy at presenting novel findings. I encourage the authors showcase a few, even without experimental validation.
- The tool cluster DNA motifs based on PWM similarity. But single base difference in motifs can be critical in determining binding affinities and targets for paralogs within the same transcription factor family. How did the authors address such subtle differences?

Here is list of major key points regarding the Tutorial

1. The tutorial is not hard to follow for the PbmC example but can still be challenging to apply on new datasets. It would be nice to separate the tutorial into two key components: 1. Preprocessing steps, and 2. SCENIC+ modeling. Current tutorial covers extensively on preprocessing steps but does not provide enough depths for SCENIC+ modeling.
2. Many users in the community already get used to tools like snapATAC/ArchR. It would be nice to build a convenient interface to interact with these tools. It would also be nice to provide support for commonly used file format, such as h5ad and Seurat objects, to make the tools easier to use.

Minor points:

Most figures are too packed. The eGRN plots such as Figure 2g look cool, but it is difficult to interpret. It might not be worthwhile to show more than once. The authors need to pick and choose materials more carefully to present in the main figures and make sure all labels are legible.

Author Rebuttal to Initial comments

Reviewer #1:

Remarks to the Author:

The manuscript describes SCENIC+, a Python package for gene regulatory network analysis using single cell profiles of gene expression and chromatin accessibility. This multimodal data has the potential to deepen our understanding of gene regulation. The development of software analysis tools for this type of data will be of interest to the research community and the software makes use of broadly adopted data structures which will allow for integration with related Python packages.

We thank the reviewer for critically assessing our manuscript. We have included additional benchmarking analyses in the revised version and better highlighted the novelty of this work.

1. Overall, the paper shows many methods and results that are very similar to those presented in previous work by the same group. Much of the new work appears to be analysis which are suggestive of interesting biological findings, but without the necessary follow-up experiments to provide solid scientific evidence. The software development for this project seems to make some useful algorithms more readily available to the community. Nevertheless, by covering such as broad range of topics, given today's high level of

activity in single cell algorithm development, the manuscript does not appear to be particularly innovative and convincing in any particular aspect. The core of the paper is on gene regulatory analysis using multimodal single cell data. In revisions of this manuscript the focus might be kept on this aspect, taking care to explicitly present the modeling assumptions along with unbiased evaluations of these assumptions, including sound benchmarking standards. It is unfortunate that many of the examples presented in the paper are not based on true multimodal data, but are generated using single cell RNA-seq and ATAC-seq data from separate experiments. This suggests there is no additional value in multiomic experiments instead of experiments on separate modalities. The manuscript might include a comment about the value (or not) of true multiomic data.

We believe our manuscript introduces several novelties that can be of interest:

1. SCENIC+ is among the first methods to derive eGRNs from single-cell multiome data. Note that only 2 other methods have been published in a journal (Pando and FigR), which we describe have shortcomings compared to SCENIC+. The other two methods we included in our benchmark (GRaNIE and CellOracle) are only available as a preprint without peer review. In addition, all four other methods have only been tested on one or two data sets, while in this work we apply SCENIC+ across species (human, mouse, fly) and tissues (e.g. brain, cancer, cell lines).
2. We curated the largest collection of transcription factor motifs, with more than 40,000 position weight matrices, covering more than 1500 human TFs. By clustering this collection with a new two-step clustering strategy, we derive sets of motifs per transcription factor. We then show that maintaining this diversity, by the use of hidden markov models, greatly outperforms the use of ‘archetype’ motifs (whereby each TF has only one consensus motif). We also make this entire collection freely available at https://resources.aertslab.org/cistarget/motif_collections/v10nr_clust_public/.
3. We developed a new Python package for cisTopic (Bravo et al., Nat Methods 2019), that scales to very large data sets and contains a series of new functionalities. pycisTopic can also

- be used separately to analyze scATAC-seq data. In this version of this manuscript, we compared pycisTopic against other state-of-the-art methods, namely ArchR and Signac. We show that pycisTopic is more robust to data sparsity and limited sample sizes, and that regulatory topics recapitulate better functional enhancers compared to the current standard, which relies on the calculation of Differentially Accessible Regions (DARs).
4. We developed a new Python package for motif discovery, called pycisTarget, that allows the user to choose among three different motif enrichment methods, including the widely used Homer method. PycisTarget implements our cisTarget algorithm in Python for easy integration with the SCENIC+ workflow. In addition, we have developed a new algorithm, called Differentially Enriched Motifs (DEM), to identify motifs differentially enriched between sets of regions. We include now a case study to showcase DEM, in which we identify partner TFs of SOXE factors in melanoma, oligodendrocytes and astrocytes.
 5. While SCENIC+ reuses previously existing algorithms for TF-gene and region-gene inference, namely Arboreto, it includes novel regulon binarization, target prioritization and visualization approaches. Arboreto has been already shown to be the best performer for predicting TF-gene relationships, and in this version of the manuscript, we show that Arboreto is also the best approach to infer region-gene links compared to other regression methods. We showcase our new target prioritization strategy in Fig 3, which we believe will become a valuable asset to explore the regulatory genome and select enhancers for further testing.
 6. We have designed and performed the first benchmark for eGRN inference methods, which has been extensively expanded in this version. We believe this benchmark can serve as a baseline to benchmark new methods and new models that may include additional data for eGRN inference.
 7. We introduce the new concept of GRN velocity. To our knowledge, this is the first approach that exploits eGRNs and the lag between TF expression and expression of the target genes to predict the directionality of a cell in a differentiation trajectory. In this version we include a

comparison with other velocity methods. scVelo measures the ratio of spliced and unspliced mRNA of a gene, while MultiVelo integrates the accessibility around the gene locus in the model. Yet, none of these approaches incorporates TF input, and overall only provide global directionality values (one vector per cell), while GRN velocity assesses TF-specific directionality (one vector per cell and TF). We have included a comparison between our GRN velocity and these velocity methods in the eye disc case study to show that they provide different insights. In addition, this approach permits to identify *Meis1* as a novel (and conserved) master regulator of oligodendrocyte differentiation.

8. To our knowledge, this is the first study in which eGRNs inferred from single-cell multiomics data (in which not only genes, but also target regions and target genes can be compared) are compared across species. We found 60 TFs involved in cell-type specific eGRNs conserved between the species, out of which 8 have not been reported before. We believe this resource serves as a proof-of-concept study and could be further exploited to assess differential regulatory target conservation between TFs, or to prioritize conserved sequences to develop cell-type specific vectors.

We agree that the question about using multimodal data as input for GRN inference, compared to matched scATAC-seq and scRNA-seq from the same sample is relevant for our work. We already provided SCENIC+ analysis of four ‘true’ multiome data sets (PBMC, mouse cortex, human cortex and human cerebellum) and three paired scATAC-seq and scRNA-seq data sets (ENCODE cell lines simulated data, melanoma cell lines, and *Drosophila* eye-antennal disc). We consider the possibility to use SCENIC+ on both multiome and paired data valuable for the community, since multiome data are not always available (and are sometimes of lower quality) compared to individually obtained modalities that can be paired.

As suggested, we now provide additional comments about their respective values (Fig S7), as well as a new in-depth comparison between multiome and matched data. To specifically test whether eGRN inference is affected depending on whether data is derived from multiome experiments or paired data from independent experiments, we have now run SCENIC+ using as input our in-house multiome mouse

cortex data and using the two layers of the multiome as independent paired datasets from the same tissue. We evaluated four different strategies to perform the computational pairing of the ATAC and RNA modalities. We particularly tested the pairing methods from FigR (Matched-1), SCENIC+ (Matched-2) and ScoMAP (Matched-3). As control, we also created a data set with random pairing between cells (i.e., not taking cell types/states into account, named Random). Technically, FigR (Matched-1) relies on Seurat/Signac’s CCA integration, which allows cell pairing without the need of independently annotating the cells in each layer. SCENIC+ (Matched-2) makes use of the cell labelling to match cells of the same cell type between the layers. ScoMAP (Matched-3) uses the same approach as SCENIC+, but it allows mapping cells based on pseudotime/pseudospacial order within a cell type. We used this technique for the cortical layers and oligodendrocytes. When comparing the distances between paired cells across modalities, we observe that the three pairing methods result in pairing between close neighbors, in comparison with the random mapping (Fig R1.1).

Figure R1.1. Comparison of methods to pair independent scATAC-seq and scRNA-seq data sets. a. SCENIC+ gene based UMAP for the mouse cortex multiome data set (19,485 cells). **b.** SCENIC+ gene based UMAP showing 500 random pairs between cells. Blue and yellow represent the cells paired by their scATAC-seq and scRNA-seq profiles, respectively. **c.** Distribution of the distances between the paired cells in the PCA space.

After the ATAC-RNA matching we ran SCENIC+ on the paired data sets and compared the predicted regulons/TFs, target genes, and target regions; and we tested whether the predicted networks could recover the different cell states. Using random pairing we found no regulons, showing that transcriptomes need to match with the corresponding underlying chromatin accessibility for correct eGRN inference. In terms of transcription factors, and after selecting high quality regulons (with at least 35 target genes and a correlation > 0.4 between the gene-based and region-based AUC values), we identified 73 TFs that were found either using the true multiome or any of the data sets computationally paired as input for SCENIC+ (Fig R1.2, Fig R1.3).

Figure R1.2. UpsetR plot showing the overlap between the TFs recovered using the true multiome or data paired with different methods as input for SCENIC+.

Overall, the different matching techniques recovered 73%-77% of the regulons inferred using the true multiome data, with most of the relevant cell-type specific TFs found by all the methods (Fig R1.3). Regulons uniquely found by one or a couple of the methods mostly correspond to general transcription factors, such as Nfya, Klf, or Sp factors.

Figure R1.3. Comparison of TFs recovered using the true multiome or paired data as input for SCENIC+. Heatmap showing whether a TF is found across different types of input data (present: green; absent: red). Key cell type specific TFs are highlighted in bold, with the matching color of the cell type in which they are present (light pink: oligodendrocyte precursors, pink: oligodendrocytes, navy blue: excitatory neurons, green: microglia, astrocytes: purple, brown: inhibitory neurons).

We next compared the predicted target regions and target genes between the true multiome and matched data as input for SCENIC+. Overall, we found a good correspondence between all approaches, with an average Jaccard index of 0.5 and 0.6 for the predicted target regions and target genes for shared TFs, respectively. While we found that some regulons also overlap with other regulons (such as Fli1, Runx1 and Spi1), this is also the case when comparing multiome regulons, suggesting that their overlap is due to common targets rather than noise introduced from using the pairing data (Fig R1.4).

Figure R1.4. Comparison between target regions and genes when using true multiome or paired data as input for SCENIC+. Heatmap showing the overlap between different regulons inferred using different types of input data for SCENIC+. The top row shows the overlap between predicted target genes, while the bottom row shows the overlap for the predicted target regions. Columns and rows correspond to the shared TFs across the different input data.

Lastly, to validate that the inferred eGRNs can recapitulate the different cell states in the mouse brain, we performed dimensionality reduction on the regulon AUC scores matrix. We found that regardless of using true multiome or matched data as input for SCENIC+, all the methods are able to cluster the cells by cell type (Fig R1.5). In Matched-2 and Matched-3, we used a virtual template of the cortex to map the different cell types, with 6,210 virtual cells (pixels, see *Methods*). We observe that even rare populations in this data set, like OPC and PVM (0.77% and 0.70% of the population, respectively) are clearly distinguishable when using SCENIC+ with these data sets as input, showing that SCENIC+ is able to infer *bona fide* eGRNs even from small populations.

Fig R1.5. *eGRN-based dimensionality reduction using true multiome or paired data as input for SCENIC+. eGRN-based dimensionality reductions when using true multiome (Multiome, 19,485 cells), or paired data (Matched-1, 18,435 cells; Matched-2, 6,210 cells; Matched-3, 6,210 cells) colored by cell type are shown.*

Altogether, we find this analysis an important contribution to the revised manuscript, as it suggests that eGRN inference is as accurate when using true multiome or well paired data.

Among the different pairing approaches we have tested, we did not observe drastic differences. Nevertheless, pairing of independent scATAC-seq and scRNA-seq data sets is not a trivial task, especially

in complex and/or variable tissues (e.g. tumors); and depends on data processing steps by the user. For pairing with SCENIC+ (Matched-2) and ScoMAP (Matched-3), cell type annotation is required in both layers; which is particularly challenging on the scATAC-seq data as features are regions rather than genes. We therefore disagree with the reviewer that this means that multiome data “is not needed”, since the matching challenge is *de facto* not needed with multiome data. However, provided a correct matching, we agree that paired independent data are as powerful to infer eGRNs.

We include this analysis in Fig S7, and detailed explanations of the analysis in the methods section and in the results section “Validation and benchmark of SCENIC+ predictions using ENCODE data”.

2. SCENIC+ provides a Python module, `pycisTopic` for inference of transcription factor (TF) binding sites from TF motif data. The module facilitates motif-based analysis using a compendium of motifs collected from a large number of individual motif resources along with algorithms for inferring TF binding using this collection. In particular, the suggested approach is based on Cluster Buster (Frith et al, Nucleic Acids Research, 2003). The Aerts lab has published multiple papers based on a similar strategy of collecting and integrating TF binding motifs, for example Janky et al. PLoS Comput. Biol., 2014, and the paper describing SCENIC (Aibar, Nature Methods, 2017). The area under recovery curve appears to be the same as presented in the first SCENIC paper. It seems the contribution in the current manuscript is mainly in the increased scale of the motif collection and perhaps the reimplementing of old software into a new Python package. A clearer description of what is novel in this work would be helpful.

We thank the reviewer for the suggestion to clearly describe what is novel compared to our previously published methods such as iRegulon and SCENIC. It is correct that the motif enrichment step in SCENIC+ (called `pycisTarget`, not `pycisTopic` as stated by the reviewer) is based on the same statistic (Area Under the recovery Curve) as we used in iRegulon and SCENIC, since we find this to be a robust statistic, and re-inventing motif enrichment does not constitute the main novelty of SCENIC+.

The `pycisTarget` package for motif discovery actually provides three motif enrichment approaches, namely `cisTarget` (Area Under the recovery Curve), Differentially Enriched Motifs (DEM, based on the

non-parametric Wilcoxon rank sum test to compare a foreground and background set of sequences), and Homer (interfacing with the Homer installation). Compared to the previously existing implementations of cisTarget in iRegulon and SCENIC, pycisTarget allows for multiprocessing of several region sets and streamlined usage within the SCENIC+ workflow. The introduction of DEM is useful for the discovery of partner TFs between close populations or different target regions of the same TF. We illustrate the functionality of DEM to identify partner TFs of SOXE factors in human melanoma, oligodendrocytes, and astrocytes. SOXE factors are key in these systems (SOX10 in melanoma and oligodendrocytes, and SOX9 in astrocytes, respectively), and despite binding the same motif, they bind different regions in the three cell types (Fig R2.1). As cisTarget uses as background other regions in the genome (or consensus peaks), motif enrichment analysis of SOXE cistromes in melanoma, oligodendrocytes and astrocytes with these methods mostly identifies highly enriched SOXE motifs (Fig R2.2), with NES 31.6, 22.31 and 31.85, respectively. On the other hand, DEM allows to compare sets of regions in a pairwise manner, allowing to directly assess differences between the sets of SOXE target regions. DEM reveals NFI and HOX motifs on SOXE target regions in astrocytes, OLIG (E-box) in oligodendrocytes, and AP1, RUNX, TFAP2 and MITF motifs in melanoma, in agreement with literature (Minnoye et al., 2020) (Fig R2.3, R2.4). While other methods for eGRN inference from single cell multiomics data only rely on motif scanning approaches (which are prone to false positives; see comment 5), the implementation of both (complementary) strategies within SCENIC+ permits to identify overrepresented motifs in topics/DARs using the genome (cisTarget) or specific sets of regions (DEM), such as DARs in other cell types, as background; from which TF cistromes are inferred. **We have added this analysis as Fig S5.**

Fig R2.1. Overlap between regions in SOXE cistromes in melanoma, oligodendrocytes and astrocytes.

Fig R2.2. cisTarget on SOX cistromes. Top 30 motifs identified by cisTarget using regions in the SOX cistromes inferred in melanoma, oligodendrocytes and astrocytes clustered using motifStack. Colors indicate the TF family of the motifs (in this case, SOX).

***Fig R2.3. DEM on SOX cistromes.*** *Top 30 motifs identified by DEM using regions in the SOX cistromes inferred in melanoma, oligodendrocytes, and astrocytes clustered using motifStack. Colors indicate the different TF families for the motifs.*

Fig R2.4. Differential motif enrichment with DEM between SOXE regions in different cell types reveals cell-type specific partner TFs. Ternary plot showing enrichment scores of motifs found in the melanoma, oligodendrocyte, and astrocyte SOX regions. Each corner represents a cell-type-specific SOX topic, dots represent enriched motifs, and axes represent average enrichment scores for each topic. The colors of the dots are used to indicate the TF family to which the motifs belong.

The second novelty is the SCENIC+ motif collection, which the reviewer correctly refers to as an increase in scale. However, we believe that this is a profound novelty, since the motif collection and the motif-TF annotation determines (for any of the eGRN methods we benchmarked) which TFs can ultimately be identified. The SCENIC+ motif collection includes 5,450 new motifs compared to the previous version of the motif collection (SCENIC motif collection), and importantly, we have developed a new strategy to reduce the redundancy of the motif collection, that improves the performance of cisTarget and DEM (Fig R2.5). We resolve a long-standing question in the field, namely whether using one archetype motif per TF is sufficient.

We ran cisTarget and DEM in peaks from 308 TF ChIP-seq tracks from ENCODE, using different motif collections: the previous cisTarget collection (SCENIC), and the new motif collection before clustering

(unclustered), using only archetypes for cluster scoring (archetype), and using all motif clusters for scoring (clustered). While there are no big differences between the unclustered collections, we found that our clustered collection, together with our (new) strategy to use all motifs per TF rather than one archetype per TF, results in a better recovery of the putative TF motifs. We believe that this motif database, as well as our motif enrichment approaches, are important contributions for the gene regulation field; and are an important aspect of the performance of SCENIC+. We have included this analysis in Figure S5.

Fig R2.5. Comparison of motif enrichment approaches and databases. *a.* Recovery of TFs from ENCODE ChIP-seq data using different motif enrichment methods, namely Homer, pycisTarget and DEM; and databases. The unclustered databases includes all annotated motifs before clustering (singlets), the archetype databases use the consensus motifs of the clusters based on STAMP, and the clustered databases uses the motif clusters, scoring regions using all motifs in the cluster. SCENIC indicates that our previous motif collection (unclustered) was used. The x-axis shows the positions in which the TF targeted in the ChIP-seq experiment can be found, and the y-axis shows the cumulative number of TFs that are found in that position. *b.* Number of times the corresponding TF motif for the ChIP-seq track was recovered in the first position using different motif enrichment methods and motif collections.

3. Although the motif compendium might be a useful resource, the resulting motif compendium itself does not appear to have been made available online. Instead results of mappings to various selected regions of the genome are provided, which reduces the value of this work. In addition, the reported reimplementations of Cluster-Buster does not appear to be available as source code, but as some sort of undefined binary file. Perhaps there are other links to the motif collection and source code which were not clear on the website?

Thank you for pointing this out. We have made the motif collection available at https://resources.aertslab.org/cistarget/motif_collections/. The source code for our cluster-buster implementation is available at https://github.com/ghuls/cluster-buster/tree/change_f4_output, as listed at the repository including the instructions to generate cisTarget databases (https://github.com/aertslab/create_cisTarget_databases). **We have added these links in the manuscript and github pages.**

4. The third step of SCENIC+ is TF to target gene prediction, making use of chromatin accessibility data. This part seems important and would benefit from a better description. It appears that this part is also reliant on previous approaches developed in the Aerts lab but the description does not clearly specify the new developments. There is also work by other labs that do a similar type of analysis (for example Duren, Z. et al Proc. Natl. Acad. Sci. U. S. A, 2018). What are the advantages and disadvantages of the SCENIC+ approach relative to other methods that try to infer GRNs using gene expression and chromatin accessibility data?

Reviewer 2 (see comment 4) had a similar remark regarding the description of this step, **we have adapted the method section at line 1016-1025** from:

“eGRN compilation: First TF-to-region-to-gene modules are generated by intersecting the binarized region-to-gene relationships with TF-to-region cistromes by region name. Next these modules are pruned by running GSEA, gsea_compute function from the GSEAPy python package (v0.10.8), using the TF-to-

gene importance scores, for all TFs and genes in the module, as ranking and genes of the module as gene set. The leading-edge genes along with the binarized regions linked to those genes are retained, generating eRegulons. This analysis is run separately for TF-to-gene and region-to-gene relationships with positive and negative correlation coefficients, for a total of four GSEA runs. By default, eRegulons with less than 10 predicted target genes or obtained from region-to-gene relationships with a negative correlation coefficient are discarded.”

To:

“eRegulon creation: We define an eRegulon as a TF together with its target regions and genes. To generate this, information from gene expression, region accessibility and motif enrichment are combined. For each TF, TF-region-gene triplets are generated by taking all regions that are enriched for a motif annotated to the TF and all genes linked to these regions, based on the binarized region-to-gene links (see “Calculating TF-gene and region-gene relationships”). However, we only want to include genes, and the regions linked to them, in the final eRegulon if they are significantly co-expressed with the TF. To determine this, we use Gene Set Enrichment Analysis (GSEA). Here, all genes are ranked based on the TF-gene importance scores and an enrichment analysis of the set of genes in the triplet compared to the overall ranking of the genes is computed, using the gsea_compute function from the GSEAPy python package (v0.10.8). Finally, only the genes at the top of the ranking, known as the leading edge are retained in the final eRegulon. This analysis is run separately for TF-gene and region-to-gene relationships with positive and negative correlation coefficients, for a total of four GSEA runs. By default, eRegulons with less than 10 predicted target genes or obtained from region-to-gene relationships with a negative correlation coefficient are discarded.”

Regarding the question on how SCENIC+ relates to other state-of-the-art methods:

We include a supplementary note (Supplementary Note 2) and a table (Fig R4.1) comparing SCENIC+ with other state-of-the-art methods for inference of (enhancer) gene regulatory networks from single-cell data. PECA (Duren et al. (2017), Duren et al. (2020)) has not been included in the comparison as it uses paired bulk RNA-seq and DNase-seq data. The PECA model co-optimizes the probabilities of TF binding, enhancer-gene interaction and gene expression using Expectation-maximization (EM) for parameter estimation. This EM step can be very resource intensive when applied to single cell data sets,

in which thousands of cells are profiled, compared to bulk (i.e. the initial PECA model was trained using 25 samples). Compared to our large motif collection, the binding strength of a TF to a region is estimated in PECA based on motif scanning with 557 PWMs. In addition, PECA does not account for data sparsity as it is designed for bulk data.

Feature	SCENIC	CellOracle	Pando	FigR	GRaNIE	SCENIC+
Input	scRNA-seq	scRNA-seq and bulk/scATAC-seq	scRNA-seq and scATAC-seq	scRNA-seq and scATAC-seq	Bulk RNA-seq and bulk ATAC-seq	scRNA-seq and scATAC-seq
# Motifs	49,054 ¹	8,049 ²	1,590	1,141	768	49,054 ¹
# TFs (Human)	1,553	1,022	1,372	1,141	681	1,553
TF-Region	No	No ³	Yes	No	Yes	Yes
Region-Gene	No	Yes	No	Yes	Yes	Yes
TF-Gene	Yes	Yes	Yes	Yes	Yes	Yes
Output GRN	TF-Gene	TF-Gene	TF-Region-Gene	TF-Gene	TF-Region-Gene	TF-Region-Gene
GRN selection	NA	P-value < 0.01	P-value < 0.01	IZ-score1 > 0.5	FDR < 0.2	Leading edge
Repression	No	No	Yes	Yes	Yes	Yes
Network activity	Yes	No	No	No	No	Yes
Network specificity	Yes	Yes	No	No	No	Yes
Network visuals	No	No	No	Yes	Yes	Yes
Perturbation	No	Yes	No	No	No	Yes
Differentiation	No	No	No	No	No	Yes

We used SCENIC(+) using cisTarget clustered motif collection, with 12,935 clusters¹
 We used CellOracle with gimmemotifs clustered motif collection (default), with 1,795 clusters²
 This information is not directly provided in the workflow but it is possible to derive it using the internal motif hits in the regions and the selected region-gene and TF-gene relationships³

Figure R4.1. Table comparing state-of-the-art methods for GRN inference at single cell resolution.

The inputs for the third step of SCENIC+ are the scRNA-seq count matrix, the TF cistromes derived from pycisTarget and the imputed accessibility matrix. Compared to other methods, our TF cistromes are derived using statistical motif enrichment using the SCENIC+ motif collection, instead of motif scanning approaches, that are prone to false positives (see comment 5). To account for the high sparsity of the scATAC-seq data, we use pycisTopic to perform drop-out imputation instead of modelling from the raw fragment matrix.

To assess TF-gene (based on TF-gene expression) and region-gene (based on region accessibility and gene expression) relationships we use linear models and Gradient Boosting Machines (to account for non-linear relationships), which have been the top performers in TF-gene inference benchmarks (Pratapa et al. 2020). These steps are also performed in GRaNIE (only using linear models); however, GRaNIE uses as input the raw fragment matrix, resulting overall in less relationships being detected. FigR also uses the raw fragment matrix to infer enhancer-gene correlations, grouping regions into Domains Of Regulatory Chromatin (DORCs), that are smoothed over the cells *a posteriori*. TF-DORC relationships are assessed *a*

posteriori based on the correlation between the expression of the TF and the gene and the presence of the TF motif in the DORC regions. CellOracle on the other hand uses Cicero (Pliner et al., 2018) to link enhancers to genes, based only on co-accessibility (based on the raw counts matrix) between the enhancer and the *promoter* of the potential target gene, rather than using the actual gene expression. Pando co-optimizes TF-enhancer-gene triplets using linear models, using the scATAC-seq raw counts matrix as well.

We then use a leading-edge approach to determine the optimal cut-offs for binarizing the TF-region, TF-gene and region-gene scores into regulons. For this, we perform GSEA using the TF-gene scores as rankings and the TF cistromes (converted into gene sets based on the enhancer-gene relationships) as gene sets and take the leading edge to form the regulon. Contrary, in CellOracle, FigR, and GRaNIE, the user must specify arbitrary thresholds to determine the regulons as a set. While a proof-of-concept of these steps have been shown in our previous work (TF-gene inference (Aibar et al., 2018), enhancer-gene inference (Bravo et al., 2020), eGRN assembly (Janssens et al., 2022)); we now provide a standalone package that streamlines the whole workflow and includes additional features for downstream processing.

A key limitation of this strategy is that we discretize the continuous distributions that represent the TF-region, region-gene, and TF-gene affinities, losing quantitative information. To solve this limitation, we have now added a new triplet scoring approach, in which TF-region, region-gene and TF-gene distributions can be queried and used to prioritize the best TF targets in a regulon. **We further illustrate its application in Fig 3, and the new result section: “SCENIC+ prioritizes functional enhancers in ENCODE cell lines.”, and comment 8.**

5. The illustration of SCENIC+ on PBMC multiome data lacks a description of how the method adds insights into the system that would not be observed through other approaches. For example, the master regulators of B cells, T cells, dendritic cells and monocytes are identified by SCENIC+, Would these regulators not be identifiable by other tools such as chromVAR, Signac, or the original SCENIC software?

We provided such a comparison for the ENCODE cell line case study, where we investigated which TFs were identified by SCENIC+, CellOracle (Kamimoto et al., 2020), GRaNIE (Kamal et al., 2021), FigR (Kartha et al., 2022), Pando (Fleck et al., 2022), and SCENIC (Van de Sande et al., 2020). Following the reviewer’s suggestion, we now expanded this comparison to the PBMC case study and included chromVAR (Schep et al., 2017) and Signac (Stuart et al. 2021) as additional baseline methods. Note, however, that chromVar and Signac do not predict eGRNs, and analyze only the scATAC-seq data, but we agree that it is interesting to test whether these methods can identify key TFs per cell type.

To test whether other methods can identify the same TFs on PBMC data compared to SCENIC+ we reran the analysis on the scATAC-seq part using ArchR and Signac, for both methods we assessed motif enrichment analysis using a standard differential test in differentially accessible regions (hypergeometric test; adjusted p-value < 0.01) and using the ChromVAR deviation enrichment test. Both procedures are detailed in the tutorials of ArchR and Signac. For ArchR, both motifs as well as ChIP-seq tracks were used as features. All methods can identify the same set of TFs as SCENIC+, however both ArchR and Signac find more TFs (Fig. R5.1).

Figure R5.1. Comparison of TFs recovered using ChromVar and Signac. Heatmap showing whether a TF is found across different methods (present: green; absent: red). Signac and ArchR were run using different options. (1) DEM: Differentially Enriched Motifs or ChIP-seq tracks in differentially accessible regions, (2) ChromVAR deviations.

To test whether the extra TFs identified by Archr and/or Signac are relevant (since they may represent false positive predictions) we ran a Gene Ontology enrichment analysis on TFs identified by ArchR and Signac, on TFs found exclusively by Signac, or on TFs found by all methods including SCENIC+. Only the TFs identified by SCENIC+ (and the other methods), but not the extra TFs identified only by Signac and or ArchR, are enriched for terms related to immune cells (Fig. R5.2). From this we can conclude that the TFs identified by SCENIC+ can also be identified by Signac and/or Archr. However, Signac and ArchR find extra TFs which are most probably false positive predictions – this is somehow expected since these methods do not perform an integration with scRNA-seq data to verify the expression of the TF and the target genes. SCENIC+ is thus more stringent as it exploits TF and target gene expression (i.e. only TFs for which the accessibility of the target regions correlate significantly with the expression of the TF and its target genes are retained), and such filtering is not done in ArchR nor Signac.

Figure R5.2. TFs found exclusively by Signac or ArchR are not enriched for immune related gene ontology terms. Gene ontology enrichment analysis using gprofiler on TFs found exclusively by Signac, Archr or all methods including SCENIC+. Top 10 enriched Human Protein Atlas and Human Phenotype terms are shown for each gene set.

One of the key outputs of SCENIC+, next to the identification of master regulators of different cell types, is the accurate identification of the genomic target regions for these TFs. To test how SCENIC+ compares

to Signac and ArchR in this aspect we assessed precision, recall and F1-scores on cistromes obtained by all methods. For this, we collected as ground truth ChIP-seq data in cell lines that are relevant for the PBMC data (BLaER1; B cell precursor, GM08714; B lymphocyte, GM12878; B lymphocyte, GM12891; B lymphocyte, GM12892; B lymphocyte, HL-60; promyeloblast, K562; hematopoietic cancer and NB4; blood cancer). Both Signac (using the hypergeometric test) and SCENIC+ outperformed the other methods based on the F1-metric and both methods have the highest precision compared to other methods (Fig R14). Signac using the ChromVAR method has the highest recall, with Signac (using the hypergeometric test) and SCENIC+ ranking second (Fig R5.3). From this we conclude that SCENIC+ is able to identify master regulators of PBMCs, which are relevant *and* is able to identify target regions of these regulators with high accuracy. Other methods, like Signac and ArchR, are also able to do this but are more prone to false positive predictions, and this both on the aspect of finding relevant TFs and predicting target regions which are bound by the TFs in question.

Figure R5.3. Precision, recall and F1-scores for predicting ChIP-seq data. Violin and jitter plot showing precision, recall and F1-scores for predicting ChIP-seq data based on cistromes obtained from SCENIC+, Signac and ArchR. Consensus peaks for which the ChIP-seq coverage was greater or equal to the 95th percentile of ChIP-seq coverage across all consensus peaks were considered as positive peaks for a certain ChIP-seq track. All ChIP-seq data the following cell lines from encode were used for this analysis: BLaER1, GM08714, GM12878, GM12891, GM12892, HL-60, K562 and NB4. The ChIP-seq track with the highest F1-score was used when more than one ChIP-seq track was present for a single TF. Signac and ArchR were run using different options. (1) DEM: Differentially Enriched Motifs or ChIP-seq tracks in differentially accessible regions, (2) ChromVAR deviations. x-axis is sorted by median F1-score, prediction or recall. Differences in mean were assessed using the two-tailed Mann-Whitney U test and p values were adjusted using the Benjamini-Hochberg procedure. y-axis is in log₁₀ scale.

Given that Signac and ArchR solely depend on chromatin accessibility data, while SCENIC+ uses both chromatin accessibility data as well as gene expression data, we also wanted to compare the performance of SCENIC+ with other methods that rely on both modalities. For this, we also re-ran the PBMC analysis using CellOracle, FigR, Pando and GRaNIE. The analysis using GRaNIE (developed for bulk ATAC+RNA data) required too much memory (more than 2 TB) and therefore is not included (tested on a machine with 72 cores Intel(R) Xeon(R) Platinum 8360Y CPU @ 2.40GHz and 2 TB of memory). As suggested by the reviewer, we also included SCENIC in this analysis. First, we assessed whether the other methods could recover the same TFs as SCENIC+. When testing whether TFs identified by SCENIC+ could also be recovered by other methods, Pando was the most similar (Fig. R5.4). Other methods miss important TFs, including PAX5 (not found by SCENIC and FigR), CEBPA (not found by SCENIC and CellOracle) and RUNX3 (not found by CellOracle) (Fig. R5.4). In this case Pando identified 136 extra TFs that were not identified by SCENIC+ (Fig. R5.5), these TFs are also enriched for immune related Gene Ontology terms (Fig. R5.6). So, in this case SCENIC+ might have missed some relevant TFs.

Figure R5.4. Comparison of TFs recovered using different (e)GRN inference methods. Heatmap showing whether a TF is found across different (e)GRN inference methods (present: green; absent: red). Only TFs found by SCENIC+ are shown.

Figure R5.5. Comparison of TFs recovered using different (e)GRN inference methods. Heatmap showing whether a TF is found across different (e)GRN inference methods (present: green; absent: red). All found TFs are shown.

Figure R5.6. Gene ontology enrichment analysis on TFs found exclusively by Pando and CellOracle. Gene ontology enrichment analysis using gprofiler on TFs found exclusively by Pando, CellOracle or all methods including SCENIC+. Top 10 enriched Human Protein Atlas and Human Phenotype terms are shown for each gene set. FigR is not included in the figure because no gene ontology terms were enriched. No significant GO terms were found for TFs found exclusively by SCENIC and FigR.

Next, we assessed the cell type specificity of the predicted target genes (Pando, CellOracle, SCENIC and FigR) and target regions (Pando and CellOracle) of each TF identified by the different methods,

compared to SCENIC+. For this, we calculated AUC values for predicted target genes and regions using AUCell (Aibar et al., 2018) and visualized these values on heatmaps (Fig. R5.7). The AUC values obtained by Pando were most similar to the ones obtained by SCENIC+, the values obtained by FigR were almost random compared to SCENIC+ (Fig. R5.8).

Figure R5.7. Comparison of cell type specificity of regulons inferred by different (e)GRN inference methods. Heatmap-dotplot showing scaled target gene AUC values using a color-scale and target region AUC values (if present) using dot-size scale. Regulons which were not recovered by one of the methods (compared to the regulons recovered by SCENIC+) are greyed out.

Figure R5.8. Comparison of AUC values of *eRegulons* inferred by different (*e*)GRN inference methods. Boxplot showing spearman correlation of target gene (red) and target region (blue) AUC values of *eRegulons* inferred by different (*e*)GRN inference methods.

Next, we tested how SCENIC+ compares to other GRN inference methods that predict individual target regions for each TF (namely Pando and CellOracle) on the aspect of predicting accurate target regions. To this end, we assessed precision, recall and F1-scores on target regions obtained by all methods, using as ground truth ChIP-seq data in cell lines which are relevant for the PBMC data (BLaER1; B cell precursor,

GM08714; B lymphocyte, GM12878; B lymphocyte, GM12891; B lymphocyte, GM12892; B lymphocyte, HL-60; promyeloblast, K562; hematopoietic cancer and NB4; blood cancer).

On average, SCENIC+ has an equal precision compared to Pando, but an order of magnitude higher than CellOracle. However, the recall of Pando is the lowest compared to SCENIC+ and CellOracle. Therefore, as measured by the F1-metric, SCENIC+ significantly outperforms Pando (Fig. R5.9).

Figure R5.9. Comparison of the accuracy of target regions recovered per regulon by different eGRN inference methods. Violin and jitter plot showing maximum F1 score and corresponding precision and recall per transcription factor comparing the recovered target regions versus ChIP-seq data of the same transcription factor, the maximum is taken across all possible thresholds of the ChIP-seq data. P-values calculated using two-tailed Mann-Whitney U test.

To further demonstrate this point, we visualized the ChIP-seq signal of some key PBMC related master regulators (BCL11A, CEBPB, PAX5 and EBF1 in GM12878) on the union of genomic regions identified by CellOracle, Pando and SCENIC+ (Fig. R5.10). From this, the predicted target regions of SCENIC+ are mostly located at the top of the ranking of the ChIP-seq coverage while the predicted target regions of Pando and CellOracle are either more located towards the bottom or they are very sparse (Fig. R5.10). As mentioned in comment 4, a key limitation to the SCENIC+ approach in the original submission was that quantitative information about TF-to-region, region-to-gene and TF-to-gene importance was lost by discretizing these continuous distributions. For this reason, we have now developed a new algorithm to

rank all the TF-region-gene triplets identified by SCENIC+. This ranking is defined as the aggregated ranking of TF-gene importance, region-to-gene importance and TF-region importance. In this context we define the TF-region importance, for a given TF-region pair, as the maximum ranked position of motif scores, for the region, over all motifs annotated to the TF. The TF-region (TF2R) importance recapitulates the ranking of ChIP-seq signal well, with the most highly ranked regions having a higher ChIP-seq signal compared to lower ranked regions (Fig. R5.10). Therefore, this TF2R score can be used by the user to adapt the precision-recall balance, which we believe is a valuable addition. This is further illustrated in comment 8.

We have added these new analyses as Figure S9 and added the following lines to the manuscript

“(…) to test whether the same conclusions can be drawn from a real single-cell multiome dataset, we also compared the results of SCENIC+, SCENIC, CellOracle, Pando and FigR on the PBMC data set. GRaNIE was not present in this comparison as it was not scalable to this larger dataset. The GRNs from all methods were able to recapitulate all biological cell states, except for the GRN from FigR (Fig S8a,c). SCENIC+ and Pando performed best in terms of identifying biologically relevant TFs, with Pando finding additional TFs compared to SCENIC+ on this data set (Fig S9f,h). On the other hand, SCENIC+ had a higher precision and recall in terms of identified target regions of these TFs (Fig S9j). Note that even when only scATAC-seq is used to predict TFs per cell type (e.g., ArchR, Signac), the majority of the known cell type specific TFs could still be identified, showing that motif discovery is a powerful means for cell type specific TF prediction (Fig S9e), however with a large amount of false positive predictions (Fig S9g,i).”

Figure R5.10. ChIP-seq coverage on predicted target regions in different methods. Coverage of *BCL11A*, *CEBPB*, *PAX5* and *EBF1* in GM12878 across the union of the target regions identified by CellOracle, Pando or SCENIC+ (y-axis), using 10 bins across the individual target regions (x-axis). y-axis is sorted by the maximum ChIP-seq signal across the 10 bins. ChIP-seq coverage is scaled based on the 5th (min) and 95th (max) percentile across all consensus regions in the PBMC dataset. The binary heatmap shows whether a region is found (black bar) or not (white bar) by a certain method. On the right of each heatmap the TF to region (TF2R) ranking of SCENIC+ is shown, a lower ranking represents a higher likelihood that the region contains a motif of the TF, curve is smoothed by taking the moving average across the regions.

6. In the validation of SCENIC+ on ENCODE data the first task presented is the prediction of regulatory TFs recovered by several methods. In the main text it would be good to define what recovering TFs means. It could mean any of the following: TFs that are expressed in a certain cell type, TFs that are actively regulating genes in a cell type, TFs that define chromatin accessibility, TFs that are necessary to define the cell states, or TFs that were chosen by ENCODE for unknown reasons. The last is not meaningful as the TF selection might be biased towards the research interests of the investigators and the availability of ChIP antibodies.

We agree that the word “recovery” may be confusing here. In the original version we simply meant the number of TFs identified/found by each tool. We now replaced the following paragraph (lines 205 – 211):

“We first assessed how many TFs were recovered by each method, and how many target regions and genes were predicted per TF (Fig 3c, S5a-e). SCENIC+ identifies 178 TFs across the eight cell lines (Fig S5, S6).”

With:

“(…) SCENIC+ identified regulons for 178 TFs (Fig 3b, S10). SCENIC, GRaNIE, FigR and Pando identified fewer regulons (for 108, 39, 71 and 157 TFs respectively). CellOracle identified regulons for more TFs (235). For each eRegulon, SCENIC+ predicts on average 471 target genes and 1,152 target regions (Fig 3b).”

For each method we selected high quality regulons based on the default thresholds given in the tutorials of the tools. For SCENIC+, high quality regulons were selected based on the correlation between gene-based regulon AUC and region-based regulon AUC (> 0.7). This resulted in 178 regulons, with a median of 437 genes and 774 regions per regulon. For CellOracle, we filtered links with the *filter_links* function using $p=0.001$, *weight='coef_abs'*, *threshold_number=2000*. This resulted in 157 regulons, with a median of 49 genes and 50 regions per regulon. For Pando, we kept triplet with adjusted p-value < 0.01 , resulting in 235 regulons, with a median of 43 genes and 42 regions per regulon. For FigR, we kept peak-gene correlations with p-value < 0.05 and kept TF-DORC associations with $\text{abs}(\text{score}) > 1$, resulting in 10,757 TF-gene pairs. This resulted in 71 regulons, with a median of 39 genes per regulon. For GRaNIE, we used 0.2 as FDR threshold to filter the eGRN. This resulted in 39 regulons, with a median of 176 genes and 68 regions per regulon. For SCENIC, we found 108 regulons with a median of 322 genes. For all methods we used a search space of 150kb for inferring region-gene relationships and kept regulons with at least 25 target genes.

We also refer to “recovery” of TFs related to Figure 3, panel d in the manuscript. Here, to assess TF recovery in the ENCODE cell lines, we ranked 309 TFs based on the number of peaks found by ChIP-seq in the Deeply Profiled Cell Lines. That is, TFs that are bound to DNA in the cell lines will be on the top of the ranking; while non-functional and not meaningful TFs that may have been chosen by ENCODE for unknown reasons will be located on the bottom on the ranking. To assess the recovery of TFs by the methods, we use the Area Under the Curve (AUC) on the top part of the ranking (Top 40 TFs). This approach enables us to distinguish methods by whether they mostly find meaningful TFs (hits on the top

of the ranking; i.e. TFs with a high number of binding sites in the cell lines of interest), or also irrelevant TFs (hits on the bottom of the ranking; i.e. TFs with a low number of binding sites in the cell lines of interest) (Fig R6.1a). We also observed that TFs on the bottom tend to be more broadly expressed/less cell type specific (such as NFYA, SP2, ELK1; with low Tau value), while TFs on the top of the ranking are more cell type specific (high Tau value) (Fig R6.1b). Importantly, at the top of the ranking we find key known TFs in the cell lines, such as CEBPA, HNF4A, FOXA2, EBF1, PAX5, SPI1, GATA1, GATA2, TEAD3, WT1, GRHL2; in agreement with literature.

As a complementary approach independent of TF ChIP-seq and its availability, we ranked TFs based on their maximum LogFC among the cell lines. This resulted in a ranking with 374 TFs, in which top TFs are more differentially expressed among the cell lines (Fig R6.1c). In both approaches, SCENIC+, Pando and SCENIC recovered a large fraction (36%, 32% and 15%, respectively) of bound TFs/differentially expressed TFs; while GRaNIE and FigR recovered little TFs. For CellOracle, we observed a limited recovery of differentially expressed TFs, as well as of most bound TFs; however, many of the not meaningful TFs (profiled by ChIP-seq by ENCODE, but with little binding sites) were recovered. In fact, we observed that TFs recovered by CellOracle are less cell type specific (with lower Tau values), in contrast to other methods (Fig R6.1d). **We have included the new analysis in Figure 3 and S11 and clarified the descriptions in the text and the methods section.**

Fig R6.1. TF recovery performance by single-cell multiomics GRN inference methods. **a.** Cumulative TF recovery for each method using as x-axis TFs ranked based on the number of ChIP-seq peaks and AUC values per method using the top 40 TFs. **b.** GAM fitted Tau value across the TF ranking based on the number of ChIP-seq peaks. **c.** Cumulative TF recovery for each method using as x-axis TFs ranked based on the maximum LogFC across the cell lines and AUC values per method. **d.** Tau values distributions for the TFs recovered by each method.

7. In Fig 3 numbers of TF, regions etc are shown. It would be more informative to show performance metrics such as precision and recall than to show these numbers. For example, one could trivially show a method that always predicts all the TFs and all regions, but this would be meaningless.

Agreed. Together with the descriptive statistics on the number of TFs, target regions and genes, we have calculated precision-recall values for the TF recovery, recovered target regions and recovered target genes. For calculating the precision and recall curves on TF recovery, we used the rankings based on number of binding sites per TF and LogFC of the TF expression as scores, and whether the regulon was found by a method or not as the labels. The results agree with our previous observations: SCENIC+, Pando and SCENIC show the highest PRAUC values, while CellOracle, FigR and GRaNIE show less agreement. CellOracle performs better in the TF ChIP-seq based recovery benchmark, as it includes some non-meaningful TFs at the end of the ranking (Fig R7.1). **We have included this new analysis in Fig S11.**

Fig R7.1. Assessing precision and recall on TF recovery by single-cell multiomics GRN inference methods. a. Precision-recall curves using TFs found by the different methods as labels and different thresholds in the TF ChIP-seq based ranking and AUC of the curves. **b.** Precision-recall curves using TF found by the different methods as labels and different thresholds in the LogFC TF expression based ranking and AUC of the curves.

To calculate the precision and recall of the methods in predicting target regions, we have used three different standards: 1) Unibind peaks (optimal ChIP-seq peaks with TF motif), 2) ENCODE ChIP-seq coverage and 3) Enformer prediction scores (Avsec *et al.*, 2021). For the latter two, we have used as threshold on the coverage or Enformer prediction score the point in which the F1 score (that depends on both recall and precision) is maximum and used that threshold to calculate the precision and recall of the methods. Overall, the three approaches agree on the results: SCENIC+ and GRaNIE show the highest precision-recall compared to Pando and CellOracle. FigR and SCENIC could not be included in this comparison as they do not provide TF target regions (Fig R7.2). Together with the TF recovery stats

above, this nicely summarizes the advantages and novelty of SCENIC+ compared to the previously existing methods: SCENIC+ is the only method that correctly identifies TFs (for this, only Pando achieves similar accuracy), while SCENIC+ also identifies correct enhancers (for this, Pando suffers from extremely low recall at the same precision threshold). **We have included these new analyses in Fig S13.**

We are particularly enthusiastic about the comparison with Enformer predictions, as this constitutes a state-of-the-art sequence-based method that serves as an independent analysis approach, and we therefore decided to include this *in silico* validation in a main figure (Fig 4).

Finally, to assess the precision and recall of predicted target genes, we exploited TF perturbation data from ENCODE, using as threshold on the LogFC rankings the point in which the F1 score (that depends on both recall and precision) is maximum. Our results indicate that the best precision-recall is achieved by SCENIC+ and SCENIC (Fig 7.3). We have added these analyses in Fig 3g and S16. However, in this comparison we do not account for the downstream indirect effects of TF perturbation (i.e. variation of genes that are not direct targets of the perturbed TF but rather of a TF that is a target of the perturbed TF). We further described how we have tackled this situation in comment 9 (Fig R9.1, S16).

Fig R7.2. F1 score, precision and recall distributions of the predicted regions per TF using three different standards. Violin plots represent the F1 score (left), precision (middle), and recall (right) distributions of the predicted target regions per TF using different standards: ChIP-seq peaks with TF motifs (Unibind, top), ChIP-seq coverage (middle) and enformer predictions (bottom). GATA1, GATA2, HNF4A and STAT5A are indicated if the method has found the regulon. The numbers above or below the

violins indicate the number of TFs that have been found by the methods and are included in the standards. Pairwise wilcox-test p-values between distributions are shown on top of the violins.

Fig R7.3. F1 score, precision and recall distributions of the predicted genes per TF using perturbation data. Violin plots represent the F1 score (left), precision (middle), and recall (right) distributions of the predicted target genes per TF compared to TF perturbation data. The numbers above or below the violins indicate the number of TFs that have been found by the methods and have been perturbed.

8. In the assessment of the quality of the predicted target regions it appears there is a component of DNA sequence analysis used in the definition of the “silver” standard. This benchmarking approach is highly problematic for several reasons. First, there is a degree of circularity between benchmark and analysis. An analysis method that aligns well with the use of motifs in the benchmark generation will tend to do well. Second, this approach suggests that the TF binding sites that can be recognized to bind directly to DNA are of greater functional importance than the rest. The manuscript does not provide any evidence to support this claim. Third, the inability to make accurate predictions of TF binding based on sequence might be limited because current characterizations of DNA binding are not adequate. In this regard it might be helpful to contrast with approaches such as that proposed by Avsec, Žiga, et al. Nature Genetics, 2021. The most appropriate benchmarking comparison for this part is the ENCODE-DREAM in vivo Transcription Factor Binding Site Prediction Challenge <https://www.synapse.org/#!Synapse:syn6131484/wiki/402026>. This benchmark would be more objective

and enable a direct comparison with the results obtained by the best performing DREAM challenge methods.

Thank you for these insightful suggestions. We agree that the “silver” standard used in this benchmarking analysis has a component of DNA sequence analysis. This is, the Unibind database (Puig et al., 2021) was used for the TF-target-region benchmark. This is a database containing target regions for a large number of TFs based on curated ChIP-seq data. This database only contains direct target regions of each TF, as obtained by filtering ChIP-seq peaks based on scores for an enriched position weight matrix of the TF. The latter is where the sequence analysis comes in. The worry of a degree of circularity is therefore (partially) valid, although we want to note that the PWMs used in Unibind do not 100% overlap with the PWMs used by any of the methods, also each of the PWMs are slightly modified using the DAMO package (Ruan et al., 2018) before they are used for selecting direct target regions. Next to this, the algorithm used by Unibind is different from the algorithms used by the GRN inference methods. Nevertheless, we think this concern is valid and we therefore added a new analysis where we used databases that do not use any sequence analysis (see below), as another “silver standard”.

Regarding the comment whether TFs that directly bind DNA are of greater functional importance compared to TFs that bind DNA in an indirect manner, we actually do not make any statement. Clearly, the task of SCENIC+ and all the other eGRN inference methods in the benchmark is to predict genomic binding sites that are directly bound by the TF. In other words, the goal is to integrate scATAC-seq, scRNA-seq, and DNA sequence information. For this reason, we believe it is appropriate to benchmark against a database that contains direct binding sites for several TFs.

Regarding the third suggestion, to compare the SCENIC+ predictions to predictions made by a deep learning model trained on sequence alone: we find this is a very interesting suggestion and therefore **we added a new analysis where we compare SCENIC+ and other eGRN inference methods to Enformer predictions (Avsec et al., 2021, Fig S13, Fig 4).**

Finally, regarding the comment whether the “ENCODE-DREAM *in vivo* Transcription Factor Binding Site Prediction Challenge” represents the most appropriate ground truth: we do agree that it is important to use a standardized ground truth. However, the data used in this DREAM challenge, both the training

data (RNA-seq and DNA-seq data from ENCODE) as well as the ground truth data (ChIP-seq from ENCODE), is actually very similar to the data that we used here to benchmark SCENIC+. In the DREAM challenge, the RNA-seq and ATAC-seq data from the ENCODE cell lines is presented as “training data”, while the ChIP-seq data from ENCODE is considered as ground truth. This is also how we present our ENCODE case study and benchmark analysis. The difference is that the data we used to benchmark SCENIC+ is more recent (the DREAM challenge data is all from before 2017) and more extensive (it also contains Hi-C and RNA-seq after TF perturbation data). In addition, the DREAM challenge relies on the ability to make predictions on left out data. SCENIC+, nor the other GRN inference methods included in the benchmark, are designed to do this. These eGRN methods can only “filter”/rank data based on their likelihood of being a TF target region. For this reason, we decided to stick to our initial use of the more recent ENCODE data for benchmarking, and to include Enformer predictions as an independent validation.

Below we describe the results of our new analyses that address these different comments.

To assess the performance of the target region predictions of SCENIC+, GRaNIE, Pando and CellOracle based on ChIP-seq data, without the use of DNA sequence, we obtained all high-quality ChIP-seq datasets from ENCODE, performed on one of the deeply profiled cell lines used in our benchmark (i.e. MCF7, HepG2, PC3, GM12878, Panc1, IMR90, HCT116 and K562), these datasets were used to score the consensus peaks and these scores were used as ground truth scores. Similarly, all consensus peaks were scored using Enformer and scores related to the Enformer classes of the ChIP-seq datasets in the deeply profiled cell lines were used as ground truth. Next, we calculated precision, recall and the F1 metric for each tool. For this, the ChIP-seq/Enformer scores were thresholded in such a way to obtain the highest F1 metric per TF for each tool.

In these tasks, and in line with the analyses above, both SCENIC+ and GRaNIE performed the best both on precision as recall (Fig. R7.2). However, SCENIC+ recovered a larger number of TFs. The results obtained by using either Unibind, ChIP-seq or Enformer as ground truth are very similar and the conclusions are the same (Fig. R7.2). Indeed, the F1 scores obtained from ChIP-seq or Enformer are highly correlated (Fig. R8.1). **These analyses are included in Fig S13.**

Figure R8.1. Comparison of F1 scores obtained by comparing predicted target regions to Enformer and ChIP-seq. Scatter plot showing maximum F1 score across all possible thresholds of ChIP-seq coverage over all consensus peaks (x-axis) and all possible thresholds of Enformer scores over all consensus peaks (y-axis) for all TFs identified by SCENIC+, GRaNIE, Pando and CellOracle. Pearson correlation coefficients is shown on top.

As an example, we visualized the ChIP-seq signal and Enformer score of two key HepG2 related master regulators (HNF1A and FOXA1) and two key K562 master regulators (GATA2 and STAT5A) on the union of genomic regions identified by GRaNIE, CellOracle, Pando and SCENIC+ (Fig. R8.2). The predicted target regions of SCENIC+ are mostly located at the top of the ranking of the Enformer score while the predicted target regions of GRaNIE, Pando and CellOracle are either more located towards the bottom or very sparse (Fig. R8.2). Also, GRaNIE yields predicted target regions that have a high ChIP-seq signal but a low Enformer score. We speculate that this latter observation is due to the fact that Enformer makes use of gene expression and other data modalities to predict ChIP-seq signal. This might point to the fact that these regions are not functional enhancers, even though they are bound by the TF. The TF-region (TF2R) score of SCENIC+ recapitulates the ranking of Enformer scores well, with the most highly ranked regions having higher Enformer scores compared to lower ranked regions (Fig. R8.2).

As mentioned in comment 5, our new TF-region score is used together with the TF-gene and region-gene score to rank each TF-region-gene triplet. This triplet score/ranking can be used to increase the precision of SCENIC+ (obviously resulting in a decreased recall) (Fig. R8.3). **We have added an analysis very similar to this example as Figure 4 to the manuscript.**

In conclusion, we again find that Pando does not perform well for enhancer identification. Note also that GRaNIE, although it performs well for TF-region predictions, cannot compete with SCENIC+ to identify

TFs (see above), and is computationally highly inefficient, preventing its use on the PBMC case study (as it was designed for bulk data).

Figure 8.2. Actual ChIP-seq coverage and ChIP-seq coverage predicted by Enformer on predicted target regions of different methods. Actual coverage (left) and Enformer predicted coverage (right) of HNF4A ChIP-seq in HepG2, FOXA1 ChIP-seq in HepG2, GATA2 ChIP-seq in K562 and STAT5A ChIP-seq in K562 across the union of the target regions identified by GRaNIE, Pando, SCENIC+ and CellOracle, (y-axis), using 5 bins across the individual target regions (x-axis). Actual ChIP-seq coverage and Enformer predicted ChIP-seq coverage is scaled based on the 5th (min) and 95th (max) percentile (based on Enformer predicted ChIP-seq coverage) across all consensus regions in the DPCL dataset. The binary heatmap shows whether a region is found (black bar) or not (white bar) by a certain method. On the right of each heatmap the TF to region (TF2R) ranking of SCENIC+ is shown, a lower ranking represents a higher likelihood that the region contains a motif of the TF, curve is smoothed by taking the moving average across the regions.

Figure R8.3. Precision and recall in relationship to triplet score. Scatter plot showing average precision and recall of predicted target regions of eRegulons compared to ChIP-seq data. Precision and recall are shown for the top 10%, 20%, 40%, 60%, 80% and 100% (x-axis) of the eRegulon edges based on the triplet score.

9. In the evaluation of the TF target gene by knockdown experiment it would be helpful to provide more information, including the TF, the true TF-regulated gene set, the prediction and the summary result. In addition, if the TF binding has been assayed by ENCODE it would be of interest to compare the performance of SCENIC+ using motifs with TF ChIP-seq data. Some insights into this evaluation would also be of value, for example which types of TFs are well predicted and what aspects of SCENIC+ were most important in making these predictions.

Regarding the comment about TF-target gene evaluation, we have added an analysis where we evaluate the TF-gene relationships based on *in silico* TF perturbation predictions (as detailed the melanoma section of the manuscript). For this, we computed the predicted LogFC of gene expression upon perturbation and compared the predictions with the observed effects. Correlation between the observed and the predicted perturbation variation shows that SCENIC+ networks are the most accurate in predicting the perturbation effects, compared to other methods (Fig R9.1). As an illustration, we show the comparison between the predicted and observed effects upon GATA1 (activator) and ARID3A (repressor) perturbation, where we

can also predict effects for genes which are predicted to be indirectly targeted by the TF (in grey, after 5 iterations). **We provide a supplementary table (Table S1) indicating the observed LogFC in each perturbation compared to the predicted effect of the perturbation using the GRNs inferred by the methods. We have also added this new analysis to Fig S16 and state in the text: “To better account for indirect effects of the TF-knockdown experiment due to direct interactions between TFs (i.e. the perturbed TF regulates another TF, Fig S16d) or cooperativity (i.e. the TFs target the same genes, Fig S16e), we also performed in silico TF perturbations based on the GRNs inferred by the different methods (see Methods). While only a fraction of the variation in gene expression can be explained by any of the GRNs, eGRNs inferred by SCENIC+ agree the best with the experimental data (Fig S16f, Table S1).”**

Figure R9.1. Comparison between simulated perturbation effects and TF knockdown experiments from ENCODE. **a.** Spearman correlation between predicted LogFC with in silico TF perturbation for each method versus the observed LogFC changes upon TF perturbation. **b.** Comparison between predicted and observed LogFC changes upon GATA1 KD. Dots in red indicate genes in the GATA1 regulon. **c.** Comparison between predicted and observed LogFC changes upon ARID3A KD. Dots in red indicate genes in the ARID3A regulon, predicted as repressor.

Regarding the comment about comparing SCENIC+ using ChIP-seq data and SCENIC+ using motifs, we have now implemented a new approach to score ChIP-seq bigwig files in order to build cisTarget rankings, available at https://github.com/aertslab/create_cisTarget_databases. This now provides the interesting option to use ChIP-seq tracks in the cistrome inference step, rather than (or next to) motifs. Using the ENCODE cell lines, we compared the performance of SCENIC+ in different settings,

concerning the cistrome inference step: 1) Using the motif collection, 2) Using ENCODE ChIP-seq tracks (on these cell lines) and 3) Combining information from both motifs and ChIP-seq tracks. We then assessed the quality of the GRNs based on the TFs identified, target regions and target genes, as in our previous benchmarks. Our results indicate that (as expected) the number of TFs identified when using only ChIP-seq tracks is lower, because it depends on the availability of ChIP-seq data for the TFs. Overall, the number of predicted target regions and genes is also lower compared to using motifs or compared to using a combination of motifs and ChIP-seq tracks (Fig R9.2, Fig R9.3). When validating the predicted target regions, we observed that the ChIP-seq track-based analysis has a slightly higher precision when using ChIP-seq and Unibind peaks as standard (ChIP-seq peaks with the TF motif), but a lower recall. However, when using the Enformer predictions as standard, the motif-based analyses have a slightly higher accuracy, likely because Enformer also uses the DNA sequence (Fig R9.4). Finally, for the evaluation of the target genes (TF perturbation data), the motif-based analyses result in a better recovery (Fig R9.5). Overall, using only ChIP-seq tracks results in sparser eGRNs, and we did not observe a substantial difference between using the motif databases on its own or using both motif and ChIP-seq data combined.

We included these results in Figure S18.

Fig R9.2. Descriptive statistics of eGRNs. Boxplot showing the number of genes per regulon (left), the number of target region per regulon (middle), and the number of high quality regulons (right) per approach.

Fig R9.3. Recovery of marker TFs across approaches. Cumulative TF recovery for each method using as x-axis TFs ranked based on the maximum LogFC across the cell lines and AUC values per method (left) and precision-recall curves (right).

Fig R9.4. F1 score, precision and recall distributions of the predicted regions per TF using three different standards. Violin plots represent the F1 score (left), precision (middle), and recall (right) distributions of the predicted target regions per TF using different standards: ChIP-seq peaks with TF motifs (Unibind, top), ChIP-seq coverage (middle) and enformer predictions (bottom). GATA1, GATA2, HNF4A and STAT5A are indicated if the method has found the regulon. The numbers above or below the

violins indicate the number of TFs that have been found by the methods and are included in the standards. Pairwise wilcox-test p-values between distributions are shown on top of the violins.

Fig R9.5. F1 score, precision and recall distributions of the predicted genes per TF using perturbation data. Violin plots represent the F1 score (left), precision (middle), and recall (right) distributions of the predicted target regions per TF compared to TF perturbation data. The numbers above or below the violins indicate the number of TFs that have been found by the methods and have been perturbed.

10. Although the paper does some GRN benchmarking, many of the benchmarks are not entirely convincing. It would be of interest to adapt some of the benchmarks proposed by Pratapa et al Nature Methods, 2020 to multomic data.

To further strengthen our benchmarks, as we find this an important aspect of our work, we included several additional analyses:

1. **TF recovery evaluation:** To define the key TFs that any method should ideally recover, we used two approaches: based on 1) the number of genomic regions bound by a TF (i.e. the more physical target regions a TF has, the more important it is) and 2) TF expression (i.e. if a TF is a marker gene in the cell line, it is likely important). The two approaches are complementary. The first approach can only be applied to TFs that have been profiled by ENCODE, while the second

may disregard general TFs. In both analyses, SCENIC+, Pando and SCENIC recovered the most relevant TFs. We observed that CellOracle tends to find more general TFs (with lower expression Tau values), likely because it infers GRNs on a cell type per cell type basis, without using other populations as background. This will result in the same patterns for general TF and marker TF in the cell type. On the other hand, GRaNIE finds few, but highly cell type specific TFs (with high expression Tau values). We describe the approaches in detail in comments 6, 7 and 8 and in Fig 3 and S11 in the manuscript.

2. **Target region evaluation:** To assess the quality of the predicted target regions we now use three different standards: 1) ChIP-seq peaks with the TF motif (Unibind), 2) ChIP-seq coverage (as in Pratapa *et al.*), and 3) Enformer predictions (Avsec *et al.*, 2021). The three analyses showed that target regions predicted by SCENIC+ and GRaNIE achieve the best precision-recall. These analyses are further described in comments 7 and 8 and in Fig 3 and S13 in the manuscript. In addition, we now also included a comparison of the predicted target regions with STARR-seq enhancer reporter data, finding that SCENIC+ predictions have the highest activities, and likely represent functional enhancers (Fig R10.1, S13).

Figure R10.1. Comparison of Enhancer activity on DPCL target regions identified by different tools. Violin plot showing distribution of maximum enhancer activity as measured using STARR-seq data from ENCODE on regions in K562, HepG2, HCT116 and MCF7. Difference in mean is

assessed using two-tailed Mann-Whitney U test. P-values are adjusted using the Benjamini-Hochberg procedure. The x-axis is sorted by the median average distance.

3. **Target gene evaluation**: To assess the quality of the predicted target genes, we use TF perturbation data (as in Pratapa *et al.*). In our previous versions, we directly compared the variability of the predicted target genes upon perturbation. However, a key challenge of this analysis is that TF perturbation can lead to indirect downstream changes, that will be considered as false negatives in the regulon (i.e. genes that change but are not included in the regulon). To tackle this challenge, we performed *in silico* TF perturbations based on the GRNs inferred by all the methods over 5 iterations, allowing to model the indirect changes upon TF perturbation. This analysis shows that SCENIC+ GRNs can predict the effects of TF perturbation the best. These analyses are described in comment 7 and 9 (Fig S16).
4. **GRN-based cell state identification**: Inspired by Pratapa *et al.*, we now also assess the overall GRN quality based on whether regulon enrichment scores can recapitulate the different cell states (based on clustering and dimensionality reduction visualization; Fig 3 and Fig S8). In addition, we used Pratapa *et al.* approach to simulate cells from defined GRNs (BoolODE), finding that SCENIC+ GRN generated cells better resemble the observed states (Fig S22). We further describe these analyses in comment 11 (Fig R11.3, R11.4).

11. One of the analyses shown by Pratapa et al relates to the number of stable states consistent with the gene regulatory networks. Can it be shown that the numbers of stable states derived from the networks presented in the manuscript are consistent with the observed biological systems?

We appreciate this suggestion. To assess whether the predicted GRNs for each of the tools corresponds with the number of stable states in the biological data we performed two new analyses.

First, we calculated AUC values based on predicted target genes and regions for each TF using the AUCCell method (Aibar *et al.*, 2018, Van de Sande *et al.*, 2020) followed by dimensionality reduction based on either target gene or target region AUC values or the combination of both. This eGRN-based

dimensionality reduction was previously already included in SCENIC+, and the eGRN AUCell matrix had been used to generate the dimensionality reductions in all the test cases with SCENIC+. To now also compare this result with other methods, we performed this approach for multiple data sets (i.e. the ENCODE cell lines data, the PBMC data, and the melanoma cell line data), and assessed whether this type of dimensionality reduction can recapitulate the cell types/states that were identified based on gene expression (in case of PBMCs) or genotype (in case of the cell lines).

For the ENCODE cell lines, we scored the predicted gene-based eGRNs with AUCell and then performed PCA to assess whether the two first PCs could recapitulate the different cell lines. We found that SCENIC+ was the most accurate method for this task, while CellOracle and Pando showed the lowest performance (Fig R11.1). **We include this analysis as Figure 3c.**

Fig R11.1. Evaluation of the eGRN-based dimensionality reduction on ENCODE cell lines. PCA plots based on the AUC values calculated using the predicted eGRNs by each of the methods (left) and Adjusted Rand Index (ARI) values, calculated using the cell line used to generate each cell as ground-truth.

For the PBMC dataset, most methods were able to separate the main cell types based on the AUC values (Fig R11.2). Except for FigR for which the dimensionality reduction is very noisy and CellOracle which seems to over-cluster the data. We speculate that this latter observation can be explained by the fact that

CellOracle performs GRN inference *per* individual cell type and is therefore able to pick up slight variations within a cell type, while other methods do the inference on a whole dataset level. The analysis using GRaNIE required too much memory and therefore could not be included (tested on a machine with 72 cores Intel(R) Xeon(R) Platinum 8360Y CPU @ 2.40GHz and 2 TB of memory). Pando performed better on the PBMC data set compared to the ENCODE data set, which could be due to using a less stringent threshold on the network compared to the ENCODE data set ($\text{padj} < 0.05$ versus $\text{padj} < 0.01$). We added this analysis to the manuscript (Fig S8) and added the following line to the PBMC section of the manuscript: “*tSNE dimensionality reduction on eRegulon enrichment scores is able to separate the main biological cell states, suggesting that the inferred eGRN recapitulates the cell states*”.

For the melanoma dataset, again most methods were able to separate the main cell types based on dimensionality reductions on the AUC values (Fig R11.3), except for GRaNIE for which the target gene AUC values are noisy, therefore also affecting the dimensionality reduction based on both target gene and region AUC values. FigR performs better on this dataset compared to the PBMC dataset. **We added this analysis to the manuscript (Fig S8).**

Figure R11.2. Comparison of dimensionality reductions based on eRegulon AUC scores for different methods in the PBMC and mm-line dataset. a. tSNE dimensionality reduction for target gene AUC, target region AUC and the combination of target gene and region AUC values, calculated using AUCCell, for (e)Regulons identified by FigR, Pando, SCENIC, SCENIC+ and CellOracle in the PBMC dataset. **b.** PCA dimensionality reduction for target gene AUC, target region AUC and the combination of target gene and region AUC values, calculated using AUCCell, for (e)Regulons identified by GRaNIE, FigR, Pando, SCENIC, SCENIC+ and CellOracle in the MM-line dataset. **c-d.** Adjusted Rand Index (ARI) quantifying how well cell type are separated based on the AUC-scores for the PBMC (c) and MM-line (d) datasets.

Second, we tested whether the inferred GRN on its own is sufficient to recapitulate the main biological states of the biological system. For this, we converted the TF-TF edges obtained from the inferred GRNs of the different tools into a boolean network, which was consequently used to simulate gene expression matrices for 500 cells using BoolODE (Pratapa et al., 2020). Interestingly, by taking the top 25% of the edges of SCENIC+, based on the triplet score (see comment 4), BoolODE was able to generate the main stable melanoma cell states (Fig R11.3). Of note, these states are (at least in part) driven by copy number variation, nonetheless the network itself is able to discover these states. In this task SCENIC+ was the best performer, with on average a smaller distance between each simulated cell and its nearest *real* cell neighbors compared to other methods (Fig R11.4). CellOracle was not able to recapitulate the mesenchymal state while the simulated cells obtained from the networks inferred by FigR and GRaNIE were more scattered (Fig. R11.3, R11.4). Pando was excluded from this analysis because the runtime exceeded 48 hours on a machine with 72 cores (Intel(R) Xeon(R) Platinum 8360Y CPU @ 2.40GHz) and 2 TB of memory. The reason for this is that Pando has many genes with an excessively large number of TF as predicted regulators and BoolODE scales exponentially with this number. **We have added this analysis to the manuscript (Fig S22) and added the following line to the melanoma section** “Furthermore, a boolean model (Pratapa et al. 2020) using the top 25% of the TF-to-TF edges of the inferred network on its own was able to recapitulate the main melanoma cell states (Fig S22).”

Figure R11.3. PCA-dimensionality reduction of simulated cells by the boolODE method (Pratapa et al., 2020) co-clustered with real cells for different eGRN inference methods. Boolean networks were generated from gene regulatory networks inferred from SCENIC+, CellOracle, FigR and GRaNIE. For SCENIC+ the top 10%, 25% and 50% of edges based on the triplet score were used. 500 cells were simulated using the boolODE method using a simulation time of 20 and the hill activation function. Simulated cells were co-embedded in PCA space with real cells after Harmony batch effect correction.

Figure R11.4. Distance between simulated cells and real cells. Violin and jitter plot of the average distance of each simulated cell to its three nearest neighbors in the first 2 principal components of PCA space. Difference in mean is assessed using two-tailed Mann-Whitney U test. And p-values are adjusted using the Benjamini-Hochberg procedure. x-axis is sorted by the median average distance.

12. The melanoma phenotype switching is derived from an earlier scRNA-seq study by the same group and new scATAC-seq data for the same 10 melanoma cell lines were generated for the current study. Although the manuscript describes many TFs discovered by SCENIC+, many of these were already discovered and reported in the earlier study and the manuscript does not clearly demonstrate what was novel in the SCENIC+ finding and what was the important difference between the SCENIC+ findings and SCENIC. In addition, it would be helpful to include a comparison with methods such as chromVAR and Signac for this section.

It is true that many TFs discovered by SCENIC+ were already described in previous publications (e.g. Wouters et al., 2020), which to us was reassuring. What is novel here, and what is the central to enhancer-GRN (compared to classical GRN inference), is that now, for each of the predicted TFs, the genomic target regions of these TFs are identified. This was impossible using SCENIC. In addition, we provide

additional novelties in the manuscript: the identification of target genes is now improved, because they are linked to the enhancers; the eGRN allows for Boolean network simulations; the eGRN allows for the prediction of perturbations; and the eGRN allows for the prediction of trajectories (GRN velocity).

In **Fig S21e** we also show that the predicted genomic regions (which was not possible with SCENIC) can be used for the inference of TF cooperativity (illustrated in the figure as the cooperativity between TFAP2A, RUNX3, MITF and SOX10). We show that these regions are highly enriched for ChIP-seq signal of corresponding TFs (**Fig S21f**).

To further illustrate this point, we show below that regions that are predicted to be targeted by any TF have on average a higher enhancer activity as measured by STARR-seq (Mauduit *et al*, 2021) (Fig R12.1a; **Fig S21i**). Also, when we rank all TF-gene-region triplets by generating an aggregated ranking based on the TF-gene importance score, region-gene importance score and maximum TF-region ranking based motif score (maximum across all motifs annotated to that TF) and take the maximum score for each region, we show that this score correlates with the enhancer activity of that region (Fig R12.1b; **Fig S21j**). Thus, using the predictions from SCENIC+ it is possible to identify active enhancers in a biological system of interest. This is of interest to researchers wanting to perform massively enhancer reporter assays (MPRAs), as SCENIC+ can be used in this context to select regions with a higher likelihood of being active compared to random ATAC-seq peaks.

We are aware that this conclusion is based on a relatively low sample size, for that reason we repeated this analysis in the Deeply Profiled cell lines from ENCODE, where additional STARR-seq data is available. **We summarize these results in an extra figure that we will add to the manuscript (Figure 4).**

Figure R12.1. Comparison of STARR-seq signal in regions in eRegulons vs regions not in regulons and correlation between triplet score and STARR-seq signal. **a.** Scatter and jitter plot showing enhancer activity as measured by the STARR-seq method (Mauduit et al., 2021) in regions targeted by any of the regulons and regions targeted by None. **b.** Scatter plot comparing enhancer activity as measure by the STARR-seq method (y-axis) (Mauduit et al., 2021) to the minimum of the triplet score over all TFs targetting and genes targetted by the region (x-axis). Labels of the regions are according to the labels in (Mauduit et al., 2021). Difference in mean is assessed using two-tailed Mann-Whitney U test. And p-values are adjusted using the Benjamini-Hochberg procedure. x-axis is sorted by the median average distance.

To test whether chromVAR (through the ArchR package) and Signac can identify the same TFs as SCENIC+ on this melanoma case study we re-ran the analysis on the scATAC-seq part using both methods. We assessed motif enrichment analysis using a standard differential test in differentially accessible regions (hypergeometric test; adjusted p-value < 0.01) and using the ChromVAR deviation enrichment test. Both procedures are detailed in the tutorials of ArchR and Signac. For ArchR, both motifs as well as ChIP-seq tracks were used as features. All methods can indeed also identify the same set of TFs as SCENIC+, however both ArchR and Signac find more TFs (Fig. R12.2). The extra TFs identified by Signac and ArchR have on average a lower expression compared to the TFs identified by SCENIC+ (Fig R12.3). These extra TFs probably represent false positives, due to the fact that Signac and Chromvar only make use of the scATAC-seq data, while SCENIC+ uses both scATAC-seq and scRNA-

seq data. Due to the lack of a large amount of ChIP-seq data on these cell lines and the absence of indicative Gene Ontology terms we can, however, not test this further.

Figure R12.2. Comparison of TFs recovered using ChromVar and Signac. Heatmap showing whether a TF is found across different methods (present: green; absent: red). Signac and ArchR were run using different options. (1) DEM: Differentially Enriched Motifs or ChIP-seq tracks in differentially accessible regions, (2) ChromVAR deviations.

Figure R12.3. Comparison of expression TFs found exclusively by Signac and ArchR. Boxplots showing distribution of the expression (log CPM) of TFs found by all methods including SCENIC+ (SCENIC+), by Signac using differentially enriched motifs but not SCENIC+ (Signac DEM), by Signac using ChromVAR deviations but not SCENIC+ (Signac ChromVar) or by ArchR using differentially enriched motifs or ChromVAR deviations but not SCENIC+ (ArchR motif DEM / ChromVar). Difference in mean is assessed using two-tailed Mann-Whitney U test. And p-values are adjusted using the Benjamini-Hochberg procedure. x-axis is sorted by the median average distance.

We also compared SCENIC+ with other (e)GRN inference methods on the melanoma cell line dataset. We re-ran the melanoma cell line analysis using CellOracle, FigR, Pando, GRaNIE and SCENIC. First, we assessed whether the other methods could recover the same TFs as SCENIC+. When testing whether TFs identified by SCENIC+ could also be recovered by other methods, Pando was again the most similar (Fig. R12.4). Other methods miss important TFs, including SOX10 (not found by FigR), SOX6 and SOX5 (not found by FigR and CellOracle). GRaNIE only identifies MITF, SOX10 and SOX9 (Fig. R12.4).

Figure R12.4. Comparison of TFs recovered using different (e)GRN inference methods. Heatmap showing whether a TF is found across different e(GRN) inference methods (present: green; absent: red). Only TFs found by SCENIC+ are shown.

Next, we wanted to assess the cell type specificity of the predicted target genes (Pando, CellOracle, SCENIC, FigR and GRaNIE) and target regions (Pando, CellOracle and GRaNIE) of each TF identified by the different methods, compared to SCENIC+. For this, we calculated AUC values for predicted target genes and regions using AUCCell (Aibar et al., 2018; Van de Sande, 2020) and visualized these values on heatmaps (Fig. R12.5). The target gene AUC values of SCENIC and FigR are most similar to those of SCENIC+, while for Pando the target region AUC values are the most similar to SCENIC+ (Fig. R12.5).

We added part of this analysis as supplementary figures (Fig S21) to the manuscript and added the following lines to the melanoma section: “FigR, CellOracle and GRaNIE could not identify many of these regulators (Fig S21h). More importantly, SCENIC+ now identifies the genomic target regions of these TFs, some of which have been previously shown that they function as active enhancers (Fig S21i-j).”

Figure R12.5. Comparison of cell type specificity of regulons inferred by different (e)GRN inference methods. Heatmap-dotplot showing scaled target gene AUC values using a color-scale and target region AUC values (if present) using dot-size scale. Regulons which were not recovered by one of the methods (compared to the regulons recovered by SCENIC+) are greyed out.

Figure R12.6. Comparison of AUC values of eRegulons inferred by different (e)GRN inference methods. Boxplot showing spearman correlation coefficient of target gene (red) and target region (blue) AUC values of eRegulons inferred by different (e)GRN inference methods.

13. The analysis with CellOracle was not particularly informative as it did not make the case for SCENIC+, which seems to be topic of the manuscript.

The purpose of this analysis was to illustrate one of the downstream analyses that can be done using the output of SCENIC+. We believe that it does make the case for SCENIC+ in the sense that SCENIC+ is

able to identify the correct target genes of TFs, so that they can recapitulate the effects of TF-knockdown. We illustrated this for the knockdown of SOX10 in the manuscript. Also, this section shows that SCENIC+, using *in silico* perturbation analysis, is able to prioritize TFs for their ability to maintain a certain cell state (i.e. when you would knockdown this TF the state of the cell changes a lot).

14. The section on conservation between human and mouse cortex also seems tangential to the main subject of the paper. The results presented in this section could reflect aspects of biology, or certain biases in the analysis that are conserved between human and mouse. Separating the two might require some deeper analysis, which would go beyond the scope of the current paper.

Thank you for the critical assessment of this section. We consider this type of evaluation, using comparative genomics, to represent a more biological benchmark compared to the previous (statistical) benchmarks against other methods. We basically assess whether between two independent data sets, from different species/genomes and techniques (10x multiomics and SNARE-seq), SCENIC+ is able to identify the same cell types and TFs, and to further assess eGRN conservation (at the level of target regions and target genes). To our knowledge, this is the first time such study is done with eGRNs inferred from single-cell multiomics data, in which not only genes, but also target regions and target genes can be compared. In addition, we provide independent resources for mouse and human that can be explored at UCSC and SCoPe, which could be of interest for the neurobiology community. Furthermore, with the analysis of the mouse cortex, we also show SCENIC+'s applicability in the mouse genome, apart from human (several data sets) and fly (eye-antennal disc).

First, we show that SCENIC+'s eGRN-based dimensionality reduction can identify the same cell types in both data sets (Fig 6). Second, we found that for approximately 50% of the TFs high quality regulons can be identified in both species (60 out of 125 and 142, respectively), and that orthologous regulons are enriched in the corresponding cell types. Of these, 52 have been described to be involved in the different cortex cell types, while the remaining 8 can be considered novel candidates. These include Smad3 in the upper cortical layers, Pparg and Bhlhe40 in L4, Etv5 and Nfat5 in the L5/6, Thrb and Pbx1 in L6 and Meis1 in oligodendrocytes. **We describe this in the manuscript:** *“While 52 out of the 60 conserved TFs have been previously described to be important for the cell types of the cortex, the remaining 8 can be*

considered novel candidates found by SCENIC+. These include Smad3 in the excitatory neurons of the upper cortical layers, Pparg and Bhlhe40 in L4 excitatory neurons, Etv5 and Nfat5 in L5/6 excitatory neurons, Thrb and Pbx1 in L6 excitatory neurons, and Meis1 in oligodendrocytes (Fig 6c, S23a-b)."

As to whether these results reflect aspects of biology, or certain biases in the analysis we make the following statement in the manuscript:

"To assess whether species-specific TFs are either not conserved or missed in one of the two analyses (i.e. due to data sparsity and/or the applied filters along the workflow) we converted the predicted target genes in the mouse regulons to human orthologs (based on ENSEMBL) and scored these mouse specific regulons in the human data set. For these regulons, we found high correlations of cell type specificity scores across species, with a median correlation of 0.79 (with 51 regulons having a correlation above 0.6), indicating that these regulators are likely conserved, but missed in the human analysis. For example, while Pou3f1/POU3F1 and Fezf2/FEZF2, previously described regulators of L5 PT and L5/6 neurons respectively⁵⁶, were only found in the mouse analysis; the human-based mouse-regulons are enriched in the corresponding cell types in the human data set, matching with the expression of these TFs (Fig S23d)."

With this analysis, we illustrate that while regulons with strong signals are identified in both cases, the quality of the technique used to generate the input data and the thresholds can have an impact on the regulons that can be identified with confidence, which we believe is of relevance for future users of the tool. **We further elaborate on this topic in a new benchmark using the simulated data from the ENCODE cell lines where we evaluate the performance of SCENIC+ depending on the number of cells and coverage (Fig S19).**

We then further explore to what extent the network is conserved between species. This is, for both species, we identified target regions that are bound by orthologous TFs and assessed to what extent orthologous TFs target orthologous genes. Overall, we found little conservation, with only 28% and 6% of the target genes and regions conserved. Though it is difficult to assess whether this is due to technical reasons (e.g. data sparsity, technique) or biological (i.e. evolutionary turnover), we were able to prioritize which regulons are the most conserved, with the inhibitory neuron regulons as the most conserved overall. We identify 4,798 and 8,318 conserved TF-region and TF-gene relationships respectively. Given

the sparsity of direct TF-enhancer and TF-target gene relationships in the literature, the majority of these conserved interactions are novel. Importantly, in combination with our GRN velocity approach, we identify Meis1 as a key regulator of oligodendrocyte maturation, which we show is conserved between mouse and human. **These observations are now described in the text.**

We believe this resource could be further exploited to assess differential regulatory target conservation between TFs, or to prioritize conserved sequences to develop cell-type specific vectors, and we now discuss these applications of single-cell eGRNs in the discussion. Nevertheless, in this study we focus on the validation/proof-of-concept of the tool and providing public resources for users to explore (and expand on) SCENIC+ insights, and the latter applications are out of the scope of this study.

15. GRN velocity is a concept which might be worth expanding on. Related ideas and methods are reported by Ma et al, Cell, 2020 and ref 69 (Li et al). and it would be of interest to compare the results produced by the respective methods. The paper does not present algorithms that have been developed specifically for multiomic data latent space analysis (for example Gong et al, Genome Biology, 2021, Minoura et al. Cell Reports Methods, 2022; Lynch et al, Nature Methods, 2022; Lin et al. scJoint: Nature Biotech, 2022). As the pseudotime analysis depends on the latent space the trajectory inference, to what degree do the results of the GRN velocity change depend on these upstream processing steps?

Thank you, we also find that eGRN-based prediction of cell state dynamics is novel and interesting. The approach we present here relies on pseudotime ordering performed using the eGRN enrichment in the cells inferred using single cell multiomics data. The methods proposed by the reviewer focus on the integration scRNA-seq and scATAC-seq in the same latent space [Cobolt (Gong et al., 2021), scMM (Minoura et al., 2022), MIRA (Lynch et al., 2022), scJoint (Lin et al., 2022)]; however, cells are not mapped in a one-to-one basis, and in the case of single-cell multiomics data, we would end up with two data points per cell. These approaches could be compared with other approaches for data integration such as CCA, and then used for data pairing using the FigR approach to mimic single-cell multiomics data (as we have shown in comment 1). Using CCA together with FigR's cell pairing algorithm, we have shown that SCENIC+ results are comparable to those obtained using true multiomics data. We believe that these

methods could be compared with other algorithms to integrate independent single cell omics modalities; nevertheless, this is not the purpose of SCENIC+.

GRN velocity describes the differences between the expression of a TF and the changes in its target regions and/or target genes. In Ma *et al.*, the authors compare the lag between chromatin accessibility around a gene (e.g. gene activity) and the observed gene expression, without including TF information. Unfortunately, the code to reproduce the Ma *et al.* analysis is not available in the manuscript. Velocity methods on the other hand, such as scVelo, measure the lag between unspliced and spliced mRNAs per gene, without incorporating information of chromatin or regulating TFs. Recently, Li *et al.* presented MultiVelo, which combines the two latter concepts. MultiVelo uses the lag between chromatin accessibility around a gene, unspliced mRNA expression, and spliced mRNA to order the cells. To illustrate the conceptual differences between GRN velocity and to further assess whether our eGRN based pseudotime ordering agrees with velocity-based ordering, we have now included scVelo and MultiVelo analyses for the eye antennal disc data set (where developmental dynamics are most pronounced). While scVelo and MultiVelo return point values of directionality (one ‘pseudotime’ measurement per cell), GRN velocity provides one vector per cell and per TF, in which each value indicates the directionality to which the activation of a regulon pushes a cell (Fig R15.1). Importantly, we observe that both scVelo and MultiVelo correctly predict the differentiation path, including the branching event between photoreceptors and interommatidial cells.

Figure R15.1. Comparison between the outputs of our eGRN approach (directionality vectors per regulon, 8 regulon-specific vector maps are shown as example) and velocity approaches (global directionality vectors).

Figure R15.2. *scVelo* velocity per gene. The first plot shows the unspliced versus spliced ratio, the second plot shows the velocity in each cell for that gene and the third plot shows the gene expression. There can be a lag between the splicing and the eGRN velocity, RNA velocity can be positive or negative in the cells where the eGRN velocity indicates TF activity, and velocity patterns can be random.

While *scVelo* permits to visualize the velocity of each gene in a cell, these values do not always correspond to the eGRN velocity, as they are based on the splicing of the mRNA of the TF rather than the difference between TF expression and its target genes (Fig R15.2). For instance, while the velocities of

Optix and lz genes are indeed higher in the cells where the eGRN velocity predicts their TF activity; for Onecut and gl, scVelo velocity is higher in the cells right before the ones (in the differentiation trajectory) where we observe the strongest eGRN velocity. This may recapitulate the processing between TF expression and splicing prior to the activation of its target genes. In other cases, the velocity seems to have a rather random pattern (e.g. ey) or is negative (e.g. ato, sens), which indicates downregulation of splicing, in the cells where the eGRN velocity shows the strongest TF activity.

As the reviewer suggests, the eGRN velocity depends on preprocessing steps, namely on having an accurate pseudotime order of the cells. In our study, we have used eGRN enrichment (AUCCell values) to perform dimensionality reduction and pseudotime ordering, resulting in the expected ordering of the cells and cell types based on the previously known differentiation steps, and as validated with PAGA (Fig R15.3). We recommend the users to follow this approach. To illustrate the effect of the pseudotime order, we also inferred eGRN velocities using scVelo pseudotime, which are noisier, especially in the cell types anterior to the morphogenetic furrow (i.e. more mixing between pseudotime values between differentiation stages rather than a gradual increase, Fig R15.3). While the eGRN velocities follow similar trends, we observe more sparseness of the arrows when using the noisier velocity pseudotime (Fig R15.4). Altogether, while having an accurate pseudotime is required for the inference of accurate eGRN velocities, we provide a recommended workflow, using the eGRN-based pseudotime ordering, that we have tested both in unbranched (oligodendrocytes) and branched (eye disc) trajectories. More details on this approach are available in the tutorials at <https://scenicplus.readthedocs.io/en/latest/>

We have included these analyses in Fig 7 and Fig S28 and added following paragraph to the manuscript:

“To assess whether velocity methods could give the same insights as eGRN velocity, we compared our approach with RNA velocity (based on the splicing of the mRNA of the TF, Fig S28a). For instance, while the RNA velocities of Optix and lz genes are indeed higher in the cells where the GRN velocity predicts their TF activity; for Onecut and gl, RNA velocity is higher earlier in differentiation trajectory compared to where the eGRN velocity is highest. This may recapitulate the processing between TF expression and splicing prior to the activation of its target genes. In other cases, RNA velocity patterns seem to have a rather random pattern (e.g. ey) or they are negative (e.g. ato, sens), which indicates downregulation of splicing, in the cells where the eGRN velocity shows the strongest TF activity. Importantly, we also

calculated eGRN velocities using the eGRN-based pseudotime and the RNA velocity pseudotime, finding that the latter results in sparser differentiation arrows (Fig S28b-d).”

Fig R15.3. Differentiation trajectories and pseudotime ordering. a. PAGA transition graph between cell states along oligodendrocyte differentiation. **b.** PAGA transition graph between cell states along the eye disc differentiation. **c.** Eye disc UMAP colored by eGRN-based pseudotime. **d.** Eye disc UMAP colored by velocity based pseudotime.

Fig R15.4. Comparison between eGRN velocities inferred using the eGRN based pseudotime (top) and the velocity pseudotime (bottom).

16. In the fly retina analysis of GRN velocity single cell ATAC-seq and RNA-seq data are not generated from the same cells using multimodal analysis. This seems problematic as there might be differences between the gene expression and the chromatin states of the cells which would be hidden by the matching procedure. Again, no comparison is made with other methods. It is not clear that the GRN velocity approach was important in this analysis, or whether similar conclusions could be reached using alternative methods.

To map the scATAC-seq and the scRNA-seq data, we used ScoMAP, which maps differentiating cell types based on the ranked pseudotime between the layers rather than projecting cells into a common space (by calculating gene activities from the scATAC-seq data) and matching neighbors. As a ranking based approach, the cells are assigned by order, rather than closest neighbors based on gene expression and gene activities, which tends to disregard the potential lag effects (because it requires the gene expression and gene accessibility patterns to be the closest for the mapping). We validated the mapping between the layers extensively in Bravo et al (2020) using enhancer activity data from Janelia Farm and stainings. In addition, MultiVelo, which uses both chromatin accessibility and gene expression, was able to unsupervisedly infer the differentiation trajectory from the paired data (Fig R15.1).

As we describe in comment 15, eGRN velocity does provide complementary insights compared to scVelo and MultiVelo, as it indicates the lag between the expression of a TF and the activation of its target genes rather than splicing, or accessibility and splicing, of the gene. Both scVelo and MultiVelo focus in inferring the *overall* differentiation trajectory (as a unique vector map), while eGRN velocity measures *the relevance of each TF* to push cells in a certain direction. We further illustrate the conceptual differences in Fig R15.2, where we show that the velocities inferred by scVelo do not always agree with the eGRN velocity (i.e. there can be a lag between the splicing and the eGRN velocity, velocity can be positive or negative in the cells where the eGRN velocity indicates TF activity, and velocity patterns can be random). **We have included these analyses in Fig 7 and Fig S28.**

17. In the spatial GRN analysis, true multiomic data was once again not used. In this section GRNs are inferred then mapped onto spatial data. This section does seem to add much to the manuscript.

There may be a misunderstanding here, because these analyses did use true multiomic data, and as explained in comment 1, four of the seven case studies in our manuscript use true multiomic data. For the spatial GRN analysis we used the 10x single-cell RNA+ATAC *multiome* data set (<https://www.10xgenomics.com/resources/datasets/frozen-human-healthy-brain-tissue-3-k-1-standard-1-0-0>) as well as our newly generated 10x single-cell RNA+ATAC *multiome* data of the mouse cortex. These data were used to infer eGRNs in the cerebellum and mouse cortex, respectively. eGRNs were then

mapped onto the spatial templates. While we believe this approach is interesting to visualize GRN enrichment in spatial templates, we have moved this figure and its description to supplement (Fig S24).

Reviewer #2:

Remarks to the Author:

González-Blas and colleagues propose a novel computational method (SCENIC+) for inference of enhancer-based gene regulatory networks (eGRNs) from multimodal (scRNA- and scATAC-seq) single cell sequencing. For this, they leverage expertise and previous methods from the own group such as the use of cisTopic for denoising sparse single cell ATAC-seq data, a meta-database of TF motifs (Motif Collection database/cisTargets), methods to characterize cis-regulatory regions (pyCisTarget & DEM) and association of TF to enhancers to genes (based on Boosting trees/Arboreto). These are all combined together as SCENIC+. Some of the individual components (DEM, Motif Collection database, Velocity) are novel. Some components were improved, i.e. cisTopic has a faster implementation/estimation algorithms. This is altogether a very important framework for inference of regulatory networks. A few recent competing methods (FigR, GRaNIE, Inferelator, scMega) provide equivalent core features (estimation eGRNs), while SCENIC+ includes additional aspects as methods for in silico perturbation and differentiation analysis. The method is evaluated in distinct data sets including ENCODE chromatin data (used for benchmarking), PMBC multimode data, melanoma cell lines, human/mouse cortex cells, fly eye-antennal disc and mammalian brain with mostly convincing results.

The method is overall complex including many sub-methods and options. Some important details were missing, and descriptions of some methods were hard to follow. As a judge this as a method manuscript, focus could be given in method description/benchmarking, and less in the interesting but mostly anecdotal data analyses. For example, estimation of differentiation velocities and spatial mapping are interesting, but there is lack of methods details and rigorous evaluation. This makes it hard to conclude their value. Overall, this manuscript would benefit from focus on main contributions.

We thank the reviewer for the positive assessment of our work and the useful suggestions to improve the manuscript and the usability of SCENIC+. We have improved the text and the descriptions of the method and the analyses performed. We include now several additional benchmarks, and better describe the relevance and novelties found through the analyses.

1. The reported improvements on cisTopic are welcomed (efficient algorithm and additional model selection approaches), but some important method descriptions are missing. For example, how is a final topic decided from the four available model selection criteria (lines 824-834 & Fig. S2b)? Authors should also give more details on how to choose the topic binarization (lines 843-851). For example, in which type of analysis "balanced region sets" are more desirable? Finally, it is unclear if topics are altogether used by SCENIC+, as analysis seen to focus on cell types/DARs. Further explanation and a table enumerating how these (and other main parameters /methodological choices) were set per data set would be helpful.

Thank you for the suggestion to improve the method descriptions.

1) Regarding the selection of the best model in cisTopic (lines 824-834) **we added the follow paragraph:**

“The best model (i.e. the one with the optimal number of topics) is the model with the smallest number of topics where coherence (Minmo_2011) and Log-likelihood are maximized and Arun_2010 and Cao_Juan_2009 are minimized.”

2) Regarding topic binarization (lines 843-851) **we have adapted the following paragraph:**

Original manuscript: *“We have included several binarization methods (applicable for topic-region and cell-topic distributions): 'otsu' (Otsu, 1979)107, 'yen' (Yen et al., 1995)108, 'li' (Li & Lee, 1993)109, 'aucell' (Van de Sande et al., 2020)20 or 'ntop' (Taking the top n regions per topic). Otsu and Yen's methods work well in topic-region distributions; however, for some downstream analyses, it may be convenient to use 'ntop' to have balanced region sets. By default, pycisTopic uses Otsu for binarization.”*

Revised version: “We have included several binarization methods (applicable for topic-region and cell-topic distributions): 'otsu' (Otsu, 1979)¹⁰⁷, 'yen' (Yen et al., 1995)¹⁰⁸, 'li' (Li & Lee, 1993)¹⁰⁹, 'auzell' (Van de Sande et al., 2020)²⁰ and 'ntop' (Taking the top n regions per topic). Otsu and Yen's methods work well for topic-region distributions; however, for some downstream analyses, it may be convenient to use 'ntop' to have balanced region sets (e.g. training of classification models). By default, pycisTopic uses Otsu for binarization. The reason for this is that Otsu, empirically, results in the largest number of regions per topic while limiting the amount of “noise” for motif enrichment analysis.”

3) Regarding whether topics are used in the SCENIC+ analysis: yes, topics are used to run SCENIC+ by default (as is detailed in the methods section). To showcase the importance of using topics in SCENIC+, we now show that regulatory topics recapitulate better functional enhancers compared to the current standard, which relies on the calculation of Differentially Accessible Regions (DARs) (**Fig 1d**). **To make this clearer we have added Table S5**, enumerating the parameters used for each analysis including whether these parameters are default or not.

2. pyCisTarget is rather complex part of SCENIC+. It consists of 3 main components: a novel TF motif db, methods for motif enrichment (DEM, homer and cisTarget) and a cisTarget database.

2.1 The nature of cisTarget database is unclear and hard to follow. In my understanding, this is a matrix encoding region vs. motif enrichment scores. How are the regions defined? I Aren't this data set specific? Can regions derived in a data set (PBMCs) be used in other data (cortex cells)? Or do authors recommend using pre-defined SCREEN or whole genome regions? This should be clarified (and again choices for each analyzed data explained).

We apologize that these details were missing from the Methods section. **We now added more detailed descriptions of the cisTarget method and databases (Supplementary Note 1)**.

1) The motif databases are indeed matrices encoding region versus motif (rankings or motifs scores, to use with cisTarget and DEM, respectively). As for the regions, there are two options.

These can be either dataset specific consensus peaks or a pre-defined set of regions (for example, we provide precomputed databases on the SCREEN regions). In the latter case, the predefined regions are intersected with dataset specific regions internally, however this approach usually results in the loss of certain regions that are not included in the SCREEN database. We certainly do not recommend using regions derived from a certain data set (e.g., PBMCs) in other systems (e.g., brain cortex).

2) Regarding scoring and ranking: a score is obtained for each region and each motif by using Cluster-Buster (Frith et al., 2003) (i.e. these scores are *not* enrichment values, but log likelihood ratios obtained from the Hidden Markov Model). Next, a ranking is generated for each motif across all regions (i.e. the region with the highest score for a certain motif gets the value 0 for that motif, the second region the value 1 and so on). This rankings database is used by the cisTarget algorithm where an Area Under the recovery Curve (AUC) and normalized enrichment score (NES) is calculated (similar to gene set enrichment analysis) to assess how much a foreground set of regions is enriched in the top of a ranking (the AUC and NES are calculated for each motif). A NES is obtained for a certain motif in this algorithm when a significant subset of the input regions is located at the top of the ranking of that motif.

Besides the rank database, we also keep a raw score database that is used in the DEM algorithm, where a Wilcoxon rank-sum test is used to compare the motif scores on a set of foreground regions to the scores on a set of background regions.

In general, we recommend using dataset specific databases. That way the result is independent on how much the ATAC-seq peaks overlap with the SCREEN regions. Generating such a database requires a fair amount of resources, and we provide the necessary scripts and tutorials to generate them. For a faster analysis, we provide the precomputed SCREEN databases.

These details are now included in the Methods section and Supplementary Note 1.

2.2. Authors mention the detection of CRMs (or cluster of motifs) using Cluster-Buster (line 131), whose scores seem to be based on individual motifs (see point 2.1 and figure 1d). This needs clarification.

CRMs can be homotypic or heterotypic (Lifanov et al., 2003). It is correct that the CRM scores calculated in cisTarget represent *homotypic* CRMs, thus for each TF motif we calculate a CRM score. The reason they represent clusters is that the HMM detects clusters of motif instances within a given region (i.e., multiple binding sites for the same TF can be identified, and more instances yield higher log-likelihood scores). The HMM also has the advantage that multiple PWMs can be included as hidden states. In the original Cluster-Buster paper, this principle was used to insert PWMs from different TFs (to detect *heterotypic* CRMs). We exploit this principle to use multiple PWMs for the same TF (i.e., from the same motif cluster). We show in our motif benchmark analysis that this new idea yields a higher accuracy for motif enrichment, since the HMM can identify the optimal path through the background and PWM states, allowing for the identification of the best TF binding sites, using a pool of slightly different PWMs.

We clarified this in Fig 1f and the Methods section.

2.3 In their evaluation (Fig. 1F) CisTarget outperforms DEM. So, it is unclear why DEM is needed in SCENIC+.

The benchmark shown in Figure 1f is meant to identify enriched motifs from ChIP-seq regions. DEM is designed to identify differentially enriched regions between two different region sets. cisTarget uses the whole genome as background, whereas DEM requires to define a set of background regions. In this case, we used as background regions sampled from other ChIP-seq tracks (5,000 regions), which may not be ideal. While cisTarget works better in this setting, we now include a case study to showcase the use of DEM to identify differential partner TFs of SOXE factors in melanoma, oligodendrocytes, and astrocyte (Reviewer 1 – comment 2). In this

case study, cisTarget is only able to identify SOXE motif, while DEM identifies the motifs that are different between the different region sets. We believe that cisTarget and DEM are complementary approaches, and thus, we use them together in the SCENIC+ pipeline. We further describe this in Fig S5.

2.4 The benchmarking of distinct motif databases and representation is an interesting aspect (Fig. 1f and Sup. S3). Fig. 1F has strong results but underlying experiments are unclear. It is hard to follow some of the analysis in Sup. S3, as for example several of the terms [(u),(c),(p) in panel h) are not clearly defined. Since, this database is a SCENIC+ novel aspect, more details on the benchmarking are needed.

Briefly, to benchmark the different motif enrichment techniques included in pycisTarget and the approaches to build databases, we assessed the recovery of target TFs using 309 ChIP-seq data sets from the Deeply Profiled Cell Lines collection from ENCODE (Table S4). These tracks have also been included in Unibind (Puig et al., 2021) and present the TF motif, showing that their quality is good enough to find the target TF motif back. Motif enrichment was performed using Homer, cisTarget and DEM. For the latter two, three different approaches for creating the motif databases were used: 1) generating a database without clustering the motif collection and only retaining annotated motifs (24,309 motifs, named as unclustered (u)), 2) generating a database using a single consensus motif (or archetype) for each of the STAMP motif clusters (named as archetype, (a)) and 3) generating a database by scoring regions using the ensemble of motifs in a cluster (names as clustered, (c)). In addition, since our motif collection contains licensed motifs from Transfac Pro, we also benchmarked cisTarget and DEM using a publicly shareable clustered collection (using all PWMs for scoring but removing the Transfac Pro motifs), finding equal TF recovery and comparable scores in regions, named as public (p)). Our results show that using cisTarget (or DEM) with the clustered collection (using all PWMs for scoring) recovers the highest number of target TF motifs in the top positions, followed by Homer. Note that DEM is meant to be used to compare foreground and background region sets, and in this analysis, we used

regions in other tracks as background. We found that scoring with archetypes performs worse compared to using all PWMs in the cluster, especially for clusters where there is more variability. Finally, we also compared the target regions (or cistromes) recovered by each approach with those reported by Unibind as ground truth, finding that both cisTarget and DEM have higher precision and recall compared to Homer (Fig S4).

We have added to the legend of the supplementary figure the meaning of the abbreviations:
unclustered (u), archetype (a) and clustered (c), public (p).

3. Another major contribution of SCENIC+ is the creation of the eGRNs. Authors should formalize or schematized the description of eRegulon (line 1016-1022), which is unclear. Also, some important aspects (need for a specificity score - line 1037) is not justified in the manuscript.

Regarding the description of an eRegulon **we adapted line 1016-1025** from:

“eGRN compilation: First TF-to-region-to-gene modules are generated by intersecting the binarized region-to-gene relationships with TF-to-region cistromes by region name. Next these modules are pruned by running GSEA, gsea_compute function from the GSEAPy python package (v0.10.8), using the TF-to-gene importance scores, for all TFs and genes in the module, as ranking and genes of the module as gene set. The leading-edge genes along with the binarized regions linked to those genes are retained, generating eRegulons. This analysis is run separately for TF-to-gene and region-to-gene relationships with positive and negative correlation coefficients, for a total of four GSEA runs. By default, eRegulons with less than 10 predicted target genes or obtained from region-to-gene relationships with a negative correlation coefficient are discarded.”

To:

“eRegulon creation: We define an eRegulon as a TF together with its target regions and genes. To generate this, information from gene expression, region accessibility and motif enrichment are combined. For each TF, TF-region-gene triplets are generated by taking all regions that are enriched for a motif

annotated to the TF and all genes linked to these regions, based on the binarized region-to-gene links (see “Calculating TF-gene and region-gene relationships”). However, we only want to include genes, and the regions linked to them, in the final eRegulon if they are significantly co-expressed with the TF. To determine this, we use Gene Set Enrichment Analysis (GSEA). Here, all genes are ranked based on the TF-gene importance scores and an enrichment analysis of the set of genes in the triplet compared to the overall ranking of the genes is computed, using the gsea_compute function from the GSEAPy python package (v0.10.8). Finally, only the genes at the top of the ranking, known as the leading edge are retained in the final eRegulon. This analysis is run separately for TF-gene and region-to-gene relationships with positive and negative correlation coefficients, for a total of four GSEA runs. By default, eRegulons with less than 10 predicted target genes or obtained from region-to-gene relationships with a negative correlation coefficient are discarded.”

The Regulon Specificity Score (RSS) is used to identify marker regulons that differentiate between clusters or cell types. We use the RSS for plotting regulon enrichment (as it normalizes the AUCCell values) and to prioritize regulons per cell type (to select the most specific regulons for plotting, or to prioritize the differentiation velocities). We have added this explanation in the methods section.

“(…) The Regulon Specificity Score (RSS) is used to identify marker regulons that differentiate between clusters or cell types. We use the RSS for plotting regulon enrichment (as it normalizes the AUCCell values) and to prioritize regulons per cell type (to select the most specific regulons for plotting, or to prioritize the differentiation velocities).”

4. The extensive benchmarking (Fig. 3) is welcome, but some aspects are difficult to. What does the recovery of TFs (F3c & S5a) mean? Are these true positives? What is the set of positives here? What are “marker gene TFs” (line 209)? Altogether, CellOracle (or Pando according to fig. 3) does perform better in absolute TF numbers. The justification is based on lack of cell specificity of CellOracle, whoever shouldn't the benchmarking approach account for cell specificity? True positives are only TFs predicted in the correct cellular context, i.e. cell type used for the ChIP-seq experiment? Authors should improve their benchmarking approach to account of cell specificity of TFs.

Thank you for pointing out how we can improve the clarity of the benchmark results.

1) Regarding the recovery of TFs (Fig 3c):

We agree that the word “recovery” is confusing here. We meant the number of TFs identified/found by each tool. In other words, they are not the number of true positives, just the raw number of TFs (regulons) found. Finding a higher or lower number of TFs is not per se indicative for the performance of the tools. For this reason, **we replaced the following paragraph** (lines 205 – 211):

“We first assessed how many TFs were recovered by each method, and how many target regions and genes were predicted per TF (Fig 3c, S5a-e). SCENIC+ identifies 178 TFs across the eight cell lines (Fig S5, S6).”

With:

“In this data set, SCENIC+ identified regulons for 178 TFs (Fig 3b, S10). SCENIC, GRaNIE, FigR and Pando identified fewer regulons (for 108, 39, 71 and 157 TFs respectively). CellOracle identified regulons for more TFs (235).”

2) Regarding the question about marker gene TFs (line 209): with this we mean TFs that are significantly differentially expressed across cell types.

3) Regarding the question about the performance based on the absolute number of TFs and the cell specificity of CellOracle’s predictions. As stated above (Reviewer 1 – Comment 6), the absolute number of TFs does not directly reflect the performance of the tool. We included these numbers to provide a notion to the reader on the number of TFs each tool can identify and to what extent these TFs are relevant. We noticed that CellOracle finds a large subset of TFs that are not found by any other tool (see Fig S11d-f, S12). Most of these TFs are very general regulators (i.e. not cell type specific). For that reason, we included the explanation of the lack of cell type specificity of CellOracle. To make this clearer, **we replaced the following paragraph:**

“The latter can be explained because CellOracle does not focus on cell type specific TFs, but also predicts broadly expressed TFs (e.g., GABPA, YY1, SPI1), while the other methods, primarily SCENIC+

and GRaNIE, predominantly predict cell type specific TFs (showing higher tau values for specificity) (Fig 3d, S5e).”

With:

“CellOracle showed little recovery of differentially expressed TFs, while it recovers a large fraction of non-cell line specific TFs (e.g. GABPA, YY1, SPI). In contrast to other methods, CellOracle infers eGRNs per cluster rather than using other cells as background, which makes CellOracle less sensitive to pick up cell type specific signals, while being more sensitive to features that are ubiquitously present in all cells. Accordingly, TFs exclusively identified by CellOracle have the lowest cell line specificity scores (based on tau value, see Methods), while the other methods, primarily SCENIC+ and GRaNIE, predominantly predict cell type specific TFs (Fig S11e-f, S12).

In conclusion, this part of the text was not really part of the benchmarking itself but more a summary stating what kind of TFs each tool is able to find.

5. Benchmarking results on TF-region, Region-Gene and TF-gene are mostly convincing. Authors should discuss however the best precision of Granie of the TF-region problem. It is also unclear why SCENIC is evaluate with Rho and Importance in figure 3f (and Rho is also not defined anywhere). This should be justified. Also, the interpretation of the NES vs. FC plots in panel 3g is unclear. Why has STAT5 a negative NES and HOXB9 a positive after their KO? Repressor TFs? This should be explained in the text.

Reviewer 1 also suggested to improve the benchmark analyses.

a. We have now expanded the TF-region benchmark using different standards: 1) Regions from Unibind (ChIP-seq peaks with the TF motif present), 2) ChIP-seq peaks and 3) Enformer predictions (Avsec et al., 2022) (Fig R5.1 and **Fig S13**). In the three comparisons both SCENIC+ and GRaNIE perform well, without significant differences in the F1 metric. While both methods can predict bona fide target regions, SCENIC+ predicts 3x more TFs (178 versus 39, Fig3). In addition, GRaNIE was designed for bulk

ATAC+RNA data, and not for single-cell data; consequently, it is highly inefficient and could not successfully be run on the PBMC data set. **In the text we have added:** “Overall, the predicted target regions of SCENIC+ and GRaNIE have the highest precision-recall, followed by Pando and CellOracle (Fig 3e, S13a-d). Nevertheless, SCENIC+ identifies three times more TFs than GRaNIE.”

b. To infer region-gene links, we use both the correlation between the imputed accessibility of the target regions in the search space and the expression of the gene (ρ) and the GBM importance to assess non-linear relationships. While by default we use ρ to classify region-gene interactions as positive or negative (keeping only positively correlated interactions) and the GBM scores to prune true interactions (using BASC for automatic thresholding), the user can also choose to use ρ for pruning. We believe that it is of interest to show that both metrics correlate with Hi-C data, with the ρ scores being even slightly more correlated (possibly due to modelling linear relationships). We removed the correlation-based evaluation from the main figure (still present in supplementary **Fig S15** but using the label “spearman” instead of “ ρ ”) because by default it is not used in SCENIC+ and **we have added the following sentence to the manuscript:** “Furthermore, we compared the performance of the region-gene inference models included in SCENIC+, namely Gradient Boosting Machines (GBMs), Random Forest (RF) and Spearman correlation, to seven other regression models (see Methods). We found that Spearman correlation, GBMs and RFs perform well, as previously observed in TF-gene inference benchmarks (Fig S15b,d,e).”

Fig R5.1. F1 score, precision and recall distributions of the predicted regions per TF using three different standards. Violin plots represent the F1 score (left), precision (middle), and recall (right) distributions of the predicted target regions per TF using different standards: ChIP-seq peaks with TF motifs (Unibind, top), ChIP-seq coverage (middle) and enformer predictions (bottom). GATA1, GATA2, HNF4A and STAT5A are indicated if the method has found the regulon. The numbers above or below the

violins indicate the number of TFs that have been found by the methods and are included in the standards. Pairwise wilcox-test p -values between distributions are shown on top of the violins.

c. Regarding the TF knockdown benchmark: we observe that if a TF is a repressor (with a negative regulon, meaning that the TF expression is anticorrelated with target region accessibility and gene expression, e.g. HOXB9, ARID3A), the target genes are upregulated upon TF knock-down (with a positive NES). If the TF is an activator, upon knockdown the target genes are downregulated (e.g. STAT5A, GATA1). However, many of these TFs are targets of other TFs, thus the knockdown of one TFs will indirectly affect the expression of the target genes of downstream TFs. For instance, in Fig R5.2, we show that SCENIC+ predicts that STAT5A directly regulates the repressors ARID3A, BHLHE40, HOXB4 and HOXB9. STAT5A knockdown causes indeed the downregulation of these TFs (shown by TF LogFC), and therefore the target genes of these repressor TFs are upregulated (derepressed). While SCENIC+ predicts that STAT5A does not regulate GATA1 and GATA2, we observe that they share common target genes, which can explain the observed downregulation of their target genes (but not TF expression, as shown by TF LogFC). **We have clarified this in the figure legend (Fig S16).**

Fig R5.2. TF knockdown perturbation and downstream effects. a. TF-gene (TF) network showing key regulators of K562. **b.** Example of the knockdown of STAT5A in K562 showing the GSEA $-\log_{10}$ adjusted p -value and NES of the key regulons in K562. The dots are colored by the LogFC of the TF expression.

d. We have added a new analysis where we evaluate the predicted TF-gene relationships based on *in silico* TF perturbations (as detailed in the melanoma section of the manuscript). For this, we computed the

predicted LogFC of gene expression upon perturbation and compared the predictions with the observed effects. Correlation between the observed and the predicted perturbation variation shows that SCENIC+ networks are the most accurate in predicting the perturbation effects (Fig R5.3 and Fig S16). As an illustration, we show the comparison between the predicted and observed effects upon GATA1 (activator) and ARID3A (repressor) perturbation, where SCENIC+ can also predict indirect effects for downstream genes that are not predicted to be direct targets of the TF (in grey, after 5 iterations).

Figure R5.3. Comparison between simulated perturbation effects and TF knockdown experiments from ENCODE. **a.** Spearman correlation between predicted LogFC with *in silico* TF perturbation for each method versus the observed LogFC changes upon TF perturbation. **b.** Comparison between predicted and observed LogFC changes upon GATA1 KD. Dots in red indicate genes in the GATA1 regulon. **c.** Comparison between predicted and observed LogFC changes upon ARID3A KD. Dots in red indicate genes in the ARID3A regulon, predicted as repressor.

6. The *in-silico* perturbation analysis is an interesting approach. Here, authors could reinforce the novelty of the proposed method in relation to CellOracle (linear vs. non-linear?). The need for several interactions needs to be better justified in the main text (predicted TF expression is used in interactions?). Also, can the model predict which TFs have best perturbation effects (or why only SOX10 and ZEB1 were chosen)? Could this approach be benchmarked more systematically by using the ENCODE data (explored in Fig. 3)?

Thank you for the appreciation of this section. The purpose of this analysis was to illustrate potential downstream analyses that can be done using the output of SCENIC+. Such analyses depend on the quality of the inferred eGRN, thus as eGRN inference improves, we expect that downstream applications become more performant. The method we used for the *in silico* TF perturbations is indeed not particularly novel compared to CellOracle, but we show that the SCENIC+ eGRNs are indeed well suited for perturbation simulations.

To clarify the need for several interactions (we assume the reviewer means the need for several iterations of gene expression simulation) **we adapted line 313** from:

“To include indirect effects, we predict new gene expression values for several iterations.”

To:

“To account for indirect effects (i.e. TFs targeting other TFs), perturbed gene expression values are propagated over several iterations. Here, the predicted TF expression of the previous iteration is used to predict the gene expression matrix of the following iteration.”

Regarding the question about which TFs have the best perturbation effects: yes this is possible, and this we show in Figure 5g. Here, we predicted the effect of knocking-out each TF in turn. In the heatmap we show the strength of each TF, as their ability to switch cells along the melanocytic-mesenchymal axis after their knock-out. The top 10 TFs for switching cells to the mesenchymal fate and the top 10 TFs for switching cells to the melanocytic fate are shown. SOX10 and ZEB1 were chosen as two examples for the “arrow-plots” (Figure 5f) because they were predicted to have very strong effects according to the screen described above.

Regarding the question of benchmarking this technique more systematically based on ENCODE data, we assessed the correlation between the log Fold Change of real TF knockdowns versus the *in silico* predicted LogFC, as described in the previous comment (Fig R5.3). Regarding to this question **we have added the following line to the manuscript**: *“This type of modeling can partially explain direct and indirect changes upon TF perturbation in the deeply profiled cell lines from ENCODE (Fig. S16f).”*

7. The mouse/human eGRN is neat but descriptive. It is unclear if any new finding comes from this analysis.

We agree that it is interesting to elaborate on the novelty of the findings in the human/mouse brain analysis. From a comparative genomics point of view, we consider the predictions that are shared between human and mouse, for corresponding cell types between the species, to be high-confidence findings. **We now explain this in more detail in the text.**

1) 60 TFs are identified by SCENIC+ in both the human and mouse cortex; of these, 52 have been described to be involved in the different cortex cell types. The remaining 8 can be considered novel candidates. These include Smad3 in the upper cortical layers excitatory neurons, Pparg and Bhlhe40 in L4 excitatory neurons, Etv5 and Nfat5 in L5/6 excitatory neurons, Thrb and Pbx1 in L6 excitatory neurons and Meis1 in oligodendrocytes.

2) Besides these 60 TFs, we identify additional TFs that are identified by SCENIC+ in only one species, but after investigating the homologous regions and target genes, we conclude that these are likely also conserved in their function; this shows a novel approach to predict additional TF regulons that are not identified through the default settings. For example, while Pou3f1/POU3F1 and Fezf2/FEZF2, previously known regulators of L5 PT and L5/6 neurons respectively, were only found in the mouse analysis (10x multiomics); the human-based mouse regulons are enriched in the corresponding cell types in the human data set, matching with the TFs expression.

3) For these conserved TFs, we identify 4,798 and 8,318 conserved TF-region and TF-gene relationships respectively. Given the sparsity of direct TF-enhancer and TF-target gene relationships in the literature, the majority of these conserved interactions are novel.

4) We provide these eGRNs as resources for mouse and human that can be explored at UCSC and SCoPe, which could be of interest for the neurobiology community. To our knowledge, such a resource does not exist yet, and is thus also novel.

5) In the analysis of oligodendrocyte differentiation with our GRN velocity approach, we identified Meis1 as a novel regulator of oligodendrocyte differentiation. Meis1 has been previously described to be

involved in early neurogenesis and hematopoietic differentiation (Isogai et al., 2022), but not in oligodendrocyte maturation. In addition, we also find this TF conserved in the human cortex, with 20% of the target genes conserved between human and mouse.

We have added more detailed descriptions of these novelties to the results and discussion sections of the manuscript.

Besides the biological novelty of conserved eGRNs, we also find this case study important to illustrate that SCENIC+ works well on mouse data sets (the other case studies are either human or Drosophila).

8. The GRN velocity is an interesting (and novel) contribution. However, more details are needed. For example, I miss how the trajectory analysis was performed in the oligodendrocytes data and how the sub-trajectories (“OPC to NFOL”, “NFOL Maturation” and NFOL to OL”) are defined. Authors need to describe this. Velocity fields of Olig2 and Bcl6 point towards the center of the OPC cluster. What does this mean? The Drosophila eye analysis is more complete/convincing, possibly due to a better capture of cells at distinct differentiation stages.

We provide a detailed tutorial of the analysis at: https://scenicplus.readthedocs.io/en/latest/Differentiation_tutorial_unbranched_RTD.html and **better describe this in the methods section:** “*Oligodendrocyte cells (Oligodendrocytes Precursor Cells and oligodendrocytes) were subsetted from our in-house mouse cortex data set, resulting in 4,435 cells. The eRegulon AUC matrix was processed using Scanpy (v1.8.2) and the embedding-based pseudotime was derived using the diffmap and dpt functions. Differentiation arrows were inferred for cells above the 70% quantile in TF expression and default parameters. Prioritization of differentiation velocities was done using the Regulon Specificity Score (RSS) metric using arrow lengths in each cell for each regulon as input values.*” (i.e. OPC to NFOL are the regulons that result in the longest arrows in OPC, NFOL maturation, in NFOL, and NFOL to OL in oligodendrocytes).

With regards to the velocity vectors pointing towards the center OPC cluster, this is because arrows are only drawn where there are cells, and arrows can only point to other cells. Given that there are more cells

at the center of the cluster compared to the border and given that all cells of this cluster have similar pseudotime values (rather than a gradual gradient) there is a higher likelihood that the arrows at the border will point towards the center, if both the TF and the target genes are expressed in the same cluster of cells. In the figures the average of several arrows is shown, making this bias even stronger while canceling out the, mostly random, arrows in the center of the cluster. On the other hand, if the TF is expressed in OPC, but target genes in NFOL, the arrows will point to the NFOL cluster. We illustrate this by showing the arrows per cell (without averaging), for Olig2 (OPC maintenance) and Sox10 (OPC to NFOL) (Fig R8.1). For Olig2 we observe strong velocities for the outer cells towards the center and random arrows in the middle, while for Sox10 all arrows point towards NFOL. **We clarified this in the manuscript as follows:** “(...) This revealed a set of TFs (*Olig2*, *Bcl6* and *Prrx1*) that maintain OPC specification.”

Figure R8.1. Oligodendrocyte differentiation UMAP colored by pseudotime, together with GRN velocity arrows (per cell (without averaging by neighbour), for 100 selected cells) for Olig2 and Sox10.

9. How is the differentiation force defined? Method (line 1716) describe the use of a “penalization curve”, but no further details are given. In the scheme (Fig. 6a), red curves anticipate the blue curves but, in the examples/real data (Fig. 6g), we also observe the opposite. How do the model deals with this? Are only positive values are considered? What would the negative vectors mean? Authors need to both formalize the method and include a benchmarking more like analysis of this novel methodology.

1) To answer the questions about method details, **we improved the Methods section** (see: “GRN velocity calculation”). The procedure works as follows. We first order a set of differentiating cells by pseudotime.

Next, for each TF, we fit a standardized generalized additive model (GAM) along the pseudotime axis, one for TF expression, and one for its target genes (or regions) AUC values, using pyGAM (v0.8.0). We then map each cell in a certain quantile of the GAM TF expression model with the cell in the same quantile of the GAM regulon AUC curve (posterior in the pseudotime axis). If there is no posterior cell in that quantile, the cell is mapped to itself. We define the differentiation force of a cell as the distance from the TF expression curve to its matching cell in the regulon AUC curve. We only consider positive interactions, as negative relationships would indicate that the target genes are expressed before the TF gene expression. While this could be the case for repressors, these curves tend to be noisier, as they can also depend on the presence of activators.

When a dataset represents multiple differentiation paths, we apply the same strategy to each path, and then average the two (or more) arrows available for cells found in more than one path. In addition, since we standardize the data prior to fitting the GAM model, we introduce a penalization curve standardized on the whole data set. This will prevent that if a TF (or its target) is not present in the branch, false arrows will not be drawn if the TF is not expressed compared to the rest of the data set. For instance, in Fig R9.1 we show the curves for *svp(+)*. This TF is not expressed in the INT branch, however, because we standardize the gene expression values between 0 and 1 in each branch, we observe a peak in gene expression. The penalization curve (as a grey dotted line), which is scaled along all possible paths, does not agree with the TF expression path, and in those points where it disagrees the GRN velocity is set to 0 as it represents an artifact. Importantly, GRN velocity is only inferred for those cells in which the TF expression is above a certain threshold (by default, above the 70% quantile, horizontal grey line). As the reviewer points out, there are some points where the blue curve precedes the red one, but only in those areas with low TF expression, which represent artifacts. In the case of multiple paths with overlapping cells, the final velocities are drawn as the sum of all the possible GRN velocities across the paths. **We have improved the explanations of the GRN velocities approach in the methods section and formalized the method as suggested.**

Fig R9.1. Representation of svp dynamics along the two paths in the eye disc differentiation. The grey horizontal line represents the TF expression threshold for arrows to be drawn. For cells below this threshold, the GRN velocity values are set to 0, as they correspond to cells with low TF expression. The grey dotted line represents the penalization curve, which is the GAM fitted curve drawn using the standardized data across all possible path for the cells in that path. Those points where the penalization and the TF expression curve disagree (by default, by 0.03 in the y-axis) are considered artifacts (i.e. the TF gene seems to be expressed even if there is no or low expression in the path, due to the standardization of the TF curve in that specific path). The red curve represent the GAM fitted curve using the standardized TF expression data (along the path), and the red curve represents the GAM fitted curve using the standardized AUCell values (along the path), which represent the enrichment of the target genes.

2) Regarding the question about benchmarking GRN velocity: to our knowledge there is no other method that yields a TF-based differentiation force based on gene regulation. We have included a comparison of GRN velocity with RNA velocity and MultiVelo (Reviewer 1 – Comment 7, Fig 7, S28). To our knowledge, GRN velocity is the first approach that exploits eGRNs and the lag between TF expression and expression of the target genes to predict the directionality of a cell. RNA velocity (scVelo) measures the ratio of spliced and unspliced mRNA of a gene, while MultiVelo integrates the accessibility around the gene locus in the model. Yet, none of these approaches includes TF input, and overall only provide global directionality values (one vector per cell), while we can assess TF-specific directionality (one vector per cell and per TF). GRN velocity permits to identify key known drivers of oligodendrocyte and

eye disc differentiation, which is reassuring. For instance, upon knockdown of atonal (ato) in the eye disc, the morphogenetic furrow still forms, but no photoreceptors are generated (Jarman et al. 2019). GRN velocity predicts atonal to be key for the transition from morphogenetic furrow cells towards the generation of photoreceptors and interommatidial cells. On the other hand, in glass (gl) mutants, photoreceptor precursors are formed and begin neuronal differentiation, but they never express photoreceptor-specific markers (Moses et al., 1989). Accordingly, GRN velocity predicts glass to be key in the differentiation of early progenitors. Knockdown of lozenge (lz), a TF that GRN velocity predicts to be key to differentiate towards interommatidial cells, results in a smaller eye, with reduced pigmentation and lack of hairs possibly due to the lack of interommatidial pigment and bristle cells (Brehme and Demerec, 1942). With regards to oligodendrocyte differentiation, Zeb2 knockdown results in the lack of formation of mature oligodendrocytes, and the expression of progenitor related TFs such as Sox2 (Quintes et al., 2016). Accordingly, GRN velocity predicts Zeb2 as a key driver of oligodendrocyte maturation. Note that GRN velocity also provided new insights, such as the identification of Meis1 as a novel (and conserved) master regulator of oligodendrocyte differentiation. **These analyses are described in Fig 7.**

10. The spatial analysis of eGRNs is interesting, but its value is unclear (besides visualization). For example, if one plots the cellular deconvolution results of the cell types with LHX7, SOX9 specific TF activity, would one not obtain similar results as shown in panel Fig. 7C? What novel information can be gained here?

Although we believe this approach is interesting to visualize GRN enrichment in spatial templates, **we have moved this figure and its description to supplement (Fig S24)** as it indeed does not add a lot of methodological (nor biological) novelty.

11. We were able to successfully install/run the method in linux. In a mac, we encountered issues with package dependencies (annoy package).

We thank the reviewer for trying to install SCENIC+ and running the tool, this is highly appreciated. We are aware of the installation difficulties with the Annoy package, these are mostly compilation errors unrelated to SCENIC+. Also, SCENIC+ is designed to be run in a Linux environment and is untested in different systems. **We adapted the Readme.md file of SCENIC+ accordingly.**

Minor points:

Line 66 - The statement that scATAC-seq is similar ChIP-seq accuracy is based on pre-print (23). Authors should tune this down.

We have removed this statement and now only say: *“With single-cell chromatin accessibility data available, for example obtained by scATAC-seq, the accuracy of TF binding site predictions can be improved significantly²³”*

Line 73 - Authors should also cite previous works introducing some of these analyses, i.e. no reference to Cicero, which first explore peak-gene links in scATAC-seq.

We now cite Cicero in the text (Pliner et al. 2018).

Line 103 - Mixture of HMMs have been previously explored for a similar problem. Please cite 10.1093/bioinformatics/bti1001) (and discuss any relevant distinction).

We added the following sentences:

“Previous studies have shown that instead of one single consensus PWM per TF, using a mixture of two subclasses of PWMs yields better results (Hannenhalli et al., 2005). We hypothesized that making use of the information present in a large multitude of PWMs (per TF) would increase performance.”

Line 104-108 – This phrase is complex and content rich (i.e. what is leading edge analysis?). Authors should improve this.

We have added a Supplementary Note 1, explaining the motif databases and motif enrichment approaches in detail.

Also, we clarified the methods section by adapting line 956 from:

“To obtain the target regions for each TF, the motif-based cistromes of motifs annotated to that TF are combined.”

To:

“To obtain the target regions for each motif (motif-based cistrome) the regions at the top of the ranking (leading edge) are retained. The top of the ranking is defined by an automated thresholding method which retains regions with a ranking below the rank at max, which is defined by the following formula:

$$\text{RankAtMax} = \max (rcc_{\text{motif}} - [\mu(rcc_{\text{all motifs}}) + 2 \cdot SD(rcc_{\text{all motifs}})])$$

With:

rcc_{motif} : *the recovery curve of the motif of interest.*

$\mu(rcc_{\text{all motifs}})$ *the average recovery curve over all motifs.*

$SD(rcc_{\text{all motifs}})$ *the standard deviation of the recovery curve over all motifs.*

To obtain target region for each TF, the motif-based cistromes of motifs annotated to the TF are merged.”

Line 112 - From the method description I got the impression that GBMs are used to find relations (and Person is only used for finding signals), but the main text indicates correlation are also important. Please, clarify.

We revised this line as follows:

Original manuscript:

“Region-to-gene (default up to 150kb from the transcription start site) and TF-to-gene relationships are inferred using Pearson correlation and a tree-based regression approach, namely Gradient Boosting Machines (GBMs), to assess both linear and non-linear relationships”

Revised version:

“Region-gene (default up to 150kb from the transcription start site) and TF-gene relationships are inferred using a tree-based regression approach, namely Gradient Boosting Machines (Moerman et al., 2019). The direction of interaction (activating/repressing) is inferred using linear correlations.”

Line 197 – Are these “TF” ChIP-seq?

Correct, these are 309 high quality *TF* ChIP-seq experiments. We added the word “TF” when referring to TF ChIP-seq.

Line 259 - How are TF ranked? By increasing or decreasing number of bindings?

The TFs are ranked descending, i.e largest number of binding sites at the left side of the x-axis, smallest number of binding sites at the right side of the x-axis. We clarified the legend.

Line 998 –The phrase regarding TF-to-TF regulation is unclear. Please clarify.

The following sentence:

“To infer autoregulation the TF-to-TF importance is set to the maximum importance score across genes for that TF added with the value of $1E-5$.”

has been replaced by:

“The TF itself is not included in the initial TF-gene relationship calculation (otherwise it would skew the importance scores of the other genes). Therefore, in order to be able to infer autoregulation (TFs regulating their own expression) the importance score of the TF itself is set to the maximum importance score across all genes added with an arbitrary small value of $1E-5$, in order to put the TF at the top of its own ranking.”

Line 1104 - The definition of archetypes is unclear here.

The following piece of sentence:

“We found that scoring with archetype performs worse compared to using all PWMs in the cluster, (...)”

has been replaced with:

“We found that scoring with a single average/consensus motif per cluster (i.e. archetype motif (Vierstra et al., 2020)) performs worse compared to using all PWMs in the cluster, (...)”

Fig. 4e – Colors of the boxplot are difficult to discern.

We changed the colors to more contrasting ones and added extra annotations to the figure (Fig 5e).

Supplement – Several links in the supplement pdf are not working. Please revise.

We have tested all the URLs included in the supplementary material (from Adobe Acrobat Pro), and all functioned correctly. We are happy to recheck this if the exact conflicting links can be provided.

Supplementary Figure 5 – What is R and L (after SCENIC + (C)).

“R” means Rho (i.e. region-gene links are assessed based on the correlation between the accessibility of the regions and the expression of the target genes), and “I” means Importance (i.e. region-gene links are assessed based on the GBM importance based on a model where the accessibility of the regions is used to

predict the gene expression). We have updated the legend to include the meaning of these abbreviations (Fig S17).

Reviewer #3:

Remarks to the Author:

Manuscript by Gonz.lez-Blas et al described a valuable computation tool for inferring enhancers and gene regulatory networks based on single cell multiomic datasets. The tool overall can be very useful for mining large scale single cell datasets and providing great insights on how cell types are regulated via binding of transcription factor and their targets. The database with curation of 30K+ motifs and motif-TF annotation is an incredible resource for the community. Overall, the authors did a great job of presenting this remarkable tool and showcase various interesting applications. On other hands, some improvements are necessarily to enhance usability of this tool and presentation of this manuscript.

We thank the reviewer for the positive assessment of our work and the useful suggestions to improve the manuscript and the usability of SCENIC+.

Here is list of major key points regarding the Manuscript

1. PycisTopic is proposed as a new tool for both enhancer discovery and co-accessibility of enhancers, which is also possible via other popular methods including ArchR. Since these early processing steps play a pivotal role in the final eGRN it would be useful to benchmark pycisTopic against current methods. Essentially an analysis like Figure 1f for enhancer discovery would be informative to the reader/user.

Even though cisTopic has been benchmarked against other methods by us (Bravo et al., 2019) and by other groups (Chen et al., 2019), we agree that Signac and ArchR were not included in those benchmarks. Therefore, we have now compared pycisTopic against Signac and ArchR for two tasks: 1) cell type discovery and 2) enhancer discovery. For cell type discovery we simulated several scATAC-seq data sets from the Deeply Profiled Cell Lines, with different coverages (20K, 10K, and 3K fragments per cell) and

numbers of cells (25K, 10K, 1K, and 80 cells). While all methods perform well with large numbers of cells, their robustness is affected as the data set decreases in size (Fig R1.1, R1.2). ArchR and Signac are most affected, while pycisTopic is more robust. We also compared these in five of the datasets, namely PBMC, melanoma, human cerebellum, Drosophila eye-antennal disc and mouse brain. For PBMC, melanoma and human cerebellum all methods were able to identify the expected cell types (as defined from the scRNA-seq layer). In the Drosophila eye-antennal disc, a dynamic system in which most cells are undergoing differentiation in the eye or the antennal part, we noticed that while pycisTopic is able to group all cells by cell type, ArchR and Signac tend to cluster low coverage cells together rather than with their corresponding cell type (Fig R1.3, encircled). We then compared the capacity of the methods to correct the strong batch effects in the mouse brain data set (due to different preparation protocols). PycisTopic batch correction (which relies on using Harmony (Korsunsky et al., 2019) on the cell-topic distributions) resulted in cell clustering by cell type. Signac (based on the integrated LSI approach) was also able to cluster the cells by cell type; however, oligodendrocytes are still affected by the batch effect. For ArchR (based on harmony correction on the LSI components), batch effect correction results in the loss of resolution of the excitatory and inhibitory neurons subtypes, and is not strong enough to correct the batch effect of astrocytes (Fig R1.4). Altogether, pycisTopic provides the best integration of the samples, without losing the real biological signal. While the three methods tend to work well for large datasets with good coverage, pycisTopic is more robust to data sparsity, low numbers of cells and batch effects.

Fig R1.1 Adjusted Rand Index for pycisTopic, Signac and ArchR in simulated datasets with different coverage per cell (3K, 10K, or 20K fragments per cell) and number of cells, using as ground truth the bulk label from which cells were simulated.

Fig R1.2 UMAPs from pycisTopic, Signac and ArchR in simulated datasets with different coverage per cell (3K, 10K, or 20K fragments per cell) and number of cells, using as ground truth the bulk label from which cells were simulated.

Fig R1.3 Dimensionality reduction from pycisTopic, Signac and ArchR in scATAC-seq datasets of PBMC, melanoma cell lines, cerebellum, and eye-antennal disc (EAD). The circle on the eye-antennal disc plots for Signac and ArchR correspond to low coverage cells that are intermixed instead of grouped with the corresponding cell type.

Fig R1.4 UMAPs from *pycisTopic*, *Signac* and *ArchR* in the 10x multiomics mouse cortex data set (*scATAC-seq*). For each method, dimensionality reduction was performed before batch correction per sample (uncorrected) and after (corrected). In each case, UMAPs are shown colored by cell type (left, as defined from the *scRNA-seq* layer) and sample (right). The different sample preparation protocols are indicated below, an extensively explained on the methods section.

Next, we compared *pycisTopic* to *Signac* and *ArchR* regarding enhancer discovery. For this, we made use of genome-wide STARR-seq data (which measures enhancer activity) in HCT116, MCF7, K562 and HepG2 and assessed the enrichment of candidate enhancers identified by running *pycisTopic*, *Signac* and *ArchR* on the simulated *scATAC-seq* data set from the Deeply Profiled Cell Lines (4k cells and 20k fragments per cell; see Fig 3). To define candidate enhancers, *Signac* and *ArchR* rely on Differentially Accessible Regions (DARs), while *pycisTopic* uses both DARs and topics (sets of co-accessible regions). Comparing the different sets of candidate enhancers, cell line specific topics from *pycisTopic* are the most highly enriched for STARR-seq activities in all cell lines except K562 (Fig R1.5). The enrichment for DARs varies across the four cell lines (Fig R1.5). We hypothesize that regulatory topics are more robust because they are derived from unsupervised clustering of the entire data set (using topic modelling), while DARs require defining foreground and background cells. For instance, if two cell types share accessible

regions, these will be allocated to an independent shared topic, while unique regions will be allocated to the putative unique and separate topics. When inferring DARs, and because most methods use as background all other cells that are assigned to other cell types, the shared regions will be classified as DARs because the signal in the background will be diluted. Altogether, regulatory topics recover the largest number of functional enhancers per cell type. We have included these analyses in Fig 1c and Fig S3.

Fig R1.5. Recovery of activate enhancers using candidate enhancers identified by pycisTopic, Signac and ArchR. *a.* Recovery curves for top 5K Differentially Accessible Regions (DARs) identified by Signac, pycisTopic and ArchR and top 5K regions in the cell line specific topics identified by pycisTopic. Genome-wide STARR-seq in HCT116, MCF7, K562 and HepG2 is ranked in descending order (x-axis) when a region of the ranking is found in a region set an increasing step along the y-axis is taken. Dashed line represents the top 10% of the ranking. *b.* Barplots showing Area Under the recovery Curve (AUC) for top 5k Differentially Accessible Regions (DARs) identified by Signac, pycisTopic and ArchR and top 5k cell line specific topics identified by pycisTopic up until 10% of the ranking. AUC value is scaled by dividing by the maximum possible AUC at 10% of the ranking. Promoter regions were excluded from the analysis.

2. The illustration of SCENIC+ on PBMC data captures 40% of genes with a proximal enhancer that regulates expression. Recent studies have used HiC to look at the 3D chromatin organization for PBMC populations including Monocytes. Using the HiC data as an external validation for region-gene links should be done.

We thank the reviewer for the suggestion and agree that this statement in the manuscript could make use of some experimental validation using publicly available data. We were not able to find the high quality Hi-C dataset in monocytes as suggested by the reviewer. However, the Hi-C dataset on GM12878, that we also used for the benchmarking section, is suitable, given that GM12878 is supposed to be a B-lymphocyte-like cell line.

We first compared the number of regions linked to each gene as predicted by SCENIC+ and measured by Hi-C. SCENIC+ predicts that 90% of genes have between 1 and 10 enhancers while Hi-C predicts that 60% of genes have between 1 and 10 enhancers (Fig R2.1). Next, we assessed the number of regions that are most likely regulating the most proximal gene (Fig R2.1 “nth. gene per region”). SCENIC+ predicts that 49% of enhancers are most likely, based on the region-to-gene importance score, to regulate the most proximal gene while Hi-C predicts that 76% of enhancers are most likely, based on the Hi-C score, to regulate the most proximal gene. Note that this number is slightly different from the number in the manuscript. Here, we look at the distribution of the rank, based on absolute distance for each region and gene with the highest region-to-gene score while in the manuscript we showed the distribution for all region-to-gene combinations. Based on this analysis we reasoned that only showing the distribution for the most likely region-to-gene links is more informative, therefore we adapted the panel and the numbers in the manuscript. Lastly, even though the percentages obtained from SCENIC+ predictions and Hi-C measurements are different the overall distributions are very similar (Fig R2.1).

Secondly, we assessed the correlation between Hi-C scores and region-to-gene scores obtained from SCENIC+, for B-cell marker genes (Fold change > 1.5 and adjusted pvalue < 0.05). Similar to the conclusions in the benchmarking analysis, we observe a significantly higher correlation between Hi-C scores and region-to-gene scores compared to randomly shuffled region-gene pairs and this correlation is higher for the region-to-gene correlation coefficient than for the region-to-gene importance score. We

added this revised analysis as a supplemental figure (Fig S9c) and changed the following lines in the manuscript:

“9,347 (40 %) enhancers are predicted to regulate only the closest gene” is changed to “11,533 (49 %) enhancers are predicted to most likely regulate the most proximal gene, based on the region-to-gene importance score”

We added the following sentence: “These distributions are similar to the distributions obtained from Hi-C data (Fig S9c).”

Figure R2.1. Comparison of region-gene links predicted by SCENIC+ and measured by Hi-C. **Left:** Distribution of the number of regions linked to each gene as predicted by SCENIC+ (top) and measured by Hi-C in GM12878 (bottom; using a minimum score of 10). **Right:** distribution of the rank, based on absolute distance, for each region and the gene with the highest region-to-gene SCENIC+ importance score (top) and the gene with the highest Hi-C score in GM12878 (bottom; using a minimum score of 10).

Figure R2.2. Correlation of Hi-C score and SCENIC+ region-to-gene scores. Boxplots showing the distribution of Spearman correlation coefficients between Hi-C scores in GM12878 (with a minimum score of 10) and region-to-gene gradient boost importance score (GBM) and region-to-gene correlation coefficients (rho) as calculated by SCENIC+ for B-cell marker genes (. Random controls are obtained by shuffling the gradient boost importance scores (GBM_rnd) and correlation coefficients (rho_rnd). Difference in the mean to the random control is assessed using the Mann-Whitney U test.

3. Limited correlation of HiC interactions in the 5 cell lines as shown in Figure 3f as well as an overall higher number of region-gene links could point towards more false-positive interactions being identified. How does SCENIC+ address the presence of noisy interactions in the eGRN building process? For example, do multiple models (including Gradient Boosting Machine regression) agree on region-to-gene

importance or binary links? Comparison of different regression models to calculate TF-to-gene and region-to-gene would be worthwhile.

We agree with the reviewer that the combination of low correlation with Hi-C data and high number of region-gene links could suggest that a high number of false positives are being identified. However, we would like to comment that even though SCENIC+ has the largest number of region-to-gene links identified, compared to other eGRN inference methods, it still has the highest correlation with Hi-C scores compared to those other methods. In other words, at much higher recall, the number of false positives is still lower compared to other methods. It is known that HiC has limited sensitivity to identify specific enhancer-target interactions, particularly when the peaks are close to the target gene (Fiorillo et al., 2021), and therefore we expect a relatively high number of predicted interactions that are not confirmed by HiC data.

Regarding the question on how SCENIC+ addresses the presence of noisy interactions in the eGRN building process, in the modeling this is addressed in two ways. First, only region-to-genes links that pass the GSEA step are retained (i.e. regions that are linked to genes that are significantly co-expressed with the TF). Second, multiple thresholding methods are used to binarize the region-to-gene links and the GSEA step is run separately for all these thresholds. These thresholds are all based on the rank of the region-to-gene scores and not on the actual values itself (i.e., by taking the top quantiles, top number of regions or using BASC (Hopfensitz et al., 2012)). The final eRegulon is obtained by merging the filtered links from all binarization methods.

Regarding the question whether multiple models agree on the region-to-gene importance or binary links, calculating region-to-gene importance scores based on the accessibility vector of regions and the expression vector of genes across cells is novel and only possible because of the existence of single-cell multiome readouts. For this reason, this has not been benchmarked extensively and comparing multiple models is a relevant question. To do this, we calculated region-to-gene relationships using the models included in the SCENIC+ package (i.e. Spearman rank correlation, Genie3 (RF) and GRNBoost2 (GBM)) and compared this to all linear regression models implemented in scikit-learn (Pedregosa et al., 2011) (Elastic Net (ENET), Lasso, Support Vector Machine (SVM) with linear kernel, Ridge, Least-Angle

Regression (LARS), Stochastic Gradient Descent (SGD)). We compared the region-to-gene scores to Hi-C data on IMR90, GM12878, HCT116, HepG2 and K562. The region-to-gene scores obtained using Spearman rank correlation is the highest followed by Genie3 and GRNBoost2 (Fig R3.1). By default, SCENIC+ makes use of a combination of GRNBoost2 (for the region-to-gene score) and Spearman rank correlation (to separate regions which negatively and positively correlate with gene expression). GRNBoost2 is preferred because its computational efficiency is significantly better compared to Genie3, with minimal loss of accuracy. Next, we compared the region-to-gene scores and the binarized relationships obtained by several regression models. Even though the scores obtained by the methods are very different (Fig R3.2a), the binarized region-to-gene relationships do not differ a lot (Fig R3.2b; minimal Jaccard index is 0.55).

On the question of comparing different models for TF-to-gene relationships, contrary to calculating region-to-gene relationships, methods for calculating TF-to-gene relationships have been extensively benchmarked. The method used by SCENIC+, GRNBoost2, is one of the best methods in these studies (Pratapa et al., 2020; Moerman et al., 2019).

These analyses are included in Fig S15.

Figure R3.1. Correlation of Hi-C data with region-to-gene scores using several regression models. Boxplots showing the distribution of correlation between Hi-C scores and region-to-gene scores, for top

100 marker genes for each of the cell lines where Hi-C is available (IMR90, GM12878, HCT116, HepG2 and K562) calculated using several regression models. The sklearn implementation (Pedregosa et al. 2012) for all the models was used with default parameters, except for GBM and RF for which the parameters as described in Moerman et al. 2019 were used. Spearman: Spearman Rank Correlation coefficient, RF: Random Forrest regression (Genie3 (Moerman et al., 2019)), GBM: Gradient Boosting Machine regression (GRNBoost2 (Moerman et al., 2019)), ENET: Elastic Net, LASSO: Lasso regression, SVM: Support Vector Machine regression (using linear kernel), RIDGE: Ridge regression, LARS: Least-Angle regression, SGD: Stochastic Gradient Descent regression. *p*-values on top of the boxplots are calculated using the two-sided Mann-Whitney *U* test testing the difference in mean between the distribution in the figure and the distribution of correlation coefficients obtained by shuffling the Hi-C scores.

Figure R3.2. Comparison of region-to-gene scores and binarized links obtained from calculating region-to-gene relationships using several regression models. **a.** Clustered heatmap showing the correlation between region-to-gene scores obtained from several regression models. **b.** Clustered heatmap showing jaccard index of binarized region-to-gene links obtained from several regression models.

4. It is unclear how scalable the method is to number of cells and number of clusters. It will be nice to present a complexity analysis and running time analysis.

To assess to what extent the number of cells and coverage impacts the running time of each of step, we simulated several scATAC-seq data sets from the Deeply Profiled Cell Lines, with different coverages (20K, 10K, and 3K fragments per cell) and numbers of cells (25K, 10K, 1K, and 80 cells). The minimum preprocessing steps with pycisTopic to run the SCENIC+ workflow are: topic modelling, dimensionality reduction (visualization), topic binarization (to perform motif enrichment for cistrome formation), dropout imputation, and DAR inference to perform motif enrichment for cistrome formation). The most time-consuming step is the topic modelling, which increases with the number of cells and the coverage (as more assignments must be done per iteration). Note that the running time of this step also depends on the topic space that the user wants to explore (i.e. how many models are trained). We find that the number of topics per model does not affect the running time (i.e. running a model with 30 topics will take the same time as running a model with 50 topics), as we show in Fig 1b. The other steps (dimensionality reduction, topic binarization, imputation and DAR inference) are mostly affected by the number of cells, but not by the coverage (Fig R4.1, Fig R4.2). These analyses were run in an Intel(R) Xeon(R) Platinum 8360Y [IceLake], with 300gb memory and 20 cores.

Fig R4.1 Time complexity of the preprocessing steps with pycisTopic using simulated datasets with different coverages and number of cells. **a.** Running time (as seconds) per step. The running time for the topic modelling corresponds to the average over topics. **b.** Running time per topic model as a function of the number of topics specified.

Fig R4.2. Time complexity of the preprocessing steps with pycisTopic as a function of the number of cells grouped by coverage.

For *pycisTarget*, we observed that DEM tends to take longer than *cisTarget* on the same region sets. While the number of cells (especially comparing the data set with 80 cells versus the rest) and the coverage seem to have an effect, this is mostly related to the impact that the number of cells and coverage has on the number DARs and regions per topic (see comment 5). At all coverages we observe that the data set with 80 cells results in a lower recall of regions, and thus, shorter running times in *pycisTarget* (with both DEM and *cisTarget*). In the data sets ranging from 1K to 25K cells we did not observe significant differences in the number of regions per topic, while for DARs the data set with 1K regions resulted in slightly less recovered DARs compared to the 10K and 25K data sets. Note that the running time of this step will also multiply by the number of topics (topics) or cell types/clusters (DAR sets)

defined. These analyses were run in an Intel(R) Xeon(R) Platinum 8360Y [IceLake], with 300gb memory and 20 cores.

Fig R4.3 Average running times per region set for cisTarget and DEM using topics and DARs versus number of cells.

Fig R4.4 Distributions of the number of regions per topic and DARs across data sets.

To evaluate the run times of SCENIC+, we simulated scRNA-seq data with different coverages that were paired with the matching scATAC-seq data sets, with low (5K reads per cell), medium (20K reads per cell) and high (50K reads per cell) coverages. We observed that most of the steps scale with the number of cells (e.g. create object, filtering genes/regions, region-gene link inference, building AUCell ranking). The most time-consuming step is the inference of the TF-to-gene links, which can take up to 20 hrs for the largest high coverage data set. A potential solution to reduce the running time of this step is to only use TFs for which cistromes are available (although this would impede using the same links if changes are made in the motif enrichment/cistromes). Altogether, both the size of the data set and the quality/coverage of the data will have an impact in the overall running time of the tool. These analyses were run in an Intel(R) Xeon(R) Platinum 8360Y [IceLake], with 1000gb memory and 20 cores. **We describe these analyses in Fig S6.**

Fig R4.5. Time complexity of the SCENIC+ steps as a function of the number of cells grouped by coverage.

5. It is unclear how sensitive this tool is to analyze fine resolution cell types. How many cluster-specific peaks are needed to identify cluster specific eRegulons? Some power analysis can be very helpful here.

Like our analyses performed under comment 4, we simulated single-cell multiomics datasets, using the ENCODE cell lines data, with different numbers of cells and coverages to assess the sensitivity of SCENIC+. Specifically, we simulated a low coverage data set, with 3K scATAC-seq fragments per cells and 5K scRNA-seq reads per cell; a medium coverage data set, with 10K scATAC-seq fragments per cells and 20K scRNA-seq reads per cell; and a high coverage data set with, with 20K scATAC-seq fragments per cells and 50K scRNA-seq reads per cell. From each of these data sets, we derived 4 data sets with different numbers of cells, 80, 1,000, 10,000 and 25,000 cells, respectively. We then compared the results using our benchmark data.

Overall, we observe an increase in the identification of high-quality regulons (and an increase in the number of regions and genes per regulon) with an increase in coverage and increase in number of cells (Fig 5.1). There is however a strong difference between the results on the sample with low coverage (3K scATAC-seq fragments per cell and 5K scRNA-seq reads per cell) compared to all the rest. This is likely related to the sensitivity of pycisTopic on such low quality data (see comment 1). Overall, the data sets with 10K and 25K cells achieved similar results.

Fig 5.1 Number of TFs for which SCENIC+ identified a high-quality regulon (left), number of genes per regulon (middle) and number of regions per regulon.

With regards to TF recovery, we observed a similar trend (i.e. increased TF recovery with increasing coverage and number of cells), especially in comparison to TF expression (Fig 5.2).

With regards to TF target region predictions, we observed that precision is maintained, while recall varies depending on the coverage and the number of cells (using Unibind peaks as silver standard in Fig 5.3, results using Enformer and ChIP-seq peaks show similar results). For both middle and high coverage (10K/20K scATAC-seq fragments per cell and 25K/50K scRNA-seq reads per cell, respectively) results are similar.

Regarding region-to-gene relationships, we find that both the number of cells and the coverage have an impact on the overall number of links (also related to the effect that coverage and sample size have on the number of regions in topics and DARs, see comment 4). However, in comparison to Hi-C data, we do not observe biases due to coverage (Fig 5.4).

Finally, with regards to TF-to-gene inference we again observe an effect in quality due to the coverage and the number of cells, but again the analyses using middle and high coverage achieve very similar results (Fig 5.5). Altogether, we can conclude that SCENIC+ should best be avoided on samples with low coverage and low number of cells (80), while in all other conditions the results are robust given that data sets with 10K/25K cells and middle/high coverage achieve comparable results.

We think this is an interesting parameter for SCENIC+ users and **included this in the results and Fig S19.**

Fig R5.2. TF recovery performance by SCENIC+ upon variations in coverage and sample size. Cumulative TF recovery for each method using as x-axis TFs ranked based on the number of ChIP-seq peaks and AUC values per method using the top 40 TFs (left) and cumulative TF recovery for each method using as x-axis TFs ranked based on the maximum LogFC across the cell lines and AUC values per method (right).

Fig R5.3. F1 score, precision and recall distributions of the predicted regions per TF using Unibind regions as standard.

Fig R5.4. Number of region-gene links inferred (non-transparent links indicate that the links are included in the final eGRN) and boxplot showing the correlation with the Hi-C links for the top 100 marker genes for each of the cell line where Hi-C is available (IMR90, GM12878, HCT116, HepG2 and K562).

Fig R5.5. F1 score, precision and recall distributions of the predicted genes per TF using perturbation data. Boxplots representing the F1 score (left), precision (middle), and recall (right) distributions of the predicted target genes per TF compared to TF perturbation data.

6. The paper focused a lot on the confirmative analysis, showing previously categorized regulons, but very shy at presenting novel findings. I encourage the authors showcase a few, even without experimentally validation.

We thank the reviewer for this encouraging suggestion. We include below a summary of the novel biological insights that SCENIC+ provided through the case studies. These findings are now better described in the text, including:

- In the cortex study we identified 60 TFs conserved between mouse and human. We now compared these with literature and found that 52/60 have been previously described to be involved in the different cortex cell types. The remaining 8 TFs can be considered novel, conserved regulators. These include Smad3 in the upper cortical layers, Pparg and Bhlhe40 in L4, Etv5 and Nfat5 in L5/6, Thrb and Pbx1 in L6 and Meis1 in oligodendrocytes. For these conserved TFs, we identify 4,798 and 8,318 conserved TF-region and TF-gene relationships, respectively. Given the sparsity of direct TF-enhancer and TF-gene relationships in the literature, the majority of these conserved interactions are novel.
- In the PBMC study, SCENIC+ identified 63 key regulators, 62,011 TF-gene and 146,583 TF-region interactions. Through different benchmark analyses and comparison with other tools, we found that SCENIC+ GRN is the most complete and accurate network compared to other methods. In addition, we show that SCENIC+ can be used to explore TF cooperativity.
- In the melanoma case study, SCENIC+ identified 42 high quality activators. This resulted in the most comprehensive GRN for melanoma to date, including 13,691 and 28,819 TF-gene and TF-region interactions, respectively. By simulating TF perturbations using this network, SCENIC+ identified potential novel regulators of EMT in melanoma, namely MXI1 and ZNF487, which is now added more explicitly to the text.
- In the eye-antennal disc, we identified Cut (ct) as a transcriptional repressor acting in the outer antennal ring (Fig S27). Among its target genes, we found several eye disc progenitor TFs, such as ey, eya, Optix and toy, among others. This suggests that ct is necessary for the development of the antenna by inhibiting the eye fate, and that has indeed been shown by genetics (Wang et al. 2012); however, it was not known that Cut works by directly repressing these master TFs for eye development.
- In the analysis of oligodendrocyte differentiation with our GRN velocity approach, we identified Meis1 as a novel regulator of oligodendrocyte differentiation. Meis1 has been previously described to be involved in early neurogenesis and hematopoietic

differentiation (Isogai et al., 2022), but not in oligodendrocyte maturation. As previously mentioned, we also found this TF conserved in the human cortex, with 20% of the target genes conserved between human and mouse.

- In the cerebellum data set, we identify a novel TF of granule cells, EMX1. EMX1 has been previously shown to have an effect in the size and morphology of the cerebellum and hippocampus, but to our knowledge, has not been linked before as a key regulator of granule cells (Kobeissy et al., 2016).

7. The tool cluster DNA motifs abased on PWM similarity. But single base difference in motifs can be critical in determining binding affinities and targets for paralogs within the same transcription factor family. How did the authors address such subtle differences?

We agree that this is a challenging problem that all tools for predicting TF binding sites based on PWMs face. A similar (or even worse) problem exists for paralog TFs for which there are no (or not very clear) differences in their motifs. We discuss how this can lead to ambiguous predictions in cases where the expression of the TF is exactly anticorrelated (Supplementary Note 3). Yet another challenge is that the same TF can often have multiple PWMs that are not all exactly the same. SCENIC+ addresses this problem in two ways. First, the motif collection is clustered based on PWM similarity, as the reviewer states in the question. However, the two-step clustering strategy that is used (i.e. Leiden clustering on PWM similarities calculated using TomTom followed by STAMP clustering), together with the magnitude of the motif collection, often result in separate cluster for motifs which are different only in small ways (for example, cluster 193.3 and cluster 193.2 both contain E-box motifs however the consensus sequence of the motifs in cluster 193.3 is CAGCTG while it is CATATG in 193.2; or cluster 169.2, cluster 119.7, 117.2 that all contain GATA motifs with different consensus sequences respectively: GATAA, GAT and GATAACGATC). Second, for each cluster all TF annotations, both based on orthology and direct experimental evidence, are retained for all motifs within a cluster. Based on the expression of these TFs SCENIC+ is able to identify the correct TF along with the correct target regions (i.e. the regions for which the target genes are significantly co-expressed with the TF).

We illustrate this for the paralogs GATA1 and GATA3. Regions enriched for a subset of motifs annotated to these TFs highly overlap, all these motifs have the consensus sequence GATA or GATAA (Fig R7.1. top cluster of motifs). Next to this there is another subset of motifs that show little overlap, these motifs have very distinct consensus sequences (GAT, GATCTTATC, GATAACAA, GATAAGGA) or are dimer motifs (Fig R7.1 bottom cluster of motifs). The motifs of GATA1 and GATA3 are not clearly distinct (Fig R7.1) and therefore there is a large overlap of the regions enriched for motifs annotated to these TFs (average Jaccard index of 0.12). However, the expression of these two TFs is very distinct, with GATA1 expressed in K562 and GATA3 in MCF7. For this reason, SCENIC+ is able to distinguish regions bound by either GATA1 or GATA3. The Jaccard index of GATA1 and GATA3 predicted target regions is therefore only 0.02.

Fig R7.1. Overlap of regions enriched for GATA1 and GATA3 motifs. Heatmap showing Jaccard index of regions, in the Deeply Profiled Cell Lines, enriched for motifs annotated to GATA1 and GATA3 motifs. Color on the left indicates whether a motif is annotated for GATA1, GATA3 or both.

To conclude, SCENIC+ relies strongly on gene expression to separate the target regions of paralog TFs. A limitation to this approach is that separating the target regions of paralog TFs that are co-expressed is more difficult. However, the latter might still be possible if the motifs annotated to these paralog TFs are clearly distinct (this is often rare).

Fig R7.2. Gene expression of GATA1 and GATA3 in Deeply Profiled Cell Lines. UMAP of Deeply Profiled Cell Lines colored by the expression of GATA1 (red) and GATA3 (green).

Here is list of major key points regarding the Tutorial

1. The tutorial is not hard to follow for the Pbmcc example but can still be challenging to apply on new datasets. It would be nice to separate the tutorial into two key components: 1. Preprocessing steps, and 2.

SCENIC+ modeling. Current tutorial covers extensively on preprocessing steps but does not provide enough depths for SCENIC+ modeling.

We want to thank the reviewer for checking out the tutorial. The feedback is well appreciated; therefore, **we adapted the tutorial as suggested**. It is now split into two parts with more extensive documentation on the SCENIC+ modeling part. The new tutorial is available in the following link: https://scenicplus.readthedocs.io/en/development/pbmc_multiome_tutorial.html. We also include detailed step-by-step tutorials for the human cerebellum data sets at: https://pycistopic.readthedocs.io/en/latest/Single_sample_workflow-RTD.html, https://pycistarget.readthedocs.io/en/latest/pycistarget_scenic%2B_human_brain.html, and https://scenicplus.readthedocs.io/en/latest/Scenicplus_step_by_step-RTD.html.

2. Many users in the community already get used to tools like snapATAC/ArchR. It would be nice to build a convenient interface to interact with these tools. It would also be nice to provide support for commonly used file format, such as h5ad and Seurat objects, to make the tools easier to use.

We strongly agree with these points.

On the point to provide an interface to data objects obtained from tools like snapATAC, ArchR or R the version of cisTopic **we added the following two sections to the frequently asked questions (FAQ) page of the pycisTopic read the docs** (<https://pycistopic.readthedocs.io/en/latest/faqs.html>).

1. *“I have a cisTopic analysis in R, can I still use pycisTopic/SCENIC+?”*
2. *“I have an analysis with another tool (e.g. Signac/ArchR), can I still use pycisTopic/SCENIC+?”*

On the point to provide support to commonly used file formats. **We are currently working on converting our codebase to be able to support MuData** (<https://github.com/scverse/mudata>) as the main data object. Currently, we implemented functions to export the cisTopic, cisTarget and SCENIC+ analysis to MuData. See following GitHub commits for more information:

For pycisTopic:

<https://github.com/aertslab/pycisTopic/commit/00563281c445482ea9fadbc0b245d7144c5304a>

<https://github.com/aertslab/pycisTopic/commit/4ab13e352d363bd6ee16dbf183f469b929000e1d>

For pycistarget:

<https://github.com/aertslab/pycistarget/commit/c8485f1637c7812394ba8aeeb01e96c8dd788d35>

<https://github.com/aertslab/pycistarget/commit/dbf1d4a0b31aa5164889e396fffe1562948b59f5>

For SCENIC+:

<https://github.com/aertslab/pycistarget/commit/dbf1d4a0b31aa5164889e396fffe1562948b59f5>

We are currently still working to convert internal functions (as opposed to only the output) to this format.

In this context we also want to comment that the input into SCENIC+ is already an h5ad data format for the gene expression side of the data. Also, we already provide functions to export to the loom file format, to UCSC genome browser compatible formats and to Cytoscape compatible formats.

Minor points:

Most figures are too packed. The eGRN plots such as Figure2g look cool, but it is difficult to interpret. It might not be worthwhile to show more than once. The authors need to pick and choose materials more carefully to present in the main figures and make sure all labels are legible.

We have better selected the materials shown in the PBMC, ENCODE cell lines benchmark and GRN velocity figures, thus emphasizing more the key points. We have made sure that all **main and supplementary figures are clearly legible**.

Decision Letter, first revision:

Our ref: NMETH-A50180A

16th Mar 2023

Dear Dr. Aerts,

Thank you for submitting your revised manuscript "SCENIC+: single-cell multiomic inference of enhancers and gene regulatory networks" (NMETH-A50180A). It has now been seen by the original referees and their comments are below. The reviewers find that the paper has improved in revision, and therefore we'll be happy in principle to publish it in Nature Methods, pending minor revisions to satisfy the referees' final requests and to comply with our editorial and formatting guidelines.

TRANSPARENT PEER REVIEW

ORCID

Sincerely,

Lin Tang, PhD
Senior Editor
Nature Methods

Reviewer #1 (Remarks to the Author):

The revised manuscript contains several new analyses that convincingly demonstrate the good performance of SCENIC+ on several metrics. The descriptions of the methods and results are much clearer. New biological findings from the eGRN velocity analysis demonstrate the utility of the method. While benchmarking of GRNs is complicated by a lack of gold standards the combination of functional assays does provide convincing evidence that SCENIC+ is performing better than the tested alternatives.

The authors have done an excellent work in addressing the questions and I think the paper has been greatly improved.

Some minor points remain to be addressed:

1. The motif database and integration methods are significant contributions. There seems to be a difference in the motif data provided for download and numbers reported in Fig R4.1 etc. There are 10,250 motifs available for download in v10nr_clust_public/singletons and 49,054 in Fig R4.1 or 12,932 clusters. Maybe this is related to the Transfac motifs? Clearer organization of files and better descriptions would be helpful. For example, the documentation could link the processing to what is in the directories, saying something like "we did x analysis resulting in n motifs which can be found in y directory."

2. There seems to be a mix-up in terminology, are the lowest ranking triplets the best or the worst? 352 The best triplets are those with the lowest aggregated ranking (Fig 4a). 359 the top 10% triplets indeed have a significantly higher ChIP-seq signal, compared to regions 360 of lower-ranked triplets, and this is also true for Enformer predictions (Fig 4b).

3. In the discussion section:

840 Another challenge is the validation of enhancer-gene
841 relationships, for which we used ENCODE Hi-C data, which has a limited resolution of 5kb.
842 Indeed, often more than one candidate enhancer is present within 5kb. Comparison with higher
843 resolution data sets such as microC or promoter Capture Hi-C, when these techniques
844 become more widely applied, may provide a better standard for this task.

Whether or not Hi-C or related data can resolve this issue is a controversial, see the introduction in Zuin et al. "Nonlinear control of transcription through enhancer-promoter Interactions." Nature 604.7906 (2022): 571-577."

4. There are several recent papers related to gene regulation analysis using single cell multiome data, including the ones below. A careful check of recent publications to create a complete list would be appropriate.

789 The combination of both layers of information, namely TF binding sites and
790 gene expression, at single-cell resolution, then lead to the concept of enhancer-driven-GRN
791 (eGRN)^{3,24,47-50}.

Duren, Zhana, et al. "Regulatory analysis of single cell multiome gene expression and chromatin accessibility data with scREG." *Genome biology* 23.1 (2022): 1-19.

Lynch, Allen W., et al. "MIRA: Joint regulatory modeling of multimodal expression and chromatin accessibility in single cells." *Nature Methods* 19.9 (2022): 1097-1108.

Persad, Sitara, et al. "SEACells: Inference of transcriptional and epigenomic cellular states from single-cell genomics data." *bioRxiv* (2022): 2022-04.

Tian, Yijun, Alex Soupir, and Liang Wang. "Multiomic single cell sequencing identified regulatory elements in prostate cells." *Cancer Research* 82.12_Supplement (2022): 782-782.

5. Typo:

485 melanoma lines. a. Principal Component Analysis (PCA) projection of 936 pseudo multiome cells based

Reviewer #2 (Remarks to the Author):

Authors provide now an extensively revised version of the manuscript describing SCENIC+. The method description and benchmarking part have greatly improved. Methodological choices are well motivated and individual components of this complex framework are well described. Altogether, I am mostly satisfied with the authors responses. However, a few additional questions remain open and should be addressed (see below).

While I judged the methodological contributions, benchmarking and velocity analysis to be very strong, the manuscript is still long, complex and diverse. Authors are encouraged to focus on the strongest contributions of this work.

Major points

Results from Fig. S9i indicate that Signac has better F1 accuracy. This should be described in the main text. It would be also interesting to compare the results from CellOracle and Pandos with frameworks using only scATAC-seq data (Signac and archR). Panels S9i and S9j. It would be really important to the readership to understand in which cases scATAC-seq is enough or when multimodal data (scRNA-seq) is relevant. This important aspects should be strengthened in the text.

Line 759 - The "TF expression threshold" is a very important parameter for the velocity analysis, as this indicates, which cells have velocity vectors. The manuscript text does not indicate how this important parameter defined. There is only a description in the reply letter (70% top quantile). Is this parameter used in all studies showed here? What happens if larger or smaller values are selected?

Minor points

Line 114 - "DEM is more sensitive for comparing 115 sets of regions with similar motif content (Fig

S5)." I could not see DEM evaluation in S5. This is maybe in S4? Please clarify

Line 124 - Authors could provide intervals of running times in the text. Exact running times are not easy to read from plots.

Line 125 - "Importantly, we found that eGRN inference with SCENIC+ is as accurate when using multiome or well-paired scATAC-seq and scRNA-seq in the same tissue". Consider rephrasing. This analysis does not shown accuracy but rather that prediction are equivalent when sub-sampling is done.

Line 187 - "and accurate genomic target regions of these regulators," Authors should also consider re-writing as accuracy is not extensively explored in this section of the manuscript.

Line 229/Fig. 3C - Please display individual ARI values at the figure panels.

This text and panels in lines 310-318 do not go together. The text mentions results from panels S8-S9, while S19 was introduced a few lines before. I would expect supplement panels to be introduced in order. Also. some passages in the text do not correspond to the results. "On the other hand, SCENIC+ had a higher precision and 314 recall in terms of identified target regions of these TFs". Plot is S9j shown that SCENIC has a better recall but is not the method with highest precision.

Line 383 - Predictions on 4E-F are nice, but only computational. Authors could validate these (or consider including these results in the supplement only).

Line 680 - The velocity fields from some factors in figure 7b look odd. Maybe using a distinct 2-dimensional map (PCA or diffusion maps as in SCENIC tutorial instead of UMAP) would make the 2D visualization better. UMAPs are not suitable for representation of differentiation trajectories.

Reviewer #3 (Remarks to the Author):

The manuscript has substantially improved with better presentation, and the tutorial is now a lot easier to follow. The reviewers have addressed most of my concerns.

One remaining concern is, for benchmark purpose, the authors test on datasets with max 25K cells. More recent multiome datasets can be substantially bigger than that, and it is still unclear how scalable this method is. The authors reported the running time, but did not comment on memory usage. The manuscript mentioned it is tested on machines with 300, and 1000 gb memory. It will be nice if the authors can provide estimate of the amount of memory needed based on the size/complexity of the input datasets, and provide the guidance.

Minor comments:

Figure 1f: Please list total of ChIP-Seq datasets used in the benchmark.

Figure 2d: Please order the cell types to reflect their similarities. It is good to see that similarities of cell types based on UMAP reflect the similarities based on TF motif enrichment.

Figure 2g: Ebf1 ChIP-Seq signal is low. The authors may explain why this is a particularly convincing EBF binding site?

Author Rebuttal, first revision:

Reviewer #1 (Remarks to the Author):

The revised manuscript contains several new analyses that convincingly demonstrate the good performance of SCENIC+ on several metrics. The descriptions of the methods and results are much clearer. New biological findings from the eGRN velocity analysis demonstrate the utility of the method.

While benchmarking of GRNs is complicated by a lack of gold standards the combination of functional assays does provide convincing evidence that SCENIC+ is performing better than the tested alternatives.

The authors have done an excellent work in addressing the questions and I think the paper has been greatly improved.

We thank the reviewer for the constructive feedback and are happy to read that he/she is of the opinion that the work has been greatly improved.

Some minor points remain to be addressed:

1. The motif database and integration methods are significant contributions. There seems to be a difference in the motif data provided for download and numbers reported in Fig R4.1 etc. There are 10,250 motifs available for download in v10nr_clust_public/singletons and 49,054 in Fig R4.1 or 12,932 clusters. Maybe this is related to the Transfac motifs? Clearer organization of files and better descriptions would be helpful. For example, the documentation could link the processing to what is in the directories, saying something like “we did x analysis resulting in n motifs which can be found in y directory.”

We provide a schematic of the workflow in Fig S2. Briefly, we first pool all the motifs across 29 collections, which results in a total of 49,504 motifs. We then merge identical motifs across collections, resulting in 32,765 motifs. Before clustering, we removed desso and factorbook motifs, as they are derived from Deep Learning models and do not have direct annotations, and

motifs with $IC < 5$, resulting in 22,476 motifs. 1,265 of these motifs are dimers and are not included in the clustering step as well as 9,685 singlets, that were not similar enough to any motif. After clustering the remaining motifs, we obtained 1,985 clusters (containing at least 2 motifs). In total this results in 12,932 files. However, as Transfac Pro motifs are licensed, clusters (or singlets) solely formed by Transfac Pro motifs cannot be shared, resulting in 10,250 files that form part of the 'public' motif collection. Nevertheless, in the cases where the cluster is formed by additional non-Transfac Pro motifs, we keep the sharable motifs and the annotations of the Transfac Pro motif/s to the cluster.

Note that we provide precomputed databases in which the 12,932 files are used, since this does not require to share the licensed motifs. In addition, we show in the manuscript that the public motif collection achieves similar results compared to the full motif collection.

At https://resources.aertslab.org/cistarget/motif_collections/v10nr_clust_public/ we provide a total of 10,250 files (singlets, dimers and cluster) that can be publicly shared from the motif collection. In each file, all motifs forming part of the cluster are provided. Across all the clusters, a total of 17,995 motifs are provided (after removal of 4,481 Transfac Pro motifs)

At <https://resources.aertslab.org/cistarget/databases/> we provide the precomputed databases for mouse, human and fly, in which scores/rankings for the 12,932 files are provided.

We have included descriptions regarding this topic on the website, see https://resources.aertslab.org/cistarget/motif_collections/.

Figure R1. SCENIC+ motif collection clustering approach.

2. There seems to be a mix-up in terminology, are the lowest ranking triplets the best or the worst?

352 The best triplets are those with the lowest aggregated ranking (Fig 4a).
359 the top 10% triplets indeed have a significantly higher ChIP-seq signal, compared to
regions
360 of lower-ranked triplets, and this is also true for Enformer predictions (Fig 4b).

We apologize for the confusion in terminology. The triplets are ranked from best to worst. For this reason, the ones with the lowest numerical value are the best triplets. We adapted the manuscript and now refer to the best triplets as the ones on the top of the ranking or as triplets that are more highly ranked.

3. In the discussion section:

840 Another challenge is the validation of enhancer-gene
841 relationships, for which we used ENCODE Hi-C data, which has a limited resolution of
5kb.
842 Indeed, often more than one candidate enhancer is present within 5kb. Comparison with
higher
843 resolution data sets such as microC or promoter Capture Hi-C, when these techniques
844 become more widely applied, may provide a better standard for this task.

Whether or not Hi-C or related data can resolve this issue is a controversial, see the introduction in Zuin et al. "Nonlinear control of transcription through enhancer–promoter Interactions." Nature 604.7906 (2022): 571-577."

We agree with the reviewer's comment and added the following line to the manuscript: "Hi-C has a limited resolution and the relationship between physical enhancer-promoter distance and gene expression is still unclear and warrants further research." Where we also refer to the relevant literature.

4. There are several recent papers related to gene regulation analysis using single cell multiome data, including the ones below. A careful check of recent publications to create a complete list

would be appropriate.

789 The combination of both layers of information, namely TF binding sites and
790 gene expression, at single-cell resolution, then lead to the concept of enhancer-driven-GRN
791 (eGRN)^{3,24,47–50}.

Duren, Zhana, et al. "Regulatory analysis of single cell multiome gene expression and chromatin accessibility data with scREG." *Genome biology* 23.1 (2022): 1-19.

Lynch, Allen W., et al. "MIRA: Joint regulatory modeling of multimodal expression and chromatin accessibility in single cells." *Nature Methods* 19.9 (2022): 1097-1108.

Persad, Sitara, et al. "SEACells: Inference of transcriptional and epigenomic cellular states from single-cell genomics data." *bioRxiv* (2022): 2022-04.

Tian, Yijun, Alex Soupir, and Liang Wang. "Multiomic single cell sequencing identified regulatory elements in prostate cells." *Cancer Research* 82.12_Supplement (2022): 782-782.

We agree with the reviewer that creating a complete overview would be interesting. However, due to limitations on the number of references we can include in the manuscript this is not possible. Furthermore, we are of the opinion that we do cite the most important prior works.

Notwithstanding this, we did screen recent publications and below is a comprehensive list of publications related to multiomics gene regulatory network inference:

- <https://doi.org/10.1093/nargab/lqac023>
- <https://doi.org/10.1101/2022.12.15.520582>
- <https://doi.org/10.1038/s41540-020-00148-4>
- <https://doi.org/10.1186/s13073-021-00908-9>
- <https://www.genome.org/cgi/doi/10.1101/gr.271080.120>
- <https://doi.org/10.1126/sciadv.abl7393>
- <https://doi.org/10.1101/2022.06.15.496239>
- <https://doi.org/10.1038/s41467-023-36559-0>
- <https://doi.org/10.1093/bib/bbad011>
- <https://doi.org/10.12688/f1000research.130530.1>
- <https://doi.org/10.1093/bioadv/vbad003>
- <https://doi.org/10.1101/2023.02.03.527081>
- <https://doi.org/10.1016/j.cell.2021.07.039>
- <https://doi.org/10.1093/bioinformatics/btac117>
- <https://doi.org/10.1038/s41593-021-01002-4>

- <https://doi.org/10.1038/s41587-022-01284-4>
- <https://doi.org/10.1101/2022.07.25.501350>
- <https://doi.org/10.1101/2022.08.10.503335>
- <https://doi.org/10.1038/s41592-022-01595-z>

5. Typo:

485 melanoma lines. a. Principal Component Analysis (PCA) projection of 936 pseudo mutliome cells based

We fixed the typo.

Reviewer #2 (Remarks to the Author):

Authors provide now an extensively revised version of the manuscript describing SCENIC+. The method description and benchmarking part have greatly improved. Methodological choices are well motivated and individual components of this complex framework are well described. Altogether, I am mostly satisfied with the authors responses. However, a few additional questions remain open and should be addressed (see below).

While I judged the methodological contributions, benchmarking and velocity analysis to be very strong, the manuscript is still long, complex and diverse. Authors are encouraged to focus on the strongest contributions of this work.

We thank the reviewer for the constructive feedback and are happy to read that he/she is of the opinion that the work has greatly improved. We shortened the manuscript and made all sections less complex. The diversity of this work is one of its strong suites, describing the variety of analysis that can be done and research questions that can be answered using SCENIC+.

Major points

Results from Fig. S9i indicate that Signac has better F1 accuracy. This should be described in the main text. **It would be also interesting to compare the results from CellOracle and Pandos with frameworks using only scATAC-seq data (Signac and archR). Panels S9i and S9j. It would be really important to the readership to understand in which cases scATAC-seq is enough or when multimodal data (scRNA-scATAC-seq) is relevant. This important aspects should be strenghtened in the text.**

We want to clarify that the difference between the F1-metric of SCENIC+ and Signac is non-significant in this analysis (p-value equal to 0.89), so they perform equally well on this task. It would indeed be interesting to compare the results from other tools, like CellOracle and Pando, to tools using only scATAC-seq data, like Signac and ArchR. However, we believe this goes beyond the scope of this manuscript, which is not focused on benchmarking all available methods.

We adapted the following paragraph, see underlined additions:

“Note that even when only scATAC-seq is used to identify TFs per cell type (e.g., ArchR, Signac), the majority of known cell type specific TFs could still be recovered with accurate target region predictions (Extended Data Fig. 7g, h), showing that motif discovery is a powerful means for cell type specific TF prediction. However, using scATAC-seq alone resulted in large amounts of false positive TF predictions, representing TFs that are not expressed in the cell type but have a similar motif (Extended Data Fig. 7i).”

The last sentence stresses the importance of the additional information provided by gene expression data.

Line 759 - The "TF expression threshold" is a very important parameter for the velocity analysis, as this indicates, which cells have velocity vectors. The manuscript text does not indicate how this important parameter defined. There is only a description in the reply letter (70% top quantile). Is this parameter used in all studies showed here? What happens if larger or smaller values are selected?

We agree that the TF expression threshold is an important parameter in the velocity analysis. In all analyses, as detailed in the methods section, we use the same threshold (70% top quantile). This parameter indicates that only for cells above the 70% quantile (or as specified by the user) of the TF expression curve, velocity vectors will be calculated. When this parameter is set too low, the differentiation potential will be calculated for cells in which the TF is not or lowly expressed, adding noise to the analysis. We set the value by default to the 70% quantile because, below this threshold, for most of the cells, the expression of targets seems to precede gene expression, which given the low expression of the TF, we interpret as noise (Fig. R2). When this parameter is set to high, the differentiation potential may be disregarded for cells in which the TF is already expressed, although not to maximum levels, which can mask the dynamics between TF expression and target activation.

Figure R2. Representation of svp dynamics along the two branches of the eye-antennal disc differentiation. The grey horizontal line represents the TF expression threshold.

Minor points

Line 114 - "DEM is more sensitive for comparing 115 sets of regions with similar motif content (Fig S5)." I could not see DEM evaluation in S5. This is maybe in S4? Please clarify

Figure S5 is in fact the correct figure. In this figure we illustrate that the Differentially Enriched Motifs (DEM) analysis can be used to analyze differences in regions with very similar motif content. In the figure we use the example of SOXE target regions in different cell types: melanoma, astrocytes and oligodendrocytes. cisTarget, which uses all consensus peaks in the genome as background, is only able to detect enrichment for SOX-motifs but not for their co-factors (see panel b). On the other hand, DEM, which uses the SOXE target regions of the two other cell types as background, is able to detect enrichment for other motifs besides SOX (see

panel c). These other motifs represent cell type specific co-factors. For that reason, we stated that DEM is more *sensitive* compared to cisTarget to detect these extra motifs. To avoid confusion we adapted the sentence, see underlined change:

“Both the cisTarget and DEM algorithm outperform Homer (Fig. 1f, Extended Data Fig. 2d), with the DEM algorithm enabling detection of differential motifs in sets of regions with similar motif content (Extended Data Fig. 2i-j).”

Line 124 - Authors could provide intervals of running times in the text. Exact running times are not easy to read from plots.

We now summarize the results of this analysis as follows in the text: “The overall running time and memory of the workflow ranges from 1 hour and 21Gb to 44 hours and 461Gb for the smallest and largest tested dataset (Extended Data Fig. 3).”

Line 125 - "Importantly, we found that eGRN inference with SCENIC+ is as accurate when using multiome or well-paired scATAC-seq and scRNA-seq in the same tissue". Consider rephrasing. This analysis does not shown accuracy but rather that prediction are equivalent when sub-sampling is done

.

Thank you for this suggestion, the word “accurate” was indeed not well suited. This analysis is not showcased in the final article version due to limitations on the amount of material that can be accommodated.

Line 187 - "and accurate genomic target regions of these regulators," Authors should also consider re-writing as accuracy is not extensively explored in this section of the manuscript.

Agreed. We removed the word “accurate”.

Line 229/Fig. 3C - Please display individual ARI values at the figure panels.

We have added the individual ARI values to the figure.

This text and panels in lines 310-318 do not go together. The text mentions results from panels S8-S9, while S19 was introduced a few lines before. I would expect supplement panels to be introduced in order. Also, some passages in the text do not correspond to the results. "On the other hand, SCENIC+ had a higher precision and 314 recall in terms of identified target regions of these TFs". Plot is S9j shown that SCENIC has a better recall but is not the method with highest precision.

We agree that the panels should follow the order in the text. We now introduce the supplementary figures in order:

"Finally, to test whether the same conclusions can be drawn from a real single-cell multiome data set we repeated the benchmark analyses on the PBMC data (Extended Data Fig. 7). GRaNIE is not present in this benchmark as it was developed for bulk data sets and did not scale to this larger (10K cells) data set. GRNs from all methods were able to recapitulate all biological cell states, except the GRN inferred by FigR (Extended Data Fig. 7d). SCENIC+ and Pando performed best in terms of identifying biologically relevant TFs, with Pando finding additional TFs compared to SCENIC+ (Extended Data Fig. 7e, j). SCENIC+ had the highest precision and recall in terms of target region predictions (Extended Data Fig. 7f). Note that even when only scATAC-seq is used to identify TFs per cell type (e.g., ArchR⁴⁴, Signac⁴⁵), the majority of known cell type specific TFs could still be recovered with accurate target region predictions (Extended Data Fig. 7g, h), showing that motif discovery is a powerful means for cell type specific TF prediction. However, using scATAC-seq alone resulted large amounts of false positive TF predictions, representing TFs that are not expressed in the cell type but have a similar motif (Extended Data Fig. 7i)."

For Supplementary Fig. 7f, we would like to comment that the difference between CellOracle and SCENIC+ in terms of precision is not significantly different (p-value of 0.94). In other words, both SCENIC+ and CellOracle are in fact the two methods with the highest precision in this analysis.

Line 383 - Predictions on 4E-F are nice, but only computational. Authors could validate these (or consider including these results in the supplement only).

We want to comment that panel e of this figure already has some experimental validation. That is SCENIC+ predicts a specific region to be bound by HNF4A, FOXA2 and CEBPB and in this panel we show that this region indeed has high ChIP-seq signal for those TFs. Experimentally validating the *in silico* saturation mutagenesis shown in panel f would be very interesting but is

however beyond the scope of this paper. This panel also serves as a validation, although a computational one, showing that the predictions of an independent computational tool (Enformer) agree, to a certain extent, with the predictions of SCENIC+. We removed the panel from the manuscript due to space constraints.

Line 680 - The velocity fields from some factors in figure 7b look odd. Maybe using a distinct 2-dimensional map (PCA or diffusion maps as in SCENIC tutorial instead of UMAP) would make the 2D visualization better. UMAPs are not suitable for representation of differentiation trajectories.

We favor to use the UMAP representation for consistency (as it is the same one used in Fig. 5 as well), and we find it clearer compared to PCA or diffusion maps. Using PCA, we observed that most PCs, except the first, are affected by the sample of origin (Fig. R3). On the diffusion map representation for this data, cells tend to be on top of each other (especially towards OL), which makes it hard to visualize the arrows (Fig. R3, R4). Overall, the conclusions are the same despite the dimensionality reduction approach used.

Figure R3. Diffusion map and PCA representation of the oligodendrocyte differentiation data set.

Figure R4. Differentiation arrows on UMAP, PCA and diffusion map projections.

Reviewer #3 (Remarks to the Author):

The manuscript has substantially improved with better presentation, and the tutorial is now a lot easier to follow. The reviewers have addressed most of my concerns.

We thank the reviewer for the constructive feedback and are happy to read that the presentation and tutorials have improved.

One remaining concern is, for benchmark purpose, the authors test on datasets with max 25K cells. More recent multiome datasets can be substantially bigger than that, and it is still unclear how scalable this method is. The authors reported the running time, but did not comment on memory usage. The manuscript mentioned it is tested on machines with 300, and 1000 gb memory. It will be nice if the authors can provide estimate of the amount of memory needed based on the size/complexity of the input datasets, and provide the guidance.

Thank you for the suggestion. We have measured the maximum memory peak in the three steps of the workflow (minimum steps) for the simulated data sets. Overall, we see that memory usage increases exponentially with the number of cells (Fig. R5).

Figure R5. Maximum memory usage in the minimum steps of the workflow upon different numbers of cells and coverage.

We have summarized the results of this analysis as:

“The overall running time and memory of the workflow ranges from 1 hour and 21Gb to 44 hours and 461Gb for the smallest and largest tested dataset (Extended Data Fig. 3).”

Minor comments:

Figure 1f: Please list total of ChIP-Seq datasets used in the benchmark.

In total, we used 309 ChIP-seq data sets. We have added this information to the figure legend. These exact data sets used are listed in Supplementary Table 4.

Figure 2d: Please order the cell types to reflect their similarities. It is good to see that similarities of cell types based on UMAP reflect the similarities based on TF motif enrichment.

We have reordered the cells based on their similarities of averaged target region and gene AUC values (Fig. R6), the same values as were used to generate the tSNE. We thank the reviewer for this suggestion as the plot indeed is more informative with this change.

Figure. R6. Dendrogram based on Euclidian distance between cell types based on averaged target gene and target region AUC.

Figure 2g: Ebf1 ChIP-Seq signal is low. The authors may explain why this is a particularly convincing EBF binding site?

We agree that the EBF1 ChIP-seq signal is low for the predicted target region. This might represent a false positive prediction from SCENIC+, even though there is some signal. To improve the clarity of this panel we have now added boxes indicating called peaks on the ChIP-seq signal. Indeed, the predicted EBF1 target region is not called as a peak. In conclusion, this is not the most convincing EBF1 target region. However, we think this panel is still useful to illustrate the output from SCENIC+ highlighting that many, but not all, predicted target regions of SCENIC+ have ChIP-seq signal for the corresponding TFs.

In this context we adapted the text, see underlined addition:

“In particular, for B cells SCENIC+ suggests cooperativity between EBF1(+), PAX5(+) and POU2F2/AF1(+) (Fig. 2e), with a strong overlap of most of their predicted target enhancers with EBF1, PAX5 and POU2F2 ChIP-seq data (Fig. 2f-g).”

Final Decision Letter:

6th Jun 2023

Dear Professor Aerts,

I am pleased to inform you that your Article, "SCENIC+: single-cell multiomic inference of enhancers and gene regulatory networks", has now been accepted for publication in Nature Methods. Your paper is tentatively scheduled for publication in our September print issue, and will be published online prior to that. The received and accepted dates will be 19th Aug 2022 and 6th Jun 2023. This note is intended to let you know what to expect from us over the next month or so, and to let you know where to address any further questions.

Once your paper is typeset, you will receive an email with a link to choose the appropriate publishing options for your paper and our Author Services team will be in touch regarding any additional information that may be required.

Please note that *Nature Methods* is a Transformative Journal (TJ). Authors may publish their research with us through the traditional subscription access route or make their paper immediately open access through payment of an article-processing charge (APC). Authors will not be required to make a final decision about access to their article until it has been accepted. [Find out more about Transformative Journals](https://www.springernature.com/gp/open-research/transformative-journals)

Authors may need to take specific actions to achieve [compliance with funder and institutional open access mandates](https://www.springernature.com/gp/open-research/funding/policy-compliance-faqs). If your research is supported by a funder that requires immediate open access (e.g. according to [Plan S principles](https://www.springernature.com/gp/open-research/plan-s-compliance)) then you should select the gold OA route, and we will direct you to the compliant route where possible. For authors selecting the subscription publication route, the journal's standard licensing terms will need to be accepted, including [self-archiving policies](https://www.springernature.com/gp/open-research/policies/journal-policies). Those licensing terms will supersede any other terms that the author or any third party may assert apply to any version of the manuscript.

Your paper will now be copyedited to ensure that it conforms to Nature Methods style. Once proofs are generated, they will be sent to you electronically and you will be asked to send a corrected version within 24 hours. It is extremely important that you let us know now whether you will be difficult to contact over the next month. If this is the case, we ask that you send us the contact information (email, phone and fax) of someone who will be able to check the proofs and deal with any last-minute problems.

If, when you receive your proof, you cannot meet the deadline, please inform us at rjsproduction@springernature.com immediately.

Once your manuscript is typeset and you have completed the appropriate grant of rights, you will receive a link to your electronic proof via email with a request to make any corrections within 48 hours. If, when you receive your proof, you cannot meet this deadline, please inform us at rjsproduction@springernature.com immediately.

Once your paper has been scheduled for online publication, the Nature press office will be in touch to confirm the details.

Content is published online weekly on Mondays and Thursdays, and the embargo is set at 16:00 London time (GMT)/11:00 am US Eastern time (EST) on the day of publication. If you need to know the exact publication date or when the news embargo will be lifted, please contact our press office after you have submitted your proof corrections. Now is the time to inform your Public Relations or Press Office about your paper, as they might be interested in promoting its publication. This will allow them time to prepare an accurate and satisfactory press release. Include your manuscript tracking number NMETH-A50180B and the name of the journal, which they will need when they contact our office.

About one week before your paper is published online, we shall be distributing a press release to news organizations worldwide, which may include details of your work. We are happy for your institution or funding agency to prepare its own press release, but it must mention the embargo date and Nature Methods. Our Press Office will contact you closer to the time of publication, but if you or your Press Office have any inquiries in the meantime, please contact press@nature.com.

Nature Portfolio journals [encourage authors to share their step-by-step experimental protocols](https://www.nature.com/nature-research/editorial-policies/reporting-standards#protocols) on a protocol sharing platform of their choice. Nature Portfolio 's Protocol Exchange is a free-to-use and open resource for protocols; protocols deposited in Protocol Exchange are citable and can be linked from the published article. More details can found at www.nature.com/protocolexchange/about.

Please note that you and any of your coauthors will be able to order reprints and single copies of the issue containing your article through Nature Portfolio 's reprint website, which is located at <http://www.nature.com/reprints/author-reprints.html>. If there are any questions about reprints please send an email to author-reprints@nature.com and someone will assist you.

Please feel free to contact me if you have questions about any of these points. Thank you very much for publishing your paper at Nature Methods!

Best regards,

Lin Tang, PhD
Senior Editor
Nature Methods